# LoRA-One: One-Step Full Gradient Could Suffice for Fine-Tuning Large Language Models, Provably and Efficiently

Yuanhe Zhang [1 2]    Fanghui Liu [2 3]    Yudong Chen [4]

## Abstract

This paper explores how theory can guide and enhance practical algorithms, using Low-Rank Adaptation (LoRA) (Hu et al., 2022) in large language models as a case study. We rigorously prove that, under gradient descent, LoRA adapters align with specific singular subspaces of the one-step full fine-tuning gradient. This result suggests that, by properly initializing the adapters using the one-step full gradient, subspace alignment can be achieved immediately—applicable to both linear and nonlinear models. Building on our theory, we propose a theory-driven algorithm, *LoRA-One*, where the linear convergence (as well as generalization) is built and incorporating preconditioners theoretically helps mitigate the effects of ill-conditioning. Besides, our theory reveals connections between *LoRA-One* and other gradient-alignment-based methods, helping to clarify misconceptions in the design of such algorithms. *LoRA-One* achieves significant empirical improvements over LoRA and its variants across benchmarks in natural language understanding, mathematical reasoning, and code generation. Code is available at: https://github.com/YuanheZ/LoRA-One.

## 1. Introduction

How to efficiently approximate or learn nonlinear models is a central question in large-scale machine learning, especially in the era of large language models (LLMs) (Brown et al., 2020; Thoppilan et al., 2022). Fine-tuning (Dodge et al., 2020) aims to make LLMs perform well on new tasks

---

[1]Department of Statistics, University of Warwick, UK. [2]Department of Computer Science, University of Warwick, UK. [3]Centre for Discrete Mathematics and its Applications (DIMAP), University of Warwick, UK. [4]Department of Computer Sciences, University of Wisconsin-Madison, USA. Correspondence to: Fanghui Liu <fanghui.liu@warwick.ac.uk>.

*Proceedings of the 42$^{nd}$ International Conference on Machine Learning*, Vancouver, Canada. PMLR 267, 2025. Copyright 2025 by the author(s).

while retain the knowledge from pre-trained models. For scalability, we expect that fine-tuning can be conducted with low computation/memory cost, i.e., parameter-efficient fine-tuning (PEFT) (Houlsby et al., 2019; Han et al., 2024).

One typical PEFT strategy is Low-Rank Adaptation (LoRA) (Hu et al., 2022), which learns an approximation of the unknown feature shift $\Delta$ by two low-rank matrices $\boldsymbol{A}$ and $\boldsymbol{B}$ with rank $r$, i.e. $\Delta \approx \boldsymbol{AB}$ under the following initialization (denoted by index 0):

$$[\boldsymbol{A}_0]_{ij} \sim \mathcal{N}(0, \alpha^2) \quad [\boldsymbol{B}_0]_{ij} = 0, \quad \alpha > 0. \quad \text{(LoRA-init)}$$

To improve the performance in the downstream tasks, various LoRA-based algorithms have been proposed based on, e.g., refined initialization (Li et al., 2025), learning rates (Hayou et al., 2024), efficiency (Kopiczko et al., 2024), and gradient information (Meng et al., 2024; Wang et al., 2024).

Although LoRA is conceptually simple, its optimization dynamics are inherently nonlinear and non-convex. There is few theoretical understanding of its behavior, e.g., optimization from *lazy-training* regime (Jang et al., 2024; Malladi et al., 2023; Liu et al., 2025) to *non-lazy training* regime (Kim et al., 2025) and generalization guarantees in some simplified settings (Dayi & Chen, 2024). It still remains unclear how (low-rank) gradient updates in LoRA evolve and which subspaces LoRA will converge to. More importantly, given the application-driven nature of LoRA, a rigorous theoretical understanding should not only explain its behavior but also inform practical algorithm design. ***The goal of this work is to enhance LoRA's empirical performance through theoretically grounded insights***. To this end, we address two key questions at the intersection of theory and practice:

- *Q1: How to characterize low-rank dynamics of LoRA and the associated subspace alignment in theory?*

- *Q2: How can our theoretical results contribute to algorithm design for LoRA in practice?*

### 1.1. Contributions

In this work, we theoretically investigate the behavior of gradient descent (GD) update of LoRA parameters $(\boldsymbol{A}_t, \boldsymbol{B}_t)$ and identify the subspaces they align with. Our theory identifies the ***optimal*** initialization strategies in the perspective

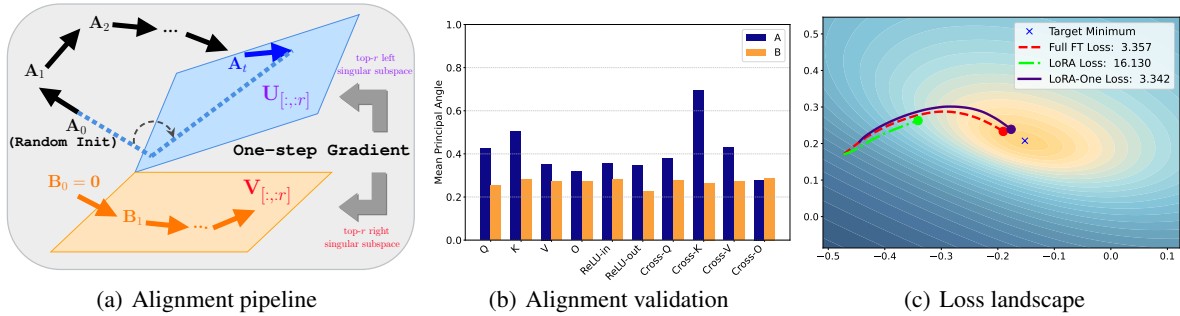

(a) Alignment pipeline      (b) Alignment validation      (c) Loss landscape

Figure 1: (a) Illustration of the alignment behavior of LoRA with the certain singular subspaces. (b) The mean principle angles within each layer class is estimated for the alignment of fine-tuning a T5 base model (Raffel et al., 2020) on MRPC using LoRA. The principal angle measures the distance between the projection matrices of the top-$r$ (LoRA rank) singular subspace of the LoRA matrices and that of the one-step gradient. Note that the principal angles are approximately 1 at initialization and then decrease for alignment. (c) Comparison of trajectories among full fine-tuning (Full FT), LoRA, and *LoRA-One* under gradient descent. We use a two-layer neural network pretrained on odd-labeled data and then fine-tune on even-labeled MNIST data. Experimental details are presented in Appendix G.1.

of subspace alignment, and we find that it also performs well on some real-world datasets. We term this initialization as **spectral initialization**, which leverages the information of one-step full gradient, leading to the theoretical grounded algorithm, *LoRA-One*. This algorithm incorporated into several architectures achieves promising performance on natural language processing (NLP), reasoning tasks when compared to LoRA and its variants. Our contributions from theory (see Table 1 for summary) to practice are:

**i) Alignment and algorithm design principles:** We start by analyzing LoRA for fine-tuning a multi-output linear model. Denoting one-step gradient of full fine-tuning as $G^\natural$, we prove that the gradient update aligns $A_t$ with the left top-$r^*$ singular subspace of $G^\natural$ while $B_t$ always stays in a right top-$r^*$ singular subspace w.r.t. $G^\natural$ as shown in Fig. 1(a), where $r^*$ is the rank of $\Delta$. This alignment phenomenon is also empirically verified in real-world fine-tuning tasks, see Fig. 1(b). Based on the alignment results, we compute the singular value decomposition (SVD) of $G^\natural = \widetilde{U}_{G^\natural} \widetilde{S}_{G^\natural} \widetilde{V}_{G^\natural}^\top$. The alignment can be directly achieved at the certain initialization strategy, termed as *spectral initialization*

$$A_0 = \sqrt{\gamma} \left[ \widetilde{U}_{G^\natural} \right]_{[:,1:r]} \left[ \widetilde{S}_{G^\natural}^{1/2} \right]_{[1:r]},$$
$$B_0 = \sqrt{\gamma} \left[ \widetilde{S}_{G^\natural}^{1/2} \right]_{[1:r]} \left[ \widetilde{V}_{G^\natural} \right]_{[:,1:r]}^\top, \quad \text{(Spectral-init)}$$

where $\gamma$ is a tuning parameter. By (Spectral-init), we theoretically ensure that $\|A_0 B_0 - \Delta\|_F$ is sufficiently small at beginning, see the theoretical results for linear models in Section 3.2 and nonlinear models in Section 4, respectively. It demonstrates *the sufficiency of using one-step full gradient*, which can be numerically verified on several real-world benchmarks, serving as the algorithm principle.

**ii) Global convergence and generalization guarantees:** Under spectral initialization, continuing gradient descent (GD) updates for $(A_t, B_t)$, we further establish the linear convergence rate of $\|A_t B_t - \Delta\|_F$ for both linear and nonlinear models. This linear rate, however, is sensitive to the condition number $\kappa(\Delta)$ of $\Delta$, leading to unsatisfactory convergence performance if $\Delta$ is ill-conditioned. To address this issue, we rigorously show that adding preconditioners into the GD update eliminates the dependence on the condition number; see Section 3.2 and Section 4, respectively.

Moreover, our theory aims to clarify certain misunderstandings in prior algorithm designs. Specifically, it identifies the correct subspace for alignment and highlights potential limitations of previous LoRA variants based on gradient alignment—such as *LoRA-GA* (Wang et al., 2024); see the discussion in Section 5.

**iii) Performance improvement in numerical and read-world datasets:** Guided by our theory, the spectral initialization strategy (Spectral-init) leads to our theoretically grounded algorithm, *LoRA-One*. As shown in Fig. 1(c), our numerical results demonstrate that *LoRA-One*'s trajectory is close to the full fine-tuning (Full FT) and obtain lower loss than LoRA.

We conducted experimental comparisons between *LoRA-One* and standard LoRA-based algorithms across various NLP benchmarks, including natural language understanding (NLU), mathematical reasoning, and code generation tasks. For instance, using only (Spectral-init), it takes just **one second** on some NLU tasks to achieve performance comparable to LoRA which requires tens of seconds. On the HumanEval benchmark, LLaMA 2-7B fine-tuned with *LoRA-One* achieves a score of 28.66, outperforming stan-

Table 1: Main results in the main text and appendix from subspace alignment to global convergence.

| Model | Results | Algorithm | Initialization | Conclusion |
|---|---|---|---|---|
| Linear | Theorem 3.1 | GD | (LoRA-init) | Subspace alignment of $\boldsymbol{B}_t$ |
| | Theorem 3.2 | GD | (LoRA-init) | Subspace alignment of $\boldsymbol{A}_t$ |
| | Theorem 3.3 | GD | (Spectral-init) | $\|\boldsymbol{A}_0\boldsymbol{B}_0 - \Delta\|_{\mathrm{F}}$ is small |
| | Theorem C.17 | GD | (Spectral-init) | Linear convergence of $\|\boldsymbol{A}_t\boldsymbol{B}_t - \Delta\|_{\mathrm{F}}$ |
| | Theorem C.21 | Precondition GD | (Spectral-init) | Linear convergence rate independent of $\kappa(\Delta)$ |
| Nonlinear | Theorem 4.2 | Precondition GD | (Spectral-init) | Linear convergence rate independent of $\kappa(\Delta)$ |

dard LoRA (25.85) by 2.81, while maintaining almost the same time and memory costs.

**Notations** For a matrix $\boldsymbol{A}$, let $\|\boldsymbol{A}\|_{op}$ denote its operator and $\|\boldsymbol{A}\|_{\mathrm{F}}$ its Frobenius norm. Let $\odot$ denote the Hadamard (i.e., entrywise) matrix product. We use $\boldsymbol{I}_n$ to denote the $\mathbb{R}^{n \times n}$-valued identity matrix. The notation $\boldsymbol{U_A}$ denotes the left singular matrix of the compact SVD of $\boldsymbol{A}$ and $\boldsymbol{U_{A,\perp}}$ denotes the corresponding orthogonal complement. Similarly, $\boldsymbol{V_A}$ denotes the right singular matrix of $\boldsymbol{A}$ and $\boldsymbol{V_{A,\perp}}$ denotes its orthogonal complement. Let $\boldsymbol{U}_{r^*}(\boldsymbol{A})$ denote the left singular subspace spanned by the $r^*$ largest singular values of $\boldsymbol{A}$ and $\boldsymbol{U}_{r^*,\perp}(\boldsymbol{A})$ denote the left singular subspace orthogonal to $\boldsymbol{U}_{r^*}(\boldsymbol{A})$. Similarly define $\boldsymbol{V}_{r^*}(\boldsymbol{A})$ and $\boldsymbol{V}_{r^*,\perp}(\boldsymbol{A})$ for the right singular subspace. A complete list notations can be found in Table 5 of Appendix A.

### 1.2. Related Work

**Parameter-Efficient Fine-Tuning (PEFT):** LoRA (Hu et al., 2022) and its variants have received great attention for downstream applications. The variants of LoRA focus on imbalance stepsize (Hayou et al., 2024), initialization using SVD of pre-trained weights (Meng et al., 2024), gradient approximation (Wang et al., 2024; 2025) for better performance, reducing parameters (Kopiczko et al., 2024) efficiency, preconditioned algorithm (Zhang & Pilanci, 2024) for stability.

In theory, the training dynamics and generalization ability of LoRA are rarely discovered. Based on the empirical evidence of kernel behavior of LoRA in Malladi et al. (2023), the global convergence is given by Jang et al. (2024) for LoRA with rank $\mathcal{O}(\sqrt{N})$ under the *lazy training* (Jacot et al., 2018) as well as Liu et al. (2025) on PL* condition. Beyond the lazy training regime, Kim et al. (2025) study the loss landscape of LoRA as well as its implicit bias. For generalization, Dayi & Chen (2024) derive the sample/time complexity by exploring the SGD dynamics of rank-1 LoRA, related to single-index model (Arous et al., 2021). In our work, we study the dynamics of LoRA from the perspective of subspace alignment, which has some overlap with matrix sensing as below.

**Matrix Sensing under Gradient Descent:** Since LoRA performs fine-tuning using a Burer-Monterio factorization, it admits similarities with matrix sensing problems, including the symmetric matrix problem with $r = r^*$ (Li et al., 2018) and $r \geq r^*$ (Stöger & Soltanolkotabi, 2021); asymmetric problem with $r \geq r^*$ (Soltanolkotabi et al., 2023; Xiong et al., 2024). Regarding initialization, small initialization (Ding et al., 2022) and spectral initialization (Ma et al., 2021) help convergence with theoretical guarantees, which is applied to LoRA under certain specific settings (Xu et al., 2025). Besides, adding preconditioner (Zhang et al., 2021; Tong et al., 2021; Xu et al., 2023; Zhang et al., 2023; Giampouras et al., 2024; Zhu et al., 2024) is beneficial to solve the problem of ill-conditioned ground truth matrix.

Technically, for the alignment part, our theory leverages some techniques from Soltanolkotabi et al. (2023). However, the symmetrization technique used in prior work cannot be applied to decouple the GD dynamics of $(\boldsymbol{A}_t, \boldsymbol{B}_t)$, posing a challenge in analyzing their individual spectral behaviors. To overcome this limitation, we develop a novel approach that enables a detailed analysis of the distinct spectral dynamics of $\boldsymbol{A}_t$ and $\boldsymbol{B}_t$, which is one technical contribution of this work. In fact, one-step gradient information has been used in deep learning theory, demonstrating that it allows for feature learning under different stepsizes (Ba et al., 2022; Moniri et al., 2024; Cui et al., 2024; Dandi et al., 2025). Besides, for the nonlinear model part, dynamical analysis are normally based on classical gradient-based algorithm (Damian et al., 2022; Lee et al., 2024) for feature learning. Nevertheless, how such model behaves under low-rank updates under (Spectral-init) is still unclear to our knowledge.

## 2. Problem Settings

In this section, we introduce the problem setting of fine-tuning pre-trained linear and nonlinear models with the following assumptions for our theory.

### 2.1. Basic Assumptions

We consider both linear and nonlinear pre-trained models with multiple outputs and thus matrix parameters (instead

of vectors), which is consistent with LoRA in practice.

**Assumption 2.1** (Pre-trained model). For the input $\boldsymbol{x} \in \mathbb{R}^d$, we denote by $\boldsymbol{W}^\natural \in \mathbb{R}^{d \times k}$ the **known** pre-trained parameter matrix. We assume that the pre-trained model can be linear or nonlinear with $\sigma(\cdot) = \max\{0, \cdot\}$ being the (entry-wise) ReLU activation function.

$$f_{\mathrm{pre}}(\boldsymbol{x}) := \begin{cases} (\boldsymbol{x}^\top \boldsymbol{W}^\natural)^\top \in \mathbb{R}^k & \text{linear} \\ \sigma[(\boldsymbol{x}^\top \boldsymbol{W}^\natural)^\top] \in \mathbb{R}^k & \text{nonlinear} \end{cases}.$$

Note that our results can handle large dimension $d$ and $k$. For fine-tuning, we assume there exists an **unknown** low-rank feature shift $\Delta$ on $\boldsymbol{W}^\natural$ that we aim to estimate.

**Assumption 2.2.** The downstream feature matrix $\widetilde{\boldsymbol{W}}^\natural := \boldsymbol{W}^\natural + \Delta$ admits an **unknown** low-rank feature shift $\Delta \in \mathbb{R}^{d \times k}$, where $\mathrm{Rank}(\Delta) = r^* < \min\{d, k\}$.

This assumption is widely used in the literature on LoRA analysis and matrix factorization (Zhang et al., 2021; Stöger & Soltanolkotabi, 2021; Soltanolkotabi et al., 2023; Xiong et al., 2024). Next we assume the following data generation process, i.e., label-noiseless and well-behaved data.

**Assumption 2.3** (Data generation process for fine-tuning). Given the unknown $\widetilde{\boldsymbol{W}}^\natural$, the label $\widetilde{\boldsymbol{y}}$ is generated by

$$\widetilde{\boldsymbol{y}} := \begin{cases} (\widetilde{\boldsymbol{x}}^\top \widetilde{\boldsymbol{W}}^\natural)^\top \in \mathbb{R}^k, & \{\widetilde{\boldsymbol{x}}_i\}_{i=1}^N \overset{i.i.d.}{\sim} SG, & \text{linear} \\ \sigma[(\widetilde{\boldsymbol{x}}^\top \widetilde{\boldsymbol{W}}^\natural)^\top], & \{\widetilde{\boldsymbol{x}}_i\}_{i=1}^N \overset{i.i.d.}{\sim} \mathcal{N}(\boldsymbol{0}, \boldsymbol{I}_d) & \text{nonlinear} \end{cases},$$

where $SG$ denotes the probability distribution for isotropic centered sub-Gaussian random vectors. We assume that we have $N$ i.i.d training data $\{\widetilde{\boldsymbol{x}}_i, \widetilde{\boldsymbol{y}}_i\}_{i=1}^N$ for fine-tuning.

Note that the nonlinear model can be regarded as a special case of multi-index model (Damian et al., 2022; Abbe et al., 2022; Bietti et al., 2023) and Gaussian data is a common assumption in the analysis of single/multi-index models (Damian et al., 2022; Lee et al., 2024; Oko et al., 2024). We additionally assume that $d < N$, which coincides with practical settings of LoRA for LLaMA 2-7b (Touvron et al., 2023) on real-world datasets, e.g., MetaMathQA (Yu et al., 2024) and Code-Feedback (Zheng et al., 2024), where $d = 128 \sim 4096$ and $N$ is on the order of $10^5$.

### 2.2. Full Fine-tuning and LoRA

Our goal is to efficiently recover $\Delta$ by fine-tuning on the downstream data. Let the complete SVD of $\Delta \in \mathbb{R}^{d \times k}$ be

$$\Delta = \widetilde{\boldsymbol{U}} \widetilde{\boldsymbol{S}}^* \widetilde{\boldsymbol{V}}^\top := \begin{bmatrix} \boldsymbol{U} & \boldsymbol{U}_\perp \end{bmatrix} \begin{bmatrix} \boldsymbol{S}^* & \boldsymbol{0} \\ \boldsymbol{0} & \boldsymbol{0} \end{bmatrix} \begin{bmatrix} \boldsymbol{V}^\top \\ \boldsymbol{V}_\perp^\top \end{bmatrix}, \quad (1)$$

where $\widetilde{\boldsymbol{U}} \in \mathbb{R}^{d \times d}$ and $\widetilde{\boldsymbol{V}} \in \mathbb{R}^{k \times k}$ are the left and right singular matrices, and $\widetilde{\boldsymbol{S}}^* \in \mathbb{R}^{d \times k}$ is a rank-$r^*$ diagonal matrix with nonzero singular values $\{\lambda_i^*\}_{i=1}^{r^*}$. It admits

the compact SVD $\Delta = \boldsymbol{U} \boldsymbol{S}^* \boldsymbol{V}^\top$ with $\boldsymbol{U} \in \mathbb{R}^{d \times r^*}$, $\boldsymbol{V}^\top \in \mathbb{R}^{r^* \times k}$, and $\boldsymbol{S}^* \in \mathbb{R}^{r^* \times r^*}$. The left/right singular subspaces spanned by $\boldsymbol{U}$ and $\boldsymbol{V}$ play an important role in our analysis.

We write the downstream data in a compact form $\widetilde{\boldsymbol{X}} = [\widetilde{\boldsymbol{x}}_1, \cdots, \widetilde{\boldsymbol{x}}_N]^\top \in \mathbb{R}^{N \times d}$ and the label matrix $\widetilde{\boldsymbol{Y}} = [\widetilde{\boldsymbol{y}}_1 \cdots \widetilde{\boldsymbol{y}}_N]^\top \in \mathbb{R}^{N \times k}$ is generated by either linear or nonlinear target functions in Assumption 2.3. We introduce the training based on full fine-tuning and LoRA below.

**Full Fine-tuning:** We consider the following empirical risk minimization with a squared loss

$$L(\boldsymbol{W}) := \frac{1}{2N} \begin{cases} \left\| \widetilde{\boldsymbol{X}} \boldsymbol{W} - \widetilde{\boldsymbol{Y}} \right\|_{\mathrm{F}}^2 & \text{linear,} \\ \left\| \sigma(\widetilde{\boldsymbol{X}} \boldsymbol{W}) - \widetilde{\boldsymbol{Y}} \right\|_{\mathrm{F}}^2 & \text{nonlinear} \end{cases}, \quad (2)$$

where the parameter $\boldsymbol{W}$ can be learned by gradient descent (GD) initialized at $\boldsymbol{W}^\natural$, i.e., $\boldsymbol{W}_0 := \boldsymbol{W}^\natural$.

**LoRA:** It updates two low-rank matrices $\boldsymbol{A} \in \mathbb{R}^{d \times r}$, $\boldsymbol{B} \in \mathbb{R}^{r \times k}$ for efficiency with the following empirical risk

$$\widetilde{L}(\boldsymbol{A}, \boldsymbol{B}) := \frac{1}{2N} \begin{cases} \left\| \widetilde{\boldsymbol{X}}(\boldsymbol{W}^\natural + \boldsymbol{A}\boldsymbol{B}) - \widetilde{\boldsymbol{Y}} \right\|_{\mathrm{F}}^2, & \text{linear,} \\ \left\| \sigma\left(\widetilde{\boldsymbol{X}}(\boldsymbol{W}^\natural + \boldsymbol{A}\boldsymbol{B})\right) - \widetilde{\boldsymbol{Y}} \right\|_{\mathrm{F}}^2, & \text{nonlinear} \end{cases} \quad (3)$$

which can be minimized using GD with stepsize $\eta > 0$

$$\begin{aligned} \boldsymbol{A}_{t+1} &= \boldsymbol{A}_t - \eta \nabla_{\boldsymbol{A}} \widetilde{L}(\boldsymbol{A}_t, \boldsymbol{B}_t), \\ \boldsymbol{B}_{t+1} &= \boldsymbol{B}_t - \eta \nabla_{\boldsymbol{B}} \widetilde{L}(\boldsymbol{A}_t, \boldsymbol{B}_t). \end{aligned} \quad (4)$$

Since the true rank $r^*$ of $\Delta$ is unknown in LoRA, our results will cover two cases: *over-ranked* ($r \geq r^*$) and *exact-ranked* ($r = r^*$).[1] Our results allow for large $d, k$ while $r, r^* = \Theta(1)$, which coincides with common practice.

**Optimization and Generalization:** We are interested in the error $\|\boldsymbol{A}_t \boldsymbol{B}_t - \Delta\|_{\mathrm{F}}^2$ under the LoRA training dynamics. Bounds on this error also imply generalization performance, because the generalization error for a new data $(\widetilde{\boldsymbol{x}}, \widetilde{\boldsymbol{y}})$ satisfies $\mathbb{E}_{\widetilde{\boldsymbol{x}}} \left\| \widetilde{\boldsymbol{y}} - \sigma(\boldsymbol{W}^\natural + \boldsymbol{A}_t \boldsymbol{B}_t)^\top \widetilde{\boldsymbol{x}} \right\|_2^2 \leq \|\boldsymbol{A}_t \boldsymbol{B}_t - \Delta\|_{\mathrm{F}}^2$ in the nonlinear setting, with equality in the linear setting.

## 3. Analysis of LoRA under Linear Model

In this section, we establish the alignment between LoRA and one gradient of full fine-tuning. This result guides us to design new strategies for speeding up practical LoRA-based algorithms, which achieve this alignment at initialization.

We formally define the negative gradient of full fine-tuning

---

[1] In the matrix sensing/completion literature, they are often called *over-* and *exact-parameterized*, respectively.

in Eq. (2) for the linear setting after the first step as

$$\boldsymbol{G}^{\natural} := -\nabla_{\boldsymbol{W}} L(\boldsymbol{W}^{\natural}) = \frac{1}{N}\widetilde{\boldsymbol{X}}^{\top}(\widetilde{\boldsymbol{Y}} - \widetilde{\boldsymbol{X}}\boldsymbol{W}^{\natural}). \quad (5)$$

Note that $\widetilde{\boldsymbol{X}}^{\top}\widetilde{\boldsymbol{X}}$ is a non-singular square matrix (Zeng & Lee, 2024, Lemma 6). Since left multiplication by a non-singular square matrix does not change the rank (Horn & Johnson, 2012, 0.4.6 (b)), we have $\mathrm{Rank}(\boldsymbol{G}^{\natural}) = \mathrm{Rank}(\Delta) = r^*$. Then, we denote the singular values of $\boldsymbol{G}^{\natural}$ by $\{\lambda_i(\boldsymbol{G}^{\natural})\}_{i=1}^{r^*}$ in non-increasing order.

### 3.1. Alignment under LoRA Initialization

We first present the results for the alignment of $\boldsymbol{B}_t$ by recalling the notations $\boldsymbol{V}_{r^*}(\cdot)$ and $\boldsymbol{V}_{r^*,\perp}(\cdot)$.

**Theorem 3.1** (Alignment between $\boldsymbol{G}^{\natural}$ and $\boldsymbol{B}_t$). *Under assumptions in Section 2.1 for the linear setting, consider the LoRA updates* (4) *with* (LoRA-init)*. We have*

$$\left\|\boldsymbol{V}_{r^*,\perp}^{\top}\left(\boldsymbol{G}^{\natural}\right)\boldsymbol{V}_{r^*}\left(\boldsymbol{B}_t\right)\right\|_{op} = 0, \quad \forall t \in \mathbb{N}_+.$$

One can see that, due to the zero initialization of $\boldsymbol{B}_0$ in (LoRA-init), after the first GD step, it holds that $\boldsymbol{B}_1 = \eta \boldsymbol{A}_0^{\top}\boldsymbol{G}^{\natural}$, which has rank $\leq r^*$ and lies in the right top-$r^*$ singular subspace of $\boldsymbol{G}^{\natural}$. The subsequent GD dynamics of $\boldsymbol{B}_t$ is always restricted to this invariant subspace.

Next we build the alignment for $\boldsymbol{A}_t$ with the notations $\boldsymbol{U}_{r^*}(\cdot)$, $\boldsymbol{U}_{r^*,\perp}(\cdot)$ and $\kappa^{\natural}$ as the condition number of $\boldsymbol{G}^{\natural}$.

**Theorem 3.2** (Alignment between $\boldsymbol{G}^{\natural}$ and $\boldsymbol{A}_t$. Simplified version of Theorem C.9). *For the $r \geq 2r^*$ case, under assumptions in Section 2.1 for the linear setting, we consider the LoRA updates* (4) *with* $[\boldsymbol{A}_0]_{ij} \sim \mathcal{N}(0, \alpha^2)$ *in* (LoRA-init)*. Then for any constant $\theta \in (0,1)$, by taking $\alpha = \mathcal{O}\left(\theta^{\frac{3}{2}\kappa^{\natural}}d^{-\frac{3}{4}\kappa^{\natural}-\frac{1}{2}}\|\boldsymbol{G}^{\natural}\|_{op}^{\frac{1}{2}}\right)$, and running gradient descent for $t^*$ steps with*

$$t^* \lesssim \frac{\ln\left(\frac{\sqrt{d}}{\theta}\right)}{\ln\left(1 + \eta\lambda_{r^*}\left(\boldsymbol{G}^{\natural}\right)\right)}, \quad (6)$$

*we achieve the following the alignment on the left singular subspace between $\boldsymbol{G}^{\natural}$ and $\boldsymbol{A}_{t^*}$ as below*

$$\left\|\boldsymbol{U}_{r^*,\perp}^{\top}(\boldsymbol{G}^{\natural})\boldsymbol{U}_{r^*}\left(\boldsymbol{A}_{t^*}\right)\right\|_{op} \lesssim \theta, \quad (7)$$

*with probability at least $1 - C_1\exp(-d) - C_2\exp(-r) - C_3\exp(-N)$ for some constants $C_1, C_2, C_3$.*

**Remark:** The result under the $r^* \leq r < 2r^*$ case is more complex and we defer this result to Theorem C.9. The choice of $\alpha$ in Theorem 3.2 shows that after $t^* = \Theta\left(\frac{\ln d}{\lambda_{r^*}(\boldsymbol{G}^{\natural})}\right)$ in Eq. (6), the alignment can be achieved. Our results can cover the standard He-initialization (He et al., 2015) if $\|\boldsymbol{G}^{\natural}\|_{op} \geq \Omega(d^{\frac{3}{4}\kappa^{\natural}})$.

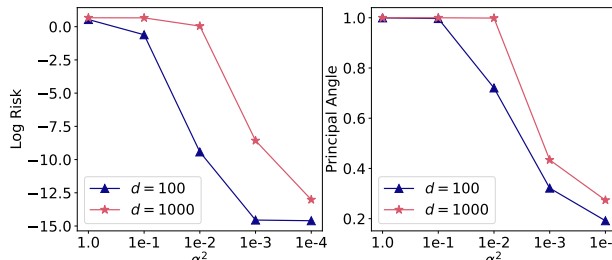

Figure 2: Under (LoRA-init), the risk and the alignment to full one-step GD of LoRA with different $\alpha^2$ and $d$, trained via GD on task (3). *Left*: the log risk under different initialization variance $\alpha^2$. The risk is defined as $\frac{1}{2}\|\boldsymbol{A}_t\boldsymbol{B}_t - \Delta\|_F^2$. *Right*: the best principal angle between the top-$r$ singular subspace of $\boldsymbol{G}^{\natural}$ and $\boldsymbol{A}_t$ during training. Smaller is closer. The principal angle is defined as $\min_t \|\boldsymbol{U}_{r^*,\perp}^{\top}(\boldsymbol{G}^{\natural})\boldsymbol{U}_{r^*}(\boldsymbol{A}_t)\|_{op}$. More experimental details can be found in Appendix G.1.

Requirement on $\|\boldsymbol{G}^{\natural}\|_{op}$ can be relaxed under smaller initialization, illustrated by Fig. 2.

The above two theorems characterize the alignment between $\boldsymbol{G}^{\natural}$ and $(\boldsymbol{A}_t, \boldsymbol{B}_t)$. Fig. 2 empirically validates Theorem 3.2 in two folds: *i)* Smaller initialization ($\alpha^2$ in the x-axis) encourages better alignment (evaluated by the principal angle), and then better generalization performance of fine-tuning (evaluated by the risk). But in practice smaller initialization would increase the training time for convergence, as a double-edge sword. *ii)* increasing $d$ leads to longer alignment time, illustrated by Eq. (6), and worse alignment performance, illustrated by the formulation of $\alpha$. Besides, we also verify this alignment in read-world applications by fine-tuning a T5 base model (Raffel et al., 2020) on MRPC using LoRA, as shown in Fig. 1(b). The mean principle angles within each layer class is computed in a similar way of Fig. 2. Our empirical results demonstrate that the principal angles decrease from around 1 because of Gaussian initialization to the value around $0.2 \sim 0.4$.

**Proof of Sketch:** Here we give a proof of sketch of Theorem 3.2. The dynamics (4) can be written as

$$\underbrace{\begin{bmatrix}\boldsymbol{A}_{t+1}\\\boldsymbol{B}_{t+1}^{\top}\end{bmatrix}}_{=:\boldsymbol{Z}_{t+1}} = \underbrace{\begin{bmatrix}\boldsymbol{I}_d & \eta\boldsymbol{G}^{\natural}\\\eta\boldsymbol{G}^{\natural\top} & \boldsymbol{I}_k\end{bmatrix}}_{=:\boldsymbol{H}}\underbrace{\begin{bmatrix}\boldsymbol{A}_t\\\boldsymbol{B}_t^{\top}\end{bmatrix}}_{=:\boldsymbol{Z}_t} + \text{nonlinear term},$$

where $\boldsymbol{H}$ is a time-independent matrix corresponding to the linear part of the dynamic $\boldsymbol{Z}_t^{\text{lin}} := \boldsymbol{H}^t\boldsymbol{Z}_0$. By Schur decomposition of $\boldsymbol{H}$ (see Lemma C.1), we can obtain the precise spectral dynamics of $\boldsymbol{Z}_t^{\text{lin}}$ and derive the alignment between $\boldsymbol{Z}_t^{\text{lin}}$ and $\boldsymbol{G}^{\natural}$, see Lemma C.5 for details. We prove that the nonlinear term is well controlled, i.e., $\|\boldsymbol{Z}_t - \boldsymbol{Z}_t^{\text{lin}}\|_{op} \leq \|\boldsymbol{A}_0\|_{op}, \forall t \leq t^*$, see Lemma C.6. Then

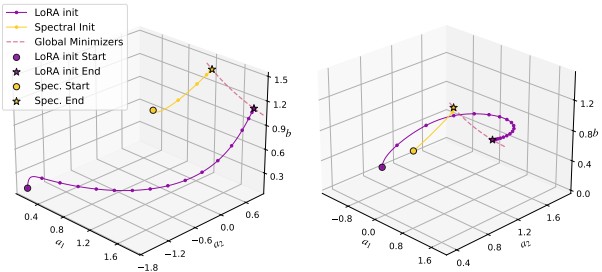

Figure 3: Comparison of the GD trajectories under (Spectral-init) and (LoRA-init) with two different starting points. See more experimental details in Appendix G.1.

the alignment between $\boldsymbol{Z}_t^{\texttt{lin}}$ and $\boldsymbol{G}^\natural$ can be successfully transferred to that of $\boldsymbol{Z}_t$, see Theorem C.9 for details.

We remark that previous work on matrix sensing (Stöger & Soltanolkotabi, 2021; Soltanolkotabi et al., 2023) via a symmetrization technique cannot be directly applied to our setting. Such symmetrization technique prevents the alignment results decoupling into two factorized matrices. We extend their technique to decouple the alignment for $\boldsymbol{A}_t$ and $\boldsymbol{B}_t$ individually via Schur decomposition of $\boldsymbol{H}$.

### 3.2. Spectral Initialization and Global Convergence

Theorem 3.2 has demonstrated the alignment on the rank-$r^*$ singular space of $\boldsymbol{G}^\natural$ and $(\boldsymbol{A}_t, \boldsymbol{B}_t)$. In other words, if we take the SVD of $\boldsymbol{G}^\natural$ and choose the certain singular subspace for initialization in (Spectral-init), we can directly achieve the alignment at this initialization and recover $\Delta$ to some extent, which is the main target of this work.

By the following standard concentration result for (sub)-Gaussian data: with probability at least $1 - 2C \exp(-\epsilon^2 N)$ for some constants $C > 0$, we have

$$\left\| \widehat{\boldsymbol{\Sigma}} - \boldsymbol{I}_d \right\|_{op} \leq \epsilon := \min \left\{ \frac{1}{2\kappa}, \frac{c}{\kappa^3} \right\} \leq \frac{1}{2}. \quad (8)$$

Recall $\kappa$ is the condition number of $\Delta$ and $\lambda_{r^*}^*$ is the $r^*$-th singular value of $\Delta$, we have the following result at the spectral initialization.

**Theorem 3.3.** *[One-step gradient can suffice] Under assumptions in Section 2.1 for the linear setting via* (Spectral-init)*, taking $\epsilon$ in Eq. (8), then with probability at least $1 - 2C \exp(-\epsilon^2 N)$ for constant $C > 0$, we have*

$$\|\boldsymbol{A}_0 \boldsymbol{B}_0 - \Delta\|_{op} \leq \epsilon \|\Delta\|_{op} \leq \frac{\lambda_{r^*}^*}{2}.$$

Theorem 3.3 demonstrates that, after one-step full gradient, i.e., using spectral initialization (Spectral-init), $\boldsymbol{A}_0 \boldsymbol{B}_0$ is able to recover $\Delta$ with small error. This is still true for

nonlinear models, see Lemma D.5 for details. Fig. 3 numerically validates that the starting points initialized by (Spectral-init) are consistently closer to the set of global minimizers, whereas those initialized by (LoRA-init) tend to be farther away across different random seeds.

Moreover, running gradient descent from points initialized by (Spectral-init) requires significantly fewer steps to reach a global minimizer, demonstrating the advantages of (Spectral-init). Due to page limit, we present the global convergence of linear models is deferred to Appendix C.2 and Appendix C.3, respectively. In Appendix C.2, we derive the linear convergence rate $\|\boldsymbol{A}_t \boldsymbol{B}_t - \Delta\|_F$. Since the convergence will be slow if the downstream feature shift $\Delta$ is ill-conditioned, i.e., $\kappa$ is large. This motivates us to add preconditioners, then the convergence rate will be independent of $\kappa$ accordingly, see Appendix C.3 for details.

## 4. Analysis of LoRA under Nonlinear Models

Now we focus on the nonlinear setting described in Section 2, where we consider the exact-rank case $r = r^*$ for delivery. We will demonstrate that $\|\boldsymbol{A}_0 \boldsymbol{B}_0 - \Delta\|_F$ is still small under the spectral initialization. Besides, the linear convergence rate of $\|\boldsymbol{A}_t \boldsymbol{B}_t - \Delta\|_F$ can still hold.

As an example, we demonstrate the equipment of precondition GD on $(\boldsymbol{A}_t, \boldsymbol{B}_t)$ for global convergence

$$\begin{aligned} \boldsymbol{A}_{t+1} &= \boldsymbol{A}_t - \eta \nabla_{\boldsymbol{A}} \widetilde{L} \left( \boldsymbol{A}_t, \boldsymbol{B}_t \right) \left( \boldsymbol{B}_t \boldsymbol{B}_t^\top \right)^{-1}, \\ \boldsymbol{B}_{t+1} &= \boldsymbol{B}_t - \eta \left( \boldsymbol{A}_t^\top \boldsymbol{A}_t \right)^{-1} \nabla_{\boldsymbol{B}} \widetilde{L} \left( \boldsymbol{A}_t, \boldsymbol{B}_t \right). \end{aligned} \quad (9)$$

Notice that here we use standard matrix inversion since we can prove that $\boldsymbol{A}_t$ and $\boldsymbol{B}_t$ stay non-singular across all $t \geq 0$. By denoting $\boldsymbol{W}_t := \boldsymbol{W}^\natural + \boldsymbol{A}_t \boldsymbol{B}_t$, we have the gradient

$$\nabla_{\boldsymbol{A}} \widetilde{L} \left( \boldsymbol{A}_t, \boldsymbol{B}_t \right) = -\boldsymbol{J}_{\boldsymbol{W}_t} \boldsymbol{B}_t^\top, \nabla_{\boldsymbol{B}} \widetilde{L} \left( \boldsymbol{A}_t, \boldsymbol{B}_t \right) = -\boldsymbol{A}_t^\top \boldsymbol{J}_{\boldsymbol{W}_t},$$

where we denote

$$\boldsymbol{J}_{\boldsymbol{W}_t} := \frac{1}{N} \widetilde{\boldsymbol{X}}^\top \left[ \sigma(\widetilde{\boldsymbol{X}} \widetilde{\boldsymbol{W}}^\natural) - \sigma(\widetilde{\boldsymbol{X}} \widetilde{\boldsymbol{W}}_t) \right] \odot \sigma'(\widetilde{\boldsymbol{X}} \widetilde{\boldsymbol{W}}_t).$$

To deliver the proof, apart from the above-mentioned assumptions in Section 2.1 for the the nonlinear setting, we also need the following assumption.

**Assumption 4.1.** We assume that **i)** $\frac{\|\widetilde{\boldsymbol{W}}^\natural\|_{op}}{\|\widetilde{\boldsymbol{w}}_m^\natural\|_2} = \mathcal{O}(1)$; **ii)** $\frac{\max\{\lambda_{r^*}^*, \|\Delta_m\|_{op}\}}{\|\widetilde{\boldsymbol{w}}_m^\natural\|_2} = \mathcal{O}\left(\frac{1}{\kappa r^*}\right)$ for $m \in [k]$.

**Remark:** The condition **i)** ensures the balance between different neurons within one layer for the downstream teacher model and the task diversity. The condition **ii)** ensures the signal of downstream feature shift is smaller than the pretrained ones approximately in the order $(\kappa r^*)^{-1}$ since the

signal of adapted weight is generally weaker than the pre-trained weight. Two conditions can be empirically observed in Appendix G.5.

Here we can show that, for the nonlinear model, LoRA training can achieve global linear convergence under (Spectral-init) via preconditioned GD in Eq. (9).

**Theorem 4.2** (Simplified version of Theorem D.10). *Under assumptions in Section 2.1 for the nonlinear setting and 4.1, with training conducted by Eq. (9) and initialization via (Spectral-init) with setting $\gamma = 2$, we take $\epsilon = \mathcal{O}\left(\frac{1}{r^* \kappa \sqrt{d}}\right)$ and $\rho \leq \frac{1}{20}$. Then choosing $\eta \in (c_\eta, 1)$ for a small constant $c_\eta > 0$, with probability at least $1 - 2Cdk \exp\left(-\epsilon^2 N\right)$ for a universal constant $C > 0$, we have*

$$\|\boldsymbol{A}_t \boldsymbol{B}_t - \Delta\|_{\mathrm{F}} \leq \left(1 - \frac{\eta}{4}\right)^t \rho \lambda_{r^*}^*, \forall t \geq 0. \quad (10)$$

**Remark:** We make three remarks here:
*i)* This theorem is based on $\|\boldsymbol{A}_0 \boldsymbol{B}_0 - \Delta\|_{\mathrm{F}} \leq \rho \lambda_{r^*}^*$ at initialization, see Lemma D.5 for details, which demonstrates that one-step full gradient can be sufficient.
*ii)* The convergence rate is independent of condition number $\kappa$ of downstream feature shift $\Delta$, demonstrating the benefits of adding preconditioners.

**Proof of Sketch** The complete proof can be found in Appendix D.2. We first compute the expectation of $\boldsymbol{J}_{\boldsymbol{W}_t}$ (see Lemma D.2) and decompose $\boldsymbol{J}_{\boldsymbol{W}_t}$ into $\frac{1}{2}\left(\boldsymbol{A}_t \boldsymbol{B}_t - \Delta\right) + \boldsymbol{\Xi}_t$, where $\boldsymbol{\Xi}_t$ is defined in Lemma D.6. The first term is the signal term which can dominate the preconditioned GD dynamics. The second term $\boldsymbol{\Xi}_t := T1 + T2$ consists of two parts (details see Lemma D.7): the first part $T1$ is the residual term from $\mathbb{E}_{\widetilde{\boldsymbol{x}}}[\boldsymbol{J}_{\boldsymbol{W}_t}]$ which vanishes due to pre-training signal dominance. For the second term $T2$, it comes from the concentration error of $\boldsymbol{J}_{\boldsymbol{W}_t}$, which can also controlled by large sample size $N$.

To handle $\|\boldsymbol{A}_t \boldsymbol{B}_t - \Delta\|_{\mathrm{F}}$, we explore its recursion relationship in Lemma D.6. The key part is to control $\left\|\left(\boldsymbol{I}_d - \boldsymbol{U}_{\boldsymbol{A}_t} \boldsymbol{U}_{\boldsymbol{A}_t}^\top\right) \Delta \left(\boldsymbol{I}_k - \boldsymbol{V}_{\boldsymbol{B}_t} \boldsymbol{V}_{\boldsymbol{B}_t}^\top\right)\right\|_{\mathrm{F}}$ (Lemma D.9) and higher order term (Lemma D.8).

## 5. Algorithm and Discussions

In this section, we present the *LoRA-One* algorithm and justify the optimality of our initialization over previous gradient alignment based algorithms for fine-tuning.

We present the implementations in Algorithm 1, which is driven by (Spectral-init) (shown in line 3-6). It coincides with the spirit of gradient alignment work, e.g., *LoRA-GA* (Wang et al., 2024), *LoRA-pro* (Wang et al., 2025), but the mechanisms for gradient alignment differ significantly, as suggested by our theory. First, *LoRA-GA* proposes the fol-

---

**Algorithm 1** LoRA-One for one specific layer

**Input:** Pre-trained weight $\boldsymbol{W}^\natural$, batched data $\{\mathcal{D}_m\}_{m=1}^T$, sampled batch data $\mathcal{B}$, LoRA rank $r$, LoRA alpha $\alpha$, loss function $L$, scaling parameter $s$

**Initialize:**
1: Compute $\nabla_{\boldsymbol{W}} L(\boldsymbol{W}^\natural)$ given $\mathcal{B}$
2: $\boldsymbol{U}, \boldsymbol{S}, \boldsymbol{V} \leftarrow \mathrm{SVD}\left(-\nabla_{\boldsymbol{W}} L(\boldsymbol{W}^\natural)\right)$
3: $\boldsymbol{S} \leftarrow \boldsymbol{S}/\boldsymbol{S}_{[0,0]}$ and $\gamma \leftarrow 1/s$
4: $\boldsymbol{A}_0 \leftarrow \sqrt{\gamma} \cdot \boldsymbol{U}_{[:,1:r]} \boldsymbol{S}_{[:r,:r]}^{1/2}$
5: $\boldsymbol{B}_0 \leftarrow \sqrt{\gamma} \cdot \boldsymbol{S}_{[:r,:r]}^{1/2} \boldsymbol{V}_{[:,1:r]}^\top$
6: Clear $\nabla_{\boldsymbol{W}} L(\boldsymbol{W}^\natural)$

**Train:**
7: **for** $t = 0, \ldots, T - 1$ **do**
8:     Compute gradients given $\mathcal{D}_{t+1}$:
    $\boldsymbol{G}_{t+1}^A \leftarrow \nabla_{\boldsymbol{A}} \widetilde{L}(\boldsymbol{A}_t, \boldsymbol{B}_t), \boldsymbol{G}_{t+1}^B \leftarrow \nabla_{\boldsymbol{B}} \widetilde{L}(\boldsymbol{A}_t, \boldsymbol{B}_t)$
9:     Update $\boldsymbol{A}_{t+1}, \boldsymbol{B}_{t+1} \leftarrow \mathrm{AdamW}\left(\boldsymbol{G}_{t+1}^A, \boldsymbol{G}_{t+1}^B\right)$
10: **end for**
**Return:** $\boldsymbol{W}^\natural + \frac{\alpha}{\sqrt{r}} \boldsymbol{A}_T \boldsymbol{B}_T$

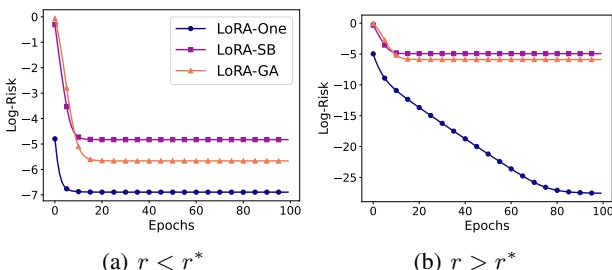

(a) $r < r^*$            (b) $r > r^*$

Figure 4: The log-risk curve under *LoRA-One*, *LoRA-SB*, and *LoRA-GA*, trained via GD on fine-tuning task (3) under: 1) under-ranked case $r < r^*$, 2) over-ranked case $r > r^*$.

lowing initialization strategy (omit the scaling parameters)

$$\boldsymbol{A}_0 \leftarrow -\left[\widetilde{\boldsymbol{U}}_{\boldsymbol{G}^\natural}\right]_{[:,1:r]}, \boldsymbol{B}_0 \leftarrow \left[\widetilde{\boldsymbol{V}}_{\boldsymbol{G}^\natural}\right]_{[:,r+1:2r]}^\top,$$

which aims to provide the best $2r$ approximation of $\boldsymbol{G}^\natural$. However, our theory indicates that $\boldsymbol{B}_t$ will align to the right-side rank-$r^*$ singular subspace of $\boldsymbol{G}^\natural$ under random initialization. However, *LoRA-GA* chooses the $(r + 1)$-th to $2r$-th singular values for $\boldsymbol{B}_0$, causing the iterates $\boldsymbol{B}_t$ to lie outside the desired subspace. As a result, the optimization may remain trapped in an undesirable subspace and fail to converge to an optimal solution, which can numerically verified by Fig. 4. Moreover, this approach subtracts the gradient for non-zero initialization and thus yields a biased estimate of $\Delta$, scaling with the model size; see further discussion in Appendix F.

Secondly, there is one concurrent work (Ponkshe et al., 2024), *LoRA-SB*, which uses the same singular subspace

for initialization but only updates $r \times r$ matrix $\boldsymbol{R}$ from the SVD of $\boldsymbol{G}^\natural$. Their intuition is to project the fine-tuning updates onto the singular subspace of first gradient step $\boldsymbol{G}^\natural$. However, the singular subspace of $\boldsymbol{G}^\natural$ normally still have distances with the ground truth, which will make their method hard to escape/rotate this subspace with limited degrees of freedom.

Our toy experiments based on (3) in Fig. 4 show that both *LoRA-GA* and *LoRA-SB* fail to find global minimizers even in a simple linear setting, whereas *LoRA-One* demonstrates significantly better generalization. More experimental details are presented in Appendix G.1.

# 6. Experiments

In this section, we conduct experiments to compare *LoRA-One* with typical LoRA based algorithms across multiple NLP benchmarks. In Section 6.1, we evaluate the ability of one-step gradient on real-world fine-tuning tasks to justify our theory on natural language understanding, i.e. Theorem 3.3 and Lemma D.5. In Section 6.2, we evaluate on mathematical reasoning, general knowledge, and code generation tasks, with more data and epochs for further evaluating math reasoning ability in Section 6.3. Furthermore, we compare the time and memory cost across different methods to illustrate our efficiency.

## 6.1. One-Step Full Gradient Could Suffice in Natural Language Understanding

We fine-tune T5 base model (Raffel et al., 2020) on a subset from GLUE (Wang et al., 2019) - MNLI, SST2, CoLA, QNLI, and MRPC. We evaluate the test performance by accuracy (%). We compare LoRA (Hu et al., 2022), *LoRA+* (Hayou et al., 2024), *P-LoRA* (Zhang & Pilanci, 2024), *PiSSA* (Meng et al., 2024), *LoRA-GA* (Wang et al., 2024), *LoRA-Pro* (Wang et al., 2025), and *LoRA-One* with rank 8. The hyperparameters are optimized for each method. More experimental details are presented in Appendix G.4.

Before experimental comparison, we first access the capacity of the one-step full gradient with its low-rank components on these real-world fine-tuning tasks. We approximate the one-step full-batch update $\boldsymbol{G}^\natural$ from full fine-tuning by a large sampled batch (2048) as $\boldsymbol{G}_B^\natural$ and a best rank-$r$ approximation of $\boldsymbol{G}^\natural$ for $r = 8$ using a smaller sampled batch (8) as $\mathcal{P}_r(\boldsymbol{G}_b^\natural)$. We optimize the learning rate, i.e. $\eta^*$, and update the base model by $\boldsymbol{W}^\natural - \eta^* \boldsymbol{G}_B^\natural$ and $\boldsymbol{W}^\natural - \eta^* \mathcal{P}_r(\boldsymbol{G}_b^\natural)$ for SST2, CoLA, QNLI, and MRPC, respectively.

The top rows of Table 2 shows that the test performance can be significantly improved over the pre-trained model by the one-step full gradient step with proper selection of stepsize. This improvement is still promising (even better) after taking the best rank-$r$ approximation with smaller

sampled batch, which is equivalent to (Spectral-init). We remark that low-rank update with small batch for CoLA and MRPC only costs *less than one second* but already matches the performance of LoRA which needs tens of seconds. Accordingly, **one-step full gradient can suffice for fine-tuning on small-scale datasets**, e.g., CoLA, MRPC.

Besides, Table 2 also shows that *LoRA-One* outperforms other LoRA-based methods on three tasks out of five and achieves the best in average. Significant gains appear on the smaller benchmarks, i.e. CoLA and MRPC. On large datasets such as MNLI and QNLI, *LoRA-Pro* performs better but is with more cost. This is because, *LoRA-pro* (Wang et al., 2025) approximates gradient from full fine-tuning at every training step while *LoRA-One* only conducts at the first step. *LoRA-pro* adds $10dr^2 + 6kr^2 + 155r^3/6$ more FLOPs than *LoRA-One* per pair of matrices for time cost. For memory cost, *LoRA-pro* on MetaMathQA100k with rank 8 costs 43.87 GB while our method only costs 21.7 GB for memory management.

## 6.2. Natural Language Generation

We fine-tune LLaMA 2-7B (Touvron et al., 2023) on: 1) 100K samples from MetaMathQA (Yu et al., 2024) and evaluate the accuracy based on two types of prompting: (a) direct prompting, (b) 8-shot Chain-of-Thought[2] (CoT) (Wei et al., 2022) prompting on GSM8K (Cobbe et al., 2021); 2) Alpaca (Taori et al., 2023) and evaluate on the MMLU (Hendrycks et al., 2021) benchmarks using direct prompting; 3) 100K samples from Code-Feedback (Zheng et al., 2024) and evaluate the PASS@1 on HumanEval (Chen et al., 2021a). We compare LoRA, *LoRA-GA*, and *LoRA-One* with rank 8. The learning rate and batch size are optimized for each method. More experimental details are presented in Appendix G.2.

Table 3 shows that *LoRA-One* consistently outperforms both vanilla LoRA and *LoRA-GA* across different tasks and prompting methods. *LoRA-One* achieves 60.44% accuracy under direct prompting—about 1.18 points higher than LoRA—and 55.88% in the few-shot CoT setting, a gain of roughly 2.52 points, indicating it not only strengthens the model's core problem-solving abilities but also its capacity for coherent, multi-step reasoning. On MMLU, *LoRA-One* also shows superior generalization on the MMLU benchmark (47.24% vs. 45.73% for LoRA), indicating improved knowledge retention across diverse domains. Finally, *LoRA-One* excels in code generation, it achieves a PASS@1 score of 28.66%, nearly 3 points higher than LoRA's 25.85% and also improving upon *LoRA-GA*, implying better adaptation to structured code synthesis tasks. Moreover, *LoRA-One* exhibits noticeably lower run-to-run variability compared to

---

[2]https://github.com/EleutherAI/lm-evaluation-harness

Table 2: Accuracy comparison on GLUE subset across typical LoRA based algorithms, as well as evaluation of the pre-training model, one-step gradient update and its low-rank approximation ($r = 8$, i.e., (Spectral-init)). Results are reported as accuracy (%) with standard deviations over 3 runs (best in **bold**). The results marked with ($*$) are sourced from Wang et al. (2024; 2025) under the same setting, and their hyper-parameter selection aligns with our search. The test accuracy on MNLI remains zero after one-step update thus not reported.

| Method | MNLI | SST-2 | CoLA | QNLI | MRPC | Avg. |
|---|---|---|---|---|---|---|
| Pre-train | | 89.79 | 59.03 | 49.28 | 63.48 | |
| One-step full gradient | | 90.94 | 69.13 | 70.35 | 68.38 | |
| $r = 8$ (low rank) | | 89.91 | 69.22 | 76.31 | 68.38 | |
| LoRA | $85.30_{\pm0.04}$ | $94.04_{\pm0.09}$ | $72.84_{\pm1.25}$ | $93.02_{\pm0.07}$ | $68.38_{\pm0.01}$ | 82.72 |
| LoRA+$^*$ | $85.81_{\pm0.09}$ | $93.85_{\pm0.24}$ | $77.53_{\pm0.20}$ | $93.14_{\pm0.03}$ | $74.43_{\pm1.39}$ | 84.95 |
| P-LoRA | $85.28_{\pm0.15}$ | $93.88_{\pm0.11}$ | $79.58_{\pm0.67}$ | $93.00_{\pm0.07}$ | $83.91_{\pm1.16}$ | 87.13 |
| PiSSA$^*$ | $85.75_{\pm0.07}$ | $94.07_{\pm0.06}$ | $74.27_{\pm0.39}$ | $93.15_{\pm0.14}$ | $76.31_{\pm0.51}$ | 84.71 |
| LoRA-GA$^*$ | $85.70_{\pm0.09}$ | $94.11_{\pm0.18}$ | $80.57_{\pm0.20}$ | $93.18_{\pm0.06}$ | $85.29_{\pm0.24}$ | 87.77 |
| LoRA-Pro$^*$ | $\mathbf{86.03}_{\pm0.19}$ | $94.19_{\pm0.13}$ | $81.94_{\pm0.24}$ | $\mathbf{93.42}_{\pm0.05}$ | $86.60_{\pm0.14}$ | 88.44 |
| LoRA-One | $85.89_{\pm0.08}$ | $\mathbf{94.53}_{\pm0.13}$ | $\mathbf{82.04}_{\pm0.22}$ | $93.37_{\pm0.02}$ | $\mathbf{87.83}_{\pm0.37}$ | **88.73** |

Table 3: Performance comparison across different methods on NLG benchmarks. Results are reported as mean with standard deviations over 5 runs (higher is better).

| ($r = 8$) | LoRA | LoRA-GA | LoRA-One |
|---|---|---|---|
| GSM8K-D | $59.26_{\pm0.99}$ | $56.44_{\pm1.15}$ | $\mathbf{60.44}_{\pm0.17}$ |
| GSM8K-CoT | $53.36_{\pm0.77}$ | $46.07_{\pm1.01}$ | $\mathbf{55.88}_{\pm0.44}$ |
| MMLU | $45.73_{\pm0.30}$ | $45.15_{\pm0.57}$ | $\mathbf{47.24}_{\pm0.20}$ |
| HumanEval | $25.85_{\pm1.75}$ | $26.95_{\pm1.30}$ | $\mathbf{28.66}_{\pm0.39}$ |

Table 4: The training time and memory cost of LoRA and *LoRA-One* from Section 6.2.

| | Training Time (Memory) | |
|---|---|---|
| ($r = 8$) | LoRA | LoRA-One |
| MetaMathQA | 6h20m (21.6GB) | 6h23m (21.7GB) |
| Alpaca | 3h22m (23.4GB) | 3h25m (23.4GB) |
| Code-Feedback | 6h24m (22.6GB) | 6h26m (22.9GB) |

the baselines, indicating better stability under (Spectral-init). Regarding the time and memory cost, *LoRA-One* takes almost the same cost as LoRA, as shown in Table 4 across all three datasets. This suggests that *LoRA-One* delivers its intended benefits—such as improved convergence stability or enhanced adaptability—without imposing any meaningful extra time or memory cost during fine-tuning.

### 6.3. Math Reasoning on Full Data and Multiple Epochs

Beyond Section 6.2, we further fine-tune LLaMA 2-7B on the complete MetaMathQA (395K) dataset for 4 epochs to access the maximum capacity of math reasoning. Here we

compare LoRA, *LoRA+*, *LoRA-GA*, and *LoRA-One* with rank 8. We evaluate the fine-tuned models on GSM8K with direct prompting. The learning rate and batch size are optimized for each method. More experimental details are presented in Appendix G.3.

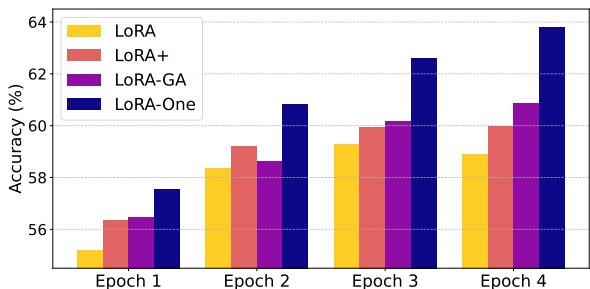

Figure 5: Accuracy comparison over epoch on GSM8K. Results are reported as mean over 2 runs (higher is better).

Fig. 5 shows that *LoRA-One* consistently leads other methods over epochs, suggesting that it scales more effectively with additional training and data. In contrast, *LoRA-GA* only shows marginal gains over LoRA and *LoRA+*.

## 7. Conclusion

This paper theoretically demonstrates how LoRA can be improved from our theoretical analysis in both linear and nonlinear models: the alignment between LoRA's gradient update $(\boldsymbol{A}_t, \boldsymbol{B}_t)$ and the singular subspace of $\boldsymbol{G}^\natural$, and adding preconditioners. Our theory derives the optimal initialization strategy for LoRA, clarifies some potential issues behind gradient alignment work, and bridge theory to practice with the promising performance of *LoRA-One*.

## Impact Statement

This paper provides theoretical understanding of low-rank adapters and proposes algorithm design for parameter-efficient fine-tuning. The target of this paper is to advance the field of Machine Learning. There might be some potential societal consequences of our work, none which we feel must be specifically highlighted here.

## Acknowledgment

Y. Zhang was supported by Warwick Chancellor's International Scholarship. F. Liu was supported by Royal Soceity KTP R1 241011 Kan Tong Po Visiting Fellowships. Y. Chen was supported in part by National Science Foundation grants CCF-2233152. We thank Yichen Wang for coding discussions, Zulip[3] for the project organization tool, and Sulis[4] for GPU computation resources.

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

# Contents

# A. Symbols and Notations

In this section, we provide a list of the symbols and notations used in a paper.

| Symbol | Dimension(s) | Definition |
|---|---|---|
| $\mathcal{N}(\boldsymbol{\mu}, \boldsymbol{\sigma})$ | - | Multivariate normal distribution with mean vector $\boldsymbol{\mu}$ and covariance matrix $\boldsymbol{\sigma}$ |
| $\mathcal{O}, o, \Omega, \Theta$ | - | Bachmann–Landau asymptotic notation |
| $\|\boldsymbol{w}\|_2$ | - | Euclidean norm of vector $\boldsymbol{w}$ |
| $\|\mathbf{M}\|_{op}$ | - | Operator norm of matrix $\mathbf{M}$ |
| $\|\mathbf{M}\|_{\mathrm{F}}$ | - | Frobenius norm of matrix $\mathbf{M}$ |
| $\langle \boldsymbol{u}, \boldsymbol{v} \rangle$ | - | Dot product of vectors $\boldsymbol{u}$ and $\boldsymbol{v}$ |
| $\mathbf{M} \odot \mathbf{N}$ | - | Hadamard product of matrix $\mathbf{M}$ and $\mathbf{N}$ |
| $\boldsymbol{W}^{\natural}$ | $\mathbb{R}^{d \times k}$ | Pre-trained weight matrix |
| $\Delta$ | $\mathbb{R}^{d \times k}$ | Downstream feature shift matrix |
| $\widetilde{\boldsymbol{W}}^{\natural}$ | $\mathbb{R}^{d \times k}$ | Downstream weight matrix $\widetilde{\boldsymbol{W}}^{\natural} = \boldsymbol{W}^{\natural} + \Delta$ |
| $\boldsymbol{G}^{\natural}$ | $\mathbb{R}^{d \times k}$ | The initial gradient matrix under full fine-tuning |
| $\boldsymbol{A}_t, \boldsymbol{B}_t$ | $\mathbb{R}^{d \times r}, \mathbb{R}^{r \times k}$ | Learnable low-rank adapters at step $t$ |
| $\boldsymbol{w}_i^{\natural}$ | $\mathbb{R}^{d}$ | $i^{\text{th}}$ column of pre-trained weight matrix $\boldsymbol{W}^{\natural}$ |
| $\widetilde{\boldsymbol{w}}_i^{\natural}$ | $\mathbb{R}^{d}$ | $i^{\text{th}}$ column of downstream weight matrix $\widetilde{\boldsymbol{W}}^{\natural}$ |
| $\boldsymbol{w}_{t,i}$ | $\mathbb{R}^{d}$ | $i^{\text{th}}$ column of adapted weight matrix $\left(\boldsymbol{W}^{\natural} + \boldsymbol{A}_t \boldsymbol{B}_t\right)$ at step $t$ |
| $\Delta_i$ | $\mathbb{R}^{d}$ | $i^{\text{th}}$ column of downstream feature matrix $\Delta$ |
| $[\boldsymbol{A}_t \boldsymbol{B}_t]_i$ | $\mathbb{R}^{d}$ | $i^{\text{th}}$ column of the product of adapters $\boldsymbol{A}_t \boldsymbol{B}_t$ |
| $\widetilde{\boldsymbol{X}}$ | $\mathbb{R}^{N \times d}$ | Downstream data matrix |
| $\widetilde{\boldsymbol{Y}}$ | $\mathbb{R}^{N \times d}$ | Downstream label matrix |
| $\widetilde{\boldsymbol{x}}_n$ | $\mathbb{R}^{d}$ | $n^{\text{th}}$ downstream data point |
| $\mathbf{M}^{-1}$ | - | Inverse of matrix $\mathbf{M}$ |
| $\mathbf{M}^{\dagger}$ | - | Pseudo-inverse of matrix $\mathbf{M}$ |
| $\lambda_i(\mathbf{M})$ | $\mathbb{R}$ | $i^{\text{th}}$ singular value of matrix $\mathbf{M}$ |
| $\lambda_i^*$ | $\mathbb{R}$ | $i^{\text{th}}$ singular value of downstream feature shift matrix $\Delta$ |
| $\kappa(\mathbf{M})$ | $\mathbb{R}$ | The condition number of matrix $\mathbf{M}$ |
| $\kappa$ | $\mathbb{R}$ | The condition number of $\Delta$: $\kappa = \lambda_{\max}^* / \lambda_{\min}^*$ |
| $\kappa^{\natural}$ | $\mathbb{R}$ | The condition number of $\mathbf{G}^{\natural}$: $\kappa^{\natural} = \lambda_{\max}\left(\mathbf{G}^{\natural}\right) / \lambda_{\min}\left(\mathbf{G}^{\natural}\right)$ |
| $\boldsymbol{U}_m(\mathbf{M})$ | - | The left singular subspace spanned by the $m$ largest singular values of the input matrix $\mathbf{M}$ |
| $\boldsymbol{U}_{m,\perp}(\mathbf{M})$ | - | The left singular subspace orthogonal to $\boldsymbol{U}_m(\mathbf{M})$ |
| $\boldsymbol{V}_m(\mathbf{M})$ | - | The right singular subspace spanned by the $m$ largest singular values of the input matrix $\mathbf{M}$ |
| $\boldsymbol{V}_{m,\perp}(\mathbf{M})$ | - | The right singular subspace orthogonal to $\boldsymbol{V}_m(\mathbf{M})$ |
| $\boldsymbol{U}_{\boldsymbol{A}}$ | - | The left singular matrix of the compact SVD of $\boldsymbol{A}$ |
| $\boldsymbol{U}_{\boldsymbol{A},\perp}$ | - | The corresponding orthogonal complement of $\boldsymbol{U}_{\boldsymbol{A}}$ |
| $\boldsymbol{V}_{\boldsymbol{A}}$ | - | The right singular matrix of the compact SVD of $\boldsymbol{A}$ |
| $\boldsymbol{V}_{\boldsymbol{A},\perp}$ | - | The corresponding orthogonal complement of $\boldsymbol{V}_{\boldsymbol{A}}$ |
| $\sigma(\cdot)$ | - | ReLU activation function |
| $\sigma'(\cdot)$ | - | The derivative of ReLU activation function |
| $\nabla_{\mathbf{w}} f(\mathbf{W})$ | - | The gradient matrix of function $f$ w.r.t. input matrix $\mathbf{W}$ |
| $\widetilde{L}(\boldsymbol{A}, \boldsymbol{B})$ | - | Loss function under LoRA fine-tuning |
| $L(\boldsymbol{W})$ | - | Loss function under full fine-tuning |
| $N$ | - | Number of downstream data |
| $d$ | - | Input dimension of the data |
| $k$ | - | Output dimension of the label |
| $\eta$ | - | Learning rates |
| $\alpha$ | - | In theory, random init. scale of $\boldsymbol{A}_0$. In algorithms, standard LoRA alpha. |

Table 5: Essential symbols and notations in this paper.

# B. Preconditioned LoRA-One

Motivated by our theory of preconditioning (see Theorem C.21 and Theorem 4.2), we propose a preconditioned variant of *LoRA-One*, termed *LoRA-One-P*. The algorithm is formally presented in Algorithm 2. *LoRA-One-P* employ the same initialization, i.e. (Spectral-init), with *LoRA-One*. For the optimizer, we use a preconditioned AdamW which introduced in (Zhang & Pilanci, 2024) instead of AdamW (Loshchilov & Hutter, 2017) in *LoRA-One*.

---

**Algorithm 2** LoRA-One-P for a specific layer

---

**Input:** Pre-trained weight $\boldsymbol{W}^\natural$, batched data $\{\mathcal{D}_m\}_{m=1}^T$, sampled batch data $\mathcal{B}$, LoRA rank $r$, LoRA alpha $\alpha$, loss function $L$, scaling parameter $s$, preconditioning parameter $\lambda$

**Initialize:**
1: Compute $\nabla_{\boldsymbol{W}} L(\boldsymbol{W}^\natural)$ given $\mathcal{B}$
2: $\boldsymbol{U}, \boldsymbol{S}, \boldsymbol{V} \leftarrow \text{SVD}\left(-\nabla_{\boldsymbol{W}} L(\boldsymbol{W}^\natural)\right)$
3: $\boldsymbol{S} \leftarrow \boldsymbol{S}/\boldsymbol{S}_{[0,0]}$
4: $\gamma \leftarrow 1/s$
5: $\boldsymbol{A}_0 \leftarrow \sqrt{\gamma} \cdot \boldsymbol{U}_{[:,1:r]} \boldsymbol{S}_{[:r,:r]}^{1/2}$
6: $\boldsymbol{B}_0 \leftarrow \sqrt{\gamma} \cdot \boldsymbol{S}_{[:r,:r]}^{1/2} \boldsymbol{V}_{[:,1:r]}^\top$
7: Clear $\nabla_{\boldsymbol{W}} L(\boldsymbol{W}^\natural)$

**Train:**
8: **for** $t = 0, \dots, T-1$ **do**
9:     Compute preconditioned gradients given $\mathcal{D}_{t+1}$:
   $\boldsymbol{G}_{t+1}^A \leftarrow \nabla_{\boldsymbol{A}} \widetilde{L}\left(\boldsymbol{A}_t, \boldsymbol{B}_t\right) \left(\boldsymbol{B}_t \boldsymbol{B}_t^\top + \lambda \boldsymbol{I}_r\right)^{-1}$,
   $\boldsymbol{G}_{t+1}^B \leftarrow \left(\boldsymbol{A}_t^\top \boldsymbol{A}_t + \lambda \boldsymbol{I}_r\right)^{-1} \nabla_{\boldsymbol{B}} \widetilde{L}\left(\boldsymbol{A}_t, \boldsymbol{B}_t\right)$
10:    Update $\boldsymbol{A}_{t+1}, \boldsymbol{B}_{t+1} \leftarrow \text{AdamW}\left(\boldsymbol{G}_{t+1}^A, \boldsymbol{G}_{t+1}^B\right)$
11: **end for**
**Return:** $\boldsymbol{W}^\natural + \frac{\alpha}{\sqrt{r}} \boldsymbol{A}_T \boldsymbol{B}_T$

---

Next, we conduct experiments to justify that *LoRA-One-P* is more robust to sub-optimal learning rate and can achieve faster convergence under suboptimal choice than *LoRA-One*. We fine-tune the T5 base model (Raffel et al., 2020) on SST-2 dataset from GLUE (Wang et al., 2019) for one epoch. To ensure a fair comparison, we fine-tune on the grid of learning rates $\{1 \times 10^{-3}, 2 \times 10^{-4}, 1 \times 10^{-4}, 5 \times 10^{-5}, 2 \times 10^{-5}, 1 \times 10^{-5}\}$ and fix other parameters to be the same as in Appendix G.4. Additionally, we set the preconditioning parameter $\lambda = 0$ in Algorithm 2 to be consistent with Section 4. For each choice of learning rate, we run 5 different seeds for both methods and record the test accuracy every 30 steps. Then, we compute the mean and 95%-confidence interval to construct the trajectory of test accuracy during fine-tuning. The results are shown in Fig. 6.

We can observe that, *LoRA-One-P* demonstrates clear advantages over *LoRA-One* across all options of tested learning rates, and these benefits manifest in two key aspects: **robustness to mis-specified learning rates** and **faster speed of convergence**.

Under overly large learning rate ($1 \times 10^{-3}$), *LoRA-One* hardly makes consistent progress and its wide confidence interval betrays unstable learning. *LoRA-One-P* makes stable improvement with a much tighter interval. This demonstrates that the preconditioners temper the volatility introduced by a too-aggressive rate, yielding both faster and more reliable gains.

When operating in the moderate range ($2 \times 10^{-4}$), both methods eventually attain strong performance and share a similar trending, indicating that *LoRA-One* achieves strong performance same as *LoRA-One-P* when the learning rate is well-specified.

At the other extreme, with the learning rates chosen too small ($1 \times 10^{-4}$ to $1 \times 10^{-5}$), the training process of *LoRA-One* gradually appears to be slow and nearly stalls. In contrast, *LoRA-One-P* not only converges more rapidly but also maintains stable, high-quality performance even when the learning rate departs substantially from its optimal setting. This **"rescue effect"** signals that *LoRA-One-P* can tolerate sub-optimal even under-scaled updates. Its expands "safe zone" for hyperparameter tuning and reduced variance across random seeds which makes it a robust and efficient choice for real-world tasks.

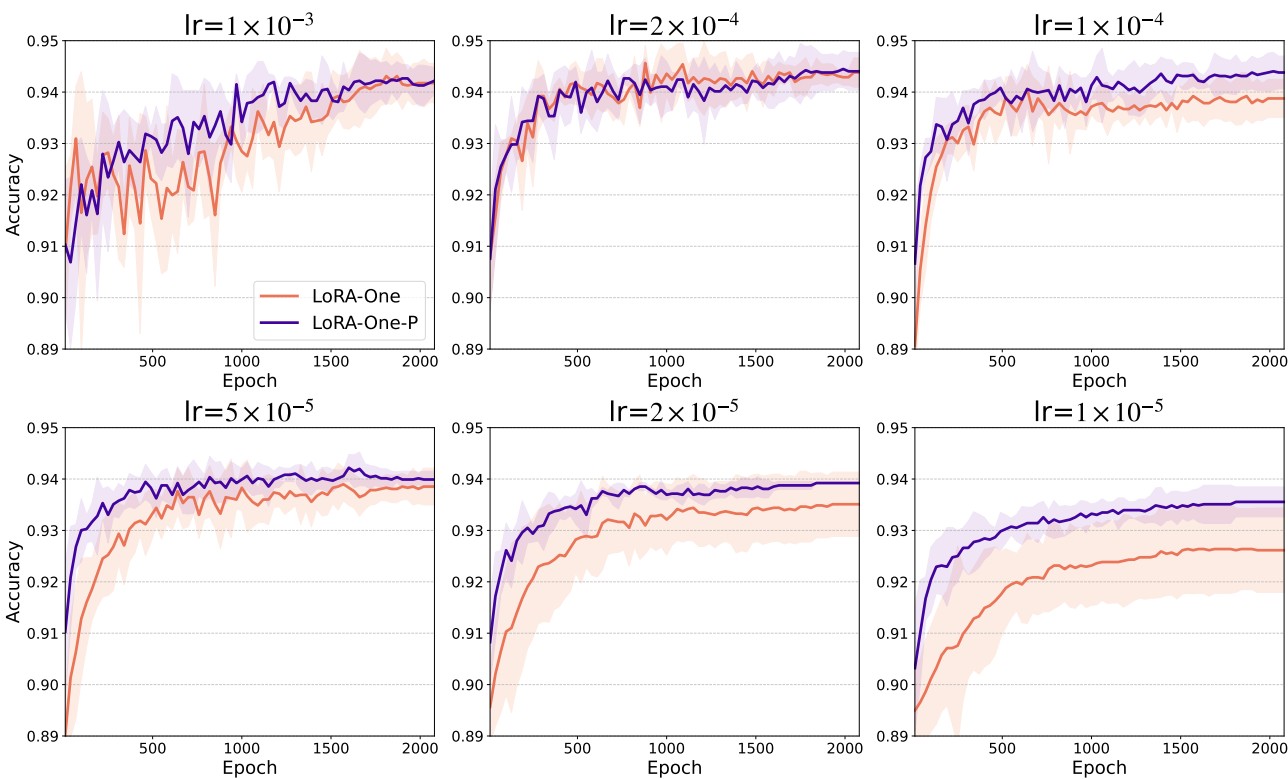

Figure 6: The trajectory of test accuracy during fine-tuning for *LoRA-One* and *LoRA-One-P* under different learning rates.

## C. Proofs for Linear Model

In Appendix C.1, we deliver the proofs for alignment in Section 3.1. In Appendix C.2, we present the proofs for the main results in Section 3.2 under spectral initialization. In Appendix C.3, we give the proofs for precondition GD.

### C.1. Proofs for LoRA under Random Initialization

Let $\widetilde{X}$ be the fine-tuned data with $\widetilde{X} \in \mathbb{R}^{N \times d}$ and the multi-output $\widetilde{Y} \in \mathbb{R}^{N \times k}$. For simplicity, we define the initial residual error $\widetilde{Y}_\Delta := \widetilde{Y} - \widetilde{X} W^\natural = \widetilde{X}\Delta$. Then, denote the negative gradient of Full Fine-tuning after the first step as

$$G^\natural = -\nabla_W L(W^\natural) = -\frac{1}{N}\widetilde{X}^\top(\widetilde{X}W^\natural - \widetilde{Y}) = \frac{1}{N}\widetilde{X}^\top\widetilde{Y}_\Delta \in \mathbb{R}^{d \times k}\,.$$

Recall the gradient update for LoRA

$$A_{t+1} = A_t - \frac{\eta}{N}\widetilde{X}^\top\left(\widetilde{X}(W^\natural + A_t B_t) - \widetilde{Y}\right)B_t^\top\,,$$

$$B_{t+1} = B_t - \frac{\eta}{N}A_t^\top\widetilde{X}^\top\left(\widetilde{X}(W^\natural + A_t B_t) - \widetilde{Y}\right)\,,$$

we rewrite it in a compact form

$$
\begin{aligned}
\begin{bmatrix} A_{t+1} \\ B_{t+1}^\top \end{bmatrix} &= \begin{bmatrix} A_t \\ B_t^\top \end{bmatrix} + \underbrace{\begin{bmatrix} \mathbf{0} & \eta G^\natural \\ \eta G^{\natural\top} & \mathbf{0} \end{bmatrix}}_{:=H}\begin{bmatrix} A_t \\ B_t^\top \end{bmatrix} - \frac{\eta}{N}\underbrace{\begin{bmatrix} \mathbf{0} & \widetilde{X}^\top\widetilde{X}A_t B_t \\ B_t^\top A_t^\top\widetilde{X}^\top\widetilde{X} & \mathbf{0} \end{bmatrix}}_{}\begin{bmatrix} A_t \\ B_t^\top \end{bmatrix} \\
&= \underbrace{\begin{bmatrix} I_d & \eta G^\natural \\ \eta G^{\natural\top} & I_k \end{bmatrix}}_{:=H}\begin{bmatrix} A_t \\ B_t^\top \end{bmatrix} - \frac{\eta}{N}\underbrace{\begin{bmatrix} \mathbf{0} & \widetilde{X}^\top\widetilde{X}A_t B_t \\ B_t^\top A_t^\top\widetilde{X}^\top\widetilde{X} & \mathbf{0} \end{bmatrix}\begin{bmatrix} A_t \\ B_t^\top \end{bmatrix}}_{:=\widehat{E}_{t+1}}\,.
\end{aligned}
\tag{11}
$$

By defining a stack iterate

$$\boldsymbol{Z}_t := \begin{bmatrix} \boldsymbol{A}_t \\ \boldsymbol{B}_t^\top \end{bmatrix}, \quad \text{and} \quad \boldsymbol{Z}_0 := \begin{bmatrix} \boldsymbol{A}_0 \\ \boldsymbol{0} \end{bmatrix} \in \mathbb{R}^{(d+k)\times r}, \tag{12}$$

we can formulate Eq. (11) as a compact form of a nonlinear dynamical system

$$\boldsymbol{Z}_{t+1} = \boldsymbol{H}\boldsymbol{Z}_t - \widehat{\boldsymbol{E}}_{t+1}, \tag{13}$$

where $\boldsymbol{H}$ is a time-independent matrix corresponding to the linear part, and $\widehat{\boldsymbol{E}}_{t+1}$ corresponds to the nonlinear part.

### C.1.1. SVD AND SCHUR DECOMPOSITION

We recall the complete SVD of $\Delta \in \mathbb{R}^{d\times k}$

$$\Delta = \widetilde{\boldsymbol{U}}\widetilde{\boldsymbol{S}}\widetilde{\boldsymbol{V}}^\top = \begin{bmatrix} \boldsymbol{U} & \boldsymbol{U}_\perp \end{bmatrix} \begin{bmatrix} \boldsymbol{S}^* & \boldsymbol{0}_{r^*\times(k-r^*)} \\ \boldsymbol{0}_{(d-r^*)\times r^*} & \boldsymbol{0}_{(d-r^*)\times(k-r^*)} \end{bmatrix} \begin{bmatrix} \boldsymbol{V}^\top \\ \boldsymbol{V}_\perp^\top \end{bmatrix}, \quad \text{where } \boldsymbol{S}^* = \mathrm{Diag}\left(\lambda_1^*, \cdots, \lambda_{r^*}^*\right).$$

Similarly, we recall the complete SVD of $\boldsymbol{G}^\natural$ as $\boldsymbol{G}^\natural = \widetilde{\boldsymbol{U}}_{\boldsymbol{G}^\natural}\widetilde{\boldsymbol{S}}_{\boldsymbol{G}^\natural}\widetilde{\boldsymbol{V}}_{\boldsymbol{G}^\natural}^\top$.

We derive the Schur decomposition of $\boldsymbol{H}$ under the special case $d = k$ in Lemma C.1 and then extend to $d \neq k$ in Lemma C.3 via zero padding on SVD in Lemma C.2.

**Lemma C.1** (Schur Decomposition of $\boldsymbol{H}$ under $d = k$). *Under assumptions in Section 2.1 for the linear setting, given $\boldsymbol{G}^\natural \in \mathbb{R}^{d\times k}$ in Eq. (5) and its complete SVD $\widetilde{\boldsymbol{U}}_{\boldsymbol{G}^\natural}\widetilde{\boldsymbol{S}}_{\boldsymbol{G}^\natural}\widetilde{\boldsymbol{V}}_{\boldsymbol{G}^\natural}^\top$, if $d = k$, then the block matrix $\boldsymbol{H}$ admits the following Schur decomposition*

$$\boldsymbol{H} = \begin{bmatrix} \boldsymbol{I}_d & \eta\boldsymbol{G}^\natural \\ \eta\left(\boldsymbol{G}^\natural\right)^\top & \boldsymbol{I}_d \end{bmatrix} = \mathbf{C}\mathbf{T}\mathbf{C}^\top,$$

*where $\mathbf{C}$ is an orthogonal matrix and $\mathbf{T}$ is a block upper triangular matrix*

$$\mathbf{C} = \frac{1}{\sqrt{2}}\begin{bmatrix} \widetilde{\boldsymbol{U}}_{\boldsymbol{G}^\natural} & -\widetilde{\boldsymbol{U}}_{\boldsymbol{G}^\natural} \\ \widetilde{\boldsymbol{V}}_{\boldsymbol{G}^\natural} & \widetilde{\boldsymbol{V}}_{\boldsymbol{G}^\natural} \end{bmatrix}, \quad \text{and} \quad \mathbf{T} = \begin{bmatrix} \boldsymbol{I}_d + \eta\widetilde{\boldsymbol{S}}_{\boldsymbol{G}^\natural} & \boldsymbol{0} \\ \boldsymbol{0} & \boldsymbol{I}_d - \eta\widetilde{\boldsymbol{S}}_{\boldsymbol{G}^\natural} \end{bmatrix}.$$

*Proof.* We prove by verifying the claim. Starting with

$$\begin{bmatrix} \widetilde{\boldsymbol{U}}_{\boldsymbol{G}^\natural} & -\widetilde{\boldsymbol{U}}_{\boldsymbol{G}^\natural} \\ \widetilde{\boldsymbol{V}}_{\boldsymbol{G}^\natural} & \widetilde{\boldsymbol{V}}_{\boldsymbol{G}^\natural} \end{bmatrix}\begin{bmatrix} \boldsymbol{I}_d + \eta\widetilde{\boldsymbol{S}}_{\boldsymbol{G}^\natural} & \boldsymbol{0} \\ \boldsymbol{0} & \boldsymbol{I}_d - \eta\widetilde{\boldsymbol{S}}_{\boldsymbol{G}^\natural} \end{bmatrix}$$

$$= \begin{bmatrix} \widetilde{\boldsymbol{U}}_{\boldsymbol{G}^\natural} + \eta\widetilde{\boldsymbol{U}}_{\boldsymbol{G}^\natural}\widetilde{\boldsymbol{S}}_{\boldsymbol{G}^\natural} & \eta\widetilde{\boldsymbol{U}}_{\boldsymbol{G}^\natural}\widetilde{\boldsymbol{S}}_{\boldsymbol{G}^\natural} - \widetilde{\boldsymbol{U}}_{\boldsymbol{G}^\natural} \\ \widetilde{\boldsymbol{V}}_{\boldsymbol{G}^\natural} + \eta\widetilde{\boldsymbol{V}}_{\boldsymbol{G}^\natural}\widetilde{\boldsymbol{S}}_{\boldsymbol{G}^\natural} & \widetilde{\boldsymbol{V}}_{\boldsymbol{G}^\natural} - \eta\widetilde{\boldsymbol{V}}_{\boldsymbol{G}^\natural}\widetilde{\boldsymbol{S}}_{\boldsymbol{G}^\natural} \end{bmatrix} =: \boldsymbol{\Xi},$$

then we can verify that

$$\frac{1}{2} \times \boldsymbol{\Xi} \times \begin{bmatrix} \widetilde{\boldsymbol{U}}_{\boldsymbol{G}^\natural}^\top & \widetilde{\boldsymbol{V}}_{\boldsymbol{G}^\natural}^\top \\ -\widetilde{\boldsymbol{U}}_{\boldsymbol{G}^\natural}^\top & \widetilde{\boldsymbol{V}}_{\boldsymbol{G}^\natural}^\top \end{bmatrix} = \begin{bmatrix} \boldsymbol{I}_d & \eta\widetilde{\boldsymbol{U}}_{\boldsymbol{G}^\natural}\widetilde{\boldsymbol{S}}_{\boldsymbol{G}^\natural}\widetilde{\boldsymbol{V}}_{\boldsymbol{G}^\natural}^\top \\ \eta\widetilde{\boldsymbol{V}}_{\boldsymbol{G}^\natural}\widetilde{\boldsymbol{S}}_{\boldsymbol{G}^\natural}\widetilde{\boldsymbol{U}}_{\boldsymbol{G}^\natural}^\top & \boldsymbol{I}_d \end{bmatrix} = \boldsymbol{H}.$$

Accordingly, we conclude the result. $\square$

Next, we consider the case of $d \neq k$.

**Case 1** ($d > k$)**:** by zero padding, $\boldsymbol{G}^\natural$ and related matrices are given by

$$\underline{\boldsymbol{G}}^\natural = \begin{bmatrix} \boldsymbol{G}^\natural & \boldsymbol{0}_{d\times(d-k)} \end{bmatrix}, \quad \underline{\boldsymbol{H}} = \begin{bmatrix} \boldsymbol{I}_d & \eta\underline{\boldsymbol{G}}^\natural \\ \eta\left(\underline{\boldsymbol{G}}^\natural\right)^\top & \boldsymbol{I}_d \end{bmatrix},$$

and for any $t \geq 0$, we have the following related matrices

$$\underline{\boldsymbol{B}}_t = \begin{bmatrix} \boldsymbol{B}_t & \boldsymbol{0}_{r\times(d-k)} \end{bmatrix}, \quad \underline{\boldsymbol{Z}}_t = \begin{bmatrix} \boldsymbol{A}_t \\ (\underline{\boldsymbol{B}}_t)^\top \end{bmatrix}, \quad \underline{\boldsymbol{Z}}_t^{\mathrm{lin}} = \begin{bmatrix} \boldsymbol{A}_t^{\mathrm{lin}} \\ (\underline{\boldsymbol{B}}_t^{\mathrm{lin}})^\top \end{bmatrix} = \underline{\boldsymbol{H}}^t\underline{\boldsymbol{Z}}_0.$$

**Case 2** ($d < k$): Similarly, by zero padding, we define

$$\underline{\boldsymbol{G}}^{\natural} = \begin{bmatrix} \boldsymbol{G}^{\natural} \\ \boldsymbol{0}_{(k-d) \times k} \end{bmatrix}, \quad \underline{\boldsymbol{H}} = \begin{bmatrix} \boldsymbol{I}_k & \eta \underline{\boldsymbol{G}}^{\natural} \\ \eta \left( \underline{\boldsymbol{G}}^{\natural} \right)^{\top} & \boldsymbol{I}_k \end{bmatrix},$$

and $\forall\, t \geq 0$, we define

$$\underline{\boldsymbol{A}}_t = \begin{bmatrix} \boldsymbol{A}_t \\ \boldsymbol{0}_{(k-d) \times r} \end{bmatrix}, \quad \underline{\boldsymbol{Z}}_t = \begin{bmatrix} \underline{\boldsymbol{A}}_t \\ (\boldsymbol{B}_t)^{\top} \end{bmatrix}, \quad \underline{\boldsymbol{Z}}_t^{\mathrm{lin}} = \begin{bmatrix} \underline{\boldsymbol{A}}_t^{\mathrm{lin}} \\ (\boldsymbol{B}_t^{\mathrm{lin}})^{\top} \end{bmatrix} = \underline{\boldsymbol{H}}^t \underline{\boldsymbol{Z}}_0.$$

Then we have the following lemma on the SVD of $\underline{\boldsymbol{G}}^{\natural}$.

**Lemma C.2.** *If $d > k$, then we have the following SVD of $\underline{\boldsymbol{G}}^{\natural}$*

$$\underline{\boldsymbol{G}}^{\natural} = \widetilde{\boldsymbol{U}}_{\boldsymbol{G}^{\natural}} \underline{\widetilde{\boldsymbol{S}}}_{\boldsymbol{G}^{\natural}} \underline{\widetilde{\boldsymbol{V}}}_{\boldsymbol{G}^{\natural}}^{\top},$$

*where*

$$\underline{\widetilde{\boldsymbol{V}}}_{\boldsymbol{G}^{\natural}} = \begin{bmatrix} \widetilde{\boldsymbol{V}}_{\boldsymbol{G}^{\natural}} & \boldsymbol{0}_{k \times (d-k)} \\ \boldsymbol{0}_{(d-k) \times k} & \boldsymbol{I}_{(d-k)} \end{bmatrix}, \quad \text{and} \quad \underline{\widetilde{\boldsymbol{S}}}_{\boldsymbol{G}^{\natural}} = \begin{bmatrix} \widetilde{\boldsymbol{S}}_{\boldsymbol{G}^{\natural}} & \boldsymbol{0}_{d \times (d-k)} \end{bmatrix}.$$

*If $d < k$, then we have the following SVD of $\underline{\boldsymbol{G}}^{\natural}$*

$$\underline{\boldsymbol{G}}^{\natural} = \underline{\widetilde{\boldsymbol{U}}}_{\boldsymbol{G}^{\natural}} \underline{\widetilde{\boldsymbol{S}}}_{\boldsymbol{G}^{\natural}} \widetilde{\boldsymbol{V}}_{\boldsymbol{G}^{\natural}}^{\top},$$

*where*

$$\underline{\widetilde{\boldsymbol{U}}}_{\boldsymbol{G}^{\natural}} = \begin{bmatrix} \widetilde{\boldsymbol{U}}_{\boldsymbol{G}^{\natural}} & \boldsymbol{0}_{k \times (k-d)} \\ \boldsymbol{0}_{(k-d) \times k} & \boldsymbol{I}_{(k-d)} \end{bmatrix}, \quad \text{and} \quad \underline{\widetilde{\boldsymbol{S}}}_{\boldsymbol{G}^{\natural}} = \begin{bmatrix} \widetilde{\boldsymbol{S}}_{\boldsymbol{G}^{\natural}} \\ \boldsymbol{0}_{(k-d) \times k} \end{bmatrix}.$$

*Proof.* The block construction does not affect the original part of the SVD. It only appends zeros to the singular values and grows the corresponding orthonormal bases as partial identity matrices appropriately. $\square$

Now we can apply Lemma C.2 for Lemma C.1 to extend to $d \neq k$ via the following lemma. The proof is direct and we omit it here.

**Lemma C.3** (Schur decomposition of $\boldsymbol{H}$ under $d \neq k$). *Given the defined block matrix $\underline{\boldsymbol{H}} \in \mathbb{R}^{2s \times 2s}$ with $s := \max\{d, k\}$, we have the following decomposition*

$$\underline{\boldsymbol{H}} = \mathbf{C} \mathbf{T} \mathbf{C}^{\top},$$

*If $d > k$,*

$$\mathbf{C} = \frac{1}{\sqrt{2}} \begin{bmatrix} \underline{\widetilde{\boldsymbol{U}}}_{\boldsymbol{G}^{\natural}} & -\underline{\widetilde{\boldsymbol{U}}}_{\boldsymbol{G}^{\natural}} \\ \underline{\widetilde{\boldsymbol{V}}}_{\boldsymbol{G}^{\natural}} & \underline{\widetilde{\boldsymbol{V}}}_{\boldsymbol{G}^{\natural}} \end{bmatrix}, \quad \mathbf{T} = \begin{bmatrix} \boldsymbol{I}_d + \eta \underline{\widetilde{\boldsymbol{S}}}_{\boldsymbol{G}^{\natural}} & \boldsymbol{0} \\ \boldsymbol{0} & \boldsymbol{I}_d - \eta \underline{\widetilde{\boldsymbol{S}}}_{\boldsymbol{G}^{\natural}} \end{bmatrix}.$$

*If $d < k$,*

$$\mathbf{C} = \frac{1}{\sqrt{2}} \begin{bmatrix} \underline{\widetilde{\boldsymbol{U}}}_{\boldsymbol{G}^{\natural}} & -\underline{\widetilde{\boldsymbol{U}}}_{\boldsymbol{G}^{\natural}} \\ \underline{\widetilde{\boldsymbol{V}}}_{\boldsymbol{G}^{\natural}} & \underline{\widetilde{\boldsymbol{V}}}_{\boldsymbol{G}^{\natural}} \end{bmatrix}, \quad \mathbf{T} = \begin{bmatrix} \boldsymbol{I}_k + \eta \underline{\widetilde{\boldsymbol{S}}}_{\boldsymbol{G}^{\natural}} & \boldsymbol{0} \\ \boldsymbol{0} & \boldsymbol{I}_k - \eta \underline{\widetilde{\boldsymbol{S}}}_{\boldsymbol{G}^{\natural}} \end{bmatrix}.$$

C.1.2. DYNAMICS OF LINEAR APPROXIMATION

The target of our proof is to demonstrate that $\widehat{\boldsymbol{E}}_{t+1}$ does not effect the dynamics too much such that the dynamics of $\boldsymbol{Z}_t$ is close to the following pseudo iterate

$$\boldsymbol{Z}_t^{\mathrm{lin}} := \boldsymbol{H}^t \boldsymbol{Z}_0 =: \begin{bmatrix} \boldsymbol{A}_t^{\mathrm{lin}} \\ (\boldsymbol{B}_t^{\mathrm{lin}})^{\top} \end{bmatrix}. \tag{14}$$

The updates of the pseudo iterate follow the trajectory of Oja's Power Method (Oja, 1982). Therefore, we aim to prove that the error between the actual iterate $\boldsymbol{Z}_t$ and the pseudo iterate $\boldsymbol{Z}_t^{\mathrm{lin}}$ is sufficiently small, which is equivalent to that the actual iterate $\boldsymbol{Z}_t$ performs a power iteration during the early steps. First, we obtain the difference between $\boldsymbol{Z}_t$ and $\boldsymbol{Z}_t^{\mathrm{lin}}$ by the following lemma.

**Lemma C.4** (Formulation of $\boldsymbol{E}_t$). *Under assumptions in Section 2.1 for the linear setting, given the nonlinear dynamical system* (13) *and its linear part* (14)*, their difference admits*

$$\boldsymbol{E}_t := \boldsymbol{Z}_t - \boldsymbol{Z}_t^{\text{lin}} = -\sum_{i=1}^{t} \boldsymbol{H}^{t-i}\widehat{\boldsymbol{E}}_i, \quad \forall t \in \mathbb{N}^+, \tag{15}$$

*where $\widehat{\boldsymbol{E}}_i$ corresponds to the nonlinear part in Eq.* (11)*.*

*Proof of Lemma C.4.* We prove it by induction. Recall the formulation of the nonlinear dynamical system $\boldsymbol{Z}_{t+1} = \boldsymbol{H}\boldsymbol{Z}_t - \widehat{\boldsymbol{E}}_{t+1}$, we start with the base case $t = 1$ such that

$$\boldsymbol{Z}_1 = \boldsymbol{H}\boldsymbol{Z}_0 - \widehat{\boldsymbol{E}}_1 = \boldsymbol{Z}_1^{\text{lin}} - \widehat{\boldsymbol{E}}_1,$$

which proves the claim. Next, we assume Eq. (15) holds for $t \geq 2$, then for $t + 1$, we have

$$\begin{aligned}
\boldsymbol{Z}_{t+1} &= \boldsymbol{H}\boldsymbol{Z}_t - \widehat{\boldsymbol{E}}_{t+1} \\
&= \boldsymbol{H}\left(\boldsymbol{Z}_t^{\text{lin}} - \sum_{i=1}^{t}\boldsymbol{H}^{t-i}\widehat{\boldsymbol{E}}_i\right) - \widehat{\boldsymbol{E}}_{t+1} \\
&= \boldsymbol{Z}_{t+1}^{\text{lin}} - \sum_{i=1}^{t}\boldsymbol{H}^{t+1-i}\widehat{\boldsymbol{E}}_i - \widehat{\boldsymbol{E}}_{t+1} \\
&= \boldsymbol{Z}_{t+1}^{\text{lin}} - \sum_{i=1}^{t+1}\boldsymbol{H}^{t+1-i}\widehat{\boldsymbol{E}}_i,
\end{aligned}$$

which proves the claim. $\square$

If $\|\boldsymbol{E}_t\|_{op}$ is sufficiently small within a certain period, e.g., $t \leq T$, then we could approximate the early dynamics by

$$\boldsymbol{Z}_{t+1} := \begin{bmatrix} \boldsymbol{A}_{t+1} \\ \boldsymbol{B}_{t+1}^\top \end{bmatrix} \approx \boldsymbol{Z}_t^{\text{lin}} := \begin{bmatrix} \boldsymbol{A}_{t+1}^{\text{lin}} \\ (\boldsymbol{B}_{t+1}^{\text{lin}})^\top \end{bmatrix} = \begin{bmatrix} \boldsymbol{A}_t^{\text{lin}} \\ (\boldsymbol{B}_t^{\text{lin}})^\top \end{bmatrix} + \begin{bmatrix} \boldsymbol{0} & \eta\boldsymbol{G}^\natural \\ \eta\boldsymbol{G}^{\natural\top} & \boldsymbol{0} \end{bmatrix}\begin{bmatrix} \boldsymbol{A}_t^{\text{lin}} \\ (\boldsymbol{B}_t^{\text{lin}})^\top \end{bmatrix},$$

via

$$\left\| \begin{bmatrix} \boldsymbol{A}_t \\ \boldsymbol{B}_t^\top \end{bmatrix} - \begin{bmatrix} \boldsymbol{A}_t^{\text{lin}} \\ (\boldsymbol{B}_t^{\text{lin}})^\top \end{bmatrix} \right\|_{op} \leq \|\boldsymbol{E}_t\|_{op}.$$

In this subsection, we will bound $\|\boldsymbol{E}_t\|_{op}$ to show that it is actually small up to the initialization. To prove it, we first conduct the dynamical analysis of $\boldsymbol{Z}_t^{\text{lin}}$ via the structure of $\boldsymbol{H}$.

**Part I: Dynamics of $\boldsymbol{Z}_t^{\text{lin}}$**

With the algebra fact above, we can derive the precise spectral dynamics of $\boldsymbol{Z}_t^{\text{lin}}$, i.e., $\boldsymbol{A}_t^{\text{lin}}$ and $\boldsymbol{B}_t^{\text{lin}}$ separately.

**Lemma C.5.** *Under assumptions in Section 2.1 for the linear setting, given the pseudo iterate* (14) *on $\boldsymbol{Z}_t^{\text{lin}}$, where two components $\boldsymbol{A}_t^{\text{lin}}$ and $\boldsymbol{B}_t^{\text{lin}}$ admit the following recursion*

$$\begin{cases}
\boldsymbol{A}_t^{\text{lin}} = \underbrace{\frac{1}{2}\widetilde{\boldsymbol{U}}_{\boldsymbol{G}^\natural}\left(\left(\boldsymbol{I}_d + \eta\widetilde{\boldsymbol{S}}_{\boldsymbol{G}^\natural}\right)^t + \left(\boldsymbol{I}_d - \eta\widetilde{\boldsymbol{S}}_{\boldsymbol{G}^\natural}\right)^t\right)\widetilde{\boldsymbol{U}}_{\boldsymbol{G}^\natural}^\top}_{:=\boldsymbol{P}_t^A}\boldsymbol{A}_0, \\[4mm]
(\boldsymbol{B}_t^{\text{lin}})^\top = \underbrace{\frac{1}{2}\widetilde{\boldsymbol{V}}_{\boldsymbol{G}^\natural}\left(\left(\boldsymbol{I}_d + \eta\widetilde{\boldsymbol{S}}_{\boldsymbol{G}^\natural}\right)^t - \left(\boldsymbol{I}_d - \eta\widetilde{\boldsymbol{S}}_{\boldsymbol{G}^\natural}\right)^t\right)\widetilde{\boldsymbol{U}}_{\boldsymbol{G}^\natural}^\top}_{:=\boldsymbol{P}_t^B}\boldsymbol{A}_0.
\end{cases}$$

*Furthermore, if $\widetilde{\boldsymbol{X}}^\top\widetilde{\boldsymbol{X}}$ is non-singular, $\boldsymbol{P}_t^A$ is a full rank matrix and singular values are 1 after the $r^*$-th order. $\boldsymbol{P}_t^B$ is a rank-$r^*$ matrix.*

*Proof.* We start with the special case $d = k$ and then discuss the case of $d \neq k$. For the case of $d = k$, we have

$$\boldsymbol{Z}_t^{\mathtt{lin}} = \boldsymbol{H}^t \boldsymbol{Z}_0 = (\mathbf{C}\mathbf{T}\mathbf{C}^\top)^t \boldsymbol{Z}_0 = \mathbf{C}\mathbf{T}^t\mathbf{C}^\top \boldsymbol{Z}_0 \,,$$

where the last equality follows from the fact that $\mathbf{C}$ is an orthogonal matrix. Next, we compute $\mathbf{T}^t$

$$\mathbf{T}^t = \begin{bmatrix} \left(\boldsymbol{I}_d + \eta\widetilde{\boldsymbol{S}}_{\boldsymbol{G}^\natural}\right)^t & \mathbf{0} \\ \mathbf{0}_{d\times d} & \left(\boldsymbol{I}_d - \eta\widetilde{\boldsymbol{S}}_{\boldsymbol{G}^\natural}\right)^t \end{bmatrix}. \tag{16}$$

Then, we can derive the following recursion

$$\begin{aligned}
\boldsymbol{Z}_t^{\mathtt{lin}} =& \boldsymbol{H}^t \boldsymbol{Z}_0 \\
=& \frac{1}{\sqrt{2}} \begin{bmatrix} \widetilde{\boldsymbol{U}}_{\boldsymbol{G}^\natural} & -\widetilde{\boldsymbol{U}}_{\boldsymbol{G}^\natural} \\ \widetilde{\boldsymbol{V}}_{\boldsymbol{G}^\natural} & \widetilde{\boldsymbol{V}}_{\boldsymbol{G}^\natural} \end{bmatrix} \begin{bmatrix} \left(\boldsymbol{I}_d + \eta\widetilde{\boldsymbol{S}}_{\boldsymbol{G}^\natural}\right)^t & \mathbf{0} \\ \mathbf{0}_{d\times d} & \left(\boldsymbol{I}_d - \eta\widetilde{\boldsymbol{S}}_{\boldsymbol{G}^\natural}\right)^t. \end{bmatrix} \times \mathbf{C}^\top \boldsymbol{Z}_0 \\
=& \begin{bmatrix} \widetilde{\boldsymbol{U}}_{\boldsymbol{G}^\natural}\left(\boldsymbol{I}_d + \eta\widetilde{\boldsymbol{S}}_{\boldsymbol{G}^\natural}\right)^t & -\widetilde{\boldsymbol{U}}_{\boldsymbol{G}^\natural}\left(\boldsymbol{I}_d - \eta\widetilde{\boldsymbol{S}}_{\boldsymbol{G}^\natural}\right)^t \\ \widetilde{\boldsymbol{V}}_{\boldsymbol{G}^\natural}\left(\boldsymbol{I}_d + \eta\widetilde{\boldsymbol{S}}_{\boldsymbol{G}^\natural}\right)^t & \widetilde{\boldsymbol{V}}_{\boldsymbol{G}^\natural}\left(\boldsymbol{I}_d - \eta\widetilde{\boldsymbol{S}}_{\boldsymbol{G}^\natural}\right)^t \end{bmatrix} \times \frac{\mathbf{C}^\top \boldsymbol{Z}_0}{\sqrt{2}} \\
=& \begin{bmatrix} \frac{1}{2}\widetilde{\boldsymbol{U}}_{\boldsymbol{G}^\natural}\left(\left(\boldsymbol{I}_d + \eta\widetilde{\boldsymbol{S}}_{\boldsymbol{G}^\natural}\right)^t + \left(\boldsymbol{I}_d - \eta\widetilde{\boldsymbol{S}}_{\boldsymbol{G}^\natural}\right)^t\right)\widetilde{\boldsymbol{U}}_{\boldsymbol{G}^\natural}^\top & * \\ \frac{1}{2}\widetilde{\boldsymbol{V}}_{\boldsymbol{G}^\natural}\left(\left(\boldsymbol{I}_d + \eta\widetilde{\boldsymbol{S}}_{\boldsymbol{G}^\natural}\right)^t - \left(\boldsymbol{I}_d - \eta\widetilde{\boldsymbol{S}}_{\boldsymbol{G}^\natural}\right)^t\right)\widetilde{\boldsymbol{U}}_{\boldsymbol{G}^\natural}^\top & * \end{bmatrix} \begin{bmatrix} \boldsymbol{A}_0 \\ \mathbf{0} \end{bmatrix} \\
=& \begin{bmatrix} \frac{1}{2}\widetilde{\boldsymbol{U}}_{\boldsymbol{G}^\natural}\left(\left(\boldsymbol{I}_d + \eta\widetilde{\boldsymbol{S}}_{\boldsymbol{G}^\natural}\right)^t + \left(\boldsymbol{I}_d - \eta\widetilde{\boldsymbol{S}}_{\boldsymbol{G}^\natural}\right)^t\right)\widetilde{\boldsymbol{U}}_{\boldsymbol{G}^\natural}^\top \boldsymbol{A}_0 \\ \frac{1}{2}\widetilde{\boldsymbol{V}}_{\boldsymbol{G}^\natural}\left(\left(\boldsymbol{I}_d + \eta\widetilde{\boldsymbol{S}}_{\boldsymbol{G}^\natural}\right)^t - \left(\boldsymbol{I}_d - \eta\widetilde{\boldsymbol{S}}_{\boldsymbol{G}^\natural}\right)^t\right)\widetilde{\boldsymbol{U}}_{\boldsymbol{G}^\natural}^\top \boldsymbol{A}_0 \end{bmatrix}.
\end{aligned}$$

Next, we extend the results above to $d \neq k$. Here we take $d > k$,

$$\begin{aligned}
\underline{\boldsymbol{B}}_t^{\mathtt{lin}} =& \frac{1}{2}\widetilde{\boldsymbol{V}}_{\boldsymbol{G}^\natural}\left(\left(\boldsymbol{I}_d + \eta\widetilde{\underline{\boldsymbol{S}}}_{\boldsymbol{G}^\natural}\right)^t - \left(\boldsymbol{I}_d - \eta\widetilde{\underline{\boldsymbol{S}}}_{\boldsymbol{G}^\natural}\right)^t\right)\widetilde{\boldsymbol{U}}_{\boldsymbol{G}^\natural}^\top \boldsymbol{A}_0 \\
=& \begin{bmatrix} \frac{1}{2}\widetilde{\boldsymbol{V}}_{\boldsymbol{G}^\natural}\left(\left(\boldsymbol{I}_d + \eta\widetilde{\boldsymbol{S}}_{\boldsymbol{G}^\natural}\right)^t - \left(\boldsymbol{I}_d - \eta\widetilde{\boldsymbol{S}}_{\boldsymbol{G}^\natural}\right)^t\right)\widetilde{\boldsymbol{U}}_{\boldsymbol{G}^\natural}^\top \boldsymbol{A}_0 & \mathbf{0}_{r\times(d-k)} \end{bmatrix},
\end{aligned}$$

which proves the claim. Lastly, we take $d < k$,

$$\begin{aligned}
\underline{\boldsymbol{A}}_t^{\mathtt{lin}} =& \frac{1}{2}\widetilde{\boldsymbol{U}}_{\boldsymbol{G}^\natural}\left(\left(\boldsymbol{I}_k + \eta\widetilde{\underline{\boldsymbol{S}}}_{\boldsymbol{G}^\natural}\right)^t + \left(\boldsymbol{I}_k - \eta\widetilde{\underline{\boldsymbol{S}}}_{\boldsymbol{G}^\natural}\right)^t\right)\widetilde{\underline{\boldsymbol{U}}}_{\boldsymbol{G}^\natural}^\top \boldsymbol{A}_0 \\
& \begin{bmatrix} \frac{1}{2}\widetilde{\boldsymbol{U}}_{\boldsymbol{G}^\natural}\left(\left(\boldsymbol{I}_d + \eta\widetilde{\boldsymbol{S}}_{\boldsymbol{G}^\natural}\right)^t + \left(\boldsymbol{I}_d - \eta\widetilde{\boldsymbol{S}}_{\boldsymbol{G}^\natural}\right)^t\right)\widetilde{\boldsymbol{U}}_{\boldsymbol{G}^\natural}^\top \boldsymbol{A}_0 \\ \mathbf{0}_{(k-d)\times r} \end{bmatrix},
\end{aligned}$$

which completes the proof.

Besides, we discuss about some properties of $\boldsymbol{P}_t^{\boldsymbol{A}}$ and $\boldsymbol{P}_t^{\boldsymbol{B}}$. Recall $\mathrm{Rank}(\boldsymbol{G}^\natural) = \mathrm{Rank}(\Delta) = r^*$, then we have

$$\lambda_{r^*+i}(\boldsymbol{P}_t^{\boldsymbol{A}}) = \frac{1}{2}\lambda_{r^*+i}\left((\boldsymbol{I}_d + \eta\widetilde{\boldsymbol{S}}_{\boldsymbol{G}^\natural})^t + (\boldsymbol{I}_d - \eta\widetilde{\boldsymbol{S}}_{\boldsymbol{G}^\natural})^t\right) = 1\,, \quad \forall\, 1 \leq i \leq (d - r^*)\,.$$

That means $\boldsymbol{P}_t^{\boldsymbol{A}} \in \mathbb{R}^{d\times d}$ is a full rank matrix and the singular values are 1 after the $r^*$-th order. However $\boldsymbol{P}_t^{\boldsymbol{B}} \in \mathbb{R}^{k\times k}$ is a rank-$r^*$ matrix. □

**Part II: Control $\|\boldsymbol{E}_t\|_{op}$**

Based on the above results, we are ready to prove that $\|\boldsymbol{E}_t\|_{op}$ is small.

**Lemma C.6.** *Under assumptions in Section 2.1 for the linear setting, with LoRA initialization (LoRA-init), given $\|\boldsymbol{A}_0\|_{op}$ and $\boldsymbol{G}^\natural$ in Eq. (5) and its largest singular value $\lambda_1(\boldsymbol{G}^\natural)$, consider the following time period*

$$t \leq t^* := \frac{\ln\left(\frac{\lambda_1(\boldsymbol{G}^\natural)}{3\|\boldsymbol{A}_0\|_{op}^2}\right)}{3\ln\left(1 + \eta\lambda_1(\boldsymbol{G}^\natural)\right)},$$

*then the following statement holds with probability at least $1 - 2C\exp(-N)$ for a universal constant $C$ over random Gaussian data*

$$\|\boldsymbol{E}_t\|_{op} \leq \|\boldsymbol{A}_0\|_{op}. \tag{17}$$

**Remark:** By choosing proper random initialization variance over $\boldsymbol{A}_0$, we can ensure $t^* > 1$ to avoid vacuous upper bound.

*Proof.* We will prove by induction. Starting from $t = 0$, this is trivially true since $\boldsymbol{Z}_0 = \boldsymbol{Z}_0^{\text{lin}}$. Next, we assume Eq. (17) holds for $t - 1$ with $t \geq 1$ and prove $\|\boldsymbol{E}_t\|_{op} \leq \|\boldsymbol{A}_0\|_{op}$. To deliver the proof, denote $a_0 := \|\boldsymbol{A}_0\|_{op}$, from Lemma C.5, we know that

$$\|\boldsymbol{A}_{t-1}^{\text{lin}}\|_{op} \leq \left(1 + \eta\lambda_1(\boldsymbol{G}^\natural)\right)^{t-1} a_0, \quad \|\boldsymbol{B}_{t-1}^{\text{lin}}\|_{op} \leq \frac{1}{2}\left(1 + \eta\lambda_1(\boldsymbol{G}^\natural)\right)^{t-1} a_0. \tag{18}$$

Besides, since $(\boldsymbol{A}_t - \boldsymbol{A}_t^{\text{lin}})$ and $(\boldsymbol{B}_t - \boldsymbol{B}_t^{\text{lin}})$ are the sub-matrices of the error term $\boldsymbol{E}_t$, our condition $\|\boldsymbol{E}_{t-1}\|_{op} \leq \|\boldsymbol{A}_0\|_{op}$ we have

$$\begin{cases} \left\|\boldsymbol{A}_{t-1} - \boldsymbol{A}_{t-1}^{\text{lin}}\right\|_{op} \leq \|\boldsymbol{E}_{t-1}\|_{op}, \\ \left\|\boldsymbol{B}_{t-1} - \boldsymbol{B}_{t-1}^{\text{lin}}\right\|_{op} \leq \|\boldsymbol{E}_{t-1}\|_{op}. \end{cases} \tag{19}$$

It implies that

$$\|\boldsymbol{A}_{t-1}\|_{op} \leq \left(1 + \eta\lambda_1(\boldsymbol{G}^\natural)\right)^{t-1} a_0 + \|\boldsymbol{E}_{t-1}\|_{op}, \quad \|\boldsymbol{B}_{t-1}\|_{op} \leq \frac{1}{2}\left(1 + \eta\lambda_1(\boldsymbol{G}^\natural)\right)^{t-1} a_0 + \|\boldsymbol{E}_{t-1}\|_{op}.$$

Besides, according to covariance matrix estimation in the operator norm in Lemma E.1, with probability at least $1 - 2C\exp(-N\epsilon^2)$ for a universal constant $C > 0$, we have (taking $\epsilon = 1$)

$$\left\|\frac{1}{N}\widetilde{\boldsymbol{X}}^\top\widetilde{\boldsymbol{X}} - \boldsymbol{I}_d\right\|_{op} \leq \epsilon = 1. \tag{20}$$

Accordingly, with probability at least $1 - 2C\exp(-N)$, $\|\widehat{\boldsymbol{E}}_t\|_{op}$ can be upper bounded by

$$\|\widehat{\boldsymbol{E}}_t\|_{op} \leq \eta\left\|\frac{1}{N}\widetilde{\boldsymbol{X}}^\top\widetilde{\boldsymbol{X}}\boldsymbol{A}_{t-1}\boldsymbol{B}_{t-1}\boldsymbol{B}_{t-1}^\top\right\|_{op} + \eta\left\|\boldsymbol{B}_{t-1}^\top\boldsymbol{A}_{t-1}^\top\frac{1}{N}\widetilde{\boldsymbol{X}}^\top\widetilde{\boldsymbol{X}}\boldsymbol{A}_{t-1}\right\|_{op}$$

$$\leq \eta(1 + \epsilon)\|\boldsymbol{A}_{t-1}\|_{op}\|\boldsymbol{B}_{t-1}\|_{op}^2 + \eta(1 + \epsilon)\|\boldsymbol{A}_{t-1}\|_{op}^2\|\boldsymbol{B}_{t-1}\|_{op} \qquad \text{[using Eq. (20)]}$$

$$\leq (1 + \epsilon)\eta\|\boldsymbol{A}_{t-1}\|_{op}\|\boldsymbol{B}_{t-1}\|_{op}\left(\|\boldsymbol{B}_{t-1}\|_{op} + \|\boldsymbol{A}_{t-1}\|_{op}\right)$$

$$\leq (1 + \epsilon)\eta\left(\|\boldsymbol{A}_{t-1}^{\text{lin}}\|_{op} + \|\boldsymbol{E}_{t-1}\|_{op}\right)\left(\|\boldsymbol{B}_{t-1}^{\text{lin}}\|_{op} + \|\boldsymbol{E}_{t-1}\|_{op}\right) \times \left(\|\boldsymbol{B}_{t-1}^{\text{lin}}\|_{op} + \|\boldsymbol{A}_{t-1}^{\text{lin}}\|_{op} + 2\|\boldsymbol{E}_{t-1}\|_{op}\right).$$
$$\text{[using Eq. (19)]}$$

Accordingly, using the upper bound of $\|\boldsymbol{A}_{t-1}^{\text{lin}}\|_{op}$ and $\|\boldsymbol{B}_{t-1}^{\text{lin}}\|_{op}$ in Eq. (18), we have

$$\|\widehat{\boldsymbol{E}}_t\|_{op} \leq (1 + \epsilon)\eta\left(\left(1 + \eta\lambda_1(\boldsymbol{G}^\natural)\right)^{t-1} a_0 + \|\boldsymbol{E}_{t-1}\|_{op}\right)\left(\frac{1}{2}\left(1 + \eta\lambda_1(\boldsymbol{G}^\natural)\right)^{t-1} a_0 + \|\boldsymbol{E}_{t-1}\|_{op}\right) \times$$

$$\left(\frac{3}{2}\left(1 + \eta\lambda_1(\boldsymbol{G}^\natural)\right)^{t-1} a_0 + 2\|\boldsymbol{E}_{t-1}\|_{op}\right)$$

$$\leq 2(1 + \epsilon)\eta\left(\left(1 + \eta\lambda_1(\boldsymbol{G}^\natural)\right)^{3t-3} a_0^3 + \|\boldsymbol{E}_{t-1}\|_{op}^3\right)$$

$$\leq 6\eta\left(1 + \eta\lambda_1(\boldsymbol{G}^\natural)\right)^{3t-3} a_0^3. \qquad \text{[from our inductive hypothesis]}$$

Then, by Lemma C.4, we can conclude that

$$
\begin{aligned}
\|\boldsymbol{E}_t\|_{op} = \left\|\sum_{i=1}^{t} \boldsymbol{H}^{t-i}\widehat{\boldsymbol{E}}_i\right\|_{op} &\leq \sum_{i=1}^{t} \|\boldsymbol{H}\|_{op}^{t-i}\|\widehat{\boldsymbol{E}}_i\|_{op} \\
&\leq 6\eta a_0^3 \times \sum_{i=1}^{t}\left(1 + \eta\lambda_1(\boldsymbol{G}^{\natural})\right)^{t+2i-3} && \text{[using Lemma C.1]} \\
&= 6\eta a_0^3 \times \left(1 + \eta\lambda_1(\boldsymbol{G}^{\natural})\right)^{t-1}\sum_{i=1}^{t}\left(1 + \eta\lambda_1(\boldsymbol{G}^{\natural})\right)^{2i-2} \\
&= 6\eta a_0^3 \times \left(1 + \eta\lambda_1(\boldsymbol{G}^{\natural})\right)^{t-1}\frac{\left(1 + \eta\lambda_1(\boldsymbol{G}^{\natural})\right)^{2t} - 1}{\left(1 + \eta\lambda_1(\boldsymbol{G}^{\natural})\right)^2 - 1} && \text{[geometric series]} \\
&\leq 6\eta a_0^3 \times \left(1 + \eta\lambda_1(\boldsymbol{G}^{\natural})\right)^{t-1}\frac{\left(1 + \eta\lambda_1(\boldsymbol{G}^{\natural})\right)^{2t+1}}{2\eta\lambda_1(\boldsymbol{G}^{\natural})} \\
&\leq 3\left(1 + \eta\lambda_1(\boldsymbol{G}^{\natural})\right)^{3t}\frac{a_0^3}{\lambda_1(\boldsymbol{G}^{\natural})}\,. && (21)
\end{aligned}
$$

Accordingly, when $t \leq t^* := \frac{\ln\left(\frac{\lambda_1(\boldsymbol{G}^{\natural})}{3\|\boldsymbol{A}_0\|_{op}^2}\right)}{3\ln(1 + \eta\lambda_1(\boldsymbol{G}^{\natural}))}$, we have

$$
\|\boldsymbol{E}_t\|_{op} \leq \|\boldsymbol{A}_0\|_{op}\,,
$$

which proves the claim. $\qquad\square$

### C.1.3. ALIGNMENT TO NEGATIVE GRADIENT OF FULL FINE-TUNING

Now we can apply Lemma C.6 to obtain
$$
\left\|\boldsymbol{A}_t - \boldsymbol{A}_t^{\texttt{lin}}\right\|_{op} \leq \|\boldsymbol{A}_0\|_{op}\,.
$$

Recall Lemma C.5, we can observe that the dynamic of $\boldsymbol{A}_t^{\texttt{lin}}$ also follows an Oja's Power Method (Oja, 1982), which aligns $\boldsymbol{A}_t^{\texttt{lin}}$'s left singular subspace to the left subspace of the initial negative gradient step $\boldsymbol{G}^{\natural}$ of full fine-tuning. We anticipate that $\lambda_{r^*}(\boldsymbol{A}_t) \gg \lambda_{r^*+1}(\boldsymbol{A}_t)$ for sufficiently large $t$. Furthermore, if $\|\boldsymbol{E}_t\|_{op}$ remains small, then the top-$r^*$ left singular subspace of $\boldsymbol{A}_t$ can closely align to $\boldsymbol{G}^{\natural}$'s. To prove this alignment, we modify Stöger & Soltanolkotabi (2021, Lemma 8.3) to obtain the following results.

**Lemma C.7.** *Under assumptions in Section 2.1 for the linear setting, recall* $\boldsymbol{P}_t^{\boldsymbol{A}} := \frac{1}{2}\widetilde{\boldsymbol{U}}_{\boldsymbol{G}^{\natural}}\left(\left(\boldsymbol{I}_d + \eta\widetilde{\boldsymbol{S}}_{\boldsymbol{G}^{\natural}}\right)^t + \left(\boldsymbol{I}_d - \eta\widetilde{\boldsymbol{S}}_{\boldsymbol{G}^{\natural}}\right)^t\right)\widetilde{\boldsymbol{U}}_{\boldsymbol{G}^{\natural}}^{\top}$ *as* $\mathbb{R}^{d\times d}$*-valued symmetric matrix in Lemma C.5, we assume that*

$$
\lambda_{r^*+1}(\boldsymbol{P}_t^{\boldsymbol{A}})\|\boldsymbol{A}_0\|_{op} + \|\boldsymbol{E}_t\|_{op} < \lambda_{r^*}(\boldsymbol{P}_t^{\boldsymbol{A}})\lambda_{\min}(\boldsymbol{U}_{r^*}^{\top}(\boldsymbol{P}_t^{\boldsymbol{A}})\boldsymbol{A}_0)\,,
$$

*that can be satisfied under certain conditions (discussed later). Then the following three inequalities hold:*

$$
\lambda_{r^*}(\boldsymbol{P}_t^{\boldsymbol{A}}\boldsymbol{A}_0 + \boldsymbol{E}_t) \geq \lambda_{r^*}(\boldsymbol{P}_t^{\boldsymbol{A}})\lambda_{\min}(\boldsymbol{U}_{r^*}^{\top}(\boldsymbol{P}_t^{\boldsymbol{A}})\boldsymbol{A}_0) - \|\boldsymbol{E}_t\|_{op}\,, \tag{22}
$$

$$
\lambda_{r^*+1}(\boldsymbol{P}_t^{\boldsymbol{A}}\boldsymbol{A}_0 + \boldsymbol{E}_t) \leq \lambda_{r^*+1}(\boldsymbol{P}_t^{\boldsymbol{A}})\|\boldsymbol{A}_0\|_{op} + \|\boldsymbol{E}_t\|_{op}\,, \tag{23}
$$

$$
\|\boldsymbol{U}_{r^*,\perp}^{\top}(\boldsymbol{P}_t^{\boldsymbol{A}})\boldsymbol{U}_{r^*}(\boldsymbol{P}_t^{\boldsymbol{A}}\boldsymbol{A}_0 + \boldsymbol{E}_t)\|_{op} \leq \frac{\lambda_{r^*+1}(\boldsymbol{P}_t^{\boldsymbol{A}})\|\boldsymbol{A}_0\|_{op} + \|\boldsymbol{E}_t\|_{op}}{\lambda_{r^*}(\boldsymbol{P}_t^{\boldsymbol{A}})\lambda_{\min}(\boldsymbol{U}_{r^*}^{\top}(\boldsymbol{P}_t^{\boldsymbol{A}})\boldsymbol{A}_0) - \lambda_{r^*+1}(\boldsymbol{P}_t^{\boldsymbol{A}})\|\boldsymbol{A}_0\|_{op} - \|\boldsymbol{E}_t\|_{op}}\,, \tag{24}
$$

*where* $\boldsymbol{U}_k(\boldsymbol{M})$ *denotes the left singular subspace spanned by the $k$ largest singular values of the input matrix $\boldsymbol{M}$ and* $\boldsymbol{U}_{k,\perp}(\boldsymbol{M})$ *denotes the left singular subspace orthogonal to* $\boldsymbol{U}_k(\boldsymbol{M})$.

This lemma can help us derive the principle angle of the left singular subspace between $\boldsymbol{A}_t^{\texttt{lin}}$ and $\boldsymbol{A}_t$. Note that the assumption comes from the necessary condition of Wedin's $\sin\theta$ theorem (Wedin, 1972). In the next lemma, we aim to derive the time threshold which can fulfill this assumption.

**Lemma C.8.** *Under assumptions in Section 2.1 for the linear setting, given $\|\boldsymbol{A}_0\|_{op}$, for any $\theta \in (0,1)$, taking*

$$t \leq \frac{\ln\left(\frac{8\|\boldsymbol{A}_0\|_{op}}{\theta \lambda_{\min}(\boldsymbol{U}_{r^*}^\top(\boldsymbol{P}_t^{\boldsymbol{A}})\boldsymbol{A}_0)}\right)}{\ln\left(1 + \eta \lambda_{r^*}\left(\boldsymbol{G}^\natural\right)\right)} ,$$

*then Eq. (24) holds with probability at least $1 - 2C\exp(-N)$ for a universal constant $C$ over random Gaussian data, i.e.*

$$\|\boldsymbol{U}_{r^*,\perp}^\top(\boldsymbol{P}_t^{\boldsymbol{A}})\boldsymbol{U}_{r^*}(\boldsymbol{P}_t^{\boldsymbol{A}}\boldsymbol{A}_0 + \boldsymbol{E}_t)\|_{op} \leq \theta .$$

**Remark:** To ensure that the $\theta$-alignment phase still falls into the early phase in Lemma C.6 for $\|\boldsymbol{E}_t\|_{op} \leq \|\boldsymbol{A}_0\|_{op}$, we need to choose proper initialization for $\boldsymbol{A}_0$. We will detail this in Theorem 3.2 later.

*Proof.* First, $\lambda_{r^*}(\boldsymbol{P}_t^{\boldsymbol{A}})$ in Lemma C.5 can be lower bounded by

$$\begin{aligned}
\lambda_{r^*}(\boldsymbol{P}_t^{\boldsymbol{A}}) &= \frac{1}{2}\lambda_{r^*}\left((\boldsymbol{I}_d + \eta\widetilde{\boldsymbol{S}}_{\boldsymbol{G}^\natural})^t + (\boldsymbol{I}_d - \eta\widetilde{\boldsymbol{S}}_{\boldsymbol{G}^\natural})^t\right) \\
&\geq \frac{1}{2}\lambda_{r^*}\left((\boldsymbol{I}_d + \eta\widetilde{\boldsymbol{S}}_{\boldsymbol{G}^\natural})^t\right) \\
&= \frac{1}{2}\left(1 + \eta\lambda_{r^*}\left(\boldsymbol{G}^\natural\right)\right)^t .
\end{aligned} \tag{25}$$

Recall Lemma C.5, we have $\lambda_{r^*+1}(\boldsymbol{P}_t^{\boldsymbol{A}}) = 1$ and Lemma C.6 with $\|\boldsymbol{E}_t\|_{op} \leq \|\boldsymbol{A}_0\|_{op}$, we define the following threshold $\gamma$ and upper bound it

$$\begin{aligned}
\gamma &:= \frac{\lambda_{r^*+1}(\boldsymbol{P}_t^{\boldsymbol{A}})\|\boldsymbol{A}_0\|_{op} + \|\boldsymbol{E}_t\|_{op}}{\lambda_{r^*}(\boldsymbol{P}_t^{\boldsymbol{A}})\lambda_{\min}(\boldsymbol{U}_{r^*}^\top(\boldsymbol{P}_t^{\boldsymbol{A}})\boldsymbol{A}_0)} \\
&\leq \frac{2\|\boldsymbol{A}_0\|_{op}}{\frac{1}{2}\left(1 + \eta\lambda_{r^*}\left(\boldsymbol{G}^\natural\right)\right)^t \lambda_{\min}(\boldsymbol{U}_{r^*}^\top(\boldsymbol{P}_t^{\boldsymbol{A}})\boldsymbol{A}_0)} \qquad \text{[using Lemma C.5, C.6]} \\
&= \exp\left(-\ln\left(1 + \eta\lambda_{r^*}\left(\boldsymbol{G}^\natural\right)\right) \cdot t\right) \cdot \frac{4\|\boldsymbol{A}_0\|_{op}}{\lambda_{\min}(\boldsymbol{U}_{r^*}^\top(\boldsymbol{P}_t^{\boldsymbol{A}})\boldsymbol{A}_0)} .
\end{aligned} \tag{26}$$

Set $\theta \in (0,1)$, let Eq.(26)$\leq \frac{\theta}{2}$, then we have that

$$\|\boldsymbol{U}_{r^*,\perp}^\top(\boldsymbol{P}_t^{\boldsymbol{A}})\boldsymbol{U}_{r^*}(\boldsymbol{P}_t^{\boldsymbol{A}}\boldsymbol{A}_0 + \boldsymbol{E}_t)\|_{op} \leq \theta .$$

The time $t$ to achieve this angle $\theta$ can be upper bounded by

$$\exp\left(-\ln\left(1 + \eta\lambda_{r^*}\left(\boldsymbol{G}^\natural\right)\right) \cdot t\right) \cdot \frac{4\|\boldsymbol{A}_0\|_{op}}{\lambda_{\min}(\boldsymbol{U}_{r^*}^\top(\boldsymbol{P}_t^{\boldsymbol{A}})\boldsymbol{A}_0)} \leq \frac{\theta}{2} ,$$

which implies that

$$t \leq \frac{\ln\left(\frac{8\|\boldsymbol{A}_0\|_{op}}{\theta \lambda_{\min}(\boldsymbol{U}_{r^*}^\top(\boldsymbol{P}_t^{\boldsymbol{A}})\boldsymbol{A}_0)}\right)}{\ln\left(1 + \eta \lambda_{r^*}\left(\boldsymbol{G}^\natural\right)\right)} .$$

Finally we conclude the proof. $\qquad\square$

**Theorem C.9.** *[Full version of Theorem 3.2] Under assumptions in Section 2.1 for the linear setting, recall $\boldsymbol{G}^\natural$ defined in Eq. (5) with its condition number $\kappa^\natural$, we consider random Gaussian initialization $\boldsymbol{A}_0 \in \mathbb{R}^{d \times r}$ with $[\boldsymbol{A}_0]_{ij} \sim \mathcal{N}(0, \alpha^2)$ in (LoRA-init), for any $\theta \in (0,1)$, let $\xi = o(1)$ be chosen such that*

$$\alpha \leq \begin{cases} \left(\frac{\theta\xi}{24r\sqrt{d}}\right)^{\frac{3\kappa^\natural}{2}}\sqrt{\frac{\lambda_1(\boldsymbol{G}^\natural)}{27d}} & \text{if } r^* \leq r < 2r^*, \\ \left(\frac{\theta}{24\sqrt{d}}\right)^{\frac{3\kappa^\natural}{2}}\sqrt{\frac{\lambda_1(\boldsymbol{G}^\natural)}{27d}} & \text{if } r \geq 2r^* . \end{cases}$$

*Then if we run gradient descent for $t^*$ steps with*

$$t^* \lesssim \begin{cases} \frac{\ln\left(\frac{24r\sqrt{d}}{\theta\xi}\right)}{\ln(1+\eta\lambda_{r^*}(\boldsymbol{G}^\natural))} & \text{if } r^* \leq r < 2r^*, \\ \frac{\ln\left(\frac{24\sqrt{d}}{\theta}\right)}{\ln(1+\eta\lambda_{r^*}(\boldsymbol{G}^\natural))} & \text{if } r \geq 2r^*, \end{cases}$$

*we have the following alignment on the left singular subspace between $\boldsymbol{G}^\natural$ and $\boldsymbol{A}_{t^*}$*

$$\left\|\boldsymbol{U}_{r^*,\perp}^\top(\boldsymbol{G}^\natural)\boldsymbol{U}_{r^*}(\boldsymbol{A}_{t^*})\right\|_{op} \lesssim \theta,$$

*with probability at least* $\begin{cases} 1 - C_1\exp(-d) - (C_2\xi)^{r-r^*+1} - C_3\exp(-r) - C\exp(-N) & \text{if } r^* \leq r < 2r^*, \\ 1 - C_4\exp(-d) - C_5\exp(-r) - C\exp(-N) & \text{if } r \geq 2r^*, \end{cases}$

*for some positive constants $C, C_1, C_2, C_3, C_4, C_5$. Here $\boldsymbol{U}_{r^*}(\boldsymbol{A}_{t^*})$ denotes the left singular subspace spanned by the $r^*$ largest singular values of $\boldsymbol{A}_{t^*}$ and $\boldsymbol{U}_{r^*,\perp}(\boldsymbol{M})$ denotes the left singular subspace orthogonal to $\boldsymbol{U}_{r^*}(\boldsymbol{M})$. Note that we can select any pair of stepsizes $(\eta, \eta)$ that satisfies the conditions $t^* > 1$, $\eta \geq \eta$, and $\zeta(\eta, \eta) = \Theta(1)$.*

*Proof.* For ease of description, we denote $\boldsymbol{A}_0 := \alpha\boldsymbol{T} \in \mathbb{R}^{d\times r}$ where $\boldsymbol{T}$ is a standard random Gaussian matrix with zero-mean and unit variance. Here we aim to choose a proper $\alpha$ to ensure that $\theta$-alignment phase in Lemma C.8 still falls into the early phase in Lemma C.6, i.e.

$$\frac{\ln\left(\frac{8\|\boldsymbol{A}_0\|_{op}}{\theta\lambda_{\min}(\boldsymbol{U}_{r^*}^\top(\boldsymbol{P}_t^A)\boldsymbol{A}_0)}\right)}{\ln(1+\eta\lambda_{r^*}(\boldsymbol{G}^\natural))} = \frac{\ln\left(\frac{\lambda_1(\boldsymbol{G}^\natural)}{3\|\boldsymbol{A}_0\|_{op}^2}\right)}{3\ln(1+\eta\lambda_1(\boldsymbol{G}^\natural))} = t^*$$

$$\Leftrightarrow \quad \ln\left(\frac{8\|\boldsymbol{A}_0\|_{op}}{\theta\lambda_{\min}(\boldsymbol{U}_{r^*}^\top(\boldsymbol{P}_t^A)\boldsymbol{A}_0)}\right) = \frac{\ln\left(1+\eta\lambda_{r^*}(\boldsymbol{G}^\natural)\right)}{3\ln\left(1+\eta\lambda_1(\boldsymbol{G}^\natural)\right)}\ln\left(\frac{\lambda_1(\boldsymbol{G}^\natural)}{3\|\boldsymbol{A}_0\|_{op}^2}\right)$$

$$\Leftrightarrow \quad \frac{8\|\boldsymbol{A}_0\|_{op}}{\theta\lambda_{\min}(\boldsymbol{U}_{r^*}^\top(\boldsymbol{P}_t^A)\boldsymbol{A}_0)} = \left(\frac{\lambda_1(\boldsymbol{G}^\natural)}{3\|\boldsymbol{A}_0\|_{op}^2}\right)^{\frac{\ln\left(1+\eta\lambda_{r^*}(\boldsymbol{G}^\natural)\right)}{3\ln\left(1+\eta\lambda_1(\boldsymbol{G}^\natural)\right)}}$$

$$\Leftrightarrow \quad \theta = \frac{8\|\boldsymbol{A}_0\|_{op}}{\lambda_{\min}(\boldsymbol{U}_{r^*}^\top(\boldsymbol{P}_t^A)\boldsymbol{A}_0)}\left(\frac{3\|\boldsymbol{A}_0\|_{op}^2}{\lambda_1(\boldsymbol{G}^\natural)}\right)^{\frac{\ln\left(1+\eta\lambda_{r^*}(\boldsymbol{G}^\natural)\right)}{3\ln\left(1+\eta\lambda_1(\boldsymbol{G}^\natural)\right)}}$$

$$= \frac{8\|\boldsymbol{T}\|_{op}}{\lambda_{\min}(\boldsymbol{U}_{r^*}^\top(\boldsymbol{P}_t^A)\boldsymbol{T})}\left(\frac{3\|\boldsymbol{A}_0\|_{op}^2}{\lambda_1(\boldsymbol{G}^\natural)}\right)^{\iota} \qquad \left[\text{by setting } \iota := \frac{\ln\left(1+\eta\lambda_{r^*}(\boldsymbol{G}^\natural)\right)}{3\ln(1+\eta\lambda_1(\boldsymbol{G}^\natural))}\right]$$

$$= \frac{8\|\boldsymbol{T}\|_{op}}{\lambda_{\min}(\boldsymbol{U}_{r^*}^\top(\boldsymbol{P}_t^A)\boldsymbol{T})}\left(\frac{3\|\boldsymbol{T}\|_{op}^2}{\lambda_1(\boldsymbol{G}^\natural)}\right)^{\iota}\alpha^{2\iota}.$$

In the next, we will discuss how to pick up $\alpha$. According to Lemma E.3, we need to consider the following two cases on the relationship between $r^*$ and $r$.

**Case 1.** $r^* \leq r < 2r^*$: by Lemma E.2 and Lemma E.3, with probability at least $1 - C_1\exp(-d) - (C_2\xi)^{r-r^*+1} - C_3\exp(-r)$ for some positive constants $C_1, C_2, C_3$, we have

$$\frac{\|\boldsymbol{T}\|_{op}}{3\sqrt{d}} \leq 1, \quad \frac{\xi}{r\lambda_{\min}(\boldsymbol{U}_{r^*}^\top(\boldsymbol{P}_t^A)\boldsymbol{T})} \lesssim 1. \tag{27}$$

Here we pick

$$\alpha \leq \left(\frac{\theta\xi}{24r\sqrt{d}}\right)^{\frac{3\kappa^\natural}{2}}\sqrt{\frac{\lambda_1(\boldsymbol{G}^\natural)}{27d}},$$

then recall Lemma C.8 on the alignment, we take $\alpha$ here

$$\left\| U_{r^*,\perp}^\top \left( -\nabla_{\boldsymbol{W}} \widetilde{L}(\boldsymbol{W}^\natural) \right) U_{r^*} (\boldsymbol{A}_{t^*}) \right\|_{op}$$

$$\leq \frac{8\|\boldsymbol{T}\|_{op}}{\lambda_{\min}(U_{r^*}^\top(\boldsymbol{P}_t^{\boldsymbol{A}})\boldsymbol{T})} \left( \frac{3\|\boldsymbol{T}\|_{op}^2}{\lambda_1(\boldsymbol{G}^\natural)} \right)^\iota \alpha^{2\iota}$$

$$= \frac{8\|\boldsymbol{T}\|_{op}}{\lambda_{\min}(U_{r^*}^\top(\boldsymbol{P}_t^{\boldsymbol{A}})\boldsymbol{T})} \left( \frac{3\|\boldsymbol{T}\|_{op}^2}{\lambda_1(\boldsymbol{G}^\natural)} \right)^\iota \left( \frac{\theta\xi}{24r\sqrt{d}} \right)^{3\kappa^\natural \iota} \left( \frac{\lambda_1(\boldsymbol{G}^\natural)}{27d} \right)^\iota$$

$$= \frac{8\|\boldsymbol{T}\|_{op}}{\lambda_{\min}(U_{r^*}^\top(\boldsymbol{P}_t^{\boldsymbol{A}})\boldsymbol{T})} \left( \frac{\|\boldsymbol{T}\|_{op}^2}{9d} \right)^\iota \left( \frac{\theta\xi}{24r\sqrt{d}} \right)^{3\kappa^\natural \iota}$$

$$\leq \frac{\|\boldsymbol{T}\|_{op}\theta\xi}{3r\sqrt{d}\lambda_{\min}(U_{r^*}^\top(\boldsymbol{P}_t^{\boldsymbol{A}})\boldsymbol{T})} \left( \frac{\|\boldsymbol{T}\|_{op}^2}{9d} \right)^\iota . \qquad \left[ \text{since } \iota \geq 1/3\kappa^\natural \text{ and } \frac{\theta\xi}{24r\sqrt{d}} \in (0,1) \right]$$

Then using Eq. (27), with probability at least $1 - C_1 \exp(-d) - (C_2\xi)^{r-r^*+1} - C_3 \exp(-r)$ for some positive constants $C_1, C_2, C_3$, we have

$$\left\| U_{r^*,\perp}^\top \left( -\nabla_{\boldsymbol{W}} \widetilde{L}(\boldsymbol{W}^\natural) \right) U_{r^*} (\boldsymbol{A}_{t^*}) \right\|_{op} \lesssim \theta .$$

And we can compute the upper bound of $t^*$ as

$$t^* = \frac{\ln \left( \frac{8\|\boldsymbol{A}\|_{op}}{\theta\lambda_{\min}(U_{r^*}^\top(\boldsymbol{P}_t^{\boldsymbol{A}})\boldsymbol{A})} \right)}{\ln \left( 1 + \eta\lambda_{r^*}(\boldsymbol{G}^\natural) \right)} \lesssim \frac{\ln \left( \frac{24r\sqrt{d}}{\theta\xi} \right)}{\ln \left( 1 + \eta\lambda_{r^*}(\boldsymbol{G}^\natural) \right)} .$$

**Case 2.** $r \geq 2r^*$: by Lemma E.2 and Lemma E.3, with probability at least $1 - C_4 \exp(-d) - C_5 \exp(-r)$ for some positive constants $C_4, C_5$, we have

$$\frac{\|\boldsymbol{T}\|_{op}}{3\sqrt{d}} \leq 1, \quad \frac{1}{\lambda_{\min}(U_{r^*}^\top(\boldsymbol{P}_t^{\boldsymbol{A}})\boldsymbol{T})} \lesssim 1 .$$

Here we pick

$$\alpha \leq \left( \frac{\theta}{24\sqrt{d}} \right)^{\frac{3\kappa^\natural}{2}} \sqrt{\frac{\lambda_1(\boldsymbol{G}^\natural)}{27d}} .$$

Similarly, we can obtain

$$\left\| U_{r^*,\perp}^\top \left( -\nabla_{\boldsymbol{W}} \widetilde{L}(\boldsymbol{W}^\natural) \right) U_{r^*} (\boldsymbol{A}_t) \right\|_{op} \leq \frac{\|\boldsymbol{T}\|_{op}\theta}{3\sqrt{d}\lambda_{\min}(U_{r^*}^\top(\boldsymbol{P}_t^{\boldsymbol{A}})\boldsymbol{T})} \left( \frac{\|\boldsymbol{T}\|_{op}^2}{9d} \right)^\iota \lesssim \theta .$$

And we can compute the upper bound of $t^*$ as

$$t^* \leq \frac{\ln \left( \frac{24\sqrt{d}}{\theta} \right)}{\ln \left( 1 + \eta\lambda_{r^*}(\boldsymbol{G}^\natural) \right)} .$$

$\square$

**Theorem C.10.** *Under assumptions in Section 2.1 for the linear setting, using the LoRA initialization for $\boldsymbol{B}_0 = \boldsymbol{0}$, then for any time-step $t \in \mathbb{N}_+$, we have*

$$\left\| V_{r^*,\perp}^\top \left( -\nabla_{\boldsymbol{W}} \widetilde{L}(\boldsymbol{W}^\natural) \right) V_{r^*} (\boldsymbol{B}_t) \right\|_{op} = 0 .$$

*Proof.* We prove by induction. Recall the complete SVD of $\Delta$ in Eq. (1) as

$$\Delta = \widetilde{U}\widetilde{S}^*\widetilde{V}^\top = \begin{bmatrix} U & U_\perp \end{bmatrix} \begin{bmatrix} S^* & 0_{r^* \times (d-r^*)} \\ 0_{(d-r^*) \times r^*} & 0_{(d-r^*) \times (d-r^*)} \end{bmatrix} \begin{bmatrix} V^\top \\ V_\perp^\top \end{bmatrix}.$$

For $t = 1$, recall $G^\natural = \frac{1}{N}\widetilde{X}^\top \widetilde{X}\Delta$ in Eq. (5), we have

$$B_1 V_\perp = \frac{\eta}{N} A_0^\top G^\natural V_\perp = \frac{\eta}{N} A_0^\top \widetilde{X}^\top \widetilde{X}\Delta V_\perp = 0_{r \times (d-r^*)}.$$

Assume $B_t V_\perp = 0_{r \times (d-r^*)}$ holds for any $t \in \mathbb{N}_+$ and $t \geq 2$, then

$$B_{t+1} V_\perp = B_t V_\perp - \frac{\eta}{N} A_t^\top \widetilde{X}^\top \widetilde{X} A_t B_t V_\perp + \frac{\eta}{N} A_t^\top G^\natural V_\perp = 0_{r \times (d-r^*)},$$

which completes the claim. $\qquad\square$

## C.2. Gradient Descent under Spectral Initialization

For notational simplicity, we denote $\widehat{\Sigma} := \frac{1}{N}\widetilde{X}^\top \widetilde{X}$ in the following content. Recall the negative gradient of Full Fine-tuning at the first step in Eq. (5), we write it here again

$$G^\natural = -\nabla_W \widetilde{L}(W^\natural) = \frac{1}{N}\widetilde{X}^\top \widetilde{Y}_\Delta = \widehat{\Sigma}\Delta = \widetilde{U}_{G^\natural}\widetilde{S}_{G^\natural}\widetilde{V}_{G^\natural}^\top. \tag{28}$$

In this section, according to Lemma E.1, the following statement

$$\left\|\widehat{\Sigma} - I_d\right\|_{op} = \epsilon \leq \min\left\{\frac{1}{2\kappa}, \frac{c}{\kappa^3}\right\} \leq \frac{1}{2}, \quad \text{for some small constant } c, \tag{29}$$

holds with probability at least $1 - 2C\exp(-\epsilon^2 N)$ for a universal constant $C > 0$. We propose the following initialization scheme (Spectral-init)

$$A_0 = \left[\widetilde{U}_{G^\natural}\right]_{[:,1:r]} \left[\widetilde{S}_{G^\natural}^{1/2}\right]_{[1:r]}, \quad B_0 = \left[\widetilde{S}_{G^\natural}^{1/2}\right]_{[1:r]} \left[\widetilde{V}_{G^\natural}\right]_{[:,1:r]}^\top.$$

First, we have the following lemma.

**Lemma C.11.** *Under assumptions in Section 2.1 for the linear setting, with spectral initialization (Spectral-init), recall* $\kappa := \lambda_1^*(\Delta)/\lambda_{r^*}^*(\Delta)$, *then with probability at least with probability* $1 - 2C\exp(-\epsilon^2 N)$ *for a universal constant* $C > 0$, *we have*

$$\|A_0 B_0 - \Delta\|_{op} \leq \epsilon\|\Delta\|_{op} \leq \frac{\lambda_{r^*}^*}{2}, \tag{30}$$

*and*

$$\lambda_{r^*}(A_0) \geq \frac{\sqrt{\lambda_{r^*}^*}}{2}, \quad \lambda_{r^*}(B_0) \geq \frac{\sqrt{\lambda_{r^*}^*}}{2}. \tag{31}$$

*Proof.* Due to $\text{rank}(G^\natural) = r^*$ and $r \geq r^*$, then $A_0 B_0 = G^\natural$. Accordingly, by Eq. (29), with probability at least $1 - 2\exp(-c\epsilon^2 N)$, we have

$$\begin{aligned}
\|A_0 B_0 - \Delta\|_{op} &\leq \|A_0 B_0 - G^\natural\|_{op} + \|G^\natural - \Delta\|_{op} \\
&= \|G^\natural - \Delta\|_{op} \\
&= \left\|\left(\widehat{\Sigma} - I_d\right)\Delta\right\|_{op} && \text{[using Eq. (28)]} \\
&\leq \left\|\widehat{\Sigma} - I_d\right\|_{op}\|\Delta\|_{op} \\
&\leq \epsilon\|\Delta\|_{op} \\
&\leq \frac{1}{2\kappa}\|\Delta\|_{op} && \text{[using Eq. (29)]} \\
&= \frac{\lambda_{r^*}^*}{2}.
\end{aligned}$$

Then, using the above result and Weyl's inequality, we have the upper bound $\lambda_{r^*}(A_0 B_0) \leq \lambda_1(A_0)\lambda_{r^*}(B_0)$ and the lower bound

$$\lambda_{r^*}(A_0 B_0) = \lambda_{r^*}(G^\natural) \geq \lambda_{r^*}(\Delta) - \left\|G^\natural - \Delta\right\|_{op} = \lambda_{r^*}(\Delta) - \|A_0 B_0 - \Delta\|_{op} \geq \frac{\lambda_{r^*}^*}{2}.$$

Now we are ready to give the lower bound of $\lambda_{r^*}(B_0)$. Because of $A_0 B_0 = G^\natural$ under spectral initialization, we have

$$\lambda_1(A_0) \leq \sqrt{\lambda_1(G^\natural)} \leq \sqrt{\left\|\widehat{\Sigma} - I_d\right\|_{op} \lambda_1(\Delta)} \leq \sqrt{\epsilon\lambda_1(\Delta)}, \quad \text{with high probability at least } 1 - 2C\exp(-\epsilon^2 N).$$

where we use $G^\natural = \widehat{\Sigma}\Delta$ and the concentration results on $\widehat{\Sigma}$. Then combining the above two inequalities, $\lambda_{r^*}(B_0)$ is lower bounded by

$$\lambda_{r^*}(B_0) \geq \frac{\lambda_{r^*}(A_0 B_0)}{\lambda_1(A_0)} \geq \frac{\lambda_{r^*}^*/2}{\lambda_1(A_0)} \geq \frac{\sqrt{\lambda_{r^*}^*}}{2},$$

by taking $\epsilon \leq \frac{1}{2\kappa}$. The lower bound of $\lambda_{r^*}(A_0)$ can be obtained similarly. $\qquad\square$

The following lemma indicates $B_t$'s GD dynamics stay in the low-dimensional target subspace under the spectral initialization.

**Lemma C.12.** *Under assumptions in Section 2.1 for the linear setting, with spectral initialization* (Spectral-init)*, during the iteration, for any $t \in \mathbb{N}^+$, we always have $B_t V_\perp = 0_{d \times (d-r^*)}$, where $V_\perp$ comes from the complete SVD of $\Delta$ in Eq. (1).*

*Proof.* We prove it by induction. First, recall the SVD of $\Delta$ in Eq. (1), we have

$$G^\natural V_\perp = \widetilde{\Sigma}\Delta V_\perp = 0_{d \times (d-r^*)},$$

and

$$\begin{aligned}
B_0 V_\perp &= \left[\widetilde{S}_{G^\natural}^{1/2}\right]_{[1:r]} \left[\widetilde{V}_{G^\natural}^\top\right]_{[:,1:r]} V_\perp \\
&= \left[\widetilde{S}_{G^\natural}^{-1/2}\right]_{[1:r]} \left[\widetilde{U}_{G^\natural}^\top\right]_{[:,1:r]} G^\natural V_\perp \\
&= \left[\widetilde{S}_{G^\natural}^{-1/2}\right]_{[1:r]} \left[\widetilde{U}_{G^\natural}^\top\right]_{[:,1:r]} \widehat{\Sigma}\Delta V_\perp \\
&= 0_{d \times (d-r^*)}.
\end{aligned}$$

Next, We prove by induction. Starting from $t = 1$, using the above two equations, we have

$$\begin{aligned}
B_1 V_\perp &= B_0 V_\perp - \frac{\eta_2}{N} A_0^\top \widetilde{X}^\top \left(\widetilde{X}(W^\natural + A_0 B_0) - \widetilde{Y}\right) V_\perp \\
&= B_0 V_\perp - \frac{\eta_2}{N} A_0^\top \widetilde{X}^\top \widetilde{X} A_0 B_0 V_\perp + \eta A_0^\top G^\natural V_\perp \\
&= 0_{d \times (d-r^*)}.
\end{aligned}$$

Assume $B_t V_\perp = 0_{d \times (d-r^*)}$ holds for any $t = 2, 3, \cdots$, then at $t + 1$, we have

$$\begin{aligned}
B_{t+1} V_\perp &= B_t V_\perp - \frac{\eta}{N} A_t^\top \widetilde{X}^\top \widetilde{X} A_t B_t V_\perp + \eta_2 A_t^\top G^\natural V_\perp \\
&= 0_{d \times (d-r^*)}.
\end{aligned}$$

Accordingly we finish the proof. $\qquad\square$

Under spectral initialization, we have already demonstrated that $A_0 B_0$ is close to $\Delta$. In the following content, we aim to track how $\|A_t B_t - \Delta\|_{op}$ behaves (in a local sense), which is a critical ingredient to study both the loss and risk of LoRA training. In this regime, there is no significant difference on setting different step-size $\eta_1$ and $\eta_2$. For ease of description, we set $\eta_1 = \eta_2 := \eta$.

Here we can characterize the operator norm of $(A_t B_t - \Delta)$ as

$$\|A_t B_t - \Delta\|_{op} = \left\|\left(A_t B_t - \Delta\right)\begin{bmatrix} V & V_\perp \end{bmatrix}\right\|_{op} \qquad \text{[by unitary invariance of operator norm]}$$

$$= \|A_t B_t V - U S^*\|_{op} \qquad \text{[by Lemma C.12]}$$

$$= \left\|\left(UU^\top + U_\perp U_\perp^\top\right)\left(A_t B_t V - U S^*\right)\right\|_{op}$$

$$= \left\|U\left(U^\top A_t B_t V - S^*\right)\right\|_{op} + \left\|U_\perp U_\perp^\top A_t B_t V\right\|_{op}$$

$$\leq \underbrace{\left\|U^\top A_t B_t V - S^*\right\|_{op}}_{\text{signal space}} + \underbrace{\left\|U_\perp^\top A_t B_t V\right\|_{op}}_{\text{complementary}}, \tag{32}$$

where the first term denotes the loss in the signal space $\|U^\top A B V - S^*\|_{op}$ and the second term denotes the complementary space decay $\|U_\perp^\top A B V\|_{op}$. Next, we need a new parametrization to track the dynamics of these two terms. Recall the complete SVD of $\Delta$ in Eq. (1) as

$$\Delta = \widetilde{U}\widetilde{S}^*\widetilde{V}^\top = \begin{bmatrix} U & U_\perp \end{bmatrix}\begin{bmatrix} S^* & 0_{r^* \times (d-r^*)} \\ 0_{(d-r^*) \times r^*} & 0_{(d-r^*) \times (d-r^*)} \end{bmatrix}\begin{bmatrix} V^\top \\ V_\perp^\top \end{bmatrix}.$$

For notational simplicity, we denote

$$A_t^U := U^\top A_t, \quad A_t^{U_\perp} := U_\perp^\top A_t, \quad B_t V := B_t^V, \quad B_t V_\perp := B_t^{V_\perp}.$$

and thus

$$R_t := (A_t B_t - \Delta)V, \quad R_t^* := A_t^U B_t^V - S^*, \quad R_t^\perp := A_t^{U_\perp} B_t^V.$$

Accordingly, Eq. (32) can be reformulated as $\|R_t\|_{op} \leq \|R_t^*\|_{op} + \|R_t^\perp\|_{op}$. By Lemma C.12, we have $B^{V_\perp} = 0_{r \times (k-r^*)}$ $\forall t \in \mathbb{N}^+$. Next, we can track $R_t^*$ and $R_t^\perp$ via the following two lemmas.

**Lemma C.13.** *Under assumptions in Section 2.1 for the linear setting, with spectral initialization (Spectral-init), we have the following reparametrized iterates*

$$A_{t+1}^U = A_t^U - \eta R_t^* \left(B_t^V\right)^\top - \eta U^\top \left(\widehat{\Sigma} - I_d\right) R_t \left(B_t^V\right)^\top, \tag{33}$$

$$A_{t+1}^{U_\perp} = A_t^{U_\perp} - \eta R_t^\perp \left(B_t^V\right)^\top - \eta U_\perp^\top \left(\widehat{\Sigma} - I_d\right) R_t \left(B_t^V\right)^\top, \tag{34}$$

$$B_{t+1}^V = B_t^V - \eta \left(A_t^U\right)^\top R_t^* - \eta \left(A_t^{U_\perp}\right)^\top R_t^\perp$$

$$- \eta \left(A_t^U\right)^\top U^\top \left(\widehat{\Sigma} - I_d\right) R_t - \eta \left(A_t^{U_\perp}\right)^\top U_\perp^\top \left(\widehat{\Sigma} - I_d\right) R_t. \tag{35}$$

*Proof.* Recall the gradient update for $A_{t+1}$, we have

$$A_{t+1} = A_t - \eta\widehat{\Sigma}\left(A_t B_t - \Delta\right)\left(B_t\right)^\top$$

$$= A_t - \eta\left(A_t B_t - \Delta\right)\left(B_t\right)^\top - \eta\left(\widehat{\Sigma} - I_d\right)\left(A_t B_t - \Delta\right)\left(B_t\right)^\top.$$

Recall $R_t := (A_t B_t - \Delta)V$ and $\Delta = U S^* V^\top$, we have

$$U^\top A_{t+1} = U^\top A_t - \eta U^\top \left(A_t B_t - \Delta\right)\left(VV^\top + V_\perp V_\perp^\top\right)\left(B_t\right)^\top$$

$$- \eta U^\top \left(\widehat{\Sigma} - I_d\right)\left(A_t B_t - \Delta\right)\left(VV^\top + V_\perp V_\perp^\top\right)\left(B_t\right)^\top$$

$$= U^\top A_t - \eta U^\top \left(A_t B_t V - \Delta V\right)\left(B_t V\right)^\top - \eta U^\top \left(\widehat{\Sigma} - I_d\right)\left(A_t B_t V - \Delta V\right)\left(B_t V\right)^\top$$

$$\text{[by Lemma C.12]}$$

$$= U^\top A_t - \eta \left(U^\top A_t B_t V - S^*\right)\left(B_t V\right)^\top - \eta U^\top \left(\widehat{\Sigma} - I_d\right) R_t \left(B_t V\right)^\top.$$

Accordingly, the recursion for $A_{t+1}^U$ is reformulated as

$$A_{t+1}^U = A_t^U - \eta R_t^* \left(B_t^V\right)^\top - \eta U^\top \left(\widehat{\Sigma} - I_d\right) R_t \left(B_t^V\right)^\top .$$

Similarly, we can obtain

$$A_{t+1}^{U_\perp} = A_t^{U_\perp} - \eta R_t^\perp \left(B_t^V\right)^\top - \eta U_\perp^\top \left(\widehat{\Sigma} - I_d\right) R_t \left(B_t^V\right)^\top .$$

Regarding the recursion for $B_{t+1}$, we can derive in a similar way

$$
\begin{aligned}
B_{t+1}V &= B_t V - \eta \left(A_t\right)^\top \widehat{\Sigma} \left(A_t B_t - \Delta\right) V \\
&= B_t V - \eta \left(A_t\right)^\top \left(UU^\top + U_\perp U_\perp^\top\right)\left(A_t B_t - \Delta\right) V \\
&\quad - \eta \left(A_t\right)^\top \left(UU^\top + U_\perp U_\perp^\top\right)\left(\widehat{\Sigma} - I_d\right)\left(A_t B_t - \Delta\right) V ,
\end{aligned}
$$

which implies

$$B_{t+1}^V = B_t^V - \eta \left(A_t^U\right)^\top R_t^* - \eta \left(A_t^{U_\perp}\right)^\top R_t^\perp - \eta \left(A_t^U\right)^\top U^\top \left(\widehat{\Sigma} - I_d\right) R_t - \eta \left(A_t^{U_\perp}\right)^\top U_\perp^\top \left(\widehat{\Sigma} - I_d\right) R_t .$$

$\square$

In the next, we are able to characterize the upper bound of $\left\|R_{t+1}^*\right\|_{op}$.

**Lemma C.14.** *Denote* $\mathcal{M}_t := \max\left\{\left\|R_t^*\right\|_{op}, \left\|R_t^\perp\right\|_{op}\right\}$, *under assumptions in Section 2.1 for the linear setting, with spectral initialization (Spectral-init), then we choose $\epsilon$ with probability at least $1 - 2C\exp(-\epsilon^2 N)$ for a universal constant $C > 0$, we have*

$$
\begin{aligned}
\left\|R_{t+1}^*\right\|_{op} \leq{}& \left(1 - \eta\left(\lambda_{r^*}^2\left(A_t^U\right) + \lambda_{r^*}^2\left(B_t^V\right)\right)\right)\mathcal{M}_t \\
& + 2\eta\epsilon\left\|B_t^V\right\|_{op}^2\mathcal{M}_t + \eta^2\left\|A_t^U\right\|_{op}\left\|B_t^V\right\|_{op}\mathcal{M}_t^2 + 2\eta^2\epsilon\left\|A_t^U\right\|_{op}\left\|B_t^V\right\|_{op}\mathcal{M}_t^2 \\
& + \eta\left\|A_t^U\right\|_{op}\left\|A_t^{U_\perp}\right\|_{op}\mathcal{M}_t + \eta^2\left\|B_t^V\right\|_{op}\left\|A_t^{U_\perp}\right\|_{op}\mathcal{M}_t^2 \\
& + 2\eta^2\epsilon\left\|B_t^V\right\|_{op}\left\|A_t^{U_\perp}\right\|_{op}\mathcal{M}_t^2 + 2\eta\epsilon\left\|A_t^U\right\|_{op}^2\mathcal{M}_t \\
& + 2\eta^2\epsilon\left\|A_t^U\right\|_{op}\left\|B_t^V\right\|_{op}\mathcal{M}_t + 4\eta^2\epsilon^2\left\|A_t^U\right\|_{op}\left\|B_t^V\right\|_{op}\mathcal{M}_t^2 \\
& + 2\eta\epsilon\left\|A_t^U\right\|_{op}\left\|A_t^{U_\perp}\right\|_{op}\mathcal{M}_t + 2\eta^2\epsilon\left\|A_t^{U_\perp}\right\|_{op}\left\|B_t^V\right\|_{op}\mathcal{M}_t^2 \\
& + 4\eta^2\epsilon^2\left\|A_t^{U_\perp}\right\|_{op}\left\|B_t^V\right\|_{op}\mathcal{M}_t^2 ,
\end{aligned}
$$

*and*

$$
\begin{aligned}
\left\|R_{t+1}^\perp\right\|_{op} \leq{}& \left(1 - \eta\left(\lambda_{\min}^2\left(A_t^{U_\perp}\right) + \lambda_{r^*}^2\left(B_t^V\right)\right)\right)\mathcal{M}_t \\
& + 2\eta\epsilon\left\|B_t^V\right\|_{op}^2\mathcal{M}_t + \eta^2\left\|A_t^U\right\|_{op}\left\|B_t^V\right\|_{op}\mathcal{M}_t^2 + 2\eta^2\epsilon\left\|A_t^U\right\|_{op}\left\|B_t^V\right\|_{op}\mathcal{M}_t^2 \\
& + \eta\left\|A_t^U\right\|_{op}\left\|A_t^{U_\perp}\right\|_{op}\mathcal{M}_t + \eta^2\left\|B_t^V\right\|_{op}\left\|A_t^{U_\perp}\right\|_{op}\mathcal{M}_t^2 \\
& + 2\eta^2\epsilon\left\|B_t^V\right\|_{op}\left\|A_t^{U_\perp}\right\|_{op}\mathcal{M}_t^2 + 2\eta\epsilon\left\|A_t^U\right\|_{op}\left\|A_t^{U_\perp}\right\|_{op}\mathcal{M}_t \\
& + 2\eta^2\epsilon\left\|A_t^U\right\|_{op}\left\|B_t^V\right\|_{op}\mathcal{M}_t + 4\eta^2\epsilon^2\left\|A_t^U\right\|_{op}\left\|B_t^V\right\|_{op}\mathcal{M}_t^2 \\
& + 2\eta\epsilon\left\|A_t^{U_\perp}\right\|_{op}^2\mathcal{M}_t + 2\eta^2\epsilon\left\|A_t^{U_\perp}\right\|_{op}\left\|B_t^V\right\|_{op}\mathcal{M}_t^2 \\
& + 4\eta^2\epsilon^2\left\|A_t^{U_\perp}\right\|_{op}\left\|B_t^V\right\|_{op}\mathcal{M}_t^2 .
\end{aligned}
\tag{36}
$$

*Proof.* Here we first track the dynamics of $\boldsymbol{R}_t^*$. We have

$$
\begin{aligned}
\boldsymbol{R}_{t+1}^* = {}& \boldsymbol{A}_{t+1}^U \boldsymbol{B}_{t+1}^V - \boldsymbol{S}^* \\
= {}& \boldsymbol{R}_t^* - \eta \boldsymbol{R}_t^* \left(\boldsymbol{B}_t^V\right)^\top \boldsymbol{B}_t^V - \eta \boldsymbol{U}^\top \left(\widehat{\boldsymbol{\Sigma}} - \boldsymbol{I}_d\right) \boldsymbol{R}_t \left(\boldsymbol{B}_t^V\right)^\top \boldsymbol{B}_t^V \\
& - \eta \boldsymbol{A}_t^U \left(\boldsymbol{A}_t^U\right)^\top \boldsymbol{R}_t^* + \eta^2 \boldsymbol{R}_t^* \left(\boldsymbol{B}_t^V\right)^\top \left(\boldsymbol{A}_t^U\right)^\top \boldsymbol{R}_t^* + \eta^2 \boldsymbol{U}^\top \left(\widehat{\boldsymbol{\Sigma}} - \boldsymbol{I}_d\right) \boldsymbol{R}_t \left(\boldsymbol{B}_t^V\right)^\top \left(\boldsymbol{A}_t^U\right)^\top \boldsymbol{R}_t^* \\
& - \eta \boldsymbol{A}_t^U \left(\boldsymbol{A}_t^{U\perp}\right)^\top \boldsymbol{R}_t^\perp + \eta^2 \boldsymbol{R}_t^* \left(\boldsymbol{B}_t^V\right)^\top \left(\boldsymbol{A}_t^{U\perp}\right)^\top \boldsymbol{R}_t^\perp + \eta^2 \boldsymbol{U}^\top \left(\widehat{\boldsymbol{\Sigma}} - \boldsymbol{I}_d\right) \boldsymbol{R}_t \left(\boldsymbol{B}_t^V\right)^\top \left(\boldsymbol{A}_t^{U\perp}\right)^\top \boldsymbol{R}_t^\perp \\
& \text{------} \\
& - \eta \boldsymbol{A}_t^U \left(\boldsymbol{A}_t^U\right)^\top \boldsymbol{U}^\top \left(\widehat{\boldsymbol{\Sigma}} - \boldsymbol{I}_d\right) \boldsymbol{R}_t \\
& + \eta^2 \boldsymbol{R}_t^* \left(\boldsymbol{B}_t^V\right)^\top \left(\boldsymbol{A}_t^U\right)^\top \boldsymbol{U}^\top \left(\widehat{\boldsymbol{\Sigma}} - \boldsymbol{I}_d\right) \boldsymbol{R}_t \\
& + \eta^2 \boldsymbol{U}^\top \left(\widehat{\boldsymbol{\Sigma}} - \boldsymbol{I}_d\right) \boldsymbol{R}_t \left(\boldsymbol{B}_t^V\right)^\top \left(\boldsymbol{A}_t^U\right)^\top \boldsymbol{U}^\top \left(\widehat{\boldsymbol{\Sigma}} - \boldsymbol{I}_d\right) \boldsymbol{R}_t \\
& \text{------} \\
& - \eta \boldsymbol{A}_t^U \left(\boldsymbol{A}_t^{U\perp}\right)^\top \boldsymbol{U}_\perp^\top \left(\widehat{\boldsymbol{\Sigma}} - \boldsymbol{I}_d\right) \boldsymbol{R}_t \\
& + \eta^2 \boldsymbol{R}_t^* \left(\boldsymbol{B}_t^V\right)^\top \left(\boldsymbol{A}_t^{U\perp}\right)^\top \boldsymbol{U}_\perp^\top \left(\widehat{\boldsymbol{\Sigma}} - \boldsymbol{I}_d\right) \boldsymbol{R}_t \\
& + \eta^2 \boldsymbol{U}^\top \left(\widehat{\boldsymbol{\Sigma}} - \boldsymbol{I}_d\right) \boldsymbol{R}_t \left(\boldsymbol{B}_t^V\right)^\top \left(\boldsymbol{A}_t^{U\perp}\right)^\top \boldsymbol{U}_\perp^\top \left(\widehat{\boldsymbol{\Sigma}} - \boldsymbol{I}_d\right) \boldsymbol{R}_t \,.
\end{aligned}
$$

Then, we take operator norm over the above equation. Hence, with probability at least $1 - 2C \exp(-\epsilon^2 N)$ for a universal constant $C > 0$, we have

$$
\begin{aligned}
\left\|\boldsymbol{R}_{t+1}^*\right\|_{op} \leq {}& \left(1 - \eta \left(\lambda_{r^*}^2 \left(\boldsymbol{A}_t^U\right) + \lambda_{r^*}^2 \left(\boldsymbol{B}_t^V\right)\right)\right) \left\|\boldsymbol{R}_t^*\right\|_{op} \\
& + \eta \epsilon \left\|\boldsymbol{B}_t^V\right\|_{op}^2 \left\|\boldsymbol{R}_t\right\|_{op} + \eta^2 \left\|\boldsymbol{A}_t^U\right\|_{op} \left\|\boldsymbol{B}_t^V\right\|_{op} \left\|\boldsymbol{R}_t^*\right\|_{op}^2 + \eta^2 \epsilon \left\|\boldsymbol{A}_t^U\right\|_{op} \left\|\boldsymbol{B}_t^V\right\|_{op} \left\|\boldsymbol{R}_t^*\right\|_{op} \left\|\boldsymbol{R}_t\right\|_{op} \\
& + \eta \left\|\boldsymbol{A}_t^U\right\|_{op} \left\|\boldsymbol{A}_t^{U\perp}\right\|_{op} \left\|\boldsymbol{R}_t^\perp\right\|_{op} + \eta^2 \left\|\boldsymbol{B}_t^V\right\|_{op} \left\|\boldsymbol{A}_t^{U\perp}\right\|_{op} \left\|\boldsymbol{R}_t^\perp\right\|_{op} \left\|\boldsymbol{R}^* t\right\|_{op} \\
& + \eta^2 \epsilon \left\|\boldsymbol{B}_t^V\right\|_{op} \left\|\boldsymbol{A}_t^{U\perp}\right\|_{op} \left\|\boldsymbol{R}_t^\perp\right\|_{op} \left\|\boldsymbol{R}_t\right\|_{op} + \eta \epsilon \left\|\boldsymbol{A}_t^U\right\|_{op}^2 \left\|\boldsymbol{R}_t\right\|_{op} \\
& + \eta^2 \epsilon \left\|\boldsymbol{A}_t^U\right\|_{op} \left\|\boldsymbol{B}_t^V\right\|_{op} \left\|\boldsymbol{R}_t\right\|_{op} + \eta^2 \epsilon^2 \left\|\boldsymbol{A}_t^U\right\|_{op} \left\|\boldsymbol{B}_t^V\right\|_{op} \left\|\boldsymbol{R}_t\right\|_{op}^2 \\
& + \eta \epsilon \left\|\boldsymbol{A}_t^U\right\|_{op} \left\|\boldsymbol{A}_t^{U\perp}\right\|_{op} \left\|\boldsymbol{R}_t\right\|_{op} + \eta^2 \epsilon \left\|\boldsymbol{A}_t^{U\perp}\right\|_{op} \left\|\boldsymbol{B}_t^V\right\|_{op} \left\|\boldsymbol{R}_t^*\right\|_{op} \left\|\boldsymbol{R}_t\right\|_{op} \\
& + \eta^2 \epsilon^2 \left\|\boldsymbol{A}_t^{U\perp}\right\|_{op} \left\|\boldsymbol{B}_t^V\right\|_{op} \left\|\boldsymbol{R}_t\right\|_{op}^2 \,.
\end{aligned}
$$

Next, we take maximum over $\left\|\boldsymbol{R}_t^*\right\|_{op}$ and $\left\|\boldsymbol{R}_t^\perp\right\|_{op}$ on the right hand side above. Recall $\mathcal{M}_t = \max\left\{\left\|\boldsymbol{R}_t^*\right\|_{op}, \left\|\boldsymbol{R}_t^\perp\right\|_{op}\right\}$,

using the fact that $\|\boldsymbol{R}_t\|_{op} \le 2\mathcal{M}_t$, we have:

$$
\begin{aligned}
\left\|\boldsymbol{R}_{t+1}^*\right\|_{op} \le {}& \left(1 - \eta\left(\lambda_{r^*}^2\left(\boldsymbol{A}_t^{\boldsymbol{U}}\right) + \lambda_{r^*}^2\left(\boldsymbol{B}_t^{\boldsymbol{V}}\right)\right)\right)\mathcal{M}_t \\
& + 2\eta\epsilon\left\|\boldsymbol{B}_t^{\boldsymbol{V}}\right\|_{op}^2\mathcal{M}_t + \eta^2\left\|\boldsymbol{A}_t^{\boldsymbol{U}}\right\|_{op}\left\|\boldsymbol{B}_t^{\boldsymbol{V}}\right\|_{op}\mathcal{M}_t^2 + 2\eta^2\epsilon\left\|\boldsymbol{A}_t^{\boldsymbol{U}}\right\|_{op}\left\|\boldsymbol{B}_t^{\boldsymbol{V}}\right\|_{op}\mathcal{M}_t^2 \\
& + \eta\left\|\boldsymbol{A}_t^{\boldsymbol{U}}\right\|_{op}\left\|\boldsymbol{A}_t^{\boldsymbol{U}_\perp}\right\|_{op}\mathcal{M}_t + \eta^2\left\|\boldsymbol{B}_t^{\boldsymbol{V}}\right\|_{op}\left\|\boldsymbol{A}_t^{\boldsymbol{U}_\perp}\right\|_{op}\mathcal{M}_t^2 \\
& + 2\eta^2\epsilon\left\|\boldsymbol{B}_t^{\boldsymbol{V}}\right\|_{op}\left\|\boldsymbol{A}_t^{\boldsymbol{U}_\perp}\right\|_{op}\mathcal{M}_t^2 + 2\eta\epsilon\left\|\boldsymbol{A}_t^{\boldsymbol{U}}\right\|_{op}^2\mathcal{M}_t \\
& + 2\eta^2\epsilon\left\|\boldsymbol{A}_t^{\boldsymbol{U}}\right\|_{op}\left\|\boldsymbol{B}_t^{\boldsymbol{V}}\right\|_{op}\mathcal{M}_t + 4\eta^2\epsilon^2\left\|\boldsymbol{A}_t^{\boldsymbol{U}}\right\|_{op}\left\|\boldsymbol{B}_t^{\boldsymbol{V}}\right\|_{op}\mathcal{M}_t^2 \\
& + 2\eta\epsilon\left\|\boldsymbol{A}_t^{\boldsymbol{U}}\right\|_{op}\left\|\boldsymbol{A}_t^{\boldsymbol{U}_\perp}\right\|_{op}\mathcal{M}_t + 2\eta^2\epsilon\left\|\boldsymbol{A}_t^{\boldsymbol{U}_\perp}\right\|_{op}\left\|\boldsymbol{B}_t^{\boldsymbol{V}}\right\|_{op}\mathcal{M}_t^2 \\
& + 4\eta^2\epsilon^2\left\|\boldsymbol{A}_t^{\boldsymbol{U}_\perp}\right\|_{op}\left\|\boldsymbol{B}_t^{\boldsymbol{V}}\right\|_{op}\mathcal{M}_t^2 .
\end{aligned}
$$

Next, we track the dynamics of $\boldsymbol{R}_t^\perp$. We have

$$
\begin{aligned}
\boldsymbol{R}_{t+1}^\perp = {}& \boldsymbol{A}_{t+1}^{\boldsymbol{U}_\perp}\boldsymbol{B}_{t+1}^{\boldsymbol{V}} \\
= {}& \boldsymbol{R}_t^\perp - \eta\boldsymbol{R}_t^\perp\left(\boldsymbol{B}_t^{\boldsymbol{V}}\right)^\top\boldsymbol{B}_t^{\boldsymbol{V}} - \eta\boldsymbol{U}_\perp^\top\left(\widehat{\boldsymbol{\Sigma}} - \boldsymbol{I}_d\right)\boldsymbol{R}_t\left(\boldsymbol{B}_t^{\boldsymbol{V}}\right)^\top\boldsymbol{B}_t^{\boldsymbol{V}} \\
& - \eta\boldsymbol{A}_t^{\boldsymbol{U}_\perp}\left(\boldsymbol{A}_t^{\boldsymbol{U}}\right)^\top\boldsymbol{R}_t^* + \eta^2\boldsymbol{R}_t^\perp\left(\boldsymbol{B}_t^{\boldsymbol{V}}\right)^\top\left(\boldsymbol{A}_t^{\boldsymbol{U}}\right)^\top\boldsymbol{R}_t^* + \eta^2\boldsymbol{U}_\perp^\top\left(\widehat{\boldsymbol{\Sigma}} - \boldsymbol{I}_d\right)\boldsymbol{R}_t\left(\boldsymbol{B}_t^{\boldsymbol{V}}\right)^\top\left(\boldsymbol{A}_t^{\boldsymbol{U}}\right)^\top\boldsymbol{R}_t^* \\
& - \eta\boldsymbol{A}_t^{\boldsymbol{U}_\perp}\left(\boldsymbol{A}_t^{\boldsymbol{U}_\perp}\right)^\top\boldsymbol{R}_t^\perp + \eta^2\boldsymbol{R}_t^\perp\left(\boldsymbol{B}_t^{\boldsymbol{V}}\right)^\top\left(\boldsymbol{A}_t^{\boldsymbol{U}_\perp}\right)^\top\boldsymbol{R}_t^\perp + \eta^2\boldsymbol{U}_\perp^\top\left(\widehat{\boldsymbol{\Sigma}} - \boldsymbol{I}_d\right)\boldsymbol{R}_t\left(\boldsymbol{B}_t^{\boldsymbol{V}}\right)^\top\left(\boldsymbol{A}_t^{\boldsymbol{U}_\perp}\right)^\top\boldsymbol{R}_t^\perp \\
& - \eta\boldsymbol{A}_t^{\boldsymbol{U}_\perp}\left(\boldsymbol{A}_t^{\boldsymbol{U}}\right)^\top\boldsymbol{U}^\top\left(\widehat{\boldsymbol{\Sigma}} - \boldsymbol{I}_d\right)\boldsymbol{R}_t \\
& + \eta^2\boldsymbol{R}_t^\perp\left(\boldsymbol{B}_t^{\boldsymbol{V}}\right)^\top\left(\boldsymbol{A}_t^{\boldsymbol{U}}\right)^\top\boldsymbol{U}^\top\left(\widehat{\boldsymbol{\Sigma}} - \boldsymbol{I}_d\right)\boldsymbol{R}_t \\
& + \eta^2\boldsymbol{U}_\perp^\top\left(\widehat{\boldsymbol{\Sigma}} - \boldsymbol{I}_d\right)\boldsymbol{R}_t\left(\boldsymbol{B}_t^{\boldsymbol{V}}\right)^\top\left(\boldsymbol{A}_t^{\boldsymbol{U}}\right)^\top\boldsymbol{U}^\top\left(\widehat{\boldsymbol{\Sigma}} - \boldsymbol{I}_d\right)\boldsymbol{R}_t \\
& - \eta\boldsymbol{A}_t^{\boldsymbol{U}_\perp}\left(\boldsymbol{A}_t^{\boldsymbol{U}_\perp}\right)^\top\boldsymbol{U}_\perp^\top\left(\widehat{\boldsymbol{\Sigma}} - \boldsymbol{I}_d\right)\boldsymbol{R}_t \\
& + \eta^2\boldsymbol{R}_t^\perp\left(\boldsymbol{B}_t^{\boldsymbol{V}}\right)^\top\left(\boldsymbol{A}_t^{\boldsymbol{U}_\perp}\right)^\top\boldsymbol{U}_\perp^\top\left(\widehat{\boldsymbol{\Sigma}} - \boldsymbol{I}_d\right)\boldsymbol{R}_t \\
& + \eta^2\boldsymbol{U}_\perp^\top\left(\widehat{\boldsymbol{\Sigma}} - \boldsymbol{I}_d\right)\boldsymbol{R}_t\left(\boldsymbol{B}_t^{\boldsymbol{V}}\right)^\top\left(\boldsymbol{A}_t^{\boldsymbol{U}_\perp}\right)^\top\boldsymbol{U}_\perp^\top\left(\widehat{\boldsymbol{\Sigma}} - \boldsymbol{I}_d\right)\boldsymbol{R}_t .
\end{aligned}
$$

Then, we take operator norm over the above equation. With probability at least $1 - 2C\exp(-\epsilon^2 N)$ for a universal constant $C > 0$, we have

$$
\begin{aligned}
\left\|\boldsymbol{R}_{t+1}^\perp\right\|_{op} \le {}& \left(1 - \eta\left(\lambda_{\min}^2\left(\boldsymbol{A}_t^{\boldsymbol{U}_\perp}\right) + \lambda_{r^*}^2\left(\boldsymbol{B}_t^{\boldsymbol{V}}\right)\right)\right)\left\|\boldsymbol{R}_t^\perp\right\|_{op} \\
& + \eta\epsilon\left\|\boldsymbol{B}_t^{\boldsymbol{V}}\right\|_{op}^2\left\|\boldsymbol{R}_t\right\|_{op} + \eta^2\left\|\boldsymbol{A}_t^{\boldsymbol{U}}\right\|_{op}\left\|\boldsymbol{B}_t^{\boldsymbol{V}}\right\|_{op}\left\|\boldsymbol{R}_t^*\right\|_{op}\left\|\boldsymbol{R}_t^\perp\right\|_{op} + \eta^2\epsilon\left\|\boldsymbol{A}_t^{\boldsymbol{U}}\right\|_{op}\left\|\boldsymbol{B}_t^{\boldsymbol{V}}\right\|_{op}\left\|\boldsymbol{R}_t^*\right\|_{op}\left\|\boldsymbol{R}_t\right\|_{op} \\
& + \eta\left\|\boldsymbol{A}_t^{\boldsymbol{U}}\right\|_{op}\left\|\boldsymbol{A}_t^{\boldsymbol{U}_\perp}\right\|_{op}\left\|\boldsymbol{R}_t^*\right\|_{op} + \eta^2\left\|\boldsymbol{B}_t^{\boldsymbol{V}}\right\|_{op}\left\|\boldsymbol{A}_t^{\boldsymbol{U}_\perp}\right\|_{op}\left\|\boldsymbol{R}_t^\perp\right\|_{op}^2 \\
& + \eta^2\epsilon\left\|\boldsymbol{B}_t^{\boldsymbol{V}}\right\|_{op}\left\|\boldsymbol{A}_t^{\boldsymbol{U}_\perp}\right\|_{op}\left\|\boldsymbol{R}_t^\perp\right\|_{op}\left\|\boldsymbol{R}_t\right\|_{op} + \eta\epsilon\left\|\boldsymbol{A}_t^{\boldsymbol{U}}\right\|_{op}\left\|\boldsymbol{A}_t^{\boldsymbol{U}_\perp}\right\|_{op}\left\|\boldsymbol{R}_t\right\|_{op} \\
& + \eta^2\epsilon\left\|\boldsymbol{A}_t^{\boldsymbol{U}}\right\|_{op}\left\|\boldsymbol{B}_t^{\boldsymbol{V}}\right\|_{op}\left\|\boldsymbol{R}_t\right\|_{op} + \eta^2\epsilon^2\left\|\boldsymbol{A}_t^{\boldsymbol{U}}\right\|_{op}\left\|\boldsymbol{B}_t^{\boldsymbol{V}}\right\|_{op}\left\|\boldsymbol{R}_t\right\|_{op}^2 \\
& + \eta\epsilon\left\|\boldsymbol{A}_t^{\boldsymbol{U}_\perp}\right\|_{op}^2\left\|\boldsymbol{R}_t\right\|_{op} + \eta^2\epsilon\left\|\boldsymbol{A}_t^{\boldsymbol{U}_\perp}\right\|_{op}\left\|\boldsymbol{B}_t^{\boldsymbol{V}}\right\|_{op}\left\|\boldsymbol{R}_t^\perp\right\|_{op}\left\|\boldsymbol{R}_t\right\|_{op} \\
& + \eta^2\epsilon^2\left\|\boldsymbol{A}_t^{\boldsymbol{U}_\perp}\right\|_{op}\left\|\boldsymbol{B}_t^{\boldsymbol{V}}\right\|_{op}\left\|\boldsymbol{R}_t\right\|_{op}^2 .
\end{aligned}
$$

Next, we take maximum over $\|\boldsymbol{R}_t^*\|_{op}$ and $\|\boldsymbol{R}_t^\perp\|_{op}$ on the right hand side above. Recall $\mathcal{M}_t = \max\left\{\|\boldsymbol{R}_t^*\|_{op}, \|\boldsymbol{R}_t^\perp\|_{op}\right\}$, using the fact that $\|\boldsymbol{R}_t\|_{op} \leq 2\mathcal{M}_t$, we have:

$$
\begin{aligned}
\left\|\boldsymbol{R}_{t+1}^\perp\right\|_{op} \leq & \left(1 - \eta\left(\lambda_{\min}^2\left(\boldsymbol{A}_t^{U_\perp}\right) + \lambda_{r^*}^2\left(\boldsymbol{B}_t^V\right)\right)\right)\mathcal{M}_t \\
& + 2\eta\epsilon\left\|\boldsymbol{B}_t^V\right\|_{op}^2\mathcal{M}_t + \eta^2\left\|\boldsymbol{A}_t^U\right\|_{op}\left\|\boldsymbol{B}_t^V\right\|_{op}\mathcal{M}_t^2 + 2\eta^2\epsilon\left\|\boldsymbol{A}_t^U\right\|_{op}\left\|\boldsymbol{B}_t^V\right\|_{op}\mathcal{M}_t^2 \\
& + \eta\left\|\boldsymbol{A}_t^U\right\|_{op}\left\|\boldsymbol{A}_t^{U_\perp}\right\|_{op}\mathcal{M}_t + \eta^2\left\|\boldsymbol{B}_t^V\right\|_{op}\left\|\boldsymbol{A}_t^{U_\perp}\right\|_{op}\mathcal{M}_t^2 \\
& + 2\eta^2\epsilon\left\|\boldsymbol{B}_t^V\right\|_{op}\left\|\boldsymbol{A}_t^{U_\perp}\right\|_{op}\mathcal{M}_t^2 + 2\eta\epsilon\left\|\boldsymbol{A}_t^U\right\|_{op}\left\|\boldsymbol{A}_t^{U_\perp}\right\|_{op}\mathcal{M}_t \\
& + 2\eta^2\epsilon\left\|\boldsymbol{A}_t^U\right\|_{op}\left\|\boldsymbol{B}_t^V\right\|_{op}\mathcal{M}_t + 4\eta^2\epsilon^2\left\|\boldsymbol{A}_t^U\right\|_{op}\left\|\boldsymbol{B}_t^V\right\|_{op}\mathcal{M}_t^2 \\
& + 2\eta\epsilon\left\|\boldsymbol{A}_t^{U_\perp}\right\|_{op}^2\mathcal{M}_t + 2\eta^2\epsilon\left\|\boldsymbol{A}_t^{U_\perp}\right\|_{op}\left\|\boldsymbol{B}_t^V\right\|_{op}\mathcal{M}_t^2 \\
& + 4\eta^2\epsilon^2\left\|\boldsymbol{A}_t^{U_\perp}\right\|_{op}\left\|\boldsymbol{B}_t^V\right\|_{op}\mathcal{M}_t^2.
\end{aligned}
$$

Finally we conclude the proof. $\qquad\square$

Before we move to the main proof, we need to establish a strict upper bound on $\boldsymbol{A}_t$ and $\boldsymbol{B}_t$.

**Lemma C.15.** *Under assumptions in Section 2.1 for the linear setting, suppose $\left\|\boldsymbol{A}_t^\top\boldsymbol{A}_t - \boldsymbol{B}_t^\top\boldsymbol{B}_t\right\|_{op} + \epsilon\|\boldsymbol{R}_t\|_{op} \leq \lambda_1^*$ and $\eta \leq \frac{1}{10\lambda_1^*}$, if $\|\boldsymbol{A}_t\|_{op} \leq 2\sqrt{\lambda_1^*}$ and $\|\boldsymbol{B}_t\|_{op} \leq 2\sqrt{\lambda_1^*}$, we choose $\epsilon$ satisfying Eq. (29), then with probability $1 - 2C\exp(-\epsilon^2 N)$ for a universal constant $C > 0$, we have*

$$
\|\boldsymbol{A}_{t+1}\|_{op} \leq 2\sqrt{\lambda_1^*}, \quad \|\boldsymbol{B}_{t+1}\|_{op} \leq 2\sqrt{\lambda_1^*}.
$$

*Proof.* Inspired by Soltanolkotabi et al. (2023), we recall the stacked iterate $\boldsymbol{Z}_t$ defined in Eq. (12) and construct an anti-iterate

$$
\underline{\boldsymbol{Z}}_t := \begin{bmatrix} \boldsymbol{A}_t \\ -\boldsymbol{B}_t^\top \end{bmatrix}.
$$

Additionally, we define a perturbation matrix

$$
\boldsymbol{\Xi}_t := \begin{bmatrix} \boldsymbol{0}_{d\times d} & \left(\widetilde{\boldsymbol{\Sigma}} - \boldsymbol{I}_d\right)\boldsymbol{R}_t \\ \boldsymbol{R}_t^\top\left(\widetilde{\boldsymbol{\Sigma}} - \boldsymbol{I}_d\right) & \boldsymbol{0}_{k\times k} \end{bmatrix}.
$$

Then, we can reformulate the recursion of $\boldsymbol{Z}_{t+1}$ as

$$
\begin{aligned}
\boldsymbol{Z}_{t+1} &= \boldsymbol{Z}_t - \eta\left(\boldsymbol{Z}_t\boldsymbol{Z}_t^\top - \underline{\boldsymbol{Z}}_t\underline{\boldsymbol{Z}}_t^\top - \boldsymbol{\Gamma}\right)\boldsymbol{Z}_t + \eta\boldsymbol{\Xi}_t\boldsymbol{Z}_t \\
&= \left(\boldsymbol{I}_{2d} - \eta\boldsymbol{Z}_t\boldsymbol{Z}_t^\top\right)\boldsymbol{Z}_t + \eta\underline{\boldsymbol{Z}}_t\underline{\boldsymbol{Z}}_t^\top\boldsymbol{Z}_t - \eta\boldsymbol{\Gamma}\boldsymbol{Z}_t + \eta\boldsymbol{\Xi}_t\boldsymbol{Z}_t,
\end{aligned}
$$

where $\boldsymbol{\Gamma}$ is defined as

$$
\boldsymbol{\Gamma} := \begin{bmatrix} \boldsymbol{0}_{d\times d} & \Delta \\ \Delta^\top & \boldsymbol{0}_{k\times k} \end{bmatrix}.
$$

Then, by the triangle inequality, with probability $1 - 2C\exp(-\epsilon^2 N)$ for a universal constant $C > 0$, we have

$$
\begin{aligned}
\|\boldsymbol{Z}_{t+1}\|_{op} \leq & \left\|\left(\boldsymbol{I}_{2d} - \eta\boldsymbol{Z}_t\boldsymbol{Z}_t^\top\right)\boldsymbol{Z}_t\right\|_{op} + \eta\left\|\underline{\boldsymbol{Z}}_t\underline{\boldsymbol{Z}}_t^\top\boldsymbol{Z}_t\right\|_{op} + \eta\|\boldsymbol{\Gamma}\boldsymbol{Z}_t\|_{op} + \eta\|\boldsymbol{\Xi}_t\boldsymbol{Z}_t\|_{op} \\
\leq & \left(1 - \eta\|\boldsymbol{Z}_t\|_{op}^2\right)\|\boldsymbol{Z}_t\|_{op} \qquad\qquad\qquad\qquad \text{[by simultaneous diagonalization]} \\
& + \eta\left\|\underline{\boldsymbol{Z}}_t\underline{\boldsymbol{Z}}_t^\top\boldsymbol{Z}_t\right\|_{op} + \eta\|\boldsymbol{\Gamma}\boldsymbol{Z}_t\|_{op} + \eta\|\boldsymbol{\Xi}_t\boldsymbol{Z}_t\|_{op} \\
\leq & \left(1 - \eta\|\boldsymbol{Z}_t\|_{op}^2\right)\|\boldsymbol{Z}_t\|_{op} + \eta\left\|\underline{\boldsymbol{Z}}_t^\top\boldsymbol{Z}_t\right\|_{op}\|\boldsymbol{Z}_t\|_{op} + \eta\lambda_1^*\|\boldsymbol{Z}_t\|_{op} + \eta\epsilon\|\boldsymbol{R}_t\|_{op}\|\boldsymbol{Z}_t\|_{op},
\end{aligned}
$$

where the last inequality follows from the fact that

$$\|\underline{Z}_t\|_{op} = \|Z_t\|_{op} \,,$$
$$\|\Gamma\|_{op} = \lambda_1^* \,,$$
$$\|\Xi_t\|_{op} = \left\|\left(\widetilde{\Sigma} - I_d\right) R_t\right\|_{op} \leq \epsilon \|R_t\|_{op} \,, \quad \text{w.h.p. } 1 - 2C\exp(-\epsilon^2 N) \,.$$

Using the assumption

$$\left\|\underline{Z}_t^\top Z_t\right\|_{op} + \epsilon \|R_t\|_{op} = \left\|A_t^\top A_t - B_t^\top B_t\right\|_{op} + \epsilon \|R_t\|_{op} \leq \lambda_1^* \,,$$

then $\|Z_{t+1}\|_{op}$ can be further bounded by

$$\|Z_{t+1}\|_{op} \leq \left(1 - \eta \|Z_t\|_{op}^2 + 2\eta\lambda_1^*\right) \|Z_t\|_{op} \,. \tag{37}$$

Denote $x = \|Z_t\|_{op}$ and $f(x) = \left(1 - \eta x^2 + 2\eta\lambda_1^*\right) x$, we have $f'(x) = 1 + 2\eta\lambda_1^* - 3\eta x^2$ and $f''(x) = -6\eta x$. Then, we know $f'(x^*) = 0$ for $x > 0$ attained at $x^* = \sqrt{\frac{1+2\eta\lambda_1^*}{3\eta}} = \sqrt{\frac{1}{3\eta} + \frac{2}{3}\lambda_1^*}$. As we pick $\eta \leq \frac{1}{10\lambda_1^*}$, then $x^* \geq 2\sqrt{\lambda_1^*}$, which implies the maximum of $f(x)$ attained at $x^* = 2\sqrt{\lambda_1^*}$ over $x \in [0, 2\lambda_1^*]$ since $\|Z_t\|_{op} \leq 2\sqrt{\lambda_1^*}$ and

$$f(2\sqrt{\lambda_1^*}) = 2(1 - 4\eta\lambda_1^* + 2\eta\lambda_1^*)\sqrt{\lambda_1^*} = 2\sqrt{\lambda_1^*} - 4\eta\lambda_1^* \leq 2\sqrt{\lambda_1^*} \,,$$

which directly implies $\|Z_{t+1}\|_{op} \leq 2\sqrt{\lambda_1^*}$. By consequence, $\|A_{t+1}\|_{op}, \|B_{t+1}\|_{op} \leq 2\sqrt{\lambda_1^*}$ if $\|A_t\|_{op}, \|B_t\|_{op} \leq 2\sqrt{\lambda_1^*}$, since $A_{t+1}$ and $B_{t+1}$ are sub-matrices of $Z_{t+1}$. $\qquad\square$

Based on the above results, we are ready to present the intermediate results on $\mathcal{M}_t$, $A_t$, $B_t$, and $\left\|A_t^{U_\perp}\right\|_{op}$.

**Lemma C.16.** *Under assumptions in Section 2.1 for the linear setting, with spectral initialization (Spectral-init), we take $\epsilon$ in data concentration as*

$$\epsilon \leq \min\left\{\frac{1}{2\kappa}, \frac{\lambda_{r^*}^*}{32\kappa(32\lambda_1^* + 128\kappa^2)}\right\} \,,$$

*and set the step-size as*

$$\eta \leq \min\left\{\frac{1}{128\kappa\lambda_1^*}, \frac{(1 - \epsilon/\kappa)}{1152\lambda_1^*}\right\} \,,$$

*then with probability at least with probability $1 - 2C\exp(-\epsilon^2 N)$ for a universal constant $C > 0$, we have that $\forall\, t \geq 0$*

$$\mathcal{M}_t \leq \frac{\lambda_{r^*}^*}{2} \tag{38}$$

$$\max\left\{\|A_t\|_{op}, \|B_t\|_{op}\right\} \leq 2\sqrt{\lambda_1^*} \,, \tag{39}$$

$$\lambda_{r^*}^*(A_t), \lambda_{r^*}^*(B_t) \geq \frac{\sqrt{\lambda_{r^*}^*}}{4\sqrt{\kappa}} \,, \tag{40}$$

$$\left\|A_t^{U_\perp}\right\|_{op} \leq \frac{32\kappa\epsilon\sqrt{\lambda_1^*}}{\lambda_{r^*}^*} \,. \tag{41}$$

*Also, we can obtain*

$$\mathcal{M}_{t+1} \leq \left(1 - \eta\frac{\lambda_{r^*}^*}{64\kappa}\right)\mathcal{M}_t \,. \tag{42}$$

*Proof.* Inspired by the matrix sensing technique from Xiong et al. (2024), we develop an inductive approach to prove the claims on our settings. First, at $t = 0$, Eq. (38)-Eq. (41) can be adopted from Lemma C.11. Next, we assume Eq. (38)-Eq. (41) hold at $t \geq 1$, recall Eq. (34), by the triangle inequality, we have

$$
\begin{aligned}
\left\| \boldsymbol{A}_{t+1}^{\boldsymbol{U}_\perp} \right\|_{op} &\leq \left( 1 - \eta \lambda_{r^*}^2 \left( \boldsymbol{B}_t^{\boldsymbol{V}} \right) \right) \left\| \boldsymbol{A}_t^{\boldsymbol{U}_\perp} \right\|_{op} + \eta \epsilon \left\| \boldsymbol{R}_t \right\|_{op} \left\| \boldsymbol{B}_t^{\boldsymbol{V}} \right\|_{op} \\
&\leq \left( 1 - \eta \lambda_{r^*}^2 \left( \boldsymbol{B}_t^{\boldsymbol{V}} \right) \right) \left\| \boldsymbol{A}_t^{\boldsymbol{U}_\perp} \right\|_{op} + 4 \eta \epsilon \mathcal{M}_t \sqrt{\lambda_1^*} \\
&\leq \left( 1 - \eta \frac{(\lambda_{r^*}^*)^2}{16\kappa} \right) \left\| \boldsymbol{A}_t^{\boldsymbol{U}_\perp} \right\|_{op} + 2 \eta \epsilon \lambda_{r^*}^* \sqrt{\lambda_1^*} \\
&\leq \left( 1 - \eta \frac{(\lambda_{r^*}^*)^2}{16\kappa} \right) \frac{32 \kappa \epsilon \sqrt{\lambda_1^*}}{\lambda_{r^*}^*} + 2 \eta \epsilon \lambda_{r^*}^* \sqrt{\lambda_1^*} \\
&\leq \frac{32 \kappa \epsilon \sqrt{\lambda_1^*}}{\lambda_{r^*}^*},
\end{aligned}
$$

which proves the Eq. (41) at $t + 1$. Next, by Lemma C.14, we have

$$
\begin{aligned}
\left\| \boldsymbol{R}_{t+1}^* \right\|_{op} &\leq \left( 1 - \eta \frac{\lambda_{r^*}^*}{8\kappa} \right) \mathcal{M}_t \\
&\quad + 8 \eta \epsilon \lambda_1^* \mathcal{M}_t + 2 \eta^2 \lambda_1^* \lambda_{r^*}^* \mathcal{M}_t + 4 \eta^2 \epsilon \lambda_1^* \lambda_{r^*}^* \mathcal{M}_t + 64 \eta \epsilon \kappa^2 \mathcal{M}_t + 32 \eta^2 \kappa^2 \lambda_{r^*}^* \mathcal{M}_t + 128 \eta^2 \epsilon^3 \kappa^2 \lambda_{r^*}^* \mathcal{M}_t \\
&\quad + 64 \eta^2 \epsilon^2 \kappa^2 \lambda_{r^*}^* \mathcal{M}_t + 8 \eta \epsilon \lambda_1^* \mathcal{M}_t + 8 \eta^2 \epsilon \lambda_1^* \mathcal{M}_t + 8 \eta^2 \epsilon^2 \lambda_1^* \lambda_{r^*}^* \mathcal{M}_t + 128 \eta \epsilon^2 \kappa^2 \mathcal{M}_t + 64 \eta^2 \epsilon^2 \kappa^2 \lambda_{r^*}^* \mathcal{M}_t \\
&= \left( 1 - \eta \frac{\lambda_{r^*}^*}{8\kappa} \right) \mathcal{M}_t \\
&\quad + \eta \Bigg\{ 16 \epsilon \lambda_1^* + 64 \epsilon \kappa^2 + 2 \eta \lambda_1^* \lambda_{r^*}^* + \eta \epsilon \left( 4 \lambda_1^* \lambda_{r^*}^* + 32 \kappa^2 \lambda_{r^*}^* + 8 \lambda_1^* \right) + 128 \epsilon^2 \kappa^2 \\
&\qquad + \eta \left( 128 \eta \epsilon^2 \kappa^2 \lambda_{r^*}^* + 8 \eta \epsilon^2 \lambda_1^* \lambda_{r^*}^* \right) + 128 \eta \epsilon^3 \kappa^2 \lambda_{r^*}^* \Bigg\} \mathcal{M}_t \\
&\leq \left( 1 - \eta \frac{\lambda_{r^*}^*}{8\kappa} \right) \mathcal{M}_t + 2 \eta \left( 16 \epsilon \lambda_1^* + 64 \epsilon \kappa^2 + 2 \eta \lambda_1^* \lambda_{r^*}^* \right) \mathcal{M}_t && \text{[due to the order dominance]} \\
&\leq \left( 1 - \eta \frac{\lambda_{r^*}^*}{16\kappa} \right) \mathcal{M}_t + 2 \eta \left( 16 \epsilon \lambda_1^* + 64 \epsilon \kappa^2 \right) \mathcal{M}_t && \left[ \text{by } \eta \leq \frac{1}{64 \kappa \lambda_1^*} \right] \\
&\leq \left( 1 - \eta \frac{\lambda_{r^*}^*}{32\kappa} \right) \mathcal{M}_t, && \left[ \text{by } \epsilon \leq \frac{\lambda_{r^*}^*}{16\kappa (32 \lambda_1^* + 128 \kappa^2)} \right]
\end{aligned}
$$

where the order dominance from the second inequality follows from the fact that $\eta$ and $\epsilon$ are sufficiently small constant such that the terms in $\mathcal{O}(\eta\epsilon), \mathcal{O}(\epsilon^2), \mathcal{O}(\eta^2 \epsilon^2), \mathcal{O}(\eta \epsilon^3)$ are significantly smaller the terms in $\mathcal{O}(\eta)$ and $\mathcal{O}(\epsilon)$.

Similarly, we can obtain

$$\left\| \boldsymbol{R}_{t+1}^{\perp} \right\|_{op} \le \left( 1 - \eta \frac{\lambda_{r^*}^*}{16\kappa} \right) \mathcal{M}_t \qquad \left[ \text{since } \lambda_{\min}\left( \boldsymbol{A}_t^{\boldsymbol{U}_\perp} \right) \ge 0 \right]$$

$$+ \eta \bigg\{ 8\epsilon\lambda_1^* + 2\eta\lambda_1^*\lambda_{r^*}^* + 4\epsilon\eta\lambda_1^*\lambda_{r^*}^* + 64\epsilon\kappa^2 + 32\eta\epsilon\kappa^2\lambda_{r^*}^* + 64\eta\epsilon\kappa^2\lambda_{r^*}^* + 128\epsilon^2\kappa^2$$

$$+ 8\eta\epsilon\lambda_1^* + 8\eta\epsilon^2\lambda_1^*\lambda_{r^*}^* + 2048\epsilon^3\frac{\kappa^3}{\lambda_{r^*}^*} + 64\eta\epsilon^2\kappa^2 + 128\eta\epsilon^3\kappa^2 \bigg\} \mathcal{M}_t$$

$$\le \left( 1 - \eta\frac{\lambda_{r^*}^*}{16\kappa} \right)\mathcal{M}_t + 2\eta\bigg\{ 8\epsilon\lambda_1^* + 2\eta\lambda_1^*\lambda_{r^*}^* + 64\epsilon\kappa^2 \bigg\}\mathcal{M}_t \qquad \text{[due to the order dominance]}$$

$$\le \left( 1 - \eta\frac{\lambda_{r^*}^*}{32\kappa} \right)\mathcal{M}_t + 2\eta\bigg\{ 8\epsilon\lambda_1^* + 64\epsilon\kappa^2 \bigg\}\mathcal{M}_t \qquad \left[ \text{by } \eta \le \frac{1}{128\kappa\lambda_1^*} \right]$$

$$\le \left( 1 - \eta\frac{\lambda_{r^*}^*}{64\kappa} \right)\mathcal{M}_t\,, \qquad \left[ \text{by } \epsilon \le \frac{\lambda_{r^*}^*}{32\kappa(32\lambda_1^* + 128\kappa^2)} \right]$$

which proves the Eq. (38) at $t + 1$.

Therefore, we can conclude that

$$\mathcal{M}_{t+1} \le \left( 1 - \eta\frac{\lambda_{r^*}^*}{64\kappa} \right)\mathcal{M}_t\,.$$

Next, assume Eq. (38)-Eq. (41) hold at $t \ge 1$, we have

$$\left( \boldsymbol{A}_{t+1}^{\top}\boldsymbol{A}_{t+1} - \boldsymbol{B}_{t+1}\boldsymbol{B}_{t+1}^{\top} \right) - \left( \boldsymbol{A}_t^{\top}\boldsymbol{A}_t - \boldsymbol{B}_t\boldsymbol{B}_t^{\top} \right) = \eta^2\boldsymbol{B}_t\left( \boldsymbol{A}_t\boldsymbol{B}_t - \Delta \right)^{\top}\widehat{\boldsymbol{\Sigma}}\widehat{\boldsymbol{\Sigma}}\left( \boldsymbol{A}_t\boldsymbol{B}_t - \Delta \right)\boldsymbol{B}_t^{\top}$$
$$+ \eta^2\boldsymbol{A}_t^{\top}\widehat{\boldsymbol{\Sigma}}\left( \boldsymbol{A}_t\boldsymbol{B}_t - \Delta \right)\left( \boldsymbol{A}_t\boldsymbol{B}_t - \Delta \right)^{\top}\widehat{\boldsymbol{\Sigma}}\boldsymbol{A}_t\,.$$

Accordingly, we can derive

$$\left\| \left( \boldsymbol{A}_{t+1}^{\top}\boldsymbol{A}_{t+1} - \boldsymbol{B}_{t+1}\boldsymbol{B}_{t+1}^{\top} \right) - \left( \boldsymbol{A}_0^{\top}\boldsymbol{A}_0 - \boldsymbol{B}_0\boldsymbol{B}_0^{\top} \right) \right\|_{op} = \sum_{i=1}^{t+1} \left\| \left( \boldsymbol{A}_i^{\top}\boldsymbol{A}_i - \boldsymbol{B}_i\boldsymbol{B}_i^{\top} \right) - \left( \boldsymbol{A}_{i-1}^{\top}\boldsymbol{A}_{i-1} - \boldsymbol{B}_{i-1}\boldsymbol{B}_{i-1}^{\top} \right) \right\|_{op}$$

$$= \sum_{i=1}^{t+1} 2\eta^2\|\widehat{\boldsymbol{\Sigma}}\|_{op}^2\|\boldsymbol{R}_{i-1}\|_{op}^2 \max\left\{ \|\boldsymbol{A}_{i-1}\|_{op}^2, \|\boldsymbol{B}_{i-1}\|_{op}^2 \right\}$$

$$= \sum_{i=1}^{t+1} 72\eta^2\mathcal{M}_{i-1}^2\lambda_1^* \qquad \text{[by Eq. (29)]}$$

$$\le \sum_{i=1}^{t+1} 18\eta^2\left( 1 - \eta\frac{\lambda_{r^*}^*}{64\kappa} \right)^{2(i-1)}(\lambda_{r^*}^*)^2\lambda_1^*$$

$$\le 18\eta^2(\lambda_{r^*}^*)^2\lambda_1^*\sum_{i=0}^{\infty}\left( 1 - \eta\frac{\lambda_{r^*}^*}{64\kappa} \right)^{2i}$$

$$\le 18\eta^2(\lambda_{r^*}^*)^2\lambda_1^*\frac{64\kappa}{\eta\lambda_{r^*}^*}$$

$$= 1152\eta\lambda_1^*\lambda_{r^*}^*\kappa$$

$$\le (1 - \epsilon/\kappa)\lambda_1^*\,. \qquad \left[ \text{by } \eta \le \frac{(1-\epsilon/\kappa)}{1152\lambda_1^*} \right]$$

Since $\left\| \left( \boldsymbol{A}_0^{\top}\boldsymbol{A}_0 - \boldsymbol{B}_0\boldsymbol{B}_0^{\top} \right) \right\|_{op} = 0$ due to the spectral initialization (Spectral-init), by triangle inequality, $\left\| \left( \boldsymbol{A}_{t+1}^{\top}\boldsymbol{A}_{t+1} - \boldsymbol{B}_{t+1}\boldsymbol{B}_{t+1}^{\top} \right) \right\|_{op} \le (1 - \epsilon/\kappa)\lambda_1^*$. Next, by Lemma C.15, we can obtain

$$\|\boldsymbol{A}_{t+1}\|_{op} \le 2\sqrt{\lambda_1^*}\,, \quad \|\boldsymbol{B}_{t+1}\|_{op} \le 2\sqrt{\lambda_1^*}\,,$$

which proves the Eq. (39) at $t + 1$. Lastly, assume Eq. (38)-Eq. (41) hold at $t \geq 1$, by Weyl's inequality, combine with $\mathcal{M}_{t+1} \leq \frac{\lambda_{r^*}^*}{2}$, we have

$$\frac{\lambda_{r^*}^*}{2} \geq \left\| \boldsymbol{A}_{t+1}^{\boldsymbol{U}} \boldsymbol{B}_{t+1}^{\boldsymbol{V}} - \boldsymbol{S}^* \right\|_{op} \geq \lambda_{r^*}^* - \lambda_{r^*}(\boldsymbol{A}_{t+1}^{\boldsymbol{U}} \boldsymbol{B}_{t+1}^{\boldsymbol{V}}) \Rightarrow \lambda_{r^*}(\boldsymbol{A}_{t+1}^{\boldsymbol{U}} \boldsymbol{B}_{t+1}^{\boldsymbol{V}}) \geq \frac{\lambda_{r^*}^*}{2} \, .$$

Again by Weyl's inequality and the Eq. (39) at time $t + 1$ we can get

$$2\sqrt{\lambda_1^*} \cdot \lambda_{r^*}(\boldsymbol{B}_{t+1}^{\boldsymbol{V}}) \geq \lambda_1(\boldsymbol{A}_{t+1}^{\boldsymbol{U}})\lambda_{r^*}(\boldsymbol{B}_{t+1}^{\boldsymbol{V}}) \geq \lambda_{r^*}(\boldsymbol{A}_{t+1}^{\boldsymbol{U}} \boldsymbol{B}_{t+1}^{\boldsymbol{V}}) \geq \frac{\lambda_{r^*}^*}{2} \Rightarrow \lambda_{r^*}(\boldsymbol{B}_{t+1}^{\boldsymbol{V}}) \geq \frac{\sqrt{\lambda_{r^*}^*}}{4\sqrt{\kappa}} \, .$$

Besides, $\lambda_{r^*}^*(\boldsymbol{A}_{t+1}^{\boldsymbol{U}})$ follows similar derivation. We prove all the claims. $\qquad\square$

**Theorem C.17.** *Under assumptions in Section 2.1 for the linear setting, with spectral initialization (Spectral-init), we take $\epsilon$ in data concentration as*

$$\epsilon \leq \min \left\{ \frac{1}{2\kappa} , \frac{\lambda_{r^*}^*}{32\kappa(32\lambda_1^* + 128\kappa^2)} \right\} \, ,$$

*and set the step-size as*

$$\eta \leq \min \left\{ \frac{1}{128\kappa\lambda_1^*} , \frac{(1 - \epsilon/\kappa)}{1152\lambda_1^*} \right\} \, , \tag{43}$$

*then with probability at least with probability $1 - 2C \exp(-\epsilon^2 N)$ for a universal constant $C > 0$, we have that $\forall t \geq 0$*

$$\|\boldsymbol{A}_t \boldsymbol{B}_t - \Delta\|_F \leq \sqrt{2r^*} \left( 1 - \eta \frac{\lambda_{r^*}^*}{64\kappa} \right)^t \cdot \lambda_{r^*}^* \, .$$

*Proof.* By Lemma C.16, with probability at least with probability $1 - 2C \exp(-\epsilon^2 N)$ for a universal constant $C > 0$, we can obtain the linear convergence of generalization risk

$$\|\boldsymbol{A}_t \boldsymbol{B}_t - \Delta\|_F \leq \sqrt{2r^*} \|\boldsymbol{A}_t \boldsymbol{B}_t - \Delta\|_{op} \qquad [\mathrm{Rank}(\boldsymbol{A}_t \boldsymbol{B}_t) = r^* \text{ by Lemma C.12 and } \mathrm{Rank}(\Delta) = r^*]$$

$$\leq \sqrt{2r^*} \left( 1 - \eta \frac{\lambda_{r^*}^*}{64\kappa} \right)^t \cdot \lambda_{r^*}^* \, ,$$

which is independent of the choice of LoRA rank $r$ if $r \geq r^*$. $\qquad\square$

**Remark:** The above convergence rate is independent of the choice of LoRA rank $r$ if $r \geq r^*$. It achieves an $\varepsilon$-risk in $\mathcal{O}\left(\kappa^3 \ln(1/\varepsilon)\right)$ iterations. The linear convergence rate heavily depends on $\kappa$.

## C.3. Preconditioned Gradient Descent under Spectral Initialization

The convergence rate in Theorem C.17 will become slow if the downstream feature shift $\Delta$ is ill-conditioned (i.e., $\kappa$ is extremely large). This motivates us to add preconditioners, which is a key technique to accelerate convergence in matrix factorization/sensing (Tong et al., 2021; Zhang et al., 2021; 2023; Jia et al., 2024). The preconditioners are derived from a Riemannian metric in Mishra et al. (2012) which originally are formulated as $(\boldsymbol{A}\boldsymbol{A}^\top)^{-1}$ and $(\boldsymbol{B}^\top \boldsymbol{B})^{-1}$. Also, there is an efficient variant, i.e. $(\boldsymbol{A}\boldsymbol{A}^\top + \lambda \boldsymbol{I}_r)^{-1}$ and $(\boldsymbol{B}^\top \boldsymbol{B} + \lambda \boldsymbol{I}_r)^{-1}$, which commonly used in practice for numerical stability via the preconditioning parameter $\lambda \geq 0$.

In the over-ranked setting ($r > r^*$), $\boldsymbol{B}_t \boldsymbol{B}_t^\top$ and $\boldsymbol{A}_t^\top \boldsymbol{A}_t$ are not necessarily invertible. Hence we add the following preconditioners to vanilla GD (4)

$$\begin{aligned}
\boldsymbol{A}_{t+1} &= \boldsymbol{A}_t - \frac{\eta}{N} \widetilde{\boldsymbol{X}}^\top \left( \widetilde{\boldsymbol{X}} \left( \boldsymbol{W}^\natural + \boldsymbol{A}_t \boldsymbol{B}_t \right) - \widetilde{\boldsymbol{Y}} \right) \boldsymbol{B}_t^\top \left( \boldsymbol{B}_t \boldsymbol{B}_t^\top \right)^\dagger \, , \\
\boldsymbol{B}_{t+1} &= \boldsymbol{B}_t - \frac{\eta}{N} \left( \boldsymbol{A}_t^\top \boldsymbol{A}_t \right)^\dagger \boldsymbol{A}_t^\top \widetilde{\boldsymbol{X}}^\top \left( \widetilde{\boldsymbol{X}} \left( \boldsymbol{W}^\natural + \boldsymbol{A}_t \boldsymbol{B}_t \right) - \widetilde{\boldsymbol{Y}} \right) \, ,
\end{aligned} \tag{Prec-GD}$$

where $\boldsymbol{M}^\dagger$ denotes the pseudo-inverse of a matrix $\boldsymbol{M}$. Such modified preconditioners are also considered in Li et al. (2025).

In the following proofs, we will prove that the LoRA fine-tuning can achieve faster linear convergence which is independent of condition number $\kappa$ under (Spectral-init) and (Prec-GD). Similar to Lemma C.12, the dynamics of $\boldsymbol{B}_t$ are still limited to the $r^*$-dimensional singular subspace $\boldsymbol{V}$ of $\Delta$ under (Spectral-init). We can verify this fact by the following lemma.

**Lemma C.18.** *For any natural number $t \geq 0$, under assumptions in Section 2.1 for the linear setting, with (Spectral-init) and (Prec-GD), we have*

$$\boldsymbol{B}_t \boldsymbol{V}_\perp = \boldsymbol{0}_{r \times (k-r^*)} .$$

*Proof.* For $t = 0$, recall the SVD of $\mathbf{G}^\natural$, i.e. $\widetilde{\boldsymbol{U}}_{\boldsymbol{G}^\natural} \widetilde{\boldsymbol{S}}_{\boldsymbol{G}^\natural} \widetilde{\boldsymbol{V}}_{\boldsymbol{G}^\natural}^\top$ in Eq. (28), we have

$$\boldsymbol{B}_0 \boldsymbol{V}_\perp = \left[\widetilde{\boldsymbol{S}}_{\boldsymbol{G}^\natural}^{-1/2}\right]_{[1:r]} \left[\widetilde{\boldsymbol{U}}_{\boldsymbol{G}^\natural}^\top\right]_{[:,1:r]} \mathbf{G}^\natural \boldsymbol{V}_\perp = \left[\widetilde{\boldsymbol{S}}_{\boldsymbol{G}^\natural}^{-1/2}\right]_{[1:r]} \left[\widetilde{\boldsymbol{U}}_{\boldsymbol{G}^\natural}^\top\right]_{[:,1:r]} \widehat{\boldsymbol{\Sigma}} \Delta \boldsymbol{V}_\perp = \boldsymbol{0}_{r \times (k-r^*)} .$$

Assume $\boldsymbol{B}_t \boldsymbol{V}_\perp = \boldsymbol{0}_{d \times (d-r^*)}$ holds for any natural number $t \geq 1$, then

$$\begin{aligned}
\boldsymbol{B}_{t+1} \boldsymbol{V}_\perp &= \boldsymbol{B}_t \boldsymbol{V}_\perp - \frac{\eta}{N} \left(\boldsymbol{A}_t^\top \boldsymbol{A}_t\right)^\dagger \boldsymbol{A}_t^\top \widetilde{\boldsymbol{X}}^\top \left(\widetilde{\boldsymbol{X}} \left(\boldsymbol{W}^\natural + \boldsymbol{A}_t \boldsymbol{B}_t\right) - \widetilde{\boldsymbol{Y}}\right) \boldsymbol{V}_\perp \\
&= \boldsymbol{B}_t \boldsymbol{V}_\perp - \eta \left(\boldsymbol{A}_t^\top \boldsymbol{A}_t\right)^\dagger \boldsymbol{A}_t^\top \widehat{\boldsymbol{\Sigma}} \left(\boldsymbol{A}_t \boldsymbol{B}_t - \Delta\right) \boldsymbol{V}_\perp \\
&= \boldsymbol{0}_{r \times (k-r^*)} , \qquad\qquad\qquad\qquad\qquad \text{[by our inductive hypothesis]}
\end{aligned}$$

which proves the claim. $\qquad\square$

We can re-formulate (Prec-GD) to be

$$\boldsymbol{A}_{t+1} = \boldsymbol{A}_t - \eta \widehat{\boldsymbol{\Sigma}} \left(\boldsymbol{A}_t \boldsymbol{B}_t - \Delta\right) \left(\boldsymbol{B}_t\right)^\top \left(\boldsymbol{B}_t \boldsymbol{B}_t^\top\right)^\dagger , \tag{44}$$

$$\boldsymbol{B}_{t+1} = \boldsymbol{B}_t - \eta \left(\boldsymbol{A}_t^\top \boldsymbol{A}_t\right)^\dagger \boldsymbol{A}_t^\top \widehat{\boldsymbol{\Sigma}} \left(\boldsymbol{A}_t \boldsymbol{B}_t - \Delta\right) . \tag{45}$$

Before we start our main proofs, we first define the following notations

- SVD of product matrix $\boldsymbol{A}_t \boldsymbol{B}_t := \mathcal{U}_t \mathcal{S}_t \mathcal{V}_t^\top$, where $\mathcal{U}_t \in \mathbb{R}^{d \times r^*}$, $\mathcal{S}_t \in \mathbb{R}^{r^* \times r^*}$, and $\mathcal{V}_t \in \mathbb{R}^{k \times r^*}$. Notice that here we employ rank-$r^*$ SVD of $\boldsymbol{A}_t \boldsymbol{B}_t$ since $\text{Rank}\left(\boldsymbol{A}_t \boldsymbol{B}_t\right) \leq r^*$ due to Lemma C.18 and $\lambda_{r^*}\left(\boldsymbol{A}_t \boldsymbol{B}_r\right) > 0$ strictly which we will obtain from Theorem C.21.

- The left compact singular matrix of $\boldsymbol{A}_t$ as $\boldsymbol{U}_{\boldsymbol{A}_t} \in \mathbb{R}^{d \times r}$.

- The right compact singular matrix of $\boldsymbol{B}_t$ as $\boldsymbol{V}_{\boldsymbol{B}_t} \in \mathbb{R}^{k \times r^*}$. Notice that here we take the top-$r^*$ right singular subspace of $\boldsymbol{B}_t$ due to Lemma C.18.

By the pseudo inverse theorem and Jia et al. (2024, Lemma 14), we can obtain

$$\boldsymbol{A}_t \left(\boldsymbol{A}_t^\top \boldsymbol{A}_t\right)^\dagger \boldsymbol{A}_t^\top = \boldsymbol{U}_{\boldsymbol{A}_t} \boldsymbol{U}_{\boldsymbol{A}_t}^\top , \tag{46}$$

$$\left(\boldsymbol{B}_t\right)^\top \left(\boldsymbol{B}_t \boldsymbol{B}_t^\top\right)^\dagger \boldsymbol{B}_t = \boldsymbol{V}_{\boldsymbol{B}_t} \boldsymbol{V}_{\boldsymbol{B}_t}^\top . \tag{47}$$

$$\left(\boldsymbol{B}_t\right)^\top \left(\boldsymbol{B}_t \boldsymbol{B}_t^\top\right)^\dagger \left(\boldsymbol{A}_t^\top \boldsymbol{A}_t\right)^\dagger \boldsymbol{A}_t^\top = \mathcal{V}_t \mathcal{S}_t^{-1} \mathcal{U}_t^\top . \tag{48}$$

**Lemma C.19.** *Denote $\boldsymbol{R}_t := \boldsymbol{A}_t \boldsymbol{B}_t - \Delta$, $\boldsymbol{\Xi} := \widehat{\boldsymbol{\Sigma}} - \boldsymbol{I}_d$, under assumptions in Section 2.1 for the linear setting, with (Prec-GD), then we have*

$$\boldsymbol{R}_{t+1} = \boldsymbol{R}_t - \eta \boldsymbol{U}_{\boldsymbol{A}_t} \boldsymbol{U}_{\boldsymbol{A}_t}^\top \boldsymbol{R}_t - \eta \boldsymbol{R}_t \boldsymbol{V}_{\boldsymbol{B}_t} \boldsymbol{V}_{\boldsymbol{B}_t}^\top - \eta \boldsymbol{U}_{\boldsymbol{A}_t} \boldsymbol{U}_{\boldsymbol{A}_t}^\top \boldsymbol{\Xi} \boldsymbol{R}_t - \eta \boldsymbol{\Xi} \boldsymbol{R}_t \boldsymbol{V}_{\boldsymbol{B}_t} \boldsymbol{V}_{\boldsymbol{B}_t}^\top + \eta^2 \widehat{\boldsymbol{\Sigma}} \boldsymbol{R}_t \mathcal{V}_t \mathcal{S}_t^{-1} \mathcal{U}_t^\top \widehat{\boldsymbol{\Sigma}} \boldsymbol{R}_t .$$

*Proof.* With Eq. (44) and Eq. (45), we can construct

$$
\begin{aligned}
\boldsymbol{R}_{t+1} &= \boldsymbol{A}_{t+1}\boldsymbol{B}_{t+1} - \Delta \\
&= \boldsymbol{A}_t\boldsymbol{B}_t - \Delta \\
&\quad - \eta\boldsymbol{A}_t\left(\boldsymbol{A}_t^\top\boldsymbol{A}_t\right)^\dagger\boldsymbol{A}_t^\top\widehat{\boldsymbol{\Sigma}}\left(\boldsymbol{A}_t\boldsymbol{B}_t - \Delta\right) \\
&\quad - \eta\widehat{\boldsymbol{\Sigma}}\left(\boldsymbol{A}_t\boldsymbol{B}_t - \Delta\right)\left(\boldsymbol{B}_t\right)^\top\left(\boldsymbol{B}_t\boldsymbol{B}_t^\top\right)^\dagger\boldsymbol{B}_t \\
&\quad + \eta^2\widehat{\boldsymbol{\Sigma}}\left(\boldsymbol{A}_t\boldsymbol{B}_t - \Delta\right)\left(\boldsymbol{B}_t\right)^\top\left(\boldsymbol{B}_t\boldsymbol{B}_t^\top\right)^\dagger\left(\boldsymbol{A}_t^\top\boldsymbol{A}_t\right)^\dagger\boldsymbol{A}_t^\top\widehat{\boldsymbol{\Sigma}}\left(\boldsymbol{A}_t\boldsymbol{B}_t - \Delta\right) \\
&= \boldsymbol{R}_t - \eta\boldsymbol{U}_{\boldsymbol{A}_t}\boldsymbol{U}_{\boldsymbol{A}_t}^\top\widehat{\boldsymbol{\Sigma}}\boldsymbol{R}_t - \eta\widehat{\boldsymbol{\Sigma}}\boldsymbol{R}_t\boldsymbol{V}_{\boldsymbol{B}_t}\boldsymbol{V}_{\boldsymbol{B}_t}^\top && \text{[by Eq. (46) and Eq. (47)]} \\
&\quad + \eta^2\widehat{\boldsymbol{\Sigma}}\boldsymbol{R}_t\mathcal{V}_t\mathcal{S}_t^{-1}\mathcal{U}_t^\top\widehat{\boldsymbol{\Sigma}}\boldsymbol{R}_t && \text{[by Eq. (48)]} \\
&= \boldsymbol{R}_t - \eta\boldsymbol{U}_{\boldsymbol{A}_t}\boldsymbol{U}_{\boldsymbol{A}_t}^\top\boldsymbol{R}_t - \eta\boldsymbol{R}_t\boldsymbol{V}_{\boldsymbol{B}_t}\boldsymbol{V}_{\boldsymbol{B}_t}^\top - \eta\boldsymbol{U}_{\boldsymbol{A}_t}\boldsymbol{U}_{\boldsymbol{A}_t}^\top\boldsymbol{\Xi}\boldsymbol{R}_t - \eta\boldsymbol{\Xi}\boldsymbol{R}_t\boldsymbol{V}_{\boldsymbol{B}_t}\boldsymbol{V}_{\boldsymbol{B}_t}^\top + \eta^2\widehat{\boldsymbol{\Sigma}}\boldsymbol{R}_t\mathcal{V}_t\mathcal{S}_t^{-1}\mathcal{U}_t^\top\widehat{\boldsymbol{\Sigma}}\boldsymbol{R}_t ,
\end{aligned}
$$

which proves the claim. $\qquad\square$

In the next, we aim to estimate the signal part $\boldsymbol{R}_t - \eta\boldsymbol{U}_{\boldsymbol{A}_t}\boldsymbol{U}_{\boldsymbol{A}_t}^\top\boldsymbol{R}_t - \eta\boldsymbol{R}_t\boldsymbol{V}_{\boldsymbol{B}_t}\boldsymbol{V}_{\boldsymbol{B}_t}^\top$.

**Lemma C.20.** *Recall $\boldsymbol{R}_t := \boldsymbol{A}_t\boldsymbol{B}_t - \Delta$, under assumptions in Section 2.1 for the linear setting, with (Prec-GD), then*

$$
\left\|\boldsymbol{R}_t - \eta\boldsymbol{U}_{\boldsymbol{A}_t}\boldsymbol{U}_{\boldsymbol{A}_t}^\top\boldsymbol{R}_t - \eta\boldsymbol{R}_t\boldsymbol{V}_{\boldsymbol{B}_t}\boldsymbol{V}_{\boldsymbol{B}_t}^\top\right\|_{\mathrm{F}} \le (1-\eta)\left\|\boldsymbol{R}_t\right\|_{\mathrm{F}} .
$$

*Proof.*

$$
\begin{aligned}
&\left\|\boldsymbol{R}_t - \eta\boldsymbol{U}_{\boldsymbol{A}_t}\boldsymbol{U}_{\boldsymbol{A}_t}^\top\boldsymbol{R}_t - \eta\boldsymbol{R}_t\boldsymbol{V}_{\boldsymbol{B}_t}\boldsymbol{V}_{\boldsymbol{B}_t}^\top\right\|_{\mathrm{F}} \\
&= \left\|\boldsymbol{R}_t\left(\boldsymbol{V}_{\boldsymbol{B}_t}\boldsymbol{V}_{\boldsymbol{B}_t}^\top + \boldsymbol{I}_k - \boldsymbol{V}_{\boldsymbol{B}_t}\boldsymbol{V}_{\boldsymbol{B}_t}^\top\right) - \eta\boldsymbol{U}_{\boldsymbol{A}_t}\boldsymbol{U}_{\boldsymbol{A}_t}^\top\boldsymbol{R}_t\left(\boldsymbol{V}_{\boldsymbol{B}_t}\boldsymbol{V}_{\boldsymbol{B}_t}^\top + \boldsymbol{I}_k - \boldsymbol{V}_{\boldsymbol{B}_t}\boldsymbol{V}_{\boldsymbol{B}_t}^\top\right) - \eta\boldsymbol{R}_t\boldsymbol{V}_{\boldsymbol{B}_t}\boldsymbol{V}_{\boldsymbol{B}_t}^\top\right\|_{\mathrm{F}} \\
&= \left\|\boldsymbol{R}_t\boldsymbol{V}_{\boldsymbol{B}_t}\boldsymbol{V}_{\boldsymbol{B}_t}^\top - \eta\boldsymbol{U}_{\boldsymbol{A}_t}\boldsymbol{U}_{\boldsymbol{A}_t}^\top\boldsymbol{R}_t\boldsymbol{V}_{\boldsymbol{B}_t}\boldsymbol{V}_{\boldsymbol{B}_t}^\top - \eta\boldsymbol{R}_t\boldsymbol{V}_{\boldsymbol{B}_t}\boldsymbol{V}_{\boldsymbol{B}_t}^\top\right\|_{\mathrm{F}} && \left[\text{since } \boldsymbol{R}_t\left(\boldsymbol{I}_k - \boldsymbol{V}_{\boldsymbol{B}_t}\boldsymbol{V}_{\boldsymbol{B}_t}^\top\right) = \boldsymbol{0} \text{ by Lemma C.18}\right] \\
&= \left\|\left(\boldsymbol{I}_d - \eta\left(\boldsymbol{I}_d + \boldsymbol{U}_{\boldsymbol{A}_t}\boldsymbol{U}_{\boldsymbol{A}_t}^\top\right)\right)\boldsymbol{R}_t\boldsymbol{V}_{\boldsymbol{B}_t}\boldsymbol{V}_{\boldsymbol{B}_t}^\top\right\|_{\mathrm{F}} \\
&= \left\|\boldsymbol{I}_d - \eta\left(\boldsymbol{I}_d + \boldsymbol{U}_{\boldsymbol{A}_t}\boldsymbol{U}_{\boldsymbol{A}_t}^\top\right)\right\|_{op}\left\|\boldsymbol{R}_t\boldsymbol{V}_{\boldsymbol{B}_t}\boldsymbol{V}_{\boldsymbol{B}_t}^\top\right\|_{\mathrm{F}} \\
&\le (1-\eta)\left\|\boldsymbol{R}_t\right\|_{\mathrm{F}} , && \left[\left\|\boldsymbol{I}_d - \eta\left(\boldsymbol{I}_d + \boldsymbol{U}_{\boldsymbol{A}_t}\boldsymbol{U}_{\boldsymbol{A}_t}^\top\right)\right\|_{op} \le 1-\eta, \text{ since } \boldsymbol{U}_{\boldsymbol{A}_t}\boldsymbol{U}_{\boldsymbol{A}_t}^\top \text{ is a rank-}r \text{ projection matrix}\right]
\end{aligned}
$$

which concludes the proof. $\qquad\square$

Finally, we have the following linear convergence under (Prec-GD) and (Spectral-init).

**Theorem C.21.** *Under assumptions in Section 2.1 for the linear setting, with (Spectral-init) and (Prec-GD), we choose*

$$
\epsilon \le \min\left\{\frac{1}{2\sqrt{r^*}\kappa}, \frac{1}{4}\right\}
$$

*and set $\eta \in \left(0, \frac{0.5-2\epsilon}{(1+\epsilon)^2}\right)$, then with probability at least $1 - 2C\exp(-\epsilon^2 N)$ for a universal constant $C > 0$, we have*

$$
\left\|\boldsymbol{A}_t\boldsymbol{B}_t - \Delta\right\|_{\mathrm{F}} \le \frac{1}{2}\left(1 - \frac{\eta}{2}\right)^t \lambda_{r^*}^* .
$$

*Proof.* We prove it by induction. We suppose the following two inductive hypothesis

$$
\lambda_{r^*}\left(\boldsymbol{A}_t\boldsymbol{B}_t\right) \ge \frac{\lambda_{r^*}^*}{2} , \tag{49}
$$

$$
\left\|\boldsymbol{A}_0\boldsymbol{B}_0 - \Delta\right\|_{\mathrm{F}} \le \frac{\lambda_{r^*}^*}{2} . \tag{50}
$$

Starting from $t = 0$, under (Spectral-init), with probability at least $1 - 2C \exp(-\epsilon^2 N)$ for a universal constant $C > 0$, we have

$$
\begin{aligned}
\left\|\boldsymbol{A}_0 \boldsymbol{B}_0 - \Delta\right\|_{\mathrm{F}} &= \left\|\boldsymbol{G}^{\natural} - \Delta\right\|_{\mathrm{F}} \\
&= \left\|\left(\widehat{\boldsymbol{\Sigma}} - \boldsymbol{I}_d\right) \Delta\right\|_{\mathrm{F}} && \text{[by Eq. (28)]} \\
&\leq \epsilon \|\Delta\|_{\mathrm{F}} \\
&\leq \epsilon \sqrt{r^*} \|\Delta\|_{op} && \left[\text{since } \mathrm{Rank}\left(\Delta\right) = r^*\right] \\
&\leq \frac{\lambda_{r^*}^*}{2}. && \left[\text{since } \epsilon \leq 1/2\sqrt{r^*}\kappa\right]
\end{aligned}
$$

Then, by Weyl's inequality, we have

$$
\lambda_{r^*}\left(\Delta\right) - \lambda_{r^*}\left(\boldsymbol{A}_0 \boldsymbol{B}_0\right) \leq \left\|\boldsymbol{A}_0 \boldsymbol{B}_0 - \Delta\right\|_{op} \leq \left\|\boldsymbol{A}_0 \boldsymbol{B}_0 - \Delta\right\|_{\mathrm{F}},
$$

which implies

$$
\lambda_{r^*}\left(\boldsymbol{A}_0 \boldsymbol{B}_0\right) \geq \frac{\lambda_{r^*}^*}{2}. \tag{51}
$$

Therefore, we verify Eq. (49) and Eq. (50) at $t = 0$. We assume Eq. (49) and Eq. (50) hold at $t = 2, 3, ...$, then by Lemma C.19, with probability at least with probability $1 - 2C \exp(-\epsilon^2 N)$ for a universal constant $C > 0$, we have

$$
\begin{aligned}
\left\|\boldsymbol{R}_{t+1}\right\|_{\mathrm{F}} &\leq \left\|\boldsymbol{R}_t - \eta \boldsymbol{U}_{\boldsymbol{A}_t} \boldsymbol{U}_{\boldsymbol{A}_t}^\top \boldsymbol{R}_t - \eta \boldsymbol{R}_t \boldsymbol{V}_{\boldsymbol{B}_t} \boldsymbol{V}_{\boldsymbol{B}_t}^\top\right\|_{\mathrm{F}} \\
&\quad + \eta \left\|\boldsymbol{U}_{\boldsymbol{A}_t} \boldsymbol{U}_{\boldsymbol{A}_t}^\top \boldsymbol{\Xi} \boldsymbol{R}_t\right\|_{\mathrm{F}} + \eta \left\|\boldsymbol{\Xi} \boldsymbol{R}_t \boldsymbol{V}_{\boldsymbol{B}_t} \boldsymbol{V}_{\boldsymbol{B}_t}^\top\right\|_{\mathrm{F}} + \eta^2 \left\|\widehat{\boldsymbol{\Sigma}} \boldsymbol{R}_t \mathcal{V}_t \mathcal{S}_t^{-1} \mathcal{U}_t^\top \widehat{\boldsymbol{\Sigma}} \boldsymbol{R}_t\right\|_{\mathrm{F}} \\
&\leq (1 - \eta) \left\|\boldsymbol{R}_t\right\|_{\mathrm{F}} && \text{[by Lemma C.20]} \\
&\quad + \eta\epsilon \left\|\boldsymbol{U}_{\boldsymbol{A}_t} \boldsymbol{U}_{\boldsymbol{A}_t}^\top \boldsymbol{R}_t\right\|_{\mathrm{F}} + \eta\epsilon \left\|\boldsymbol{R}_t \boldsymbol{V}_{\boldsymbol{B}_t} \boldsymbol{V}_{\boldsymbol{B}_t}^\top\right\|_{\mathrm{F}} + \eta^2 (1 + \epsilon)^2 \frac{\left\|\boldsymbol{R}_t\right\|_{\mathrm{F}}^2}{\lambda_{r^*}\left(\boldsymbol{A}_t \boldsymbol{B}_t\right)} && \left[\text{by } \|\boldsymbol{\Xi}\|_{op} \leq \epsilon\right] \\
&\leq (1 - \eta) \left\|\boldsymbol{R}_t\right\|_{\mathrm{F}} \\
&\quad + \eta\epsilon \left\|\boldsymbol{R}_t\right\|_{\mathrm{F}} + \eta\epsilon \left\|\boldsymbol{R}_t\right\|_{\mathrm{F}} + \eta^2 (1 + \epsilon)^2 \left\|\boldsymbol{R}_t\right\|_{\mathrm{F}} && \text{[since Eq. (49) and Eq. (50) hold at } t] \\
&= \left(1 - (1 - 2\epsilon)\eta + \eta^2 (1 + \epsilon)^2\right) \left\|\boldsymbol{R}_t\right\|_{\mathrm{F}} \\
&\leq \left(1 - \frac{\eta}{2}\right) \left\|\boldsymbol{R}_t\right\|_{\mathrm{F}}. && \left[\text{taking } \eta \leq \frac{0.5 - 2\epsilon}{(1+\epsilon)^2}\right]
\end{aligned}
$$

This implies Eq. (50) at time $t + 1$. By consequence, we can obtain Eq. (49) at time $t + 1$ again by Weyl's inequality. $\qquad \square$

**Remark:** The convergence rate is independent of the condition number of $\kappa$. The choice of stepsize $\eta$ is upper bounded by $\frac{0.5 - 2\epsilon}{(1+\epsilon)^2} \in (0, 0.5)$, which is a decreasing function of $\epsilon$. Therefore, if the condition number $\kappa$ is very large and thus $\epsilon$ is chosen as sufficiently small, then $\eta$ can reach $0.5$ and we still have a fast convergence rate independent of $\kappa$. This is particularly useful in practical fine-tuning tasks, where the adapted matrix can be highly ill-conditioned when its rank increases. We can empirically observe the ill-conditioned issues in real-world benchmarks, as shown in Appendix G.5 for more discussions.

## D. Proofs for Nonlinear Model

We deliver the proofs for nonlinear models in Section 4 here. The problem setting and results for $\|\boldsymbol{A}_0 \boldsymbol{B}_0 - \Delta\|_{\mathrm{F}}$ are presented in Appendix D.1. In Appendix D.2, we present the proofs of Theorem 4.2 as well as proofs for smoothed GD.

### D.1. Problem Settings and Spectral Initialization

Recall the pre-training model from Assumption 2.1

$$
f_{\mathrm{pre}}\left(\boldsymbol{x}\right) = \sigma\left(\boldsymbol{x}^\top \boldsymbol{W}^{\natural}\right)^\top \in \mathbb{R}^k, \quad \boldsymbol{W}^{\natural} \in \mathbb{R}^{d \times k}, \tag{52}
$$

and the downstream teacher weights from Assumption 2.2

$$\widetilde{W}^{\natural} = W^{\natural} + \Delta \in \mathbb{R}^{d \times k}, \quad \text{with } \widetilde{W}^{\natural} := \left[\widetilde{w}_1^{\natural}, \widetilde{w}_2^{\natural}, \cdots, \widetilde{w}_k^{\natural}\right].$$

The empirical loss of LoRA fine-tuning is defined as

$$\widetilde{L}\left(A_t, B_t\right) = \frac{1}{2N} \left\| \sigma\left(\widetilde{X}(W^{\natural} + A_t B_t)\right) - \sigma\left(\widetilde{X}\widetilde{W}^{\natural}\right)\right\|_F^2.$$

Next, we can derive the empirical gradients for $A_t$ and $B_t$ respectively.

$$
\begin{aligned}
\nabla_A \widetilde{L}\left(A_t, B_t\right) &= \frac{1}{N}\widetilde{X}^{\top}\left[\sigma\left(\widetilde{X}(W^{\natural} + A_t B_t)\right) - \sigma\left(\widetilde{X}\widetilde{W}^{\natural}\right)\right] \odot \sigma'\left(\widetilde{X}(W^{\natural} + A_t B_t)\right) B_t^{\top} \\
&:= \frac{1}{N}\widetilde{X}^{\top}\left[\sigma\left(\widetilde{X}(W_t)\right) - \sigma\left(\widetilde{X}\widetilde{W}^{\natural}\right)\right] \odot \sigma'\left(\widetilde{X}W_t\right) B_t^{\top} \qquad \text{[denote } W_t := W^{\natural} + A_t B_t] \\
&= -\left[\underbrace{\frac{1}{N}\widetilde{X}^{\top}\sigma\left(\widetilde{X}\widetilde{W}^{\natural}\right) \odot \sigma'\left(\widetilde{X}W_t\right)}_{:=\Gamma_{1,t}} - \underbrace{\frac{1}{N}\widetilde{X}^{\top}\sigma\left(\widetilde{X}W_t\right) \odot \sigma'\left(\widetilde{X}W_t\right)}_{:=\Gamma_{2,t}}\right] B_t^{\top} \qquad (53) \\
&:= -J_{W_t} B_t^{\top} \qquad \text{[denote } J_{W_t} := \Gamma_{1,t} - \Gamma_{2,t}]
\end{aligned}
$$

where the matrix operator $J_W : \mathbb{R}^{d \times k} \to \mathbb{R}^{d \times k}$ is formally defined as (by denoting $W_t := W^{\natural} + A_t B_t$)

$$J_W : W \to \frac{1}{N}\widetilde{X}^{\top}\left[\sigma\left(\widetilde{X}\widetilde{W}^{\natural}\right) - \sigma\left(\widetilde{X}(W)\right)\right] \odot \sigma'\left(\widetilde{X}W\right). \qquad (54)$$

Similarly, we can compute

$$
\begin{aligned}
\nabla_B \widetilde{L}\left(A_t, B_t\right) &= \frac{1}{N}A_t^{\top}\widetilde{X}^{\top}\left[\sigma\left(\widetilde{X}(W_t)\right) - \sigma\left(\widetilde{X}\widetilde{W}^{\natural}\right)\right] \odot \sigma'\left(\widetilde{X}W_t\right) \\
&= -A_t^{\top} J_{W_t}.
\end{aligned}
$$

For full fine-tuning, we consider the following empirical loss function over $K \in \mathbb{R}^{d \times k}$

$$L\left(K\right) = \frac{1}{2N}\left\|\sigma\left(\widetilde{X}K\right) - \sigma\left(\widetilde{X}\widetilde{W}^{\natural}\right)\right\|_F^2.$$

The gradient w.r.t. $K$ is

$$\nabla L\left(K\right) = \frac{1}{N}\widetilde{X}^{\top}\left[\sigma\left(\widetilde{X}K\right) - \sigma\left(\widetilde{X}\widetilde{W}^{\natural}\right)\right] \odot \sigma'\left(\widetilde{X}K\right)$$

Next, we can define the one-step negative gradient of full fine-tuning in the nonlinear case as

$$
\begin{aligned}
G^{\natural} &:= -\nabla L\left(W^{\natural}\right) \\
&= \frac{1}{N}\widetilde{X}^{\top}\left[\sigma\left(\widetilde{X}\widetilde{W}^{\natural}\right) - \sigma\left(\widetilde{X}W^{\natural}\right)\right] \odot \sigma'\left(\widetilde{X}W^{\natural}\right) \\
&= J_{W^{\natural}}. \qquad \text{[by definition of } J_W \text{ in Eq. (54)]}
\end{aligned}
$$

Additionally, we define

$$
\begin{aligned}
\Gamma_1^{\natural} &= \frac{1}{N}\widetilde{X}^{\top}\sigma\left(\widetilde{X}\widetilde{W}^{\natural}\right) \odot \sigma'\left(\widetilde{X}W^{\natural}\right), \\
\Gamma_2^{\natural} &= \frac{1}{N}\widetilde{X}^{\top}\sigma\left(\widetilde{X}W^{\natural}\right) \odot \sigma'\left(\widetilde{X}W^{\natural}\right).
\end{aligned} \qquad (55)
$$

In this section, we aim to analyze the initial properties of low-rank adapters under (Spectral-init) in a nonlinear context. The high-level proof strategy begins with examining the spectral properties of the one-step full gradient matrix, $G^{\natural}$. Unlike the linear case, the presence of nonlinearity prevents a direct analysis. To address this, we first establish the concentration of the

empirical full gradient, leveraging the fact that the empirical gradient approximates its expectation closely when the sample size is sufficiently large.

Subsequently, we utilize tools from (Brutzkus & Globerson, 2017) to derive useful properties of the expected gradients. These properties are then transferred back to the empirical gradients through concentration results. Finally, since low-rank adapters under (Spectral-init) represent the best $r$-rank approximation of $G^\natural$, we apply matrix analysis techniques to derive the desired results. Also, the concentration results in this part can serve as an important component for the later convergence analysis.

### D.1.1. COMPUTATION OF FULL POPULATION GRADIENTS

First, we can simplify $\mathbf{\Gamma}_{1,t}$ and $\mathbf{\Gamma}_{2,t}$ which defined in Eq. (53) to be

$$
\mathbf{\Gamma}_{1,t} = \frac{1}{N} \sum_{i=1}^{N} \widetilde{\boldsymbol{x}}_i \left[ \sigma\left( \widetilde{\boldsymbol{x}}_i^\top \widetilde{\boldsymbol{w}}_1^\natural \right) \sigma'\left( \widetilde{\boldsymbol{x}}_i^\top \boldsymbol{w}_{t,1} \right) \quad \dots \quad \sigma\left( \widetilde{\boldsymbol{x}}_i^\top \widetilde{\boldsymbol{w}}_k^\natural \right) \sigma'\left( \widetilde{\boldsymbol{x}}_i^\top \boldsymbol{w}_{t,k} \right) \right],
$$

and

$$
\mathbf{\Gamma}_{2,t} = \frac{1}{N} \sum_{i=1}^{N} \widetilde{\boldsymbol{x}}_i \left[ \sigma\left( \widetilde{\boldsymbol{x}}_i^\top \boldsymbol{w}_{t,1} \right) \sigma'\left( \widetilde{\boldsymbol{x}}_i^\top \boldsymbol{w}_{t,1} \right) \quad \dots \quad \sigma\left( \widetilde{\boldsymbol{x}}_i^\top \boldsymbol{w}_{t,k} \right) \sigma'\left( \widetilde{\boldsymbol{x}}_i^\top \boldsymbol{w}_{t,k} \right) \right],
$$

where $\boldsymbol{w}_{t,m}$ is the $m$-th column of $\boldsymbol{W}_t := \widetilde{\boldsymbol{W}}^\natural + \boldsymbol{A}_t \boldsymbol{B}_t$ and $\widetilde{\boldsymbol{w}}_m^\natural$ is the $m$-th column of $\widetilde{\boldsymbol{W}}^\natural$.

The following two lemmas provide the columnwise expectation of $\mathbf{\Gamma}_{1,t}$ and $\mathbf{\Gamma}_{2,t}$ respectively.

**Lemma D.1.** *Under assumptions in Section 2.1 for the nonlinear setting, for any $1 \le m \le k$, we have*

$$
\mathbb{E}_{\widetilde{\boldsymbol{x}}} \left[ \widetilde{\boldsymbol{x}} \sigma'\left( \widetilde{\boldsymbol{x}}^\top \boldsymbol{w}_{t,m} \right) \sigma\left( \widetilde{\boldsymbol{x}}^\top \widetilde{\boldsymbol{w}}_m^\natural \right) \right] = \frac{1}{2\pi} \left[ \frac{\|\widetilde{\boldsymbol{w}}_m^\natural\|_2}{\|\boldsymbol{w}_{t,m}\|_2} \sin\theta(\boldsymbol{w}_{t,m}, \widetilde{\boldsymbol{w}}_m^\natural) \boldsymbol{w}_{t,m} + \left( \pi - \theta(\boldsymbol{w}_{t,m}, \widetilde{\boldsymbol{w}}_m^\natural) \right) \widetilde{\boldsymbol{w}}_m^\natural \right], \tag{56}
$$

*and*

$$
\mathbb{E}_{\widetilde{\boldsymbol{x}}} \left[ \boldsymbol{x} \sigma'\left( \widetilde{\boldsymbol{x}}^\top \boldsymbol{w}_{t,m} \right) \sigma\left( \widetilde{\boldsymbol{x}}^\top \boldsymbol{w}_{t,m} \right) \right] = \frac{1}{2} \boldsymbol{w}_{t,m}. \tag{57}
$$

*Proof.* First, for any $1 \le m \le k$, we have

$$
\begin{aligned}
&\mathbb{E}_{\widetilde{\boldsymbol{x}}} \left[ \sigma'\left( \widetilde{\boldsymbol{x}}^\top \boldsymbol{w}_{t,m} \right) \sigma\left( \widetilde{\boldsymbol{x}}^\top \widetilde{\boldsymbol{w}}_m^\natural \right) \widetilde{\boldsymbol{x}} \right] \\
=& \frac{\partial}{\partial \boldsymbol{w}_{t,m}} \mathbb{E}_{\widetilde{\boldsymbol{x}}} \left[ \sigma\left( \widetilde{\boldsymbol{x}}^\top \boldsymbol{w}_{t,m} \right) \sigma\left( \widetilde{\boldsymbol{x}}^\top \widetilde{\boldsymbol{w}}_m^\natural \right) \right] \\
=& \frac{1}{2\pi} \left[ \frac{\|\widetilde{\boldsymbol{w}}_m^\natural\|_2}{\|\boldsymbol{w}_{t,m}\|_2} \sin\theta(\boldsymbol{w}_{t,m}, \widetilde{\boldsymbol{w}}_m^\natural) \boldsymbol{w}_{t,m} + \left( \pi - \theta(\boldsymbol{w}_{t,m}, \widetilde{\boldsymbol{w}}_m^\natural) \right) \widetilde{\boldsymbol{w}}_m^\natural \right]. \quad \text{[by Lemma E.4]}
\end{aligned}
$$

The proof for Eq. (57) is the same as that of Eq. (56), i.e.

$$
\begin{aligned}
&\mathbb{E}_{\widetilde{\boldsymbol{x}}} \left[ \sigma'\left( \widetilde{\boldsymbol{x}}^\top \boldsymbol{w}_{t,m} \right) \sigma\left( \widetilde{\boldsymbol{x}}^\top \boldsymbol{w}_{t,m} \right) \widetilde{\boldsymbol{x}} \right] \\
=& \frac{1}{2\pi} \left[ \frac{\|\boldsymbol{w}_{t,m}\|_2}{\|\boldsymbol{w}_{t,m}\|_2} \sin\theta(\boldsymbol{w}_{t,m}, \boldsymbol{w}_{t,m}) \boldsymbol{w}_{t,m} + \left( \pi - \theta(\boldsymbol{w}_{t,m}, \boldsymbol{w}_{t,m}) \right) \boldsymbol{w}_{t,m} \right] \quad \text{[by Lemma E.4]} \\
=& \frac{1}{2} \boldsymbol{w}_{t,m}.
\end{aligned}
$$

$\square$

Next, we can obtain the expected full gradients via the following lemma.

**Lemma D.2.** *Recall $\boldsymbol{W}_t := \boldsymbol{W}^\natural + \boldsymbol{A}_t \boldsymbol{B}_t$, suppose $\|\boldsymbol{A}_t \boldsymbol{B}_t - \Delta\|_F \le \rho \lambda_{r^*}^*$ for some constant $\rho \in (0,1)$, under assumptions in Section 2.1 for the nonlinear setting and Assumption 4.1, then it holds that*

$$
-\mathbb{E}_{\widetilde{\boldsymbol{x}}} \left[ \boldsymbol{J}_{\boldsymbol{W}_t} \right] = \frac{1}{2} \left( \boldsymbol{A}_t \boldsymbol{B}_t - \Delta \right) + \boldsymbol{\Psi}(t),
$$

*where $\boldsymbol{\Psi}(t)$ is defined as*

$$\boldsymbol{\Psi}(t) := (\boldsymbol{A}_t\boldsymbol{B}_t - \Delta)\,\mathbf{D}_1(t)\mathbf{D}_3(t) + \widetilde{\boldsymbol{W}}^\natural \Big[\mathbf{D}_1(t) - \mathbf{D}_2(t) - \mathbf{D}_1(t)(\boldsymbol{I}_k - \mathbf{D}_3(t))\Big],$$

*with*

$$\mathbf{D}_1(t) := \text{Diag}\left\{\sin\Big[\theta(\widetilde{\boldsymbol{w}}_1^\natural + \boldsymbol{r}_{t,1}\,,\widetilde{\boldsymbol{w}}_1^\natural)\Big]\,,\dots,\sin\Big[\theta(\widetilde{\boldsymbol{w}}_k^\natural + \boldsymbol{r}_{t,k}\,,\widetilde{\boldsymbol{w}}_k^\natural)\Big]\right\},$$

$$\mathbf{D}_2(t) := \text{Diag}\left\{\theta(\widetilde{\boldsymbol{w}}_1^\natural + \boldsymbol{r}_{t,1}\,,\widetilde{\boldsymbol{w}}_1^\natural)\,,\dots,\theta(\widetilde{\boldsymbol{w}}_k^\natural + \boldsymbol{r}_{t,k}\,,\widetilde{\boldsymbol{w}}_k^\natural)\right\},$$

$$\mathbf{D}_3(t) := \text{Diag}\left\{\frac{\|\widetilde{\boldsymbol{w}}_1^\natural\|_2}{\|\widetilde{\boldsymbol{w}}_1^\natural + \boldsymbol{r}_{t,1}\|_2}\,,\dots,\frac{\|\widetilde{\boldsymbol{w}}_k^\natural\|_2}{\|\widetilde{\boldsymbol{w}}_k^\natural + \boldsymbol{r}_{t,k}\|_2}\right\}. \tag{58}$$

*Then we have the following upper bound*

$$\frac{\|\boldsymbol{\Psi}(t)\|_{\mathrm{F}}}{\|\boldsymbol{A}_t\boldsymbol{B}_t - \Delta\|_{\mathrm{F}}} \leq \mathcal{O}\left(\frac{1}{\kappa r^*}\right).$$

*Proof.* We first give some notations here. Let $\boldsymbol{w}_{t,m}$ be the $m$-th column of $\boldsymbol{W}_t \in \mathbb{R}^{d\times k}$, $\boldsymbol{w}_m^\natural$ be the $m$-th column of $\widetilde{\boldsymbol{W}}^\natural$, $\Delta_m$ as the $m$-th of the low-rank shift $\Delta$, $[\boldsymbol{A}_t\boldsymbol{B}_t]_m$ as the $m$-th column of $\boldsymbol{A}_t\boldsymbol{B}_t$.

By Lemma D.1, we can derive $m$-th column of $-\mathbb{E}_{\widetilde{\boldsymbol{x}}}\left[\boldsymbol{J}_{\boldsymbol{W}_t}\right]$ for any $m = 1, 2, \cdots, k$ as

$$\mathbb{E}_{\widetilde{\boldsymbol{x}}}\left[\frac{1}{N}\sum_{i=1}^N \left(\sigma\left(\widetilde{\boldsymbol{x}}_i^\top\boldsymbol{w}_{t,m}\right) - \sigma\left(\widetilde{\boldsymbol{x}}_i^\top\widetilde{\boldsymbol{w}}_m^\natural\right)\right)\sigma'\left(\widetilde{\boldsymbol{x}}_i^\top\boldsymbol{w}_{t,m}\right)\widetilde{\boldsymbol{x}}_i\right]$$

$$=\mathbb{E}_{\widetilde{\boldsymbol{x}}}\left[\left(\sigma\left(\widetilde{\boldsymbol{x}}^\top\boldsymbol{w}_{t,m}\right) - \sigma\left(\widetilde{\boldsymbol{x}}^\top\widetilde{\boldsymbol{w}}_m^\natural\right)\right)\sigma'\left(\widetilde{\boldsymbol{x}}^\top\boldsymbol{w}_{t,m}\right)\widetilde{\boldsymbol{x}}\right]$$

$$=\frac{1}{2}(\boldsymbol{w}_{t,m} - \widetilde{\boldsymbol{w}}_m^\natural) - \frac{1}{2\pi}\left[\frac{\|\widetilde{\boldsymbol{w}}_m^\natural\|_2}{\|\boldsymbol{w}_{t,m}\|_2}\sin\theta(\boldsymbol{w}_{t,m}\,,\widetilde{\boldsymbol{w}}_m^\natural)\boldsymbol{w}_{t,m} - \theta(\boldsymbol{w}_{t,m}\,,\widetilde{\boldsymbol{w}}_m^\natural)\widetilde{\boldsymbol{w}}_m^\natural\right]$$

$$=\frac{1}{2}\left([\boldsymbol{A}_t\boldsymbol{B}_t]_m - \Delta_m\right) - \frac{1}{2\pi}\underbrace{\left[\frac{\|\widetilde{\boldsymbol{w}}_m^\natural\|_2}{\|\boldsymbol{w}_{t,m}\|_2}\sin\theta(\boldsymbol{w}_{t,m}\,,\widetilde{\boldsymbol{w}}_m^\natural)\boldsymbol{w}_{t,m} - \theta(\boldsymbol{w}_{t,m}\,,\widetilde{\boldsymbol{w}}_m^\natural)\widetilde{\boldsymbol{w}}_m^\natural\right]}_{\text{residual part R}}. \tag{59}$$

Denote

$$\boldsymbol{r}_{t,m} := [\boldsymbol{A}_t\boldsymbol{B}_t]_m - \Delta_m\,, \tag{60}$$

then we can write

$$\boldsymbol{w}_{t,m} = \widetilde{\boldsymbol{w}}_m^\natural + [\boldsymbol{A}_t\boldsymbol{B}_t]_m - \Delta_m = \widetilde{\boldsymbol{w}}_m^\natural + \boldsymbol{r}_{t,m}\,,$$

as a relative perturbed time-dependent vector. Next, we take it back to the residual part in Eq. (59)

$$\mathrm{R} = \frac{\|\widetilde{\boldsymbol{w}}_m^\natural\|_2}{\|\widetilde{\boldsymbol{w}}_m^\natural + \boldsymbol{r}_{t,m}\|_2}\sin\Big[\theta(\widetilde{\boldsymbol{w}}_m^\natural + \boldsymbol{r}_{t,m}\,,\widetilde{\boldsymbol{w}}_m^\natural)\Big](\widetilde{\boldsymbol{w}}_m^\natural + \boldsymbol{r}_{t,m}) - \theta(\widetilde{\boldsymbol{w}}_m^\natural + \boldsymbol{r}_{t,m}\,,\widetilde{\boldsymbol{w}}_m^\natural)\widetilde{\boldsymbol{w}}_m^\natural$$

$$=\frac{\|\widetilde{\boldsymbol{w}}_m^\natural\|_2}{\|\widetilde{\boldsymbol{w}}_m^\natural + \boldsymbol{r}_{t,m}\|_2}\sin\Big[\theta(\widetilde{\boldsymbol{w}}_m^\natural + \boldsymbol{r}_{t,m}\,,\widetilde{\boldsymbol{w}}_m^\natural)\Big]\boldsymbol{r}_{t,m}$$

$$-\frac{\|\widetilde{\boldsymbol{w}}_m^\natural + \boldsymbol{r}_{t,m}\|_2 - \|\widetilde{\boldsymbol{w}}_m^\natural\|_2}{\|\widetilde{\boldsymbol{w}}_m^\natural + \boldsymbol{r}_{t,m}\|_2}\sin\Big[\theta(\widetilde{\boldsymbol{w}}_m^\natural + \boldsymbol{r}_{t,m}\,,\widetilde{\boldsymbol{w}}_m^\natural)\Big]\widetilde{\boldsymbol{w}}_m^\natural$$

$$+\sin\Big[\theta(\widetilde{\boldsymbol{w}}_m^\natural + \boldsymbol{r}_{t,m}\,,\widetilde{\boldsymbol{w}}_m^\natural)\Big]\widetilde{\boldsymbol{w}}_m^\natural - \theta(\widetilde{\boldsymbol{w}}_m^\natural + \boldsymbol{r}_{t,m}\,,\widetilde{\boldsymbol{w}}_m^\natural)\widetilde{\boldsymbol{w}}_m^\natural\,. \tag{61}$$

Combining Eq. (59) and Eq. (61), we can write $-\mathbb{E}_{\widetilde{\boldsymbol{x}}}\left[\boldsymbol{J}_{\boldsymbol{W}_t}\right]$ in matrix form as

$$\mathbb{E}_{\widetilde{\boldsymbol{x}}}\left[\boldsymbol{J}_{\boldsymbol{W}_t}\right] = \frac{1}{2}\left(\boldsymbol{A}_t\boldsymbol{B}_t - \Delta\right) + (\boldsymbol{A}_t\boldsymbol{B}_t - \Delta)\,\mathbf{D}_1(t)\mathbf{D}_3(t) + \widetilde{\boldsymbol{W}}^\natural\Big[\mathbf{D}_1(t) - \mathbf{D}_2(t) - \mathbf{D}_1(t)(\boldsymbol{I}_k - \mathbf{D}_3(t))\Big].$$

Additionally, we define

$$\boldsymbol{\Psi}(t) := \left(\boldsymbol{A}_t\boldsymbol{B}_t - \Delta\right)\mathbf{D}_1(t)\mathbf{D}_3(t) + \widetilde{\boldsymbol{W}}^{\natural}\Big[\mathbf{D}_1(t) - \mathbf{D}_2(t) - \mathbf{D}_1(t)(\boldsymbol{I}_k - \mathbf{D}_3(t))\Big]$$

$$= \underbrace{\left(\boldsymbol{A}_t\boldsymbol{B}_t - \Delta\right)\mathbf{D}_1(t)\mathbf{D}_3(t)}_{:=\boldsymbol{\Psi}_1(t)} + \underbrace{\widetilde{\boldsymbol{W}}^{\natural}(\mathbf{D}_1(t) - \mathbf{D}_2(t))}_{:=\boldsymbol{\Psi}_2(t)} - \underbrace{\widetilde{\boldsymbol{W}}^{\natural}(\mathbf{D}_1(t)(\boldsymbol{I}_k - \mathbf{D}_3(t)))}_{:=\boldsymbol{\Psi}_3(t)} . \tag{62}$$

For notational simplicity, we drop time & column index and denote $\boldsymbol{w} := \widetilde{\boldsymbol{w}}_m^{\natural}$ and $\boldsymbol{r} := \boldsymbol{r}_{t,m}$. By condition $\|\boldsymbol{A}_t\boldsymbol{B}_t - \Delta\|_{\mathrm{F}} \leq \rho\lambda_{r^*}^*$, we have

$$\|\boldsymbol{r}\|_2 \leq \|\boldsymbol{A}_t\boldsymbol{B}_t - \Delta\|_{\mathrm{F}} \leq \rho\lambda_{r^*}^* .$$

Next, by Assumption 4.1, we can obtain

$$\frac{\|\boldsymbol{r}\|_2}{\|\boldsymbol{w}\|_2} \leq \frac{\rho\lambda_{r^*}^*}{\|\boldsymbol{w}\|_2} \leq \mathcal{O}\left(\frac{1}{\kappa r^*}\right) . \tag{63}$$

**Part I: Control the Angle $\theta(\boldsymbol{w} + \boldsymbol{r}, \boldsymbol{w})$**

We denote $\alpha := \frac{\langle\boldsymbol{r},\boldsymbol{w}\rangle}{\|\boldsymbol{w}\|_2^2}$ and $\beta := \frac{\|\boldsymbol{r}\|_2^2}{\|\boldsymbol{w}\|_2^2}$, then we can derive

$$\cos\theta(\boldsymbol{w} + \boldsymbol{r}, \boldsymbol{w}) = \frac{1 + \frac{\langle\boldsymbol{r},\boldsymbol{w}\rangle}{\|\boldsymbol{w}\|_2^2}}{\sqrt{1 + 2\frac{\langle\boldsymbol{r},\boldsymbol{w}\rangle}{\|\boldsymbol{w}\|_2^2} + \frac{\|\boldsymbol{r}\|_2^2}{\|\boldsymbol{w}\|_2^2}}} = \frac{1 + \alpha}{\sqrt{1 + 2\alpha + \beta}} ,$$

which can imply

$$\sin\theta(\boldsymbol{w} + \boldsymbol{r}, \boldsymbol{w}) = \sqrt{1 - \cos^2\theta(\boldsymbol{w} + \boldsymbol{r}, \boldsymbol{w})} = \sqrt{\frac{\beta - \alpha^2}{1 + 2\alpha + \beta}} = \Theta\left(\sqrt{\beta - \alpha^2}\right) = \mathcal{O}\left(\frac{\|\boldsymbol{r}\|_2}{\|\boldsymbol{w}\|_2}\right) . \tag{64}$$

By consequence, we can obtain

$$\theta(\boldsymbol{w} + \boldsymbol{r}, \boldsymbol{w}) = \mathcal{O}\left(\frac{\|\boldsymbol{r}\|_2}{\|\boldsymbol{w}\|_2}\right) . \tag{65}$$

Lastly, by the fact that $x - \sin x \leq \frac{x^3}{6}$ if $x \geq 0$, then we can have

$$\theta(\boldsymbol{w} + \boldsymbol{r}, \boldsymbol{w}) - \sin\theta(\boldsymbol{w} + \boldsymbol{r}, \boldsymbol{w}) = \mathcal{O}\left(\frac{\|\boldsymbol{r}\|_2^3}{\|\boldsymbol{w}\|_2^3}\right) . \tag{66}$$

**Part II: Control the Ratio $\left|1 - \frac{\|\boldsymbol{w}\|_2}{\|\boldsymbol{w}+\boldsymbol{r}\|_2}\right|$**

We can compute

$$\left|1 - \frac{\|\boldsymbol{w}\|_2}{\|\boldsymbol{w} + \boldsymbol{r}\|_2}\right| = \left|1 - \frac{1}{\sqrt{1 + 2\alpha + \beta}}\right| = \left|1 - \Theta\left(1 - \alpha - \frac{\beta}{2}\right)\right| = \mathcal{O}\left(\frac{\|\boldsymbol{r}\|_2}{\|\boldsymbol{w}\|_2}\right) . \tag{67}$$

where the second equality follows from the first order binomial approximation $\frac{1}{\sqrt{1+x}} = \Theta\left(1 - \frac{x}{2}\right)$ if $|x| \ll 1$ and we have $\frac{\|\boldsymbol{r}\|_2}{\|\boldsymbol{w}\|_2} = \mathcal{O}\left(\frac{1}{\kappa r^*}\right)$ by Eq. (63). By consequence, we can have

$$\frac{\|\boldsymbol{w}\|_2}{\|\boldsymbol{w} + \boldsymbol{r}\|_2} \leq 1 + \mathcal{O}\left(\frac{\|\boldsymbol{r}\|_2}{\|\boldsymbol{w}\|_2}\right) . \tag{68}$$

Now, we can upper bound Eq. (62) in terms of Frobenius norm by triangle inequality, i.e.

$$\|\boldsymbol{\Psi}(t)\|_{\mathrm{F}} \leq \|\boldsymbol{\Psi}_1(t)\|_{\mathrm{F}} + \|\boldsymbol{\Psi}_2(t)\|_{\mathrm{F}} + \|\boldsymbol{\Psi}_3(t)\|_{\mathrm{F}} .$$

For $\|\mathbf{\Psi}_1(t)\|_{\mathrm{F}}$, we have

$$
\begin{aligned}
\|\mathbf{\Psi}_1(t)\|_{\mathrm{F}} &\leq \max_{1 \leq m \leq k} \frac{\|\widetilde{\boldsymbol{w}}_m^\natural\|_2 \sin\left[\theta(\widetilde{\boldsymbol{w}}_m^\natural + \boldsymbol{r}_{t,1}, \widetilde{\boldsymbol{w}}_m^\natural)\right]}{\|\widetilde{\boldsymbol{w}}_m^\natural + \boldsymbol{r}_{t,m}\|_2} \|\boldsymbol{A}_t \boldsymbol{B}_t - \Delta\|_{\mathrm{F}} \\
&\leq \max_{1 \leq m \leq k} \left(1 + \mathcal{O}\left(\frac{\|\boldsymbol{r}_{t,m}\|_2}{\|\widetilde{\boldsymbol{w}}_m^\natural\|_2}\right)\right) \mathcal{O}\left(\frac{\|\boldsymbol{r}_{t,m}\|_2}{\|\widetilde{\boldsymbol{w}}_m^\natural\|_2}\right) \|\boldsymbol{A}_t \boldsymbol{B}_t - \Delta\|_{\mathrm{F}} && \text{[by Eq. (64) and Eq. (68)]} \\
&\leq \mathcal{O}\left(\max_{1 \leq m \leq k} \frac{\|\boldsymbol{r}_{t,m}\|_2}{\|\widetilde{\boldsymbol{w}}_m^\natural\|_2}\right) \|\boldsymbol{A}_t \boldsymbol{B}_t - \Delta\|_{\mathrm{F}} \leq \mathcal{O}\left(\frac{1}{\kappa r^*}\right) \|\boldsymbol{A}_t \boldsymbol{B}_t - \Delta\|_{\mathrm{F}} . && \text{[by Eq. (63)]}
\end{aligned}
$$

For $\|\mathbf{\Psi}_2(t)\|_{\mathrm{F}}$, we have

$$
\begin{aligned}
\|\mathbf{\Psi}_2(t)\|_{\mathrm{F}} &\leq \|\widetilde{\boldsymbol{W}}^\natural\|_{op} \|\mathbf{D}_1(t) - \mathbf{D}_2(t)\|_{\mathrm{F}} \\
&= \|\widetilde{\boldsymbol{W}}^\natural\|_{op} \sqrt{\sum_{m=1}^{k} \left(\sin\left[\theta(\widetilde{\boldsymbol{w}}_m^\natural + \boldsymbol{r}_{t,m}, \widetilde{\boldsymbol{w}}_m^\natural)\right] - \theta(\widetilde{\boldsymbol{w}}_m^\natural + \boldsymbol{r}_{t,m}, \widetilde{\boldsymbol{w}}_m^\natural)\right)^2} \\
&\leq \|\widetilde{\boldsymbol{W}}^\natural\|_{op} \sqrt{\sum_{m=1}^{k} \mathcal{O}\left(\frac{\|\boldsymbol{r}_{t,m}\|_2^3}{\|\widetilde{\boldsymbol{w}}_m^\natural\|_2^3}\right)^2} && \text{[by Eq. (66)]} \\
&\leq \|\widetilde{\boldsymbol{W}}^\natural\|_{op} \sqrt{\sum_{m=1}^{k} \mathcal{O}\left(\|\boldsymbol{r}_{t,m}\|_2 \max_{1 \leq i \leq k} \frac{\|\boldsymbol{r}_{t,i}\|_2^2}{\|\widetilde{\boldsymbol{w}}_i^\natural\|_2^3}\right)^2} \\
&= \mathcal{O}\left(\|\widetilde{\boldsymbol{W}}^\natural\|_{op} \max_{1 \leq i \leq k} \frac{\|\boldsymbol{r}_{t,i}\|_2^2}{\|\widetilde{\boldsymbol{w}}_i^\natural\|_2^3}\right) \|\boldsymbol{A}_t \boldsymbol{B}_t - \Delta\|_{\mathrm{F}} \\
&\leq \mathcal{O}\left(\frac{\|\widetilde{\boldsymbol{W}}^\natural\|_{op}}{\min_{1 \leq i \leq k} \|\widetilde{\boldsymbol{w}}_i^\natural\|_2} \max_{1 \leq i \leq k} \frac{\|\boldsymbol{r}_{t,i}\|_2^2}{\|\widetilde{\boldsymbol{w}}_i^\natural\|_2^2}\right) \|\boldsymbol{A}_t \boldsymbol{B}_t - \Delta\|_{\mathrm{F}} \\
&\leq \mathcal{O}\left(\frac{1}{(\kappa r^*)^2}\right) \|\boldsymbol{A}_t \boldsymbol{B}_t - \Delta\|_{\mathrm{F}} . && \text{[by Assumption 4.1 and Eq. (63)]}
\end{aligned}
$$

For $\|\mathbf{\Psi}_3(t)\|_{\mathrm{F}}$, we have

$$
\begin{aligned}
\|\mathbf{\Psi}_3(t)\|_{\mathrm{F}} &\leq \|\widetilde{\boldsymbol{W}}^\natural\|_{op} \|\mathbf{D}_1(t)(\boldsymbol{I}_k - \mathbf{D}_3(t))\|_{\mathrm{F}} \\
&\leq \|\widetilde{\boldsymbol{W}}^\natural\|_{op} \sqrt{\sum_{m=1}^{k} \left[\left(1 - \frac{\|\widetilde{\boldsymbol{w}}_m^\natural\|_2}{\|\widetilde{\boldsymbol{w}}_m^\natural + \boldsymbol{r}_{t,m}\|_2}\right) \sin\left[\theta(\widetilde{\boldsymbol{w}}_m^\natural + \boldsymbol{r}_{t,m}, \widetilde{\boldsymbol{w}}_m^\natural)\right]\right]^2} \\
&\leq \|\widetilde{\boldsymbol{W}}^\natural\|_{op} \max_{1 \leq m \leq k} \left|\left(1 - \frac{\|\widetilde{\boldsymbol{w}}_m^\natural\|_2}{\|\widetilde{\boldsymbol{w}}_m^\natural + \boldsymbol{r}_{t,m}\|_2}\right)\right| \times \sqrt{\sum_{m=1}^{k} \sin^2\left[\theta(\widetilde{\boldsymbol{w}}_m^\natural + \boldsymbol{r}_{t,m}, \widetilde{\boldsymbol{w}}_m^\natural)\right]} \\
&\leq \|\widetilde{\boldsymbol{W}}^\natural\|_{op} \max_{1 \leq m \leq k} \mathcal{O}\left(\frac{\|\boldsymbol{r}_{t,m}\|_2}{\|\widetilde{\boldsymbol{w}}_m^\natural\|_2}\right) \sqrt{\sum_{m=1}^{k} \mathcal{O}\left(\frac{\|\boldsymbol{r}_{t,m}\|_2^2}{\|\widetilde{\boldsymbol{w}}_m^\natural\|_2^2}\right)} && \text{[by Eq. (64) and Eq. (67)]} \\
&\leq \|\widetilde{\boldsymbol{W}}^\natural\|_{op} \max_{1 \leq m \leq k} \mathcal{O}\left(\frac{\|\boldsymbol{r}_{t,m}\|_2}{\|\widetilde{\boldsymbol{w}}_m^\natural\|_2}\right) \sqrt{\mathcal{O}\left(\frac{\sum_{m=1}^{k} \|\boldsymbol{r}_{t,m}\|_2^2}{\min_{1 \leq i \leq k} \|\widetilde{\boldsymbol{w}}_i^\natural\|_2^2}\right)} && \text{[due to the positivity of } \|\boldsymbol{r}_{t,m}\|_2] \\
&= \mathcal{O}\left(\frac{\|\widetilde{\boldsymbol{W}}^\natural\|_{op}}{\min_{1 \leq i \leq k} \|\widetilde{\boldsymbol{w}}_i^\natural\|_2} \max_{1 \leq m \leq k} \frac{\|\boldsymbol{r}_{t,m}\|_2}{\|\widetilde{\boldsymbol{w}}_m^\natural\|_2}\right) \|\boldsymbol{A}_t \boldsymbol{B}_t - \Delta\|_{\mathrm{F}} \\
&\leq \mathcal{O}\left(\frac{1}{\kappa r^*}\right) \|\boldsymbol{A}_t \boldsymbol{B}_t - \Delta\|_{\mathrm{F}} . && \text{[by Assumption 4.1 and Eq. (63)]}
\end{aligned}
$$

Combine the above upper bounds together, we can obtain

$$\frac{\|\boldsymbol{\Psi}(t)\|_{\mathrm{F}}}{\|\boldsymbol{A}_t\boldsymbol{B}_t - \Delta\|_{\mathrm{F}}} \leq \mathcal{O}\left(\frac{1}{\kappa r^*}\right),$$

which completes the proof. □

### D.1.2. Concentration of Empirical Gradients

In this part, we aim to provide the concentration of empirical gradient $\boldsymbol{J}_{\boldsymbol{W}_t} := \boldsymbol{\Gamma}_{1,t} - \boldsymbol{\Gamma}_{2,t} \in \mathbb{R}^{d \times k}$ in Frobenius norm. Recall $\boldsymbol{W}_t := \boldsymbol{W}^{\natural} + \boldsymbol{A}_t\boldsymbol{B}_t$ and $\boldsymbol{w}_{t,m}$ is the corresponding $m$-th column of $\boldsymbol{W}_t$, denote $\widetilde{x}_{i,j}$ as the $j$-th element of $\widetilde{\boldsymbol{x}}_i$, for notational simplicity, we define each element of $\boldsymbol{J}_{\boldsymbol{W}_t} := \boldsymbol{\Gamma}_{1,t} - \boldsymbol{\Gamma}_{2,t}$ as

$$c_{t,m}^{j}(\widetilde{\boldsymbol{x}}_i) := \left(\sigma\left(\widetilde{\boldsymbol{x}}_i^{\top}\widetilde{\boldsymbol{w}}_m^{\natural}\right) - \sigma\left(\widetilde{\boldsymbol{x}}_i^{\top}\boldsymbol{w}_{t,m}^{\natural}\right)\right)\sigma'\left(\widetilde{\boldsymbol{x}}_i^{\top}\boldsymbol{w}_{t,m}^{\natural}\right)\widetilde{x}_{i,j} \in \mathbb{R}, \quad \text{for } 1 \leq m \leq k, 1 \leq i \leq N, 1 \leq j \leq d,$$

Then, we can write $\boldsymbol{J}_{\boldsymbol{W}_t}$ in an element-wise way

$$\boldsymbol{J}_{\boldsymbol{W}_t} = \frac{1}{N}\sum_{i=1}^{N}\begin{bmatrix} c_{t,1}^{1}(\widetilde{\boldsymbol{x}}_i) & \cdots & c_{t,k}^{1}(\widetilde{\boldsymbol{x}}_i) \\ \vdots & \ddots & \vdots \\ c_{t,1}^{d}(\widetilde{\boldsymbol{x}}_i) & \cdots & c_{t,k}^{d}(\widetilde{\boldsymbol{x}}_i) \end{bmatrix} \in \mathbb{R}^{d \times k},$$

and

$$\left\|\boldsymbol{J}_{\boldsymbol{W}_t} - \mathbb{E}_{\widetilde{\boldsymbol{x}}}[\boldsymbol{J}_{\boldsymbol{W}_t}]\right\|_{\mathrm{F}}^2 = \sum_{j=1}^{d}\sum_{m=1}^{k}\left(\frac{1}{N}\sum_{i=1}^{N}c_{t,m}^{j}(\widetilde{\boldsymbol{x}}_i) - \mathbb{E}_{\widetilde{\boldsymbol{x}}}\left[c_{t,m}^{j}(\widetilde{\boldsymbol{x}})\right]\right)^2.$$

Next, we have the following lemma.

**Lemma D.3.** *For $1 \leq m \leq k$, $1 \leq j \leq d$, under assumptions in Section 2.1 for the nonlinear setting, with probability at least $1 - 2C\exp\left(-N\epsilon^2\right)$ for a universal constant $C > 0$ and $\epsilon \in (0,1)$, we have*

$$\left|\frac{1}{N}\sum_{i=1}^{N}c_{t,m}^{j}(\widetilde{\boldsymbol{x}}_i) - \mathbb{E}_{\widetilde{\boldsymbol{x}}}\left[c_{t,m}^{j}(\widetilde{\boldsymbol{x}})\right]\right| \leq C^*K^2\epsilon\|\widetilde{\boldsymbol{w}}_m^{\natural} - \boldsymbol{w}_{t,m}^{\natural}\|_2,$$

*for some absolute constant $C^* > 0$ and $K = \sqrt{8/3}$.*

*Proof.* Since $\widetilde{x}_{i,j} \sim \mathcal{N}(0,1)$ for any $1 \leq m \leq k$ and $1 \leq j \leq d$, then we have that $K := \|\widetilde{x}_{i,j}\|_{\psi_2} = \sqrt{8/3}$. By the Orlicz-based definition of subgaussian norm, the subgaussian norm of random variable is identical to its absolute value. Then, for any $\lambda \in \mathbb{R}$, we have the following moment generating function

$$\mathbb{E}\left[\exp\left(\lambda\left|\left(\sigma\left(\widetilde{\boldsymbol{x}}_i^{\top}\widetilde{\boldsymbol{w}}_m^{\natural}\right) - \sigma\left(\widetilde{\boldsymbol{x}}_i^{\top}\boldsymbol{w}_{t,m}^{\natural}\right)\right)\sigma'\left(\widetilde{\boldsymbol{x}}_i^{\top}\boldsymbol{w}_{t,m}^{\natural}\right)\right|\right)\right]$$

$$\leq \mathbb{E}\left[\exp\left(\lambda\left|\left\langle\widetilde{\boldsymbol{x}}_i, \widetilde{\boldsymbol{w}}_m^{\natural} - \boldsymbol{w}_{t,m}^{\natural}\right\rangle\right|\right)\right] \qquad \text{[by Lipschitz continuity of } \sigma \text{ and } \sigma']$$

$$\leq \mathbb{E}\left[\exp\left((C^*)^2\lambda^2\left\|\left|\left\langle\widetilde{\boldsymbol{x}}_i, \widetilde{\boldsymbol{w}}_m^{\natural} - \boldsymbol{w}_{t,m}^{\natural}\right\rangle\right|\right\|_{\psi_2}^2\right)\right], \qquad \text{[by subgaussian property]}$$

for some constant $C^* > 0$, which implies

$$\left\|\left(\sigma\left(\widetilde{\boldsymbol{x}}_i^{\top}\widetilde{\boldsymbol{w}}_m^{\natural}\right) - \sigma\left(\widetilde{\boldsymbol{x}}_i^{\top}\boldsymbol{w}_{t,m}^{\natural}\right)\right)\sigma'\left(\widetilde{\boldsymbol{x}}_i^{\top}\boldsymbol{w}_{t,m}^{\natural}\right)\right\|_{\psi_2}^2 \leq (C^*)^2\left\|\left|\left\langle\widetilde{\boldsymbol{x}}_i, \widetilde{\boldsymbol{w}}_m^{\natural} - \boldsymbol{w}_{t,m}^{\natural}\right\rangle\right|\right\|_{\psi_2}^2 = (C^*K)^2\|\widetilde{\boldsymbol{w}}_m^{\natural} - \boldsymbol{w}_{t,m}^{\natural}\|_2^2,$$

where the last inequality follows from the fact that $\|X\|_{\psi_2} = Ks$ if $X \sim \mathcal{N}(0,s^2)$. Therefore, by Vershynin (2018, Lemma 2.7.7), this implies $c_{t,m}^{j}(\widetilde{\boldsymbol{x}}_i)$ is sub-exponential with

$$B_{t,m} := \|c_{t,m}^{j}(\widetilde{\boldsymbol{x}})\|_{\psi_1} \leq \|\widetilde{x}_{i,j}\|_{\psi_2}\left\|\left(\sigma\left(\widetilde{\boldsymbol{x}}_i^{\top}\widetilde{\boldsymbol{w}}_m^{\natural}\right) - \sigma\left(\widetilde{\boldsymbol{x}}_i^{\top}\boldsymbol{w}_{t,m}^{\natural}\right)\right)\sigma'\left(\widetilde{\boldsymbol{x}}_i^{\top}\boldsymbol{w}_{t,m}^{\natural}\right)\right\|_{\psi_2} \leq C^*K^2\|\widetilde{\boldsymbol{w}}_m^{\natural} - \boldsymbol{w}_{t,m}^{\natural}\|_2. \quad (69)$$

Then, let $\epsilon_{t,m} = C^* K^2 \epsilon \|\widetilde{\boldsymbol{w}}_m^{\natural} - \boldsymbol{w}_{t,m}^{\natural}\|_2$ for $\epsilon \in (0,1)$, we can apply Bernstein's inequality for sub-exponential variables Vershynin (2018, Corollary 2.8.3)

$$\mathbb{P}\left( \left| \frac{1}{N} \sum_{i=1}^{N} c_{t,m}^j (\widetilde{\boldsymbol{x}}_i) - \mathbb{E}_{\widetilde{\boldsymbol{x}}} \left[ c_{t,m}^j (\widetilde{\boldsymbol{x}}) \right] \right| \geq \epsilon_{t,m} \right)$$

$$\leq 2C \exp\left( -N \min\left\{ \frac{\epsilon_{t,m}}{B_{t,m}}, \frac{\epsilon_{t,m}^2}{B_{t,m}^2} \right\} \right) \qquad\qquad [\text{ for some constant } C > 0]$$

$$\leq 2C \exp\left( -N\epsilon^2 \right) . \qquad\qquad [\text{by Eq. (69) and } \epsilon \in (0,1)]$$

$\square$

**Theorem D.4.** *Suppose $\epsilon \in (0,1)$, under assumptions in Section 2.1 for the nonlinear setting, then with probability at least $1 - 2Cdk \exp\left( -N\epsilon^2 \right)$ for a universal constant $C > 0$, we have*

$$\left\| \boldsymbol{J}_{\boldsymbol{W}_t} - \mathbb{E}_{\widetilde{\boldsymbol{x}}} \left[ \boldsymbol{J}_{\boldsymbol{W}_t} \right] \right\|_{\mathrm{F}} \leq C^* K^2 \sqrt{d}\epsilon \|\boldsymbol{A}_t \boldsymbol{B}_t - \Delta\|_{\mathrm{F}} ,$$

*for some absolute constant $C^* > 0$ and $K = \sqrt{8/3}$.*

*Proof.* By a union bound argument and Lemma D.3, with probability at least $1 - 2Cdk \exp\left( -N\epsilon^2 \right)$ for a universal constant $C > 0$, we have

$$\left\| \boldsymbol{J}_{\boldsymbol{W}_t} - \mathbb{E}_{\widetilde{\boldsymbol{x}}} \left[ \boldsymbol{J}_{\boldsymbol{W}_t} \right] \right\|_{\mathrm{F}}^2 = \sum_{j=1}^{d} \sum_{m=1}^{k} \left( \frac{1}{N} \sum_{i=1}^{N} c_{t,m}^j (\widetilde{\boldsymbol{x}}_i) - \mathbb{E}_{\widetilde{\boldsymbol{x}}} \left[ c_{t,m}^j (\widetilde{\boldsymbol{x}}) \right] \right)^2$$

$$\leq \sum_{j=1}^{d} \sum_{m=1}^{k} \epsilon_{t,m}^2$$

$$\leq \sum_{j=1}^{d} \sum_{m=1}^{k} (C^* K^2)^2 \epsilon^2 \|\widetilde{\boldsymbol{w}}_m^{\natural} - \boldsymbol{w}_{t,m}^{\natural}\|_2^2$$

$$= d(C^* K^2)^2 \epsilon^2 \|\boldsymbol{A}_t \boldsymbol{B}_t - \Delta\|_{\mathrm{F}}^2 ,$$

which implies

$$\left\| \boldsymbol{J}_{\boldsymbol{W}_t} - \mathbb{E}_{\widetilde{\boldsymbol{x}}} \left[ \boldsymbol{J}_{\boldsymbol{W}_t} \right] \right\|_{\mathrm{F}} \leq C^* K^2 \sqrt{d}\epsilon \|\boldsymbol{A}_t \boldsymbol{B}_t - \Delta\|_{\mathrm{F}} ,$$

which finishes the proof. $\square$

**Lemma D.5.** *Recall $\boldsymbol{G}^{\natural} := -\nabla L(\boldsymbol{W}^{\natural}) = \boldsymbol{J}_{\boldsymbol{W}^{\natural}}$, under Assumption 4.1 and assumptions in Section 2.1 for the nonlinear setting, with (Spectral-init), suppose $\epsilon \leq \frac{\rho}{3C^* K^2 \gamma \sqrt{2d} r^* \kappa}$ for some positive constant $\rho > 0$ and we set $\gamma = 2$, then with probability at least $1 - 2Cdk \exp(-N\epsilon^2)$ for a universal constant $C > 0$, it holds that*

$$\|\boldsymbol{A}_0 \boldsymbol{B}_0 - \Delta\|_{\mathrm{F}} \leq \rho \lambda_{r^*}^* .$$

*Proof.* We start with decompose $\|\boldsymbol{A}_0 \boldsymbol{B}_0 - \Delta\|_{op}$ into three components, i.e.

$$\|\boldsymbol{A}_0 \boldsymbol{B}_0 - \Delta\|_{op} \leq \underbrace{\left\| \boldsymbol{A}_0 \boldsymbol{B}_0 - \gamma \boldsymbol{G}^{\natural} \right\|_{op}}_{\text{low-rank approximation error}} + \gamma \underbrace{\left\| \boldsymbol{G}^{\natural} - \mathbb{E}_{\widetilde{\boldsymbol{x}}} \left[ \boldsymbol{G}^{\natural} \right] \right\|_{op}}_{\text{concentration error}} + \underbrace{\left\| \gamma \mathbb{E}_{\widetilde{\boldsymbol{x}}} \left[ \boldsymbol{G}^{\natural} \right] - \Delta \right\|_{op}}_{\text{population error}} . \qquad (70)$$

First, for the population error, we can use similar technique from Lemma D.2, since $\frac{\|\Delta_m\|_2}{\|\widetilde{\boldsymbol{w}}_m^{\natural}\|_2} = \mathcal{O}\left( \frac{1}{\kappa r^*} \right)$ by Assumption 4.1 for $1 \leq m \leq k$, we can obtain

$$\frac{\left\| \mathbb{E}_{\widetilde{\boldsymbol{x}}} \left[ \boldsymbol{G}^{\natural} \right] - \frac{1}{2}\Delta \right\|_{\mathrm{F}}}{\|\Delta\|_{\mathrm{F}}} \leq \mathcal{O}\left( \frac{1}{\kappa r^*} \right) .$$

Next, for the concentration error, following Theorem D.4, we replace $\boldsymbol{W}_t$ with $\boldsymbol{W}^{\natural}$ and then obtain the following concentration with the probability at least $1 - 2Cdk\exp(-N\epsilon^2)$ for a universal constant $C > 0$, we have

$$\left\|\boldsymbol{G}^{\natural} - \mathbb{E}_{\widetilde{\boldsymbol{x}}}\left[\boldsymbol{G}^{\natural}\right]\right\|_{\mathrm{F}} \leq \frac{\rho\|\Delta\|_{\mathrm{F}}}{3\sqrt{2}r^*\gamma\kappa} \leq \frac{\rho\sqrt{r^*}\|\Delta\|_{op}}{3\sqrt{2}r^*\gamma\kappa} = \frac{\rho\lambda_{r^*}^*}{3\sqrt{2r^*}\gamma}, \tag{71}$$

where $\epsilon \leq \frac{\rho}{2C^*K^2\gamma\sqrt{2d}r^*\kappa}$ for $\rho > 0$.

Lastly, we can upper bound the $(r^* + 1)$-th singular value of $\boldsymbol{G}^{\natural}$ (with scale parameter $\gamma$) which acts as the low-rank approximation error. Due to the randomness contained in $\boldsymbol{G}^{\natural}$, we decompose the $(r^* + 1)$-th singular value into two components, i.e.

$$\gamma\lambda_{r^*+1}\left(\boldsymbol{G}^{\natural}\right) \leq \underbrace{\left|\gamma\lambda_{r^*+1}\left(\boldsymbol{G}^{\natural}\right) - \lambda_{r^*+1}\left(\gamma\mathbb{E}_{\widetilde{\boldsymbol{x}}}\left[\boldsymbol{G}^{\natural}\right]\right)\right|}_{\text{concentration error}} + \underbrace{\lambda_{r^*+1}\left(\gamma\mathbb{E}_{\widetilde{\boldsymbol{x}}}\left[\boldsymbol{G}^{\natural}\right]\right)}_{\text{population error}}.$$

First, for the concentration error, we can obtain

$$\begin{aligned}
&\left|\gamma\lambda_{r^*+1}\left(\boldsymbol{G}^{\natural}\right) - \lambda_{r^*+1}\left(\gamma\mathbb{E}_{\widetilde{\boldsymbol{x}}}\left[\boldsymbol{G}^{\natural}\right]\right)\right| \\
\leq{}& \gamma\left\|\boldsymbol{G}^{\natural} - \mathbb{E}_{\widetilde{\boldsymbol{x}}}\left[\boldsymbol{G}^{\natural}\right]\right\|_{op} && \text{[by Weyl's inequality]} \\
\leq{}& \gamma\left\|\boldsymbol{G}^{\natural} - \mathbb{E}_{\widetilde{\boldsymbol{x}}}\left[\boldsymbol{G}^{\natural}\right]\right\|_{\mathrm{F}} \\
\leq{}& \frac{\rho\lambda_{r^*}^*}{3\sqrt{2r^*}}. && \text{[by Eq. (71)]}
\end{aligned}$$

Second, we can obtain the population error as

$$\begin{aligned}
\lambda_{r^*+1}\left(\mathbb{E}_{\widetilde{\boldsymbol{x}}}\left[\boldsymbol{G}^{\natural}\right]\right) &= \left|\lambda_{r^*+1}\left(\mathbb{E}_{\widetilde{\boldsymbol{x}}}\left[\boldsymbol{G}^{\natural}\right]\right) - \frac{1}{2}\lambda_{r^*+1}\left(\Delta\right)\right| && \text{[since } \mathrm{Rank}(\Delta) = r^*] \\
&\leq \left\|\mathbb{E}_{\widetilde{\boldsymbol{x}}}\left[\boldsymbol{G}^{\natural}\right] - \frac{1}{2}\Delta\right\|_{op} && \text{[by Weyl's inequality]} \\
&\leq \left\|\mathbb{E}_{\widetilde{\boldsymbol{x}}}\left[\boldsymbol{G}^{\natural}\right] - \frac{1}{2}\Delta\right\|_{\mathrm{F}} \\
&\leq \mathcal{O}\left(\frac{1}{\kappa r^*}\right)\|\Delta\|_{\mathrm{F}}.
\end{aligned}$$

Now we can have

$$\gamma\lambda_{r^*+1}\left(\boldsymbol{G}^{\natural}\right) \leq \frac{\rho\lambda_{r^*}^*}{3\sqrt{2r^*}} + \mathcal{O}\left(\frac{\|\Delta\|_{\mathrm{F}}}{\kappa r^*}\right) \leq \mathcal{O}\left(\frac{1}{\sqrt{r^*}}\right)\rho\lambda_{r^*}^*. \tag{72}$$

Therefore, combine everything together, recall Eq. (70), we can obtain

$$\begin{aligned}
\|\boldsymbol{A}_0\boldsymbol{B}_0 - \Delta\|_{op} &\leq \gamma\lambda_{r^*+1}\left(\boldsymbol{G}^{\natural}\right) + \gamma\left\|\boldsymbol{G}^{\natural} - \mathbb{E}_{\widetilde{\boldsymbol{x}}}\left[\boldsymbol{G}^{\natural}\right]\right\|_{\mathrm{F}} + \gamma\left\|\mathbb{E}_{\widetilde{\boldsymbol{x}}}\left[\boldsymbol{G}^{\natural}\right] - \frac{1}{\gamma}\Delta\right\|_{\mathrm{F}} \\
&\leq \mathcal{O}\left(\frac{1}{\sqrt{r^*}}\right)\rho\lambda_{r^*}^* + \frac{\rho\lambda_{r^*}^*}{3\sqrt{2r^*}} + \mathcal{O}\left(\frac{1}{\sqrt{r^*}}\right)\rho\lambda_{r^*}^* \leq \frac{\rho\lambda_{r^*}^*}{\sqrt{2r^*}}. \tag{73}
\end{aligned}$$

Since we work in the exact-rank case $\mathrm{Rank}\left(\boldsymbol{A}_t\boldsymbol{B}_t\right) \leq r = r^*$ with $\mathrm{Rank}(\Delta) = r^*$, then $\mathrm{Rank}(\boldsymbol{A}_0\boldsymbol{B}_0 - \Delta) \leq 2r^*$, this can imply

$$\|\boldsymbol{A}_0\boldsymbol{B}_0 - \Delta\|_{\mathrm{F}} \leq \sqrt{2r^*}\|\boldsymbol{A}_0\boldsymbol{B}_0 - \Delta\|_{op} \leq \rho\lambda_{r^*}^*,$$

which completes the proof. $\qquad\square$

## D.2. Preconditioned Gradient Descent under Spectral Initialization

Recall the loss of LoRA fine-tuning:

$$\widetilde{L}\left(\boldsymbol{A}_t,\boldsymbol{B}_t\right) = \frac{1}{2N}\left\|\sigma\left(\widetilde{\boldsymbol{X}}\left(\boldsymbol{W}^\natural + \boldsymbol{A}_t\boldsymbol{B}_t\right)\right) - \sigma\left(\widetilde{\boldsymbol{X}}\widetilde{\boldsymbol{W}}^\natural\right)\right\|_{\mathrm{F}}^2.$$

Then, we employ the following preconditioned gradient updates for LoRA fine-tuning

$$\boldsymbol{A}_{t+1} = \boldsymbol{A}_t + \eta\boldsymbol{J}_{\boldsymbol{W}_t}\boldsymbol{B}_t^\top\left(\boldsymbol{B}_t\boldsymbol{B}_t^\top\right)^{-1}, \tag{74}$$

and

$$\boldsymbol{B}_{t+1} = \boldsymbol{B}_t + \eta\left(\boldsymbol{A}_t^\top\boldsymbol{A}_t\right)^{-1}\boldsymbol{A}_t^\top\boldsymbol{J}_{\boldsymbol{W}_t}. \tag{75}$$

Similar to the linear case, we define the following notations

- SVD of product matrix $\boldsymbol{A}_t\boldsymbol{B}_t := \mathcal{U}_t\mathcal{S}_t\mathcal{V}_t^\top$, where $\mathcal{U}_t \in \mathbb{R}^{d\times r}$, $\mathcal{S}_t \in \mathbb{R}^{r^*\times r}$, and $\mathcal{V}_t \in \mathbb{R}^{k\times r}$.

- The left singular matrix of $\boldsymbol{A}_t$ as $\boldsymbol{U}_{\boldsymbol{A}_t} \in \mathbb{R}^{d\times r}$.

- The right singular matrix of $\boldsymbol{B}_t$ as $\boldsymbol{V}_{\boldsymbol{B}_t} \in \mathbb{R}^{k\times r}$.

**Lemma D.6.** *Under assumptions in Section 2.1 for the nonlinear setting, we update $\boldsymbol{A}_t$ and $\boldsymbol{B}_t$ via Eq. (74) and Eq. (75) under spectral initialization (Spectral-init), then we have the following recursion*

$$\begin{aligned}
\boldsymbol{A}_{t+1}\boldsymbol{B}_{t+1} - \Delta &= (1-\eta)\boldsymbol{U}_{\boldsymbol{A}_t}\boldsymbol{U}_{\boldsymbol{A}_t}^\top(\boldsymbol{A}_t\boldsymbol{B}_t - \Delta)\boldsymbol{V}_{\boldsymbol{B}_t}\boldsymbol{V}_{\boldsymbol{B}_t}^\top \\
&\quad + (1-\eta/2)\left(\boldsymbol{I}_d - \boldsymbol{U}_{\boldsymbol{A}_t}\boldsymbol{U}_{\boldsymbol{A}_t}^\top\right)(\boldsymbol{A}_t\boldsymbol{B}_t - \Delta)\boldsymbol{V}_{\boldsymbol{B}_t}\boldsymbol{V}_{\boldsymbol{B}_t}^\top \\
&\quad + (1-\eta/2)\boldsymbol{U}_{\boldsymbol{A}_t}\boldsymbol{U}_{\boldsymbol{A}_t}^\top(\boldsymbol{A}_t\boldsymbol{B}_t - \Delta)\left(\boldsymbol{I}_k - \boldsymbol{V}_{\boldsymbol{B}_t}\boldsymbol{V}_{\boldsymbol{B}_t}^\top\right) \\
&\quad + \left(\boldsymbol{I}_d - \boldsymbol{U}_{\boldsymbol{A}_t}\boldsymbol{U}_{\boldsymbol{A}_t}^\top\right)(\boldsymbol{A}_t\boldsymbol{B}_t - \Delta)\left(\boldsymbol{I}_k - \boldsymbol{V}_{\boldsymbol{B}_t}\boldsymbol{V}_{\boldsymbol{B}_t}^\top\right) \\
&\quad + \eta\boldsymbol{\Xi}_t\boldsymbol{V}_{\boldsymbol{B}_t}\boldsymbol{V}_{\boldsymbol{B}_t}^\top + \eta\boldsymbol{U}_{\boldsymbol{A}_t}\boldsymbol{U}_{\boldsymbol{A}_t}^\top\boldsymbol{\Xi}_t + \eta^2\boldsymbol{J}_{\boldsymbol{W}_t}\mathcal{V}_t\mathcal{S}_t^{-1}\mathcal{U}_t^\top\boldsymbol{J}_{\boldsymbol{W}_t},
\end{aligned} \tag{76}$$

*and*

$$\boldsymbol{\Xi}_t := \boldsymbol{J}_{\boldsymbol{W}_t} - \frac{1}{2}\left(\boldsymbol{A}_t\boldsymbol{B}_t - \Delta\right).$$

*Then, by choosing $\eta \in (0,1)$, we have the associated upper bound in Frobenius norm*

$$\begin{aligned}
&\|\boldsymbol{A}_{t+1}\boldsymbol{B}_{t+1} - \Delta\|_{\mathrm{F}} \\
\leq &(1-\eta)\left\|\boldsymbol{U}_{\boldsymbol{A}_t}\boldsymbol{U}_{\boldsymbol{A}_t}^\top(\boldsymbol{A}_t\boldsymbol{B}_t - \Delta)\boldsymbol{V}_{\boldsymbol{B}_t}\boldsymbol{V}_{\boldsymbol{B}_t}^\top\right\|_{\mathrm{F}} \\
&+ (1-\eta/2)\left\|\left(\boldsymbol{I}_d - \boldsymbol{U}_{\boldsymbol{A}_t}\boldsymbol{U}_{\boldsymbol{A}_t}^\top\right)(\boldsymbol{A}_t\boldsymbol{B}_t - \Delta)\boldsymbol{V}_{\boldsymbol{B}_t}\boldsymbol{V}_{\boldsymbol{B}_t}^\top + \boldsymbol{U}_{\boldsymbol{A}_t}\boldsymbol{U}_{\boldsymbol{A}_t}^\top(\boldsymbol{A}_t\boldsymbol{B}_t - \Delta)\left(\boldsymbol{I}_k - \boldsymbol{V}_{\boldsymbol{B}_t}\boldsymbol{V}_{\boldsymbol{B}_t}^\top\right)\right\|_{\mathrm{F}} \\
&+ \left\|\left(\boldsymbol{I}_d - \boldsymbol{U}_{\boldsymbol{A}_t}\boldsymbol{U}_{\boldsymbol{A}_t}^\top\right)(\boldsymbol{A}_t\boldsymbol{B}_t - \Delta)\left(\boldsymbol{I}_k - \boldsymbol{V}_{\boldsymbol{B}_t}\boldsymbol{V}_{\boldsymbol{B}_t}^\top\right)\right\|_{\mathrm{F}} \\
&+ 2\eta\|\boldsymbol{\Xi}_t\|_{\mathrm{F}} + \eta^2\left\|\boldsymbol{J}_{\boldsymbol{W}_t}\mathcal{V}_t\mathcal{S}_t^{-1}\mathcal{U}_t^\top\boldsymbol{J}_{\boldsymbol{W}_t}\right\|_{\mathrm{F}}.
\end{aligned} \tag{77}$$
$$\tag{78}$$

*Proof.* By the preconditioned update in Eq. (74) and Eq. (75), we can construct

$$
\begin{aligned}
\boldsymbol{A}_{t+1}\boldsymbol{B}_{t+1} - \Delta = {} & \boldsymbol{A}_t\boldsymbol{B}_t - \Delta \\
& - \eta \boldsymbol{J}_{\boldsymbol{W}_t}\boldsymbol{B}_t^\top \left(\boldsymbol{B}_t\boldsymbol{B}_t^\top\right)^{-1}\boldsymbol{B}_t - \eta \boldsymbol{A}_t \left(\boldsymbol{A}_t^\top\boldsymbol{A}_t\right)^{-1}\boldsymbol{A}_t^\top \boldsymbol{J}_{\boldsymbol{W}_t} \\
& + \eta^2 \boldsymbol{J}_{\boldsymbol{W}_t}\boldsymbol{B}_t^\top (\boldsymbol{B}_t\boldsymbol{B}_t^\top)^{-1}(\boldsymbol{A}_t^\top\boldsymbol{A}_t)^{-1}\boldsymbol{A}_t^\top \boldsymbol{J}_{\boldsymbol{W}_t} \\
= {} & \boldsymbol{A}_t\boldsymbol{B}_t - \Delta \\
& - \eta/2(\boldsymbol{A}_t\boldsymbol{B}_t - \Delta)\boldsymbol{B}_t^\top \left(\boldsymbol{B}_t\boldsymbol{B}_t^\top\right)^{-1}\boldsymbol{B}_t + \eta \Xi_t \boldsymbol{B}_t^\top \left(\boldsymbol{B}_t\boldsymbol{B}_t^\top\right)^{-1}\boldsymbol{B}_t \\
& - \eta/2\boldsymbol{A}_t \left(\boldsymbol{A}_t^\top\boldsymbol{A}_t\right)^{-1}\boldsymbol{A}_t^\top (\boldsymbol{A}_t\boldsymbol{B}_t - \Delta) + \eta \boldsymbol{A}_t \left(\boldsymbol{A}_t^\top\boldsymbol{A}_t\right)^{-1}\boldsymbol{A}_t^\top \Xi_t \\
& + \eta^2 \boldsymbol{J}_{\boldsymbol{W}_t}\boldsymbol{B}_t^\top (\boldsymbol{B}_t\boldsymbol{B}_t^\top)^{-1}(\boldsymbol{A}_t^\top\boldsymbol{A}_t)^{-1}\boldsymbol{A}_t^\top \boldsymbol{J}_{\boldsymbol{W}_t} \\
= {} & \boldsymbol{A}_t\boldsymbol{B}_t - \Delta \\
& - \eta/2(\boldsymbol{A}_t\boldsymbol{B}_t - \Delta)\boldsymbol{V}_{\boldsymbol{B}_t}\boldsymbol{V}_{\boldsymbol{B}_t}^\top + \eta \Xi_t \boldsymbol{V}_{\boldsymbol{B}_t}\boldsymbol{V}_{\boldsymbol{B}_t}^\top \\
& - \eta/2\boldsymbol{U}_{\boldsymbol{A}_t}\boldsymbol{U}_{\boldsymbol{A}_t}^\top(\boldsymbol{A}_t\boldsymbol{B}_t - \Delta) + \eta \boldsymbol{U}_{\boldsymbol{A}_t}\boldsymbol{U}_{\boldsymbol{A}_t}^\top \Xi_t \\
& + \eta^2 \boldsymbol{J}_{\boldsymbol{W}_t}\mathcal{V}_t\mathcal{S}_t^{-1}\mathcal{U}_t^\top \boldsymbol{J}_{\boldsymbol{W}_t}, \qquad \text{[by pseudo inverse theorem and Jia et al. (2024, Lemma 14)]}
\end{aligned}
$$

from our choice on $\Xi_t = \boldsymbol{J}_{\boldsymbol{W}_t} - \frac{1}{2}(\boldsymbol{A}_t\boldsymbol{B}_t - \Delta)$. We can continue to expand

$$
\begin{aligned}
\boldsymbol{A}_{t+1}\boldsymbol{B}_{t+1} - \Delta = {} & \left(\boldsymbol{I}_d - \boldsymbol{U}_{\boldsymbol{A}_t}\boldsymbol{U}_{\boldsymbol{A}_t}^\top + \boldsymbol{U}_{\boldsymbol{A}_t}\boldsymbol{U}_{\boldsymbol{A}_t}^\top\right)(\boldsymbol{A}_t\boldsymbol{B}_t - \Delta)\left(\boldsymbol{I}_d - \boldsymbol{V}_{\boldsymbol{B}_t}\boldsymbol{V}_{\boldsymbol{B}_t}^\top + \boldsymbol{V}_{\boldsymbol{B}_t}\boldsymbol{V}_{\boldsymbol{B}_t}^\top\right) \\
& - \eta/2\left(\boldsymbol{I}_d - \boldsymbol{U}_{\boldsymbol{A}_t}\boldsymbol{U}_{\boldsymbol{A}_t}^\top + \boldsymbol{U}_{\boldsymbol{A}_t}\boldsymbol{U}_{\boldsymbol{A}_t}^\top\right)(\boldsymbol{A}_t\boldsymbol{B}_t - \Delta)\boldsymbol{V}_{\boldsymbol{B}_t}\boldsymbol{V}_{\boldsymbol{B}_t}^\top \\
& - \eta/2\boldsymbol{U}_{\boldsymbol{A}_t}\boldsymbol{U}_{\boldsymbol{A}_t}^\top(\boldsymbol{A}_t\boldsymbol{B}_t - \Delta)\left(\boldsymbol{I}_d - \boldsymbol{V}_{\boldsymbol{B}_t}\boldsymbol{V}_{\boldsymbol{B}_t}^\top + \boldsymbol{V}_{\boldsymbol{B}_t}\boldsymbol{V}_{\boldsymbol{B}_t}^\top\right) \\
& + \eta \Xi_t \boldsymbol{V}_{\boldsymbol{B}_t}\boldsymbol{V}_{\boldsymbol{B}_t}^\top + \eta \boldsymbol{U}_{\boldsymbol{A}_t}\boldsymbol{U}_{\boldsymbol{A}_t}^\top \Xi_t + \eta^2 \boldsymbol{J}_{\boldsymbol{W}_t}\mathcal{V}_t\mathcal{S}_t^{-1}\mathcal{U}_t^\top \boldsymbol{J}_{\boldsymbol{W}_t} \\
= {} & (1 - \eta)\boldsymbol{U}_{\boldsymbol{A}_t}\boldsymbol{U}_{\boldsymbol{A}_t}^\top(\boldsymbol{A}_t\boldsymbol{B}_t - \Delta)\boldsymbol{V}_{\boldsymbol{B}_t}\boldsymbol{V}_{\boldsymbol{B}_t}^\top \\
& + (1 - \eta/2)\left(\boldsymbol{I}_d - \boldsymbol{U}_{\boldsymbol{A}_t}\boldsymbol{U}_{\boldsymbol{A}_t}^\top\right)(\boldsymbol{A}_t\boldsymbol{B}_t - \Delta)\boldsymbol{V}_{\boldsymbol{B}_t}\boldsymbol{V}_{\boldsymbol{B}_t}^\top \\
& + (1 - \eta/2)\boldsymbol{U}_{\boldsymbol{A}_t}\boldsymbol{U}_{\boldsymbol{A}_t}^\top(\boldsymbol{A}_t\boldsymbol{B}_t - \Delta)\left(\boldsymbol{I}_k - \boldsymbol{V}_{\boldsymbol{B}_t}\boldsymbol{V}_{\boldsymbol{B}_t}^\top\right) \\
& + \left(\boldsymbol{I}_d - \boldsymbol{U}_{\boldsymbol{A}_t}\boldsymbol{U}_{\boldsymbol{A}_t}^\top\right)(\boldsymbol{A}_t\boldsymbol{B}_t - \Delta)\left(\boldsymbol{I}_k - \boldsymbol{V}_{\boldsymbol{B}_t}\boldsymbol{V}_{\boldsymbol{B}_t}^\top\right) \\
& + \eta \Xi_t \boldsymbol{V}_{\boldsymbol{B}_t}\boldsymbol{V}_{\boldsymbol{B}_t}^\top + \eta \boldsymbol{U}_{\boldsymbol{A}_t}\boldsymbol{U}_{\boldsymbol{A}_t}^\top \Xi_t + \eta^2 \boldsymbol{J}_{\boldsymbol{W}_t}\mathcal{V}_t\mathcal{S}_t^{-1}\mathcal{U}_t^\top \boldsymbol{J}_{\boldsymbol{W}_t}.
\end{aligned}
$$

Based on the above formulation, suppose $\eta \in (0, 1)$, we can derive the following upper bound by triangle inequality

$$
\begin{aligned}
& \left\|\boldsymbol{A}_{t+1}\boldsymbol{B}_{t+1} - \Delta\right\|_{\mathrm{F}} \qquad\qquad\qquad\qquad\qquad\qquad\qquad\qquad\qquad\qquad\qquad\qquad\qquad (79) \\
& \le \left\|(1 - \eta)\boldsymbol{U}_{\boldsymbol{A}_t}\boldsymbol{U}_{\boldsymbol{A}_t}^\top(\boldsymbol{A}_t\boldsymbol{B}_t - \Delta)\boldsymbol{V}_{\boldsymbol{B}_t}\boldsymbol{V}_{\boldsymbol{B}_t}^\top\right\|_{\mathrm{F}} \\
& \quad + \left\|(1 - \eta/2)\left(\boldsymbol{I}_d - \boldsymbol{U}_{\boldsymbol{A}_t}\boldsymbol{U}_{\boldsymbol{A}_t}^\top\right)(\boldsymbol{A}_t\boldsymbol{B}_t - \Delta)\boldsymbol{V}_{\boldsymbol{B}_t}\boldsymbol{V}_{\boldsymbol{B}_t}^\top\right\|_{\mathrm{F}} \\
& \quad + \left\|(1 - \eta/2)\boldsymbol{U}_{\boldsymbol{A}_t}\boldsymbol{U}_{\boldsymbol{A}_t}^\top(\boldsymbol{A}_t\boldsymbol{B}_t - \Delta)\left(\boldsymbol{I}_k - \boldsymbol{V}_{\boldsymbol{B}_t}\boldsymbol{V}_{\boldsymbol{B}_t}^\top\right)\right\|_{\mathrm{F}} \\
& \quad + \left\|\left(\boldsymbol{I}_d - \boldsymbol{U}_{\boldsymbol{A}_t}\boldsymbol{U}_{\boldsymbol{A}_t}^\top\right)(\boldsymbol{A}_t\boldsymbol{B}_t - \Delta)\left(\boldsymbol{I}_k - \boldsymbol{V}_{\boldsymbol{B}_t}\boldsymbol{V}_{\boldsymbol{B}_t}^\top\right)\right\|_{\mathrm{F}} \\
& \quad + \eta \left\|\Xi_t \boldsymbol{V}_{\boldsymbol{B}_t}\boldsymbol{V}_{\boldsymbol{B}_t}^\top\right\|_{\mathrm{F}} + \eta \left\|\boldsymbol{U}_{\boldsymbol{A}_t}\boldsymbol{U}_{\boldsymbol{A}_t}^\top \Xi_t\right\|_{\mathrm{F}} + \eta^2 \left\|\boldsymbol{J}_{\boldsymbol{W}_t}\mathcal{V}_t\mathcal{S}_t^{-1}\mathcal{U}_t^\top \boldsymbol{J}_{\boldsymbol{W}_t}\right\|_{\mathrm{F}} \qquad \text{[by triangle inequality]} \\
& \le (1 - \eta)\left\|\boldsymbol{U}_{\boldsymbol{A}_t}\boldsymbol{U}_{\boldsymbol{A}_t}^\top(\boldsymbol{A}_t\boldsymbol{B}_t - \Delta)\boldsymbol{V}_{\boldsymbol{B}_t}\boldsymbol{V}_{\boldsymbol{B}_t}^\top\right\|_{\mathrm{F}} \\
& \quad + (1 - \eta/2)\left\|\left(\boldsymbol{I}_d - \boldsymbol{U}_{\boldsymbol{A}_t}\boldsymbol{U}_{\boldsymbol{A}_t}^\top\right)(\boldsymbol{A}_t\boldsymbol{B}_t - \Delta)\boldsymbol{V}_{\boldsymbol{B}_t}\boldsymbol{V}_{\boldsymbol{B}_t}^\top + \boldsymbol{U}_{\boldsymbol{A}_t}\boldsymbol{U}_{\boldsymbol{A}_t}^\top(\boldsymbol{A}_t\boldsymbol{B}_t - \Delta)\left(\boldsymbol{I}_k - \boldsymbol{V}_{\boldsymbol{B}_t}\boldsymbol{V}_{\boldsymbol{B}_t}^\top\right)\right\|_{\mathrm{F}} \\
& \quad + \left\|\left(\boldsymbol{I}_d - \boldsymbol{U}_{\boldsymbol{A}_t}\boldsymbol{U}_{\boldsymbol{A}_t}^\top\right)(\boldsymbol{A}_t\boldsymbol{B}_t - \Delta)\left(\boldsymbol{I}_k - \boldsymbol{V}_{\boldsymbol{B}_t}\boldsymbol{V}_{\boldsymbol{B}_t}^\top\right)\right\|_{\mathrm{F}} \\
& \quad + 2\eta \left\|\Xi_t\right\|_{\mathrm{F}} + \eta^2 \left\|\boldsymbol{J}_{\boldsymbol{W}_t}\mathcal{V}_t\mathcal{S}_t^{-1}\mathcal{U}_t^\top \boldsymbol{J}_{\boldsymbol{W}_t}\right\|_{\mathrm{F}}, \qquad\qquad\qquad\qquad\qquad \text{[since } \eta \in (0, 1)\text{]}
\end{aligned}
$$

which proves the claim. $\qquad\qquad\qquad\qquad\qquad\qquad\qquad\qquad\qquad\qquad\qquad\qquad\qquad\qquad\qquad\qquad$ $\square$

In order to derive the convergence rate of $\left\|\boldsymbol{A}_{t+1}\boldsymbol{B}_{t+1} - \Delta\right\|_{\mathrm{F}}$ in the above terms, we need to provide the estimation of the

following four terms

$$\|\mathbf{\Xi}_t\|_{\mathrm{F}} \ , \quad \left\|\boldsymbol{J}_{\boldsymbol{W}_t}\mathcal{V}_t\mathcal{S}_t^{-1}\mathcal{U}_t^\top\boldsymbol{J}_{\boldsymbol{W}_t}\right\|_{\mathrm{F}} \ ,$$

$$\left\|\left(\boldsymbol{I}_d - \boldsymbol{U}_{\boldsymbol{A}_t}\boldsymbol{U}_{\boldsymbol{A}_t}^\top\right)(\boldsymbol{A}_t\boldsymbol{B}_t - \Delta)\boldsymbol{V}_{\boldsymbol{B}_t}\boldsymbol{V}_{\boldsymbol{B}_t}^\top + \boldsymbol{U}_{\boldsymbol{A}_t}\boldsymbol{U}_{\boldsymbol{A}_t}^\top(\boldsymbol{A}_t\boldsymbol{B}_t - \Delta)\left(\boldsymbol{I}_k - \boldsymbol{V}_{\boldsymbol{B}_t}\boldsymbol{V}_{\boldsymbol{B}_t}^\top\right)\right\|_{\mathrm{F}} \ ,$$

$$\left\|\left(\boldsymbol{I}_d - \boldsymbol{U}_{\boldsymbol{A}_t}\boldsymbol{U}_{\boldsymbol{A}_t}^\top\right)(\boldsymbol{A}_t\boldsymbol{B}_t - \Delta)\left(\boldsymbol{I}_k - \boldsymbol{V}_{\boldsymbol{B}_t}\boldsymbol{V}_{\boldsymbol{B}_t}^\top\right)\right\|_{\mathrm{F}} \ .$$

which are important elements in Eq. (77). We firstly prove the upper bound for $\|\mathbf{\Xi}_t\|_{\mathrm{F}}$ and $\left\|\boldsymbol{J}_{\boldsymbol{W}_t}\mathcal{V}_t\mathcal{S}_t^{-1}\mathcal{U}_t^\top\boldsymbol{J}_{\boldsymbol{W}_t}\right\|_{\mathrm{F}}$ since they are relatively straightforward. After that, we will handle with the remaining three terms which are the most technical part. All of these three terms rely on the condition $\|\boldsymbol{A}_t\boldsymbol{B}_t - \Delta\|_{\mathrm{F}} \le \rho\lambda_{r*}^*$ and we will prove it by induction finally in Theorem D.10.

**Lemma D.7.** *For a positive constant $\rho \in (0,1)$, suppose $\epsilon \le \frac{\rho}{3C^*K^2\gamma\sqrt{2dr^*}\kappa}$ with $\gamma = 2$, assume $\|\boldsymbol{A}_t\boldsymbol{B}_t - \Delta\|_{\mathrm{F}} \le \rho\lambda_{r*}^*$, under assumptions in Section 2.1 for the nonlinear setting and Assumption 4.1, then with probability at least $1 - 2Cdk\exp\left(-\epsilon^2N\right)$ for a universal constant $C > 0$, we have*

$$\|\mathbf{\Xi}_t\|_{\mathrm{F}} \le \left(\mathcal{O}\left(\frac{1}{\kappa r^*}\right) + C^*K^2\sqrt{d}\epsilon\right)\|\boldsymbol{A}_t\boldsymbol{B}_t - \Delta\|_{\mathrm{F}} \ ,$$

*Proof.* Recall $\mathbf{\Xi}_t := \boldsymbol{J}_{\boldsymbol{W}_t} - \frac{1}{2}\left(\boldsymbol{A}_t\boldsymbol{B}_t - \Delta\right)$ from Lemma D.6, then with probability at least $1 - 2Cdk\exp\left(-\epsilon^2N\right)$ for a universal constant $C > 0$, we have

$$\|\mathbf{\Xi}_t\|_{\mathrm{F}} = \left\|\frac{1}{2}\left(\boldsymbol{A}_t\boldsymbol{B}_t - \Delta\right) - \mathbb{E}_{\widetilde{\boldsymbol{x}}}\left[\boldsymbol{J}_{\boldsymbol{W}_t}\right] + \mathbb{E}_{\widetilde{\boldsymbol{x}}}\left[\boldsymbol{J}_{\boldsymbol{W}_t}\right] - \boldsymbol{J}_{\boldsymbol{W}_t}\right\|_{\mathrm{F}}$$

$$\le \underbrace{\left\|\frac{1}{2}\left(\boldsymbol{A}_t\boldsymbol{B}_t - \Delta\right) - \mathbb{E}_{\widetilde{\boldsymbol{x}}}\left[\boldsymbol{J}_{\boldsymbol{W}_t}\right]\right\|_{\mathrm{F}}}_{\text{population error}} + \underbrace{\left\|\mathbb{E}_{\widetilde{\boldsymbol{x}}}\left[\boldsymbol{J}_{\boldsymbol{W}_t}\right] - \boldsymbol{J}_{\boldsymbol{W}_t}\right\|_{\mathrm{F}}}_{\text{concentration error}}$$

$$= \frac{\|\boldsymbol{\Psi}(t)\|_{\mathrm{F}}}{\|\boldsymbol{A}_t\boldsymbol{B}_t - \Delta\|_{\mathrm{F}}}\|\boldsymbol{A}_t\boldsymbol{B}_t - \Delta\|_{\mathrm{F}} + \left\|\mathbb{E}_{\widetilde{\boldsymbol{x}}}\left[\boldsymbol{J}_{\boldsymbol{W}_t}\right] - \boldsymbol{J}_{\boldsymbol{W}_t}\right\|_{\mathrm{F}} \qquad \text{[by Lemma D.2]}$$

$$\le \left(\mathcal{O}\left(\frac{1}{\kappa r^*}\right) + C^*K^2\sqrt{d}\epsilon\right)\|\boldsymbol{A}_t\boldsymbol{B}_t - \Delta\|_{\mathrm{F}} \ , \qquad \text{[by Lemma D.2 and Theorem D.4]}$$

which completes the proof. $\qquad\square$

**Lemma D.8.** *Under assumptions in Section 2.1 for the nonlinear setting, suppose $\|\boldsymbol{A}_t\boldsymbol{B}_t - \Delta\|_{\mathrm{F}} \le \rho\lambda_{r*}^*$ for a positive constant $\rho > 0$, with probability at least $1 - 2C\exp\left(-\epsilon^2N\right)$ for some constants $C > 0$, it holds that*

$$\left\|\boldsymbol{J}_{\boldsymbol{W}_t}\mathcal{V}_t\mathcal{S}_t^{-1}\mathcal{U}_t^\top\boldsymbol{J}_{\boldsymbol{W}_t}\right\|_{\mathrm{F}} \le (1+\epsilon)^2\frac{\rho}{1-\rho}\|\boldsymbol{A}_t\boldsymbol{B}_t - \Delta\|_{\mathrm{F}} \ .$$

*Proof.* First, with probability at least $1 - 2C\exp\left(-\epsilon^2N\right)$ for some constants $C > 0$, we can derive

$$\left\|\boldsymbol{J}_{\boldsymbol{W}_t}\mathcal{V}_t\mathcal{S}_t^{-1}\mathcal{U}_t^\top\boldsymbol{J}_{\boldsymbol{W}_t}\right\|_{\mathrm{F}} \le \|\boldsymbol{J}_{\boldsymbol{W}_t}\|_{\mathrm{F}}^2 \left\|\mathcal{V}_t\mathcal{S}_t^{-1}\mathcal{U}_t^\top\right\|_{op}$$

$$= \frac{\left\|\frac{1}{N}\widetilde{\boldsymbol{X}}^\top\left(\sigma\left(\widetilde{\boldsymbol{X}}(\boldsymbol{W}^\natural + \boldsymbol{A}_t\boldsymbol{B}_t)\right) - \sigma\left(\widetilde{\boldsymbol{X}}\widetilde{\boldsymbol{W}}^\natural\right)\right) \odot \sigma'\left(\widetilde{\boldsymbol{X}}(\boldsymbol{W}^\natural + \boldsymbol{A}_t\boldsymbol{B}_t)\right)\right\|_{\mathrm{F}}^2}{\lambda_r\left(\boldsymbol{A}_t\boldsymbol{B}_t\right)}$$

$$\le \frac{\left\|\frac{1}{N}\widetilde{\boldsymbol{X}}^\top\widetilde{\boldsymbol{X}}(\boldsymbol{A}_t\boldsymbol{B}_t - \Delta)\right\|_{\mathrm{F}}^2}{\lambda_r\left(\boldsymbol{A}_t\boldsymbol{B}_t\right)} \qquad \text{[by Lipschitz continuity of } \sigma, \sigma']$$

$$\le \left(\frac{1}{N}\lambda_1^2(\widetilde{\boldsymbol{X}})\right)^2\frac{\|\boldsymbol{A}_t\boldsymbol{B}_t - \Delta\|_{\mathrm{F}}^2}{\lambda_r\left(\boldsymbol{A}_t\boldsymbol{B}_t\right)}$$

$$\le (1+\epsilon)^2\frac{\rho}{1-\rho}\|\boldsymbol{A}_t\boldsymbol{B}_t - \Delta\|_{\mathrm{F}} \ , \qquad \text{[by concentration of operator norm]}$$

where the last equality follows from $r = r^*$ and

$$\lambda_r \left( \boldsymbol{A}_t \boldsymbol{B}_t \right) \geq \lambda_{r^*} (\Delta) - \| \boldsymbol{A}_t \boldsymbol{B}_t - \Delta \|_{\mathrm{F}} \geq (1 - \rho) \lambda_{r^*} (\Delta) .$$

□

With Lemma D.7 and Lemma D.8, now we can prove for the other three terms.

**Lemma D.9.** *Suppose* $\| \boldsymbol{A}_t \boldsymbol{B}_t - \Delta \|_{\mathrm{F}} \leq \rho \lambda_{r^*}^*$ *with a positive constant* $\rho \in [0 , 1/4]$*, then it holds that*

$$\left\| \left( \boldsymbol{I}_d - \boldsymbol{U}_{\boldsymbol{A}_t} \boldsymbol{U}_{\boldsymbol{A}_t}^\top \right) \Delta \left( \boldsymbol{I}_k - \boldsymbol{V}_{\boldsymbol{B}_t} \boldsymbol{V}_{\boldsymbol{B}_t}^\top \right) \right\|_{\mathrm{F}} \leq \frac{\rho}{\sqrt{1 - 8\rho^2}} \| \boldsymbol{A}_t \boldsymbol{B}_t - \Delta \|_{\mathrm{F}} ,$$

*and*

$$\left\| \left( \boldsymbol{I}_d - \boldsymbol{U}_{\boldsymbol{A}_t} \boldsymbol{U}_{\boldsymbol{A}_t}^\top \right) \Delta \boldsymbol{V}_{\boldsymbol{B}_t} \boldsymbol{V}_{\boldsymbol{B}_t}^\top + \boldsymbol{U}_{\boldsymbol{A}_t} \boldsymbol{U}_{\boldsymbol{A}_t}^\top \Delta \left( \boldsymbol{I}_k - \boldsymbol{V}_{\boldsymbol{B}_t} \boldsymbol{V}_{\boldsymbol{B}_t}^\top \right) \right\|_{\mathrm{F}} \leq \| \boldsymbol{A}_t \boldsymbol{B}_t - \Delta \|_{\mathrm{F}} .$$

*Proof.* First, we recall

$$\boldsymbol{Z}_t = \begin{bmatrix} \boldsymbol{A}_t \\ \boldsymbol{B}_t^\top \end{bmatrix} , \quad \underline{\boldsymbol{Z}}_t = \begin{bmatrix} \boldsymbol{A}_t \\ -\boldsymbol{B}_t^\top \end{bmatrix} ,$$

and define a preconditioned operator $\mathcal{P}$ and symmetrized downstream feature shift matrix $\hat{\boldsymbol{\Delta}}$ as

$$\mathcal{P}(\boldsymbol{Z}_t) := \begin{bmatrix} \boldsymbol{A}_t (\boldsymbol{A}_t^\top \boldsymbol{A}_t)^{-1} \\ \boldsymbol{B}_t^\top (\boldsymbol{B}_t \boldsymbol{B}_t^\top)^{-1} \end{bmatrix} , \quad \mathcal{P}(\underline{\boldsymbol{Z}}_t) := \begin{bmatrix} \boldsymbol{A}_t (\boldsymbol{A}_t^\top \boldsymbol{A}_t)^{-1} \\ -\boldsymbol{B}_t^\top (\boldsymbol{B}_t \boldsymbol{B}_t^\top)^{-1} \end{bmatrix} , \quad \hat{\boldsymbol{\Delta}} := \begin{bmatrix} \boldsymbol{0}_{d \times d} & \Delta \\ \Delta^\top & \boldsymbol{0}_{k \times k} \end{bmatrix} .$$

Next, we observe that

$$\frac{1}{2} \left( \boldsymbol{Z}_t \boldsymbol{Z}_t^\top - \underline{\boldsymbol{Z}}_t \underline{\boldsymbol{Z}}_t^\top \right) - \hat{\boldsymbol{\Delta}} = \begin{bmatrix} \boldsymbol{0}_{d \times d} & \boldsymbol{A}_t \boldsymbol{B}_t - \Delta \\ (\boldsymbol{A}_t \boldsymbol{B}_t - \Delta)^\top & \boldsymbol{0}_{k \times k} \end{bmatrix} ,$$

leading to

$$\left\| \frac{1}{2} \left( \boldsymbol{Z}_t \boldsymbol{Z}_t^\top - \underline{\boldsymbol{Z}}_t \underline{\boldsymbol{Z}}_t^\top \right) - \hat{\boldsymbol{\Delta}} \right\|_{op} = \| \boldsymbol{A}_t \boldsymbol{B}_t - \Delta \|_{op} , \quad \left\| \frac{1}{2} \left( \boldsymbol{Z}_t \boldsymbol{Z}_t^\top - \underline{\boldsymbol{Z}}_t \underline{\boldsymbol{Z}}_t^\top \right) - \hat{\boldsymbol{\Delta}} \right\|_{\mathrm{F}} = \sqrt{2} \| \boldsymbol{A}_t \boldsymbol{B}_t - \Delta \|_{\mathrm{F}} .$$

Based on the compact SVD of $\Delta$ in Eq. (1), we can write out the eigendecomposition of $\hat{\boldsymbol{\Delta}}$ as

$$\hat{\boldsymbol{\Delta}} = \begin{bmatrix} \boldsymbol{\Phi} & \underline{\boldsymbol{\Phi}} \end{bmatrix} \begin{bmatrix} \boldsymbol{S}^* & \boldsymbol{0}_{r^* \times r^*} \\ \boldsymbol{0}_{r^* \times r^*} & -\boldsymbol{S}^* \end{bmatrix} \begin{bmatrix} \boldsymbol{\Phi} & \underline{\boldsymbol{\Phi}} \end{bmatrix}^\top = \boldsymbol{\Phi} \boldsymbol{S}^* \boldsymbol{\Phi}^\top - \underline{\boldsymbol{\Phi}} \boldsymbol{S}^* \underline{\boldsymbol{\Phi}}^\top , \quad \text{where } \boldsymbol{\Phi} = \frac{1}{\sqrt{2}} \begin{bmatrix} \boldsymbol{U} \\ \boldsymbol{V} \end{bmatrix} , \underline{\boldsymbol{\Phi}} = \frac{1}{\sqrt{2}} \begin{bmatrix} \boldsymbol{U} \\ -\boldsymbol{V} \end{bmatrix} . \quad (80)$$

Notice that we can also obtain the SVD of $\hat{\boldsymbol{\Delta}}$ as

$$\hat{\boldsymbol{\Delta}} = \widehat{\boldsymbol{U}} \widehat{\boldsymbol{S}} \widehat{\boldsymbol{V}}^\top = \underbrace{\begin{bmatrix} \boldsymbol{\Phi} & \underline{\boldsymbol{\Phi}} \end{bmatrix}}_{:= \widehat{\boldsymbol{U}}} \underbrace{\begin{bmatrix} \boldsymbol{S}^* & \boldsymbol{0}_{r^* \times r^*} \\ \boldsymbol{0}_{r^* \times r^*} & \boldsymbol{S}^* \end{bmatrix}}_{\widehat{\boldsymbol{S}}} \underbrace{\begin{bmatrix} \boldsymbol{\Phi} & -\underline{\boldsymbol{\Phi}} \end{bmatrix}^\top}_{:= \widehat{\boldsymbol{V}}^\top} . \quad (81)$$

Notice that $\hat{\boldsymbol{\Delta}}$ is a low-rank matrix with rank-$2r^*$ because of $\mathrm{Rank}(\Delta) = r^*$. If $\frac{1}{2} \left( \boldsymbol{Z}_t \boldsymbol{Z}_t^\top - \underline{\boldsymbol{Z}}_t \underline{\boldsymbol{Z}}_t^\top \right)$ recovers $\hat{\boldsymbol{\Delta}}$, this indicates that the top-$2r^*$ subspace of $\frac{1}{2} \left( \boldsymbol{Z}_t \boldsymbol{Z}_t^\top - \underline{\boldsymbol{Z}}_t \underline{\boldsymbol{Z}}_t^\top \right)$ will align to $\hat{\boldsymbol{\Delta}}$ perfectly. Next, we can derive the projection matrix for the top-$2r^*$ subspace of $\frac{1}{2} \left( \boldsymbol{Z}_t \boldsymbol{Z}_t^\top - \underline{\boldsymbol{Z}}_t \underline{\boldsymbol{Z}}_t^\top \right)$. First, we have

$$\boldsymbol{Z}_t \mathcal{P}^\top (\boldsymbol{Z}_t) = \begin{bmatrix} \boldsymbol{A}_t (\boldsymbol{A}_t^\top \boldsymbol{A}_t)^{-1} \boldsymbol{A}_t^\top & \boldsymbol{A}_t (\boldsymbol{B}_t \boldsymbol{B}_t^\top)^{-1} \boldsymbol{B}_t \\ \boldsymbol{B}_t^\top (\boldsymbol{A}_t^\top \boldsymbol{A}_t)^{-1} \boldsymbol{A}_t^\top & \boldsymbol{B}_t^\top (\boldsymbol{B}_t \boldsymbol{B}_t^\top)^{-1} \boldsymbol{B}_t \end{bmatrix} ,$$

which can imply

$$\frac{1}{2}\boldsymbol{Z}_t\mathcal{P}^\top(\boldsymbol{Z}_t)\boldsymbol{Z}_t\boldsymbol{Z}_t^\top = \frac{1}{2}\begin{bmatrix} \boldsymbol{A}_t(\boldsymbol{A}_t^\top\boldsymbol{A}_t)^{-1}\boldsymbol{A}_t^\top & \boldsymbol{A}_t(\boldsymbol{B}_t\boldsymbol{B}_t^\top)^{-1}\boldsymbol{B}_t \\ \boldsymbol{B}_t^\top(\boldsymbol{A}_t^\top\boldsymbol{A}_t)^{-1}\boldsymbol{A}_t^\top & \boldsymbol{B}_t^\top(\boldsymbol{B}_t\boldsymbol{B}_t^\top)^{-1}\boldsymbol{B}_t \end{bmatrix}\begin{bmatrix} \boldsymbol{A}_t\boldsymbol{A}_t^\top & \boldsymbol{A}_t\boldsymbol{B}_t \\ \boldsymbol{B}_t^\top\boldsymbol{A}_t^\top & \boldsymbol{B}_t^\top\boldsymbol{B}_t \end{bmatrix}$$

$$= \frac{1}{2}\begin{bmatrix} \boldsymbol{A}_t\boldsymbol{A}_t^\top & \boldsymbol{A}_t\boldsymbol{B}_t \\ \boldsymbol{B}_t^\top\boldsymbol{A}_t^\top & \boldsymbol{B}_t^\top\boldsymbol{B}_t \end{bmatrix} = \frac{1}{2}\boldsymbol{Z}_t\boldsymbol{Z}_t^\top\,.$$

Similarly, we can derive

$$\frac{1}{2}\underline{\boldsymbol{Z}}_t\mathcal{P}^\top(\underline{\boldsymbol{Z}}_t)\underline{\boldsymbol{Z}}_t\underline{\boldsymbol{Z}}_t^\top = \frac{1}{2}\begin{bmatrix} \boldsymbol{A}_t\boldsymbol{A}_t^\top & -\boldsymbol{A}_t\boldsymbol{B}_t \\ -\boldsymbol{B}_t^\top\boldsymbol{A}_t^\top & \boldsymbol{B}_t^\top\boldsymbol{B}_t \end{bmatrix} = \frac{1}{2}\underline{\boldsymbol{Z}}_t\underline{\boldsymbol{Z}}_t^\top\,.$$

Additionally, we have

$$\frac{1}{2}\underline{\boldsymbol{Z}}_t\mathcal{P}^\top(\underline{\boldsymbol{Z}}_t)\boldsymbol{Z}_t\boldsymbol{Z}_t^\top = \boldsymbol{0}_{(d+k)\times(d+k)}\,, \quad \frac{1}{2}\boldsymbol{Z}_t\mathcal{P}^\top(\boldsymbol{Z}_t)\underline{\boldsymbol{Z}}_t\underline{\boldsymbol{Z}}_t^\top = \boldsymbol{0}_{(d+k)\times(d+k)}\,.$$

Base on the above identity, we can obtain that the subspace of $\boldsymbol{Z}_t\boldsymbol{Z}_t^\top$ is orthogonal to the subspace of $\underline{\boldsymbol{Z}}_t\underline{\boldsymbol{Z}}_t^\top$. Since $\mathrm{Rank}\left(\boldsymbol{Z}_t\boldsymbol{Z}_t^\top\right) \le r$ and $\mathrm{Rank}\left(\underline{\boldsymbol{Z}}_t\underline{\boldsymbol{Z}}_t^\top\right) \le r$, then we have that $\mathrm{Rank}\left(\boldsymbol{Z}_t\boldsymbol{Z}_t^\top - \underline{\boldsymbol{Z}}_t\underline{\boldsymbol{Z}}_t^\top\right) \le 2r^*$ since $r = r^*$. Therefore, we can construct a valid projection matrix

$$\mathbf{P}_t := \boldsymbol{Z}_t\mathcal{P}^\top(\boldsymbol{Z}_t) + \underline{\boldsymbol{Z}}_t\mathcal{P}^\top(\underline{\boldsymbol{Z}}_t)\,, \tag{82}$$

which satisfies

$$\frac{1}{2}\mathbf{P}_t\left(\boldsymbol{Z}_t\boldsymbol{Z}_t^\top - \underline{\boldsymbol{Z}}_t\underline{\boldsymbol{Z}}_t^\top\right) = \frac{1}{2}\left(\boldsymbol{Z}_t\boldsymbol{Z}_t^\top - \underline{\boldsymbol{Z}}_t\underline{\boldsymbol{Z}}_t^\top\right)\,,$$

and

$$\frac{1}{2}\left(\boldsymbol{I}_{d+k} - \mathbf{P}_t\right)\left(\boldsymbol{Z}_t\boldsymbol{Z}_t^\top - \underline{\boldsymbol{Z}}_t\underline{\boldsymbol{Z}}_t^\top\right) = \boldsymbol{0}_{(d+k)\times(d+k)}\,. \tag{83}$$

Also, we can verify that $\mathbf{P}_t$ is symmetric and $\mathbf{P}_t\mathbf{P}_t = \mathbf{P}_t$. Therefore we can conclude that $\mathbf{P}_t$ is the projection matrix which maps matrices or vectors to the top-$2r$ subspace of $\frac{1}{2}\left(\boldsymbol{Z}_t\boldsymbol{Z}_t^\top - \underline{\boldsymbol{Z}}_t\underline{\boldsymbol{Z}}_t^\top\right)$. For notational simplicity, here we fix the timestamp $t$ and denote

$$\boldsymbol{F} := \frac{1}{2\sqrt{2}}\left(\boldsymbol{Z}_t\boldsymbol{Z}_t^\top - \underline{\boldsymbol{Z}}_t\underline{\boldsymbol{Z}}_t^\top\right)\,,$$

which means

$$\left\|\boldsymbol{F} - \frac{\hat{\boldsymbol{\Delta}}}{\sqrt{2}}\right\|_{\mathrm{F}} = \|\boldsymbol{A}_t\boldsymbol{B}_t - \Delta\|_{\mathrm{F}} \le \rho\lambda_{r^*}^*\,. \tag{84}$$

Next, we define $\mathbf{P}_t := \boldsymbol{L}\boldsymbol{L}^\top \in \mathbb{R}^{(d+k)\times(d+k)}$ with

$$\boldsymbol{L} = \begin{bmatrix} \boldsymbol{U}_{\boldsymbol{A}_t} & \boldsymbol{0}_{d\times r} \\ \boldsymbol{0}_{k\times r} & \boldsymbol{V}_{\boldsymbol{B}_t} \end{bmatrix} \in \mathbb{R}^{(d+k)\times 2r}\,,$$

and $(\boldsymbol{I}_{d+k} - \mathbf{P}_t) = \boldsymbol{L}_\perp\boldsymbol{L}_\perp^\top$ where

$$\boldsymbol{L}_\perp = \begin{bmatrix} \boldsymbol{U}_{\boldsymbol{A}_t,\perp} & \boldsymbol{0}_{d\times(k-r)} \\ \boldsymbol{0}_{k\times(d-r)} & \boldsymbol{V}_{\boldsymbol{B}_t,\perp} \end{bmatrix} \in \mathbb{R}^{(d+k)\times(d+k-2r)}\,,$$

then we have

$$\left\|\boldsymbol{F} - \frac{\hat{\boldsymbol{\Delta}}}{\sqrt{2}}\right\|_{\mathrm{F}}^2 = \left\|\begin{bmatrix} \boldsymbol{L}^\top \\ \boldsymbol{L}_\perp^\top \end{bmatrix}\left(\boldsymbol{F} - \frac{\hat{\boldsymbol{\Delta}}}{\sqrt{2}}\right)\begin{bmatrix} \boldsymbol{L} & \boldsymbol{L}_\perp \end{bmatrix}\right\|_{\mathrm{F}}$$

$$= \left\|\begin{bmatrix} \boldsymbol{L}^\top\boldsymbol{F}\boldsymbol{L} - \boldsymbol{L}^\top\frac{\hat{\boldsymbol{\Delta}}}{\sqrt{2}}\boldsymbol{L} & -\boldsymbol{L}^\top\frac{\hat{\boldsymbol{\Delta}}}{\sqrt{2}}\boldsymbol{L}_\perp \\ -\boldsymbol{L}_\perp^\top\frac{\hat{\boldsymbol{\Delta}}}{\sqrt{2}}\boldsymbol{L} & \boldsymbol{L}_\perp^\top\frac{\hat{\boldsymbol{\Delta}}}{\sqrt{2}}\boldsymbol{L}_\perp \end{bmatrix}\right\|_{\mathrm{F}}^2 \qquad \text{[by Eq. (83)]}$$

$$= \left\|\boldsymbol{L}^\top\boldsymbol{F}\boldsymbol{L} - \frac{1}{\sqrt{2}}\boldsymbol{L}^\top\hat{\boldsymbol{\Delta}}\boldsymbol{L}\right\|_{\mathrm{F}}^2 + \frac{1}{2}\left\|\boldsymbol{L}_\perp^\top\hat{\boldsymbol{\Delta}}\boldsymbol{L}\right\|_{\mathrm{F}}^2 + \frac{1}{2}\left\|\boldsymbol{L}^\top\hat{\boldsymbol{\Delta}}\boldsymbol{L}_\perp\right\|_{\mathrm{F}}^2 + \frac{1}{2}\left\|\boldsymbol{L}_\perp^\top\hat{\boldsymbol{\Delta}}\boldsymbol{L}_\perp\right\|_{\mathrm{F}}^2\,. \tag{85}$$

Since $I_{d+k} - P_t = L_\perp L_\perp^\top$, then we have

$$\left\| (I_d - U_{A_t} U_{A_t}^\top) \Delta (I_k - V_{B_t} V_{B_t}^\top) \right\|_F = \frac{1}{\sqrt{2}} \left\| L_\perp^\top \hat{\Delta} L_\perp \right\|_F . \tag{86}$$

Next, by Eq. (85), we have $\left\| F - \frac{\hat{\Delta}}{\sqrt{2}} \right\|_F^2 \geq \frac{1}{2} \left\| L_\perp^\top \hat{\Delta} L \right\|_F^2 + \frac{1}{2} \left\| L^\top \hat{\Delta} L_\perp \right\|_F^2$, leading to

$$\frac{\frac{1}{2} \left\| L_\perp^\top \hat{\Delta} L_\perp \right\|_F^2}{\left\| F - \frac{\hat{\Delta}}{\sqrt{2}} \right\|_F^2} \leq \frac{\frac{1}{2} \left\| L_\perp^\top \hat{\Delta} L_\perp \right\|_F^2}{\frac{1}{2} \left\| L_\perp^\top \hat{\Delta} L \right\|_F^2 + \frac{1}{2} \left\| L^\top \hat{\Delta} L_\perp \right\|_F^2} . \tag{87}$$

The technical part is to lower bound $\left\| L_\perp^\top \hat{\Delta} L \right\|_F^2$ and $\left\| L^\top \hat{\Delta} L_\perp \right\|_F^2$, we will rely on the following decomposition which based on Eq. (81), i.e.

$$\begin{aligned}
\left\| L_\perp^\top \hat{\Delta} L \right\|_F^2 &= \left\| L_\perp^\top \hat{U} \hat{S} \hat{V}^\top L \right\|_F^2 \\
&= \left\| \left( L_\perp^\top \hat{U} \hat{S}^{1/2} \right) \left( L^\top \hat{V} \hat{S}^{1/2} \right)^\top \right\|_F^2 \\
&= \mathrm{tr} \left( \left( L^\top \hat{V} \hat{S}^{1/2} \right) \left( L_\perp^\top \hat{U} \hat{S}^{1/2} \right)^\top \left( L_\perp^\top \hat{U} \hat{S}^{1/2} \right) \left( L^\top \hat{V} \hat{S}^{1/2} \right)^\top \right) \\
&= \mathrm{tr} \left( \left( L^\top \hat{V} \hat{S}^{1/2} \right)^\top \left( L^\top \hat{V} \hat{S}^{1/2} \right) \left( L_\perp^\top \hat{U} \hat{S}^{1/2} \right)^\top \left( L_\perp^\top \hat{U} \hat{S}^{1/2} \right) \right) \\
&= \mathrm{tr} \left( \left( \hat{S}^{1/2} \hat{V}^\top L L^\top \hat{V} \hat{S}^{1/2} \right) \left( \hat{S}^{1/2} \hat{U}^\top L_\perp L_\perp^\top \hat{U} \hat{S}^{1/2} \right) \right) .
\end{aligned}$$

Notice that $\hat{S}^{1/2} \hat{V}^\top L L^\top \hat{V} \hat{S}^{1/2}$ and $\hat{S}^{1/2} \hat{U}^\top L_\perp L_\perp^\top \hat{U} \hat{S}^{1/2}$ are two positive semi-definite matrices, then by lower bound of trace of product of positive semi-definite matrices, using Weyl inequality, we have

$$\begin{aligned}
\left\| L_\perp^\top \hat{\Delta} L \right\|_F^2 &\geq \lambda_{2r^*} \left( \hat{S}^{1/2} \hat{V}^\top L L^\top \hat{V} \hat{S}^{1/2} \right) \left\| \hat{S}^{1/2} \hat{U}^\top L_\perp L_\perp^\top \hat{U} \hat{S}^{1/2} \right\|_F \\
&\geq \lambda_{r^*}^* \times \lambda_{2r^*} \left( \hat{V}^\top L L^\top \hat{V} \right) \left\| \hat{S}^{1/2} \hat{U}^\top L_\perp L_\perp^\top \hat{U} \hat{S}^{1/2} \right\|_F \\
&= \lambda_{r^*}^* \times \lambda_{2r^*} \left( \hat{V}^\top L L^\top \hat{V} \right) \left\| L_\perp L_\perp^\top \hat{U} \hat{S} \hat{U}^\top L_\perp L_\perp^\top \right\|_F ,
\end{aligned}$$

where the last equality follows from

$$\begin{aligned}
\left\| \hat{S}^{1/2} \hat{U}^\top L_\perp L_\perp^\top \hat{U} \hat{S}^{1/2} \right\|_F^2 &= \mathrm{tr} \left( \hat{S}^{1/2} \hat{U}^\top L_\perp L_\perp^\top \hat{U} \hat{S} \hat{U}^\top L_\perp L_\perp^\top \hat{U} \hat{S}^{1/2} \right) \\
&= \mathrm{tr} \left( \hat{S}^{1/2} \hat{U}^\top L_\perp L_\perp^\top \left( L_\perp L_\perp^\top \hat{U} \hat{S} \hat{U}^\top L_\perp L_\perp^\top \right) L_\perp L_\perp^\top \hat{U} \hat{S}^{1/2} \right) \\
&= \mathrm{tr} \left( \left( L_\perp L_\perp^\top \hat{U} \hat{S} \hat{U}^\top L_\perp L_\perp^\top \right) \left( L_\perp L_\perp^\top \hat{U} \hat{S} \hat{U}^\top L_\perp L_\perp^\top \right) \right) \\
&= \left\| L_\perp L_\perp^\top \hat{U} \hat{S} \hat{U}^\top L_\perp L_\perp^\top \right\|_F^2 .
\end{aligned}$$

Similarly, we have

$$\left\| L^\top \hat{\Delta} L_\perp \right\|_F^2 \geq \lambda_{r^*}^* \times \lambda_{2r^*} \left( \hat{U}^\top L L^\top \hat{U} \right) \left\| L_\perp L_\perp^\top \hat{V} \hat{S} \hat{V}^\top L_\perp L_\perp^\top \right\|_F .$$

Next, we can derive

$$\begin{aligned}
\left\| L_\perp L_\perp^\top \hat{U} \hat{S} \hat{U}^\top L_\perp L_\perp^\top \right\|_F^2 &= \left\| L_\perp L_\perp^\top \left( \Phi S^* \Phi^\top + \underline{\Phi} S^* \underline{\Phi}^\top \right) L_\perp L_\perp^\top \right\|_F^2 \qquad \text{[by Eq. (80)]} \\
&= \left\| L_\perp L_\perp^\top \underline{\Phi} S^* \underline{\Phi}^\top L_\perp L_\perp^\top \right\|_F^2 + \left\| L_\perp L_\perp^\top \Phi S^* \Phi^\top L_\perp L_\perp^\top \right\|_F^2 \\
&\quad + 2 \left\langle L_\perp L_\perp^\top \underline{\Phi} S^* \underline{\Phi}^\top L_\perp L_\perp^\top , L_\perp L_\perp^\top \Phi S^* \Phi^\top L_\perp L_\perp^\top \right\rangle ,
\end{aligned}$$

and

$$\left\| L_\perp L_\perp^\top \widehat{V} \widehat{S} \widehat{V}^\top L_\perp L_\perp^\top \right\|_{\mathrm{F}}^2 = \left\| L_\perp L_\perp^\top \left( \Phi S^* \Phi^\top + \underline{\Phi} S^* \underline{\Phi}^\top \right) L_\perp L_\perp^\top \right\|_{\mathrm{F}}^2 \qquad \text{[by Eq. (80)]}$$

$$= \left\| L_\perp L_\perp^\top \underline{\Phi} S^* \underline{\Phi}^\top L_\perp L_\perp^\top \right\|_{\mathrm{F}}^2 + \left\| L_\perp L_\perp^\top \Phi S^* \Phi^\top L_\perp L_\perp^\top \right\|_{\mathrm{F}}^2$$

$$+ 2 \left\langle L_\perp L_\perp^\top \underline{\Phi} S^* \underline{\Phi}^\top L_\perp L_\perp^\top , L_\perp L_\perp^\top \Phi S^* \Phi^\top L_\perp L_\perp^\top \right\rangle .$$

Also, we can obtain

$$\left\| L_\perp^\top \hat{\Delta} L_\perp \right\|_{\mathrm{F}}^2 = \left\| L_\perp L_\perp^\top \widehat{U} \widehat{S} \widehat{V}^\top L_\perp L_\perp^\top \right\|_{\mathrm{F}}^2$$

$$= \left\| L_\perp L_\perp^\top \left( \Phi S^* \Phi^\top - \underline{\Phi} S^* \underline{\Phi}^\top \right) L_\perp L_\perp^\top \right\|_{\mathrm{F}}^2 \qquad \text{[by Eq. (80)]}$$

$$= \left\| L_\perp L_\perp^\top \underline{\Phi} S^* \underline{\Phi}^\top L_\perp L_\perp^\top \right\|_{\mathrm{F}}^2 + \left\| L_\perp L_\perp^\top \Phi S^* \Phi^\top L_\perp L_\perp^\top \right\|_{\mathrm{F}}^2 - 2 \left\langle L_\perp L_\perp^\top \underline{\Phi} S^* \underline{\Phi}^\top L_\perp L_\perp^\top , L_\perp L_\perp^\top \Phi S^* \Phi^\top L_\perp L_\perp^\top \right\rangle .$$

Notice that the matrix inner product term is the inner product of two positive semi-definite matrices, then by trace inequality for positive semi-definite matrices, we can obtain

$$\left\langle L_\perp L_\perp^\top \underline{\Phi} S^* \underline{\Phi}^\top L_\perp L_\perp^\top , L_\perp L_\perp^\top \Phi S^* \Phi^\top L_\perp L_\perp^\top \right\rangle = \mathrm{tr} \left( \left( L_\perp L_\perp^\top \underline{\Phi} S^* \underline{\Phi}^\top L_\perp L_\perp^\top \right) \left( L_\perp L_\perp^\top \Phi S^* \Phi^\top L_\perp L_\perp^\top \right) \right) \geq 0 .$$

Then, we can claim

$$\left\| L_\perp L_\perp^\top \widehat{U} \widehat{S} \widehat{U}^\top L_\perp L_\perp^\top \right\|_{\mathrm{F}}^2 , \left\| L_\perp L_\perp^\top \widehat{V} \widehat{S} \widehat{V}^\top L_\perp L_\perp^\top \right\|_{\mathrm{F}}^2 \geq \left\| L_\perp^\top \hat{\Delta} L_\perp \right\|_{\mathrm{F}}^2 . \tag{88}$$

Next, we can obtain

$$\left\| L_\perp^\top \hat{\Delta} L \right\|_{\mathrm{F}}^2 + \left\| L^\top \hat{\Delta} L_\perp \right\|_{\mathrm{F}}^2 \geq \lambda_{r^*}^* \times \lambda_{2r^*} \left( \widehat{U}^\top L L^\top \widehat{U} \right) \left\| L_\perp L_\perp^\top \widehat{V} \widehat{S} \widehat{V}^\top L_\perp L_\perp^\top \right\|_{\mathrm{F}}$$

$$+ \lambda_{r^*}^* \times \lambda_{2r^*} \left( \widehat{V}^\top L L^\top \widehat{V} \right) \left\| L_\perp L_\perp^\top \widehat{U} \widehat{S} \widehat{U}^\top L_\perp L_\perp^\top \right\|_{\mathrm{F}}$$

$$\geq \lambda_{r^*}^* \min \left\{ \lambda_{2r^*} \left( \widehat{U}^\top L L^\top \widehat{U} \right) , \lambda_{2r^*} \left( \widehat{V}^\top L L^\top \widehat{V} \right) \right\}$$

$$\times \left( \left\| L_\perp L_\perp^\top \widehat{V} \widehat{S} \widehat{V}^\top L_\perp L_\perp^\top \right\|_{\mathrm{F}} + \left\| L_\perp L_\perp^\top \widehat{U} \widehat{S} \widehat{U}^\top L_\perp L_\perp^\top \right\|_{\mathrm{F}} \right)$$

$$\geq 2 \lambda_{r^*}^* \min \left\{ \lambda_{2r^*} \left( \widehat{U}^\top L L^\top \widehat{U} \right) , \lambda_{2r^*} \left( \widehat{V}^\top L L^\top \widehat{V} \right) \right\} \left\| L_\perp^\top \hat{\Delta} L_\perp \right\|_{\mathrm{F}} . \qquad \text{[by Eq. (88)]}$$

Then, combining the above inequality and Eq. (87), we have

$$\frac{\frac{1}{2} \left\| L_\perp^\top \hat{\Delta} L_\perp \right\|_{\mathrm{F}}^2}{\left\| F - \frac{\hat{\Delta}}{\sqrt{2}} \right\|_{\mathrm{F}}^2} \leq \frac{\left\| L_\perp^\top \hat{\Delta} L_\perp \right\|_{\mathrm{F}}^2}{2 \lambda_{r^*}^* \min \left\{ \lambda_{2r^*} \left( \widehat{U}^\top L L^\top \widehat{U} \right) , \lambda_{2r^*} \left( \widehat{V}^\top L L^\top \widehat{V} \right) \right\} \left\| L_\perp^\top \hat{\Delta} L_\perp \right\|_{\mathrm{F}}}$$

$$= \frac{\left\| L_\perp^\top \hat{\Delta} L_\perp \right\|_{\mathrm{F}}}{2 \lambda_{r^*}^* \min \left\{ \lambda_{2r^*} \left( \widehat{U}^\top L L^\top \widehat{U} \right) , \lambda_{2r^*} \left( \widehat{V}^\top L L^\top \widehat{V} \right) \right\}} .$$

Next, we will focus on the lower bound of $\lambda_{2r^*} \left( \widehat{U}^\top L L^\top \widehat{U} \right)$ and $\lambda_{2r^*} \left( \widehat{V}^\top L L^\top \widehat{V} \right)$. Due to symmetry, the technique is identical to each other, so here we only prove for $\lambda_{2r^*} \left( \widehat{U}^\top L L^\top \widehat{U} \right)$. First, $\lambda_{2r^*} \left( \widehat{U}^\top L L^\top \widehat{U} \right) = \lambda_{2r^*}^2 \left( L^\top \widehat{U} \right)$ since $\widehat{U}^\top L L^\top \widehat{U}$ is symmetric. Next, we have

$$\lambda_{2r^*}^2 \left( L^\top \widehat{U} \right) = 1 - \left\| L_\perp^\top \widehat{U} \right\|_{op}^2 ,$$

where $\left\|\boldsymbol{L}_\perp^\top \widehat{\boldsymbol{U}}\right\|_{op}$ can be upper bounded by Wedin's $\sin(\Theta)$ theorem, here we use a variant from in Chen et al. (2021b, Theorem 2.9) to obtain

$$\left\|\boldsymbol{L}_\perp^\top \widehat{\boldsymbol{U}}\right\|_{op} \leq \frac{2\left\|\boldsymbol{F} - \frac{\widehat{\boldsymbol{\Delta}}}{\sqrt{2}}\right\|_{op}}{\lambda_{2r^*}^*\left(\frac{\widehat{\boldsymbol{\Delta}}}{\sqrt{2}}\right)} \leq \frac{2\sqrt{2}\left\|\boldsymbol{F} - \frac{\widehat{\boldsymbol{\Delta}}}{\sqrt{2}}\right\|_{\mathrm{F}}}{\lambda_{r^*}^*} \leq 2\sqrt{2}\rho, \qquad \left[\text{by } \left\|\boldsymbol{F} - \frac{\widehat{\boldsymbol{\Delta}}}{\sqrt{2}}\right\|_{\mathrm{F}} \leq \rho\lambda_{r^*}^*\right]$$

which implies

$$\lambda_{2r^*}^2\left(\boldsymbol{L}^\top \widehat{\boldsymbol{U}}\right) \geq 1 - 8\rho^2. \qquad (89)$$

Therefore, we have

$$\begin{aligned}
\rho^2(\lambda_{r^*}^*)^2 = \rho^2\lambda_{2r^*}^2(\widehat{\boldsymbol{\Delta}}) &\geq \left\|\boldsymbol{F} - \frac{\widehat{\boldsymbol{\Delta}}}{\sqrt{2}}\right\|_{\mathrm{F}}^2 \\
&\geq \frac{1}{2}\left\|\boldsymbol{L}_\perp^\top \widehat{\boldsymbol{\Delta}} \boldsymbol{L}\right\|_{\mathrm{F}}^2 + \frac{1}{2}\left\|\boldsymbol{L}^\top \widehat{\boldsymbol{\Delta}} \boldsymbol{L}_\perp\right\|_{\mathrm{F}}^2 && [\text{by Eq. (85)}] \\
&\geq \lambda_{r^*}^* \min\left\{\lambda_{2r^*}\left(\widehat{\boldsymbol{U}}^\top \boldsymbol{L}\boldsymbol{L}^\top \widehat{\boldsymbol{U}}\right), \lambda_{2r^*}\left(\widehat{\boldsymbol{V}}^\top \boldsymbol{L}\boldsymbol{L}^\top \widehat{\boldsymbol{V}}\right)\right\}\left\|\boldsymbol{L}_\perp^\top \widehat{\boldsymbol{\Delta}} \boldsymbol{L}_\perp\right\|_{\mathrm{F}} \\
&\geq \frac{1}{2}\lambda_{r^*}^*\left\|\boldsymbol{L}_\perp^\top \widehat{\boldsymbol{\Delta}} \boldsymbol{L}_\perp\right\|_{\mathrm{F}}, && [\text{by Eq. (89) and } \rho \leq 1/4]
\end{aligned}$$

which implies

$$\frac{\left\|\boldsymbol{L}_\perp^\top \widehat{\boldsymbol{\Delta}} \boldsymbol{L}_\perp\right\|_{\mathrm{F}}}{\lambda_{r^*}^*} \leq 2\rho^2.$$

Finally, combining Eq. (86), we can obtain

$$\left\|\left(\boldsymbol{I}_d - \boldsymbol{U}_{\boldsymbol{A}_t}\boldsymbol{U}_{\boldsymbol{A}_t}^\top\right)\boldsymbol{\Delta}\left(\boldsymbol{I}_k - \boldsymbol{V}_{\boldsymbol{B}_t}\boldsymbol{V}_{\boldsymbol{B}_t}^\top\right)\right\|_{\mathrm{F}}^2 = \frac{1}{2}\left\|\boldsymbol{L}_\perp^\top \widehat{\boldsymbol{\Delta}} \boldsymbol{L}_\perp\right\|_{\mathrm{F}}^2 \leq \frac{\rho^2}{1 - 8\rho^2}\|\boldsymbol{A}_t\boldsymbol{B}_t - \boldsymbol{\Delta}\|_{\mathrm{F}}^2.$$

Notice that

$$\begin{aligned}
\frac{1}{2}\left\|\boldsymbol{L}_\perp^\top \widehat{\boldsymbol{\Delta}} \boldsymbol{L}\right\|_{\mathrm{F}}^2 + \frac{1}{2}\left\|\boldsymbol{L}^\top \widehat{\boldsymbol{\Delta}} \boldsymbol{L}_\perp\right\|_{\mathrm{F}}^2 &= \frac{1}{2}\left\|\boldsymbol{P}_t\widehat{\boldsymbol{\Delta}}\left(\boldsymbol{I}_{d+k} - \boldsymbol{P}_t\right)\right\|_{\mathrm{F}}^2 + \frac{1}{2}\left\|\left(\boldsymbol{I}_{d+k} - \boldsymbol{P}_t\right)\widehat{\boldsymbol{\Delta}}\boldsymbol{P}_t\right\|_{\mathrm{F}}^2 \\
&= \left\|\left(\boldsymbol{I}_d - \boldsymbol{U}_{\boldsymbol{A}_t}\boldsymbol{U}_{\boldsymbol{A}_t}^\top\right)\boldsymbol{\Delta}\boldsymbol{V}_{\boldsymbol{B}_t}\boldsymbol{V}_{\boldsymbol{B}_t}^\top\right\|_{\mathrm{F}}^2 + \left\|\boldsymbol{U}_{\boldsymbol{A}_t}\boldsymbol{U}_{\boldsymbol{A}_t}^\top\boldsymbol{\Delta}\left(\boldsymbol{I}_k - \boldsymbol{V}_{\boldsymbol{B}_t}\boldsymbol{V}_{\boldsymbol{B}_t}^\top\right)\right\|_{\mathrm{F}}^2 \\
&= \left\|\left(\boldsymbol{I}_d - \boldsymbol{U}_{\boldsymbol{A}_t}\boldsymbol{U}_{\boldsymbol{A}_t}^\top\right)\boldsymbol{\Delta}\boldsymbol{V}_{\boldsymbol{B}_t}\boldsymbol{V}_{\boldsymbol{B}_t}^\top + \boldsymbol{U}_{\boldsymbol{A}_t}\boldsymbol{U}_{\boldsymbol{A}_t}^\top\boldsymbol{\Delta}\left(\boldsymbol{I}_k - \boldsymbol{V}_{\boldsymbol{B}_t}\boldsymbol{V}_{\boldsymbol{B}_t}^\top\right)\right\|_{\mathrm{F}}^2,
\end{aligned}$$

then by the decomposition in Eq. (85), we can obtain

$$\left\|\left(\boldsymbol{I}_d - \boldsymbol{U}_{\boldsymbol{A}_t}\boldsymbol{U}_{\boldsymbol{A}_t}^\top\right)\boldsymbol{\Delta}\boldsymbol{V}_{\boldsymbol{B}_t}\boldsymbol{V}_{\boldsymbol{B}_t}^\top + \boldsymbol{U}_{\boldsymbol{A}_t}\boldsymbol{U}_{\boldsymbol{A}_t}^\top\boldsymbol{\Delta}\left(\boldsymbol{I}_k - \boldsymbol{V}_{\boldsymbol{B}_t}\boldsymbol{V}_{\boldsymbol{B}_t}^\top\right)\right\|_{\mathrm{F}}^2 \leq \left\|\boldsymbol{F} - \frac{\widehat{\boldsymbol{\Delta}}}{\sqrt{2}}\right\|_{\mathrm{F}}^2 = \|\boldsymbol{A}_t\boldsymbol{B}_t - \boldsymbol{\Delta}\|_{\mathrm{F}}^2,$$

which completes the proof. $\qquad\square$

Based on the above estimation, we are ready to deliver the linear convergence rate of $\|\boldsymbol{A}_t\boldsymbol{B}_t - \boldsymbol{\Delta}\|_{\mathrm{F}}$.

**Theorem D.10.** *Suppose* $\epsilon \leq \frac{\rho}{3C^*K^2\gamma\sqrt{2dr^*}\kappa}$ *for a positive constant* $\rho \leq \frac{1}{20}$*, we take* $\gamma = 2$ *for* (Spectral-init)*, set* $\eta \in (c_\eta, 1)$ *where* $c_\eta > 0$ *is a small constant, under assumptions in Section 2.1 for the nonlinear setting and Assumption 4.1, then with probability at least* $1 - 2Cdk\exp\left(-\epsilon^2 N\right)$ *for a universal constant* $C > 0$*, we have*

$$\|\boldsymbol{A}_t\boldsymbol{B}_t - \boldsymbol{\Delta}\|_{\mathrm{F}} \leq \left(1 - \frac{\eta}{4}\right)^t \rho\lambda_{r^*}^*.$$

*Proof.* We prove it by induction. The following hypothesis holds at $t = 0$ by Lemma D.5, i.e.

$$\|\boldsymbol{A}_t \boldsymbol{B}_t - \Delta\|_{\mathrm{F}} \leq \rho \lambda_{r^*}^* .$$

We suppose it also holds for at time $t$, then the conditions of Lemma D.7, Lemma D.8, and Lemma D.9 are fulfilled. By consequence, we can show that

$$\lambda_{r^*} (\boldsymbol{A}_t \boldsymbol{B}_t) \geq (1 - \rho)\lambda_{r^*}^* . \qquad \text{[by Weyl's inequality]}$$

Next, by Eq. (77) from Lemma D.6, under initial conditions from Lemma D.5, for time $t + 1$, we can derive

$$
\begin{aligned}
&\|\boldsymbol{A}_{t+1} \boldsymbol{B}_{t+1} - \Delta\|_{\mathrm{F}} \\
\leq &(1 - \eta) \left\|\boldsymbol{U}_{\boldsymbol{A}_t} \boldsymbol{U}_{\boldsymbol{A}_t}^\top (\boldsymbol{A}_t \boldsymbol{B}_t - \Delta) \boldsymbol{V}_{\boldsymbol{B}_t} \boldsymbol{V}_{\boldsymbol{B}_t}^\top \right\|_{\mathrm{F}} \\
&+ (1 - \eta/2) \left\|\left(\boldsymbol{I}_d - \boldsymbol{U}_{\boldsymbol{A}_t} \boldsymbol{U}_{\boldsymbol{A}_t}^\top\right) (\boldsymbol{A}_t \boldsymbol{B}_t - \Delta) \boldsymbol{V}_{\boldsymbol{B}_t} \boldsymbol{V}_{\boldsymbol{B}_t}^\top + \boldsymbol{U}_{\boldsymbol{A}_t} \boldsymbol{U}_{\boldsymbol{A}_t}^\top (\boldsymbol{A}_t \boldsymbol{B}_t - \Delta) \left(\boldsymbol{I}_k - \boldsymbol{V}_{\boldsymbol{B}_t} \boldsymbol{V}_{\boldsymbol{B}_t}^\top\right)\right\|_{\mathrm{F}} \\
&+ \left\|\left(\boldsymbol{I}_d - \boldsymbol{U}_{\boldsymbol{A}_t} \boldsymbol{U}_{\boldsymbol{A}_t}^\top\right) (\boldsymbol{A}_t \boldsymbol{B}_t - \Delta) \left(\boldsymbol{I}_k - \boldsymbol{V}_{\boldsymbol{B}_t} \boldsymbol{V}_{\boldsymbol{B}_t}^\top\right)\right\|_{\mathrm{F}} \\
&+ 2\eta \|\boldsymbol{\Xi}_t\|_{\mathrm{F}} + \eta^2 \left\|\boldsymbol{J}_{\boldsymbol{W}_t} \mathcal{V}_t \mathcal{S}_t^{-1} \mathcal{U}_t^\top \boldsymbol{J}_{\boldsymbol{W}_t}\right\|_{\mathrm{F}} \\
\leq &(1 - \eta) \|\boldsymbol{A}_t \boldsymbol{B}_t - \Delta\|_{\mathrm{F}} \\
&+ \left(1 - \eta/2 + \frac{\rho}{\sqrt{1 - 8\rho^2}}\right) \|\boldsymbol{A}_t \boldsymbol{B}_t - \Delta\|_{\mathrm{F}} && \text{[by Lemma D.9]} \\
&+ 2\eta \left(\mathcal{O}\left(\frac{1}{\kappa r^*}\right) + C^* K^2 \sqrt{d} \epsilon\right) \|\boldsymbol{A}_t \boldsymbol{B}_t - \Delta\|_{\mathrm{F}} && \text{[by Lemma D.7]} \\
&+ \eta^2 (1 + \epsilon)^2 \frac{\rho}{1 - \rho} \|\boldsymbol{A}_t \boldsymbol{B}_t - \Delta\|_{\mathrm{F}} && \text{[by Lemma D.8]} \\
\leq &\left(2 - 3\eta/2 + \frac{\rho}{\sqrt{1 - 8\rho^2}}\right) \|\boldsymbol{A}_t \boldsymbol{B}_t - \Delta\|_{\mathrm{F}} \\
&+ \eta \left(\frac{2\rho}{3\gamma\sqrt{2}r^*\kappa}\right) \|\boldsymbol{A}_t \boldsymbol{B}_t - \Delta\|_{\mathrm{F}} \\
&+ \eta^2 \left(1 + \frac{\rho}{3C^* K^2 \gamma \sqrt{2d}r^*\kappa}\right)^2 \frac{\rho}{1 - \rho} \|\boldsymbol{A}_t \boldsymbol{B}_t - \Delta\|_{\mathrm{F}} , && \left[\text{since } \epsilon \leq \frac{\rho}{3C^* K^2 \gamma \sqrt{2d}r^*\kappa}\right]
\end{aligned}
$$

with probability at least $1 - 2Cdk \exp\left(-\epsilon^2 N\right)$ for a universal constant $C > 0$. Since we take $\rho \leq \frac{1}{20}$, and $\frac{\rho}{\sqrt{1-8\rho^2}}$ is monotonically increasing, then there exists a constant $c_\eta > 0$ such that $\forall \eta \in (c_\eta, 1)$, we have

$$\|\boldsymbol{A}_{t+1} \boldsymbol{B}_{t+1} - \Delta\|_{\mathrm{F}} \leq \left(1 - \frac{\eta}{4}\right) \|\boldsymbol{A}_t \boldsymbol{B}_t - \Delta\|_{\mathrm{F}} .$$

Then, we can obtain the inductive hypothesis at $t + 1$ and prove the claim. $\qquad \square$

## E. Auxiliary Results for Proofs

In this subsection, we present some auxiliary results that are needed for our proof. First, we present the estimation of the spectral norm of random matrices. It can be easily derived from (Vershynin, 2018) and we put it here for the completeness.

**Lemma E.1.** *(Vershynin, 2018, Adapted from Theorem 4.6.1) For a random sub-Gaussian matrix $\widetilde{\boldsymbol{X}} \in \mathbb{R}^{N \times d}$ whose rows are i.i.d. isotropic sub-gaussian random vector with sub-Gaussian norm $K$, then we have the following statement*

$$\mathbb{P}\left(\left\|\frac{1}{N}\widetilde{\boldsymbol{X}}^\top \widetilde{\boldsymbol{X}} - \boldsymbol{I}_d\right\|_{op} > \delta\right) \leq 2\exp\left(-CN \min\left(\delta^2, \delta\right)\right) .$$

*for a universal constant $C$ depending only on $K$.*

**Lemma E.2.** *(Vershynin, 2010, Adapted from Corollary 5.35)  For a random standard Gaussian matrix $S \in \mathbb{R}^{d \times r}$ with $[S]_{ij} \sim \mathcal{N}(0, 1)$, if $d > 2r$, we have*

$$\frac{\sqrt{d}}{2} \leq \|S\|_{op} \leq (2\sqrt{d} + \sqrt{r}), \tag{90}$$

*with probability at least $1 - C \exp(-d)$ for some positive constants $C$.*

The following results are modified from the proof of Stöger & Soltanolkotabi (2021, Lemma 8.7).

**Lemma E.3.** *Suppose $S \in \mathbb{R}^{d \times r}$ is a random standard Gaussian matrix with $[S]_{ij} \sim \mathcal{N}(0, 1)$ and $U \in \mathbb{R}^{d \times r^*}$ has orthonormal columns. If $r \geq 2r^*$, with probability at least $1 - C \exp(-r)$ for some positive constants $C$, we have*

$$\lambda_{\min}(U^\top S) \gtrsim 1.$$

*If $r^* \leq r < 2r^*$, by choosing $\xi > 0$ appropriately, with probability at least $1 - (C\xi)^{r - r^* + 1} - C' \exp(-r)$ for some positive constants $C, C'$, we have*

$$\lambda_{\min}(U^\top S) \gtrsim \frac{\xi}{r}.$$

**Lemma E.4.** *(Brutzkus & Globerson, 2017, Lemma 3.2) Define $\theta(w, v) = \cos^{-1}\left(\frac{\langle w, v \rangle}{\|w\|_2 \|v\|_2}\right)$, then we have*

$$h(w, v) := \frac{\partial}{\partial w} \mathbf{E}_{\widetilde{x} \sim \mathcal{N}(0, I_d)} \Big[ \sigma\left(\langle w, \widetilde{x} \rangle\right) \sigma\left(\langle v, \widetilde{x} \rangle\right) \Big] = \frac{1}{2\pi} \left[ \frac{\|v\|_2}{\|w\|_2} \sin \theta(w, v) w + (\pi - \theta(w, v)) v \right].$$

## F. Detailed Comparison with LoRA-GA

LoRA-GA proposes the following initialization strategy

$$A_0 = -\frac{k^{1/4}}{c} \left[ \widetilde{U}_{G^\natural} \right]_{[:, 1:r]}, \, B_0 = \frac{k^{1/4}}{c} \left[ \widetilde{V}_{G^\natural} \right]_{[:, r+1:2r]}^\top,$$
$$W_{\text{off}}^\natural := W^\natural - \frac{\alpha}{\sqrt{r}} A_0 B_0, \tag{91}$$

where $k$ is the output dimension, $c$ is a user-specified hyperparameter in constant order. They propose to recover the one-step full gradient $G^\natural$ to the largest extent after the first LoRA update, i.e., under gradient descent with stepsize $\eta$, the adapted weight becomes

$$W_{\text{off}}^\natural + A_1 B_1 := W_{\text{off}}^\natural + \frac{\alpha}{\sqrt{r}} A_0 B_0 + \frac{\alpha}{\sqrt{r}} \left[ -\eta G^\natural B_0^\top B_0 - \eta A_0 A_0^\top G^\natural + \eta^2 G^\natural B_0^\top A_0^\top G^\natural \right]$$

$$= W^\natural + \frac{\alpha}{\sqrt{r}} \underbrace{\left[ -\eta G^\natural B_0^\top B_0 - \eta A_0 A_0^\top G^\natural + \eta^2 G^\natural B_0^\top A_0^\top G^\natural \right]}_{\texttt{update in the full parameter space}}.$$

Then, $A_0$ and $B_0$ in Eq. (91) can admit the best rank-$2r$ approximation of $G^\natural$ in terms of full parameter update as they drop the $\eta^2$-term. However, this scheme has structural limitations in various perspectives.

First, as pointed by our theory, $B_t$ will align to the right-side rank-$r^*$ singular subspace of $G^\natural$ under random initialization. That means, due to the way LoRA-GA chooses the $(r + 1)$-th to $2r$-th singular values for $B_0$, the iterate $B_t$ does not lie in the desired subspace and may not escape an undesirable subspace.

Second, for common LoRA-based algorithms including ours, the global optimization problem is to solve

$$\min_{A, B} \left\| \frac{\alpha}{\sqrt{r}} A B - \Delta \right\|_{\text{F}}^2, \tag{92}$$

which achieves the global minimum at the best rank-$r$ approximation of $\Delta$. However, under LoRA-GA in Eq. (91), the modifications to the pre-trained weight, i.e. $\boldsymbol{W}^\natural$, lead to an unfavorable optimization problem, i.e. to solve

$$\min_{\boldsymbol{A},\boldsymbol{B}} \left\| \boldsymbol{W}^\natural + \frac{\alpha}{\sqrt{r}} \left(\boldsymbol{A}\boldsymbol{B} - \boldsymbol{A}_0\boldsymbol{B}_0\right) - \widetilde{\boldsymbol{W}} \right\|_{\mathrm{F}}^2$$

$$\Leftrightarrow \quad \min_{\boldsymbol{A},\boldsymbol{B}} \left\| \frac{\alpha}{\sqrt{r}} \boldsymbol{A}\boldsymbol{B} - \left(\frac{\alpha}{\sqrt{r}} \boldsymbol{A}_0\boldsymbol{B}_0 + \Delta\right) \right\|_{\mathrm{F}}^2,$$

which achieves the global minimum at a **biased** best rank-$r$ approximation of $\Delta$, i.e. $\frac{\alpha}{\sqrt{r}}\boldsymbol{A}_0\boldsymbol{B}_0 + \Delta$, **no matter what initialization it is**. This upward bias scales in the order of $\Theta(\sqrt{k})$ as they propose the scaling to be $\sqrt{k}/c^2$ for stability and can be dominant if it has stronger signal than downstream feature $\Delta$.

Lastly, since they ignore $\eta^2$-term in the illustrative analysis, this imposes a latent assumption that the best rank-$2r$ approximation of $\boldsymbol{G}^\natural$ in terms of full parameter update only holds if the stepsize $\eta \approx 0$. This restriction is consistent with ablation results from Wang et al. (2024) that LoRA-GA is not robust under moderate/large stepsize.

In contrast, LoRA-One aligns well with our theory under correct subspace specification. We do not modify the pre-trained weight so the optimization problem remain the same as Eq. (92). Also, our method is robust to the choice of stepsizes and can undertake large stepsize to achieve faster convergence as shown in the Appendix G.2.

# G. Experimental Settings and Additional Results

In Appendix G.1, we firstly provide the experimental details of small-scale experiments in our text. Experimental settings of various NLP tasks in the main text are given by Appendix G.2, Appendix G.3, and Appendix G.4, respectively. Finally, we visualize the spectral properties of both the pre-trained weights and the difference weights after fine-tuning in Appendix G.5 to justify the validity of Assumption 4.1. All small-scale experiments were performed on AMD EPYC 7B12 CPU. All experiments for T5 base model and Llama 2-7B were performed on Nvidia A100 GPUs (40GB).

## G.1. Small-Scale Experiments

Here we give the experimental details of Fig. 1(c), Fig. 2, Fig. 3, and Fig. 4, respectively.

**Details for Fig. 1(c)** The experimental settings are sourced from Meng et al. (2024). We use 10000 odd-labeled data from MNIST (LeCun, 1998) for pre-training and 1000 even-labeled data for fine-tuning. The learning rates for Full Fine-tuning, LoRA, and LoRA-One are set to $5 \times 10^{-4}$.

**Details for Fig. 2:** We initialize $\boldsymbol{A}_0$ and $\boldsymbol{B}_0$ via (LoRA-init) over variance $\alpha^2 \in \{1, 0.1, 0.01, 0.001, 0.0001\}$. We examine for dimension $d = k = 100$ and $d = k = 1000$. We set $N = 16d$, $r^* = 4$, and $r = 8$. We construct $\Delta := \boldsymbol{U}\boldsymbol{V}^\top$ where $\boldsymbol{U} \in \mathbb{R}^{100 \times 4}$ and $\boldsymbol{V} \in \mathbb{R}^{100 \times 4}$ are obtained from the SVD of a matrix whose elements are independently sampled from $\mathcal{N}(0, 1)$. We set learning rate $\eta = \frac{1}{64}$. We run 1500 GD steps for each case.

**Details for Fig. 4:** We take $d = k = 100$ and $N = 12800$ in common. For: 1) under-ranked case $r = 4, r^* = 8$, 2) over-ranked case $r = 8, r^* = 4$. We sample each element of $\boldsymbol{W}^\natural$ independently from $\mathcal{N}(0, 1)$. We construct $\Delta := \boldsymbol{U}\boldsymbol{S}\boldsymbol{V}^\top$ where $\boldsymbol{U} \in \mathbb{R}^{100 \times r^*}$ and $\boldsymbol{V} \in \mathbb{R}^{100 \times r^*}$ are obtained from the SVD of a matrix whose elements are independently sampled from $\mathcal{N}(0, 1)$ and the diagonal values of $\boldsymbol{S}$ is the first $r^*$ elements of the dictionary $\{40, 30, 20, 10, 1, 1, 1, 0.5\}$. For LoRA-GA defined in Eq. (91), we use learning rate $\eta = 0.5$ and stable parameter 16. For LoRA-SB and LoRA-One, we use learning rate $\eta = 0.5$ and scaling parameter 1.

**Comparison on GD trajectories of Fig. 3:** Here we conduct a toy experiment to intuitively compare the GD trajectories under (Spectral-init) and (LoRA-init). We fine-tune a simple pre-trained model $y = \boldsymbol{x}^\top \boldsymbol{w}^\natural$ on downstream data generated by $\widetilde{y} = \widetilde{\boldsymbol{x}}^\top (\boldsymbol{w}^\natural + \boldsymbol{w})$, where $\boldsymbol{x}^\top, \widetilde{\boldsymbol{x}}, \boldsymbol{w}^\natural, \boldsymbol{w} \in \mathbb{R}^2$ and $y, \widetilde{y} \in \mathbb{R}$. We propose to use LoRA to fine-tune this model by $\widehat{y} = \widetilde{\boldsymbol{x}}^\top (\boldsymbol{w}^\natural + b\boldsymbol{a})$ where $\boldsymbol{a} = [a_1 \, a_2]^\top \in \mathbb{R}^2$ and $b \in \mathbb{R}$. Without loss of generality, we set $\boldsymbol{w}^\natural = \boldsymbol{0}$ and $\boldsymbol{w} = [2 \, 1]^\top$. The set of global minimizers to this problem is $\{a_1^* = 2/t, a_2^* = 1/t, b^* = t \mid t \in \mathbb{R}\}$. We generate 4 data points $(\widetilde{\boldsymbol{x}}_1, \widetilde{\boldsymbol{x}}_2, \widetilde{\boldsymbol{x}}_3, \widetilde{\boldsymbol{x}}_4)$ whose elements are independently sampled from $\mathcal{N}(0, 1)$ and calculate for $(\widetilde{y}_1, \widetilde{y}_2, \widetilde{y}_3, \widetilde{y}_4)$. We use the squared loss $\frac{1}{8} \sum_{i=1}^4 (\widetilde{y}_i - b\widetilde{\boldsymbol{x}}^\top \boldsymbol{a})^2$. For (LoRA-init), we initialize each element of $\boldsymbol{a}_0$ from $\mathcal{N}(0, 1)$ and $b_0 = 0$. Notice that the variance 1 follows from the Kaiming initialization (He et al., 2015). For (Spectral-init), we first calculate the one-step full gradient, i.e. $\boldsymbol{g}^\natural := \frac{1}{4} \sum_{i=1}^4 \widetilde{y}_i^2 \widetilde{\boldsymbol{x}}_i$. Accordingly, we initialize $\boldsymbol{a}_0 = \frac{\boldsymbol{g}^\natural}{\sqrt{\|\boldsymbol{g}^\natural\|_2}}$ and $b_0 = \sqrt{\|\boldsymbol{g}^\natural\|_2}$. Next, we run GD to train $\boldsymbol{a}$ and $b$ for

1000 steps with learning rate $\eta = 0.1$. For each initialization strategy and data generation, we run for 2 different seeds.

## G.2. Natural Language Generation

The common hyperparameters are presented in Table 6. Next, we present in the order of {MetaMathQA, Code-Feedback, Alpaca}. We search the best learning rate over $\{5 \times 10^{-4}, 2 \times 10^{-4}, 1 \times 10^{-4}, 5 \times 10^{-5}, 2 \times 10^{-5}, 1 \times 10^{-5}\}$ and batch size over $\{16, 32, 128\}$. The optimized learning rate and batch size are presented in Table 7. Additionally, the scale parameters are set to $\{128, 16, 128\}$ for LoRA-One and $\{64, 64, 64\}$ for LoRA-GA.

Furthermore, we employ the gradient approximation approach proposed by Wang et al. (2024) to replace the full-batch full gradient with stochastic full gradient using a smaller sampled batch from training data and denote the corresponding sample size as Gradient Batch Size. According to the ablation studies on spectral properties and performance under various gradient batch sizes in (Wang et al., 2024), larger gradient batch size only can yield marginal improvement, indicating that it is sufficient to use a smaller batch size for computational efficiency.

Table 6: Common hyperparameters for fine-tuning LLaMA 2-7B on MetaMathQA, Code-Feedback, and Alpaca.

| Epoch | Optimizer | $(\beta_1, \beta_2)$ | $\epsilon$ | LoRA Precision | Weight Decay |
|---|---|---|---|---|---|
| 1 | AdamW | (0.9, 0.999) | $1 \times 10^{-8}$ | FP32 | 0 |
| Warm-up Ratio | LoRA $\alpha$ | LR Scheduler | Max Length | #Runs | Gradient Batch Size |
| 0.03 | 16 | cosine | 1024 | 3 | 8 |

Table 7: Optimized hyperparameters for LoRA, LoRA-GA, and LoRA-One.

| | Batch Size | Learning Rate |
|---|---|---|
| LoRA | $\{32, 32, 32\}$ | $\{2 \times 10^{-4}, 2 \times 10^{-4}, 5 \times 10^{-5}\}$ |
| LoRA-GA | $\{32, 32, 32\}$ | $\{5 \times 10^{-5}, 5 \times 10^{-5}, 1 \times 10^{-5}\}$ |
| LoRA-One | $\{32, 32, 16\}$ | $\{2 \times 10^{-4}, 5 \times 10^{-4}, 2 \times 10^{-4}\}$ |

## G.3. Math Reasoning on Full Data and Multiple Epochs

We present the detailed values of Fig. 5 in Table 8. The common hyperparameters are same as Table 6. We search the best learning rate over $\{5 \times 10^{-4}, 2 \times 10^{-4}, 1 \times 10^{-4}, 5 \times 10^{-5}, 2 \times 10^{-5}, 1 \times 10^{-5}\}$ and batch size over $\{16, 32, 64, 128\}$. The optimized learning rate and batch size are presented in Table 9. Additionally, the scale parameter are set to 128 for LoRA-One and 64 for LoRA-GA. The imbalance parameter for LoRA+ is set to 16. The results of LoRA, LoRA+, and LoRA-GA are taken from (Wang et al., 2024) since their optimized hyperparameters align with our search.

Table 8: Performance comparison across different methods and epochs

| | Epoch 1 | Epoch 2 | Epoch 3 | Epoch 4 |
|---|---|---|---|---|
| LoRA | 55.19 | 58.37 | 59.28 | 58.90 |
| LoRA+ | 56.37 | 59.21 | 59.93 | 59.97 |
| LoRA-GA | 56.48 | 58.64 | 60.16 | 60.88 |
| LoRA-One | **57.54** | **60.84** | **62.62** | **63.80** |

## G.4. Natural Language Understanding

In Section 6.1, we have presented the experimental comparisons between Algorithm 1 and typical LoRA based algorithms. We follow the configuration of prompt tuning as Wang et al. (2024). The general hyperparameter settings are provides in Table 10. To ensure a fair comparison, we tune the learning rate via grid search over $\{1 \times 10^{-3}, 5 \times 10^{-4}, 2 \times 10^{-4}, 1 \times 10^{-4}\}$.

Table 9: Optimized hyperparameters for LoRA, LoRA+, LoRA-GA, and LoRA-One.

|          | Batch Size | Learning Rate      |
|----------|------------|--------------------|
| LoRA     | 128        | $1 \times 10^{-4}$ |
| LoRA+    | 128        | $5 \times 10^{-5}$ |
| LoRA-GA  | 128        | $5 \times 10^{-5}$ |
| LoRA-One | 128        | $2 \times 10^{-4}$ |

Additionally, the scale parameters for LoRA-One are set to be $\{128, 16, 128, 128, 64\}$ for MNLI, SST-2, CoLA, QNLI, and MRPC.

For Section 6.1, the learning rates for one-step gradient update with gradient batch size 2048 are set to be $\{0.1, 1.0, 0.05, 0.1\}$ for SST-2, CoLA, QNLI, and MRPC. The learning rates for low-rank update ($r = 8$) with gradient batch size 8 are set to be $\{1 \times 10^{-4}, 0.1, 5 \times 10^{-2}, 5 \times 10^{-2}\}$. We omit results for MNLI since the test accuracy remains at 0.0% for the first dozen steps in both full and LoRA fine-tuning, likely due to a substantial structural discrepancy between pre-training and downstream tasks.

Table 10: Common hyperparameters for LoRA fine-tuning on T5-base model.

| Epoch | Optimizer | $(\beta_1, \beta_2)$ | $\epsilon$ | Batch Size | Weight Decay | LR Scheduler |
|-------|-----------|----------------------|------------|------------|--------------|--------------|
| 1     | AdamW     | (0.9, 0.999)         | $1 \times 10^{-8}$ | 32 | 0 | cosine |

| Warm-up Ratio | LoRA Alpha | #Runs | Sequence Length | Precision | Gradient Batch Size | |
|---------------|------------|-------|-----------------|-----------|---------------------|-|
| 0.03          | 16         | 3     | 128             | FP32      | 8                   | |

### G.5. Empirical Verification of Assumption 4.1

We perform full fine-tuning for the pre-trained T5 base model (Raffel et al., 2020) on SST-2 dataset from GLUE (Wang et al., 2019) to approximately access the downstream feature matrices. To ensure better convergence, we take the hyperparameter settings which are presented in Table 11.

To validate the part i) of Assumption 4.1, we inspect the spread of the metric values presented in Fig. 7. We can clearly observe that the ratio between the operator norm of fine-tuned weight and minimum norm of neuron within each layer is bounded.

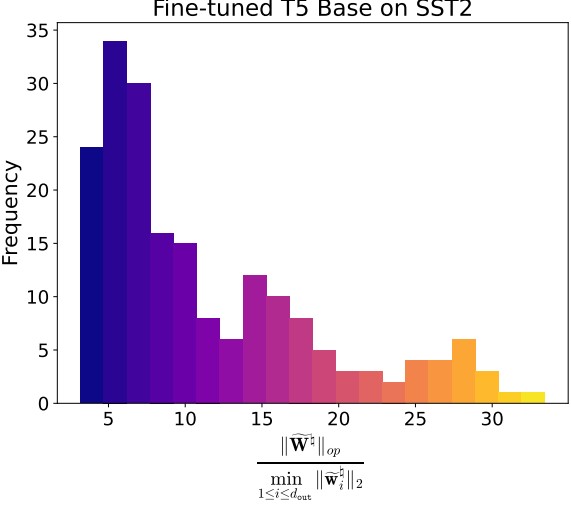

Figure 7: Histogram of the metric values defined presented on x-axis for all fine-tuned weight matrices.

To validate the part ii) of Assumption 4.1, we collect top-32 singular values for each pre-trained layer $\mathbf{W}^\natural$ of pre-trained model. After training, we collect top-32 singular values for each difference weights, i.e. $\Delta\mathbf{W} = \mathbf{W}_{\text{fine-tuned}} - \mathbf{W}^\natural$. The results are shown in Fig. 8. We observe that, across all layers, the singular values of the pre-trained weights are significantly larger than those of the difference weights. For example, the layer on the right has a pretrained operator norm exceeding 200, while its downstream operator norm is only around 4. Moreover, the singular values decrease drastically as the index increases, indicating an ill-conditioned behavior during fine-tuning.

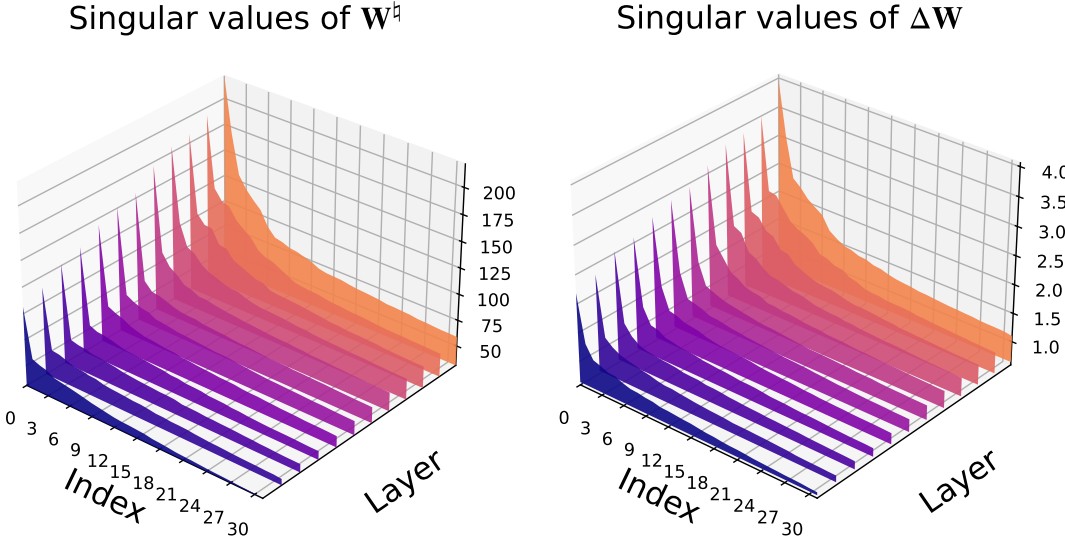

Figure 8: *Left*: top-32 singular values for each pre-trained weight matrices $\mathbf{W}^\natural$. *Right*: top-32 singular values for each difference matrices $\Delta\mathbf{W} = \mathbf{W}_{\text{fine-tuned}} - \mathbf{W}^\natural$ after full fine-tuning. The Index is ranked from the largest to the smallest singular values.

Table 11: Hyperparameters for full fine-tuning on T5-base model used for Appendix G.5.

| Epoch | Optimizer | $(\beta_1, \beta_2)$ | $\epsilon$ | Batchsize | Weight Decay | LR | LR Scheduler | Warm-up Ratio |
|-------|-----------|----------------------|------------|-----------|--------------|-----|--------------|---------------|
| 10 | AdamW | $(0.9, 0.999)$ | $1 \times 10^{-8}$ | 32 | 0.1 | $1 \times 10^{-4}$ | cosine | 0.03 |

