# OpenReview forum: "LoRA-One: One-Step Full Gradient Could Suffice for Fine-Tuning Large Language Models, Provably and Efficiently"
_ICML.cc/2025/Conference — ICML 2025 oral_

### Official Review · Reviewer_TZFT · 2025-03-11

**Overall Recommendation:** 4

**Summary:**

This paper presents a theoretical analysis of Low-Rank Adaptation (LoRA) for efficient fine-tuning of large language models. The main contributions are:
- Theoretical analysis showing that LoRA's gradient updates align with the singular subspace of the full fine-tuning gradient.
- Introduction of a spectral initialization strategy and preconditioned gradient descent to improve convergence.
- Proof of linear convergence rates for both linear and nonlinear models under certain conditions.
- Empirical validation showing improved performance over vanilla LoRA and its variants on NLP benchmarks.

## update after rebuttal

The authors provide a strong theoretical foundation with comprehensive proofs, which is crucial for the community. Therefore, I decide to raise my score to 4.

**Claims And Evidence:**

The claims made in the paper are generally supported by clear and convincing evidence:
- The theoretical alignment between LoRA updates and the singular subspace of the full gradient is demonstrated through mathematical proofs.
- The effectiveness of the proposed spectral initialization and preconditioning methods is supported by both theoretical analysis and empirical results.
- The linear convergence rates are proven under specific assumptions for both linear and nonlinear models.
- Experimental results on NLP benchmarks show consistent improvements over baseline methods.

**Essential References Not Discussed:**

The paper adequately covers relevant literature.

**Experimental Designs Or Analyses:**

The experimental designs are sound:
- The comparison against vanilla LoRA and other variants is comprehensive.
- The ablation studies help understand the contribution of different components.
- The results on multiple NLP tasks demonstrate the general effectiveness of the proposed method.

**Methods And Evaluation Criteria:**

The proposed methods, including spectral initialization and preconditioned gradient descent, make sense for improving parameter-efficient fine-tuning of large models. The evaluation criteria, using standard NLP benchmarks like GLUE tasks, are appropriate for assessing the effectiveness of fine-tuning methods.

**Other Comments Or Suggestions:**

None.

**Other Strengths And Weaknesses:**

Strengths:
- Strong theoretical foundation with comprehensive proofs
- Practical algorithm with empirical validation
- Clear improvement over existing LoRA variants

Weakness:
- The experimental validation could be expanded to more diverse tasks.

**Questions For Authors:**

None.

**Relation To Broader Scientific Literature:**

This work relates to several key areas in machine learning:
- Parameter-efficient fine-tuning methods like LoRA, Adapter, and Prefix-Tuning
- Matrix factorization and low-rank approximation theory
- Optimization for deep neural networks
- Theoretical understanding of fine-tuning large language models

**Theoretical Claims:**

I checked the proofs in the main paper.
- The alignment analysis in Section 3.1 shows that LoRA updates align with the singular subspace of the full gradient, which appears correct.
- The convergence proofs for both linear (Theorem 3.6) and nonlinear (Theorem 4.3) models follow standard optimization proof techniques and seem valid.
- The analysis of spectral initialization and preconditioning methods is mathematically rigorous.

---

> ### Author Rebuttal · Authors · 2025-03-29
>
> We greatly appreciate reviewer's efforts and constructive feedback.
>
> ---
>
> >**Q1** More diverse experimental tasks
>
> We extend our method to fine-tune T5 model on a subset of SuperGLUE [1] datasets, which is more challenging than GLUE and widely used in fine-tuning papers [2-5]. We use full fine-tuning, LoRA and LoRA-One with rank 8 for comparison. For a fair comparison, we search the optimal stepsize for each method via a large grid ranging from 1e-2 to 1e-5. The other settings are same as Appendix F.2. except the epochs of CB is set to 4 since it is a very tiny set. The final results are provided in the table below.
>
> | Data | BoolQ | CB |   COPA  |  RTE   |  WIC   | Avg. |
> | -------- | -------- | -------- | --- | --- | --- | --- |
> |     Full FT     |     $70.89_{\pm 0.02}$     |    $89.29_{\pm 0.00}$      |   $63.67_{\pm 0.94}$  |   $75.33_{\pm 0.34}$  |  $66.35_{\pm 0.32}$  | $73.11$ |
> |     LoRA     |     $70.01_{\pm 0.03}$     |     $85.12_{\pm 0.84}$     |  $61.67_{\pm 0.47}$   |   $70.88_{\pm 0.17}$  |  $65.78_{\pm 0.37}$  | $70.69$ |
> | Ours     |   $70.21_{\pm 0.09}$   | $88.10_{\pm 0.84}$  |  $65.33_{\pm 1.24}$   |   $74.61_{\pm 0.61}$  |  $68.29_{\pm 0.49}$  | $73.31$ |
>
> We also extend our method to fine-tune image classification task on Vision Transformer (ViT [6]). We fine-tune ViT using CIFAR10 and CIFAR100 [7], search optimal stepsize for each method to ensure a fair comparison. Since the convergence of LoRA is slow on CIFAR100, we also run two epochs for comparison. The results are provided in the table below.
>
> | Data (#epoch) | CIFAR10 (1) | CIFAR100 (1) | CIFAR100 (2) |
> |---|---|---|---|
> |Full FT| $98.48_{\pm 0.03}$ | $89.31_{\pm 0.18}$ | $91.73_{\pm 0.08}$ |
> |LoRA| $97.91_{\pm 0.08}$ | $76.46_{\pm 0.22}$ | $80.23_{\pm 0.12}$ |
> |Ours | $98.50_{\pm 0.04}$ | $86.68_{\pm 0.44}$ | $88.83_{\pm 0.11}$ |
>
> In both reasoning and vision classification tasks, we can observe that LoRA-One consistently outperforms LoRA and achieves comparable performance (even better) with full fine-tuning.
>
> ---
>
> Reference:
>
> [1] Wang, A., Pruksachatkun, Y., Nangia, N., Singh, A., Michael, J., Hill, F., Levy, O. and Bowman, S., 2019. Superglue: A stickier benchmark for general-purpose language understanding systems. Advances in neural information processing systems, 32.
>
> [2] Zhang, Q., Chen, M., Bukharin, A., Karampatziakis, N., He, P., Cheng, Y., Chen, W. and Zhao, T., 2023. AdaLoRA: Adaptive budget allocation for parameter-efficient fine-tuning. The Eleventh International Conference on Learning Representations.
>
> [3] Meng, F., Wang, Z. and Zhang, M., 2024. PiSSA: Principal singular values and singular vectors adaptation of large language models. Advances in Neural Information Processing Systems, 37, pp.121038-121072.
>
> [4] Kopiczko, D.J., Blankevoort, T. and Asano, Y.M., 2024. VeRA: Vector-based Random Matrix Adaptation. The Twelfth International Conference on Learning Representations.
>
> [5] Zhao, Z., Shen, T., Zhu, D., Li, Z., Su, J., Wang, X., Kuang, K. and Wu, F., 2025. Merging LoRAs like playing LEGO: Pushing the modularity of lora to extremes through rank-wise clustering. The Thirteenth International Conference on Learning Representations.
>
> [6] Dosovitskiy, A., Beyer, L., Kolesnikov, A., Weissenborn, D., Zhai, X., Unterthiner, T., Dehghani, M., Minderer, M., Heigold, G., Gelly, S. and Uszkoreit, J., 2020. An image is worth 16x16 words: Transformers for image recognition at scale. arXiv preprint arXiv:2010.11929.
>
> [7] Krizhevsky, A. and Hinton, G., 2009. Learning multiple layers of features from tiny images.

---

> > ### Comment · Reviewer_TZFT · 2025-04-02
> >
> > Thank you for your rebuttal. My concerns have been addressed and I devide to keep my score.

---

> > > ### Author Response · Authors · 2025-04-03
> > >
> > > Much appreciated for your positive support.
> > >
> > > This paper studies the training dynamics of LoRA and identifies alignment and pre-condition as key factors for accelerating convergence, with demonstrating **“strong theoretical foundation with comprehensive proofs”.** This theory guides the practice with the LoRA-One algorithm to **“achieve clear improvement over existing LoRA variants”** with massive experiments as recognized by your comments.
> > >
> > > Accordingly we sincerely appreciate your stronger support if possible. We’re happy to address your any further concerns.

---

### Official Review · Reviewer_4x8e · 2025-03-13

**Overall Recommendation:** 3

**Summary:**

This paper investigates methods to enhance the performance of Low-Rank Adaptation (LoRA). The authors make two key discoveries: (i) LoRA tends to align with a specific singular subspace in a single step, and (ii) the use of preconditioners significantly improves convergence in high-rank scenarios. Building on these insights, the authors provide rigorous theoretical guarantees for the convergence of the proposed preconditioned gradient descent algorithm. These contributions advance the understanding of LoRA and offer practical improvements for its application in high-rank settings.

This paper is theoretically rigorous and presents highly interesting insights that are valuable to the community. The analysis is well-structured, and the findings contribute meaningfully to the understanding of low-rank fine-tuning and optimization dynamics. While the theoretical focus is on gradient descent and preconditioned gradient descent, the work lays a strong foundation for further exploration. Overall, it is a compelling contribution that will likely inspire follow-up research and discussions in the field.

**Claims And Evidence:**

1. The title of this paper, "One-Step Full Gradient Suffices for Low-Rank Fine-Tuning," suggests that a single step of full gradient computation is sufficient for effective low-rank fine-tuning. However, the experimental results indicate that the performance of one-step fine-tuning is not only suboptimal but also significantly inferior to that of LoRA. This raises questions about the appropriateness of the title, as it does not accurately reflect the empirical findings. The authors should consider revising the title to better align with the actual results.

2. The model presented in Eq. (2) appears overly simplistic and may lack practical applicability, particularly in the context of modern deep learning architectures. While the theoretical results derived for this model are insightful, it remains unclear whether they extend to more complex and widely used architectures, such as transformers.

3. The authors assume that the input $X$ follows an isotropic centered sub-Gaussian distribution, which ensures that $\tilde{X}$ in Eq. (2) is approximately orthogonal. Under this assumption, the linear case in Eq. (2) becomes nearly equivalent to minimizing $||W-W^b||$, thereby reducing LoRA to a standard low-rank matrix factorization problem of the form $||AB-\Delta||$. In this simplified regime, one-step full gradient descent is indeed sufficient for convergence. However, this assumption on $\tilde{X}$ is critical to the theoretical framework and results presented in the paper. It would be important for the authors to discuss the validity and practicality of this assumption in real-world scenarios, particularly when dealing with more complex data distributions or architectures, such as transformers, where such conditions may not hold. This would help clarify the scope and limitations of their theoretical findings.

**Essential References Not Discussed:**

Not available

**Experimental Designs Or Analyses:**

The experiments presented in the paper are sufficient to support the theoretical claims within the scope of the simplified model and assumptions considered.

**Methods And Evaluation Criteria:**

Yes

**Other Comments Or Suggestions:**

Please refer to the previous section.

**Other Strengths And Weaknesses:**

1. This paper primarily focuses on analyzing the properties and convergence of gradient descent (GD) and preconditioned gradient descent for low-rank fine-tuning. However, in practice, optimizers like Adam, which incorporate adaptive learning rates and momentum, are widely used for training parameters. This creates a notable mismatch between the theoretical analysis and practical implementation. Specifically, the first and second moment estimates in Adam significantly alter the learning dynamics of the parameters, leading to behavior that may differ substantially from the theoretical results presented in this paper. To bridge this gap, the authors should consider extending their analysis to include adaptive optimization methods like Adam or provide empirical evidence demonstrating how their theoretical insights translate to such practical settings.

2. The proposed preconditioned gradient descent method appears to be identical to the algorithm presented in Zhang & Pilanci, 2024, and bears strong similarities to the LoRA_pro method introduced by Wang et al., 2024. The primary distinction lies in the spectral initialization proposed in this work. Given this overlap, the authors should place greater emphasis on discussing the novelty and significance of their initialization scheme within this learning regime. Specifically, they should clarify how the spectral initialization contributes to improved convergence, stability, or performance compared to existing methods, and provide empirical or theoretical evidence to support its importance. This would help better highlight the unique contribution of their work and its potential impact on the field.

**Questions For Authors:**

Please refer to the previous section.

**Relation To Broader Scientific Literature:**

Closely related.

**Theoretical Claims:**

The theoretical results are well-established and sufficient for the simplified model presented in Eq. (2).

---

> ### Author Rebuttal · Authors · 2025-03-28
>
> We greatly thank the reviewer's efforts and constructive comments.
>
> >**Q1** *Appropriateness of title*
>
> **A1** The title is derived from our theory: under proper initialization from one-step full gradient, we can recover $\Delta$ to large extent **at initialization**, see Prop. 3.3 (linear) and Lem. C.5 (nonlinear). This claim is also supported by our toy experiments (Fig. 4) and small-scale datasets (CoLA & MRPC): It achieves similar performance with LoRA but fewer time cost. We provide a table (displayed in anonymous link: https://imgur.com/a/i42zqCm) for comparison.
>
> We are aware of the fact that one-step full gradient is not sufficient for large-scale datasets on LLMs. Hence we continue to run LoRA-One for one epoch. We provide a table (in anonymous link: https://imgur.com/a/qiB59c2) for comparison.
>
> We will update the title according to the reviewer's feedback.
>
> ---
>
> >**Q2** *Simplistic model*
>
> **A2** Extension to complex architecture, e.g., transformer, can be found in A1 of our responses to Reviewer cf1y.
>
> We also remark that, prior works on the analysis of LoRA are generally limited to single-vector linear regression [1,2], matrix factorization [3,4], or strict assumptions [5] (rank-one shift, rank-one LoRA, frozen $B$). Unlike them, our model is more advanced, and settings (e.t.c. algorithms, rank) are standard in practice.
>
> ---
>
> >**Q3** *Data assumption*
>
> **A3** The sub-Gaussian data assumption (e.g., bounded data) is commonly used in theoretical literatures [5,6]. Even for the linear model, our setting is different from low-rank factorization problem because we allow $\tilde{X}$ to be random and can be any matrix shape (including rectangular and asymmetric matrices), which is more flexible.
>
> Our analysis can be extended to more complex data, such as structured data. For example, the linear model under subgaussian data with covariance $\Sigma$, we can track the summary statistic $||\Sigma AB - \Delta||_F$ instead of $||AB - \Delta||_F$ by reparametrization. Also, the nonlinear model can be extended to gaussian mixture data via incorporating Stein's lemma into Hermite analysis. We leave this extension as a future work.
>
> According to the reviewer's feedback, we will add a detailed discussion on the validity of our theory to practice and clearly clarify its scope and limitations in the updated version.
>
> ---
>
> >**Q4** *GD vs Adam*
>
> **A4** We understand reviewer's concern about the gap between theoretical (GD) and practical (Adam) optimizers. Before studying Adam, understanding GD is the first step, commonly used for theoretical analysis [1-4,6] including LoRA and use Adam in practice. Our theory can serve as a conceptional motivation to design the practical algorithm and we achieve significant empirical improvements (see Table 2&8 in paper and Table in A1). Extending our theory to adaptive methods will be an interesting topic.
>
> ---
>
> >**Q5** *Distinction with prior methods*
>
> **A5** In our submission, we have compared with gradient alignment based algorithms in Appendix E. According to the reviewers' suggestion, we add a detailed comparison with Zhang & Pilanci 2024 [2] & LoRA pro here and will include in the updated version.
>
> Comparing to [2]:
>
> - We identify ill-condition as a convergence bottleneck in LoRA (Thm. 3.5) and demonstrate that pre-condition resolves it (supported by Thm. 3.6&4.3, Fig. 3, Tables 10–11). [2] employ pre-condition (originated from matrix sensing) for stability but overlook ill-condition.
>
> - Our method significantly outperforms theirs (Table 2), highlighting the importance of spectral initialization.
>
> Comparing to LoRA pro:
>
> - We only need the **exact first full batch** gradient for initialization, but LoRA pro needs to **approximate** the **stochastic batch** gradient from full fine-tuning in **every training step** using more matrix operations.
>
> - LoRA pro adds $8dr^2+4kr^2+24.5 r^3$ **more** FLOPs for each $A\in R^{d\times r},B\in R^{r\times k}$.
>
> - We have **much more efficient** memory usage, e.t.c. for GSM8K, we provide a table (in anonymous link: https://imgur.com/a/yjZYUzC)  for comparison.
>
> There is a series of empirical works e.t.c. LoRA Pro and LoRA-GA introducing gradient information, we are first to provide a theoretical foundation for these heuristic methods and wish we can inspire more theoretical works.
>
> ---
>
> Reference:
> [1] Lora+. ICML24.
>
> [2] Riemannian preconditioned lora. ICML24.
>
> [3] On the crucial role of initialization for matrix factorization. ICLR25.
>
> [4] Compressible dynamics in deep overparameterized low-rank learning & adaptation. ICML24.
>
> [5] Gradient dynamics for low-rank fine-tuning beyond kernels.
>
> [6] Implicit balancing and regularization: Generalization and convergence guarantees for overparameterized asymmetric matrix sensing. COLT23.

---

### Official Review · Reviewer_cf1y · 2025-03-15

**Overall Recommendation:** 4

**Summary:**

The paper provides a theoretical analysis of LoRA fine-tuning by showing that a single full gradient step naturally aligns the LoRA updates with the top singular subspace of the full gradient, and by introducing a spectral initialization strategy, it can effectively recover the downstream low-rank target before iterative training even begins; moreover, by incorporating preconditioned gradient descent, the method eliminates dependence on the condition number of the feature shift, thereby accelerating convergence, and these insights culminate in the development of the LoRA-One algorithm, which achieve empirical improvements over traditional LoRA methods on various benchmarks.

**Claims And Evidence:**

This work makes two claims, one theoretical and one empirical.

Theoretically, this work analyzes LoRA fine-tuning applied to a 1-layer neural network without and with a non-linearity. The theory is detailed and rigorous, although it is unclear how generalizable these insights are to the practical setup of LoRA fine-tuning deep transformers.

Empirically, this work proposes the LoRA-One method, which is inspired by the 1-layer analysis, and apply it to various fine-tuning tasks (on deep transformers). The results outperform LoRA-type methods in terms of generalization performance on several benchmarks.

**Essential References Not Discussed:**

None.

**Experimental Designs Or Analyses:**

The experimental design seems to be sound. The experiments are done thoroughly, and the gains over other competing methods are consistent.

**Methods And Evaluation Criteria:**

The evaluation criteria seems to be sound.

**Other Comments Or Suggestions:**

.

**Other Strengths And Weaknesses:**

I'm not very convinced of the theoretical merits of this work: the theorem statements are not the cleanest one can hope for and it is unclear if their insights generalize to deeper architectures.

However, the experimental demonstrations are thorough and convincing. In the end, the idea of initializing the LoRA modules according to the subspaces defined by the full gradient seems like a very nice insight.

**Questions For Authors:**

.

**Relation To Broader Scientific Literature:**

The prior work seem to be appropriately referenced.

**Theoretical Claims:**

The proofs justifying the theoretical claims seem to be sound.

---

> ### Author Rebuttal · Authors · 2025-03-28
>
> We deeply appreciate the reviewer’s efforts and the positive support.
>
> ---
>
> >**Q1** *The theory is detailed and rigorous, although it is unclear how generalizable these insights are to the practical setup of LoRA fine-tuning deep transformers.*
>
> **A1** We are greatly thankful for the reviewer to point out the potential extension to transformer-based models. The attention module and depth will be very challenging. Currently, we believe our techniques can be extended to single linear attention module which considered in a variety in-context learning theory papers [1], i.e.
>
> $$
> \hat{y} = \left\langle \frac{\mathbf{W}\mathbf{X}^\top\mathbf{y}}{N}, \mathbf{x}_{\tt query}\right\rangle,
> $$
>
> where $\mathbf{W}$ is the attention matrix and $(\mathbf{X},\mathbf{y},\mathbf{x}_{\tt query})$ are data. By a reparametrization, we can admit the following matrix inner product form
>
> $$
> \hat{y} = \left\langle \frac{\mathbf{X}^\top\mathbf{y}\mathbf{x}_{\tt query}^\top}{N}, \mathbf{W}\right\rangle.
> $$
>
> Treating $\frac{\mathbf{X}^\top\mathbf{y}\mathbf{x}_{\tt query}^\top}{N}$ as a random measurement matrix, we can characterize the above model as a special variant of matrix sensing model [2] and LoRA will be a factorization approach. With this characterization. we believe our analysis can also apply. In the updated version, we will add a discussion on the possible extension to linear attention module.
>
> The original attention module in transformer are more sophisticated due to the existence of softmax activations and multiple tunable weight matrices. First, handling the nonlinearity brought by softmax needs to incorporate more theoretical tools, such as [3,4]. Second, the simultaneous training of all layers (Q,K,V) in attention is quite challenging, we might seek  tools from dynamical systems to deal with the coupling between parameters.
>
> We believe the theory can be extended to deep transformer if the single attention module can be understood thoroughly.
>
> From an empirical perspective, in our experimental part, we choose LLaMA 2 7B and T5 base model which consist of multiple attention layers and we achieve promising results (Table 2 & 8) on various NLP tasks using our theory-grounded method, which can empirically justify our insights from theory.
>
> ---
>
> >**Q2** *I'm not very convinced of the theoretical merits of this work: the theorem statements are not the cleanest one can hope for*
>
> **A2** Our theory demonstrates the subspace alignment between $(A_t, B_t)$ and one-step full gradient, see Section 3.1, and motivates us to design one certain initialization to achieve this subspace alignment, which is for convergence acceleration. According to the reviewer's feedback, we will use the simplified version for better illustration. For example, Theorem 3.2 (alignment between $A_t$ and $G^{\natural}$) can be simplified as
>
> Theorem 3.2 [Simplified]. Under standard random initialization for LoRA, after training for $t^*=\mathcal{O}(\ln d)$ steps, then we have the subspace alignment between $G^{\natural}$ and $A_t$
>
> $$
> || U^\top_{r^*,\perp}( G^{\natural}) U_{r^*}\left( A_{t^*}\right)||_{op} \mbox{~is small}, \quad w.h.p.
> $$
>
> ---
>
> Reference:
>
> [1] How Many Pretraining Tasks Are Needed for In-Context Learning of Linear Regression? ICLR24.
>
> [2] Implicit balancing and regularization: Generalization and convergence guarantees for overparameterized asymmetric matrix sensing. COLT23.
>
> [3] Training dynamics of multi-head softmax attention for in-context learning: Emergence, convergence, and optimality. COLT24.
>
> [4] On the convergence of encoder-only shallow transformers. NeurIPS23.

---

### Official Review · Reviewer_4cE1 · 2025-03-16

**Overall Recommendation:** 3

**Summary:**

This paper focuses on the learning dynamics of Low-Rank Adaptation (LoRA) and proposes improvements in initialization and gradient preconditioning.

The authors analyze both linear and nonlinear matrix factorization cases, where the objective is to minimize $\|\tilde{X}(W+AB)-\tilde{Y}\|_F^2$ using gradient descent. In the linear case, they show that the factors $(A, B)$ in vanilla LoRA align with the top singular vectors of the he first iteration's full gradient. To accelerate this process, they propose spectral initialization based on the full gradient, and prove linear convergence under such initialization. To further address ill-conditioned targets, they introduce preconditioning, leading to a condition-number-agnostic convergence rate. They also extend these results to nonlinear settings. The paper provides insights into why introducing full-gradient information benefits fine-tuning.

For practical applications, the authors propose LoRA-One, a fine-tuning method incorporating spectral initialization and preconditioned GD, achieving performance gains on the GLUE benchmark compared to LoRA+, P-LoRA, and Galore.

**Claims And Evidence:**

The paper presents both theoretical analysis and empirical validation, including:
Alignment of LoRA updates with the first-step gradient (Fig. 2), GD trajectories of LoRA-init vs. spectral-init (Fig. 4), and practical performance improvements on GLUE benchmarks.

**Essential References Not Discussed:**

None.

**Experimental Designs Or Analyses:**

The authors validate their findings through numerical experiments on matrix factorization and further demonstrate the effectiveness of their method on modern NLP tasks, as outlined in the **Summary**.

**Methods And Evaluation Criteria:**

Characterizing LoRA as a single-layer matrix factorization problem seems somewhat unrealistic. While LoRA is often viewed as a low-rank approximation of the full fine-tuning update, its optimization behaves differently in practice (e.g., https://arxiv.org/abs/2410.21228 shows that LoRA updates are nearly orthogonal to full fine-tuning updates). This raises questions about whether a unique $\Delta$ exists in such cases.

Moreover, I am skeptical about introducing full-gradient information into parameter-efficient fine-tuning (PEFT). If computing the full gradient is feasible, then full fine-tuning—given proper hyperparameter tuning and regularizations—usually serves as a strong benchmark. From my perspective, the most effective approaches in practice still tend to be either full fine-tuning with careful optimization or vanilla LoRA.

**Other Comments Or Suggestions:**

A minor typo: Line 16, "algin" → "align".

**Other Strengths And Weaknesses:**

See **Summary** and **Methods And Evaluation Criteria**.

**Questions For Authors:**

I would like to see that if LoRA-one actually leads to a better alignment to the first step gradients on the practical NLP tasks?

**Relation To Broader Scientific Literature:**

This paper may be in the interest of the general area of improving the efficiency of large models.

**Theoretical Claims:**

The theoretical analysis is conducted under a single-layer matrix factorization setting, focusing on its optimization and generalization. The theorems are mostly correct and intuitive.

---

> ### Author Rebuttal · Authors · 2025-03-28
>
> We thank reviewer's effort and constructive comments on this work. We fixed the typo on "algin" and address your concerns as below.
>
> >**Q1** Simplistic model
>
> **A1** Our formulation **can be not regarded as the matrix factorization (MF) problem**, which is strictly defined as $\min_{A, B} ||AB-\Delta||_F^2$.
>
> For linear models, our problem reduces to MF only when the downstream data matrix $\tilde{X}$ is an identity matrix (i.e., square and symmetric). However, our setting is more general: we assume $\tilde{X}$ to be random sub-Gaussian (e.g., bounded entries) and allow it to have arbitrary dimensions, including rectangular and asymmetric shapes. This makes our framework more flexible than standard MF.
>
> Our setting for nonlinear models is structurally different to MF due to the nonlinearity of ReLU.
>
> There are some prior works that use MF to study LoRA but the practical validity of their theory should be concerned. In a recent paper [1], authors analyze the dynamics of MF using Nystrom initialization and apply it to LoRA. However, such initialization in MF needs to know the complete ground truth $\Delta$ as the prior knowledge, which is unknown in real-world fine-tuning. So there is a large gap between their MF theory and practice. The MF is also considered in [2] and applied to LoRA, however the convergence and generalization guarantees are missing.
>
> Our work theoretically studies LoRA under realistic assumptions, demonstrating the subspace alignment between $(A_t, B_t)$ and one-step full gradient, and then develops a theory-grounded algorithm to achieve promising performance in practice.
>
> ---
>
> >**Q2** LoRA and FFT optimizes differently (e.g., paper). Existence of \Delta.
>
> **A2** We thank for the reviewer to point out this paper and the empirical findings in this paper are very interesting and inspiring. In our updated version, we will cite the paper above and add a discussion.
>
> Our theory follows the classical data generation process in machine learning for generalization guarantees, i.e., the label of (downstream) data is generated by a linear/nonlinear target function, corrupted by some noise. Under this setting, $\Delta$ represents a part of the target function and is therefore unique. Then different fine-tuning strategies (e.g., full FT, LoRA) can still lead to different solutions that achieve similar performance in practice, as suggested by the aforementioned arXiv paper. This phenomenon is common in deep learning.
>
> Note that, if we focus solely on optimization guarantees, our theory remains valid regardless of whether $\Delta$ is unique or not.
>
> ---
>
> >**Q3** Skeptical about introducing full-gradient info into PEFT
>
> **A3** We understand the reviewer's concern and would like to provide more details to make it clear.
>
> All experiments are conducted on a single A100 40GB GPU. Due to memory limitations, full fine-tuning is not feasible. However, our implementation employs a memory-efficient approach [3,4], allowing the computation of the first full gradient. Notably, this approach cannot extend to full fine-tuning.
>
> It's true that using one-step full gradient information requires additional (and marginal) time cost but significantly improves the accuracy. Here we provide a table (displayed in the anonymous link: https://imgur.com/a/qiB59c2) for comparison between vanilla LoRA and LoRA-One on LLaMA 2 7B. It takes additional 10 mins but the accuracy is improved with more than 5%.
>
> In fact, using additional gradient information from full fine-tuning is a trend and empirically considered in [3,5] as well.
>
> ---
>
> >**Q4** LoRA-one leads to better alignment on NLP tasks?
>
> **A4** Based on Algo. 1, LoRA-One achieves perfect alignment with first-step gradient at initialization but LoRA needs to take long time (e.g. one epoch) to achieve a weak alignment, e.g., fine-tune on MRPC via LoRA with r=8 for one epoch, using principal angle defined in Thm 3.1 & 3.2, we have
>
> | | Avg. | Min | Max |
> | - | - | - | - |
> | A | 0.399 | 0.199 | 0.854 |
> | B | 0.267 | 0.156 | 0.393 |
>
> Thanks to alignment, LoRA-One surpasses the first-step GD accuracy at the start and continues growing better (displayed in anonymous link https://imgur.com/a/BG5DZca). But LoRA needs a long time to achieve the first-step GD accuracy (Table 2). Also, LoRA-GA mismatches singular subspace with first-step gradient which is the potential reason that we consistently outperform theirs on all datasets even without precondition (Fig 3, Table 10&11).
>
> ---
>
> We expect that our clarification would help you have a better understanding on this work.
>
> ---
>
> Reference:
>
> [1] On the crucial role of initialization for matrix factorization. ICLR25.
>
> [2] Compressible dynamics in deep overparameterized low-rank learning & adaptation. ICML24.
>
> [3] Lora-GA: Low-rank adaptation with gradient approximation. NeurIPS24.
>
> [4] Full parameter fine-tuning for large language models with limited resources. ACL24.
>
> [5] LoRA-Pro: Are Low-Rank Adapters Properly Optimized? ICLR25.

---

> > ### Comment · Reviewer_4cE1 · 2025-04-02
> >
> > Thank you for the rebuttal. I appreciate the authors' efforts in both the paper and the response. I also apologize for the misstatement regarding the authors' focus on learning dynamics in linear/nonlinear regression problems under coefficient shift. I am glad to learn that incorporating the full gradient introduces a negligible time cost (Q3) and that the authors have verified the enhanced alignment brought by LoRA-One (Q4).
> >
> > However, my main concern remains whether fine-tuning large models can truly be interpreted as a regression problem and whether refining the vanilla LoRA method is still a pressing issue. Since LoRA was introduced four years ago, thousands of papers have claimed to improve upon it in an elegant and theoretically superior manner. But from my perspective, many of these modifications do not seem to withstand the test of time. Of course, this perspective may be highly biased.
> >
> > Overall, this paper is well-organized, the theoretical analysis is solid and the authors provide concrete experimental results. In light of the authors' clarifications, I have raised my score to 3.

---

> > > ### Author Response · Authors · 2025-04-04
> > >
> > > We deeply appreciate your support and are very happy to see you have a better understanding of our work.
> > >
> > > We would like to make a few more keynotes:
> > >
> > > - Regression is the first step to theoretically understand fine-tuning beyond linear models. Extension to classification based on [1] or Next Token Prediction (which is closer to LLM) based on [2] requires more efforts under different settings.
> > >
> > > - We deeply agreed with you on the concerns of validity of huge amount of modifications to LoRA. They have different benchmarks and experimental settings, so sometimes they may only work in a narrow setting. Our work can clarify misunderstandings of some heuristic algorithms and improve the performance in practice, which is the spirit and value of theory in this case (with required simplifications but acceptable).
> > >
> > > Reference:
> > >
> > > [1] Collins, L., Hassani, H., Soltanolkotabi, M., Mokhtari, A. and Shakkottai, S. Provable Multi-Task Representation Learning by Two-Layer ReLU Neural Networks. ICML 2024.
> > >
> > > [2] Thrampoulidis, C. Implicit Optimization Bias of Next-token Prediction in Linear Models. NeurIPS 2024.

---

### Decision · Program_Chairs · 2025-05-01

**Decision:**

Accept (oral)

**Comment:**

This paper presents a theoretical and empirical study of Low-Rank Adaptation (LoRA), focusing on its optimization dynamics and proposing a new method, LoRA-One, which combines spectral initialization with preconditioned gradient descent. The theoretical contribution is that LoRA updates, under gradient descent, align with top singular directions of the full gradient. LoRA-One builds upon this insight to offer accelerated convergence and improved generalization. Empirically, LoRA-One consistently outperforms LoRA and its variants (LoRA+, LoRA-GA, LoRA-Pro) across NLP tasks (GLUE, SuperGLUE) and vision benchmarks.

There are some concerns about the scope of the claims and its overlap with existing work. The title and the framing could overstate the contribution; "one-step full gradient suffices" is only true under specific assumptions and in simplified settings. The authors acknowledged this and have agreed to revise the title. Also, the proposed LoRA-One is, in the end, very similar to a prior work of Zhang & Pilanci 2024, although the present paper provides a more thorough analysis, with a focus on the ill-conditioning aspect of the problem.

In the end, I found both the theoretical and empirical aspect of this paper to be compelling. The paper builds a deeper understanding of LoRA and uses it to inform the design of a practically competitive algorithm. This is a very worthy paper for inclusion in the ICML program.